# Deep Networks Provably Classify Data on Curves

**Tingran Wang**
Columbia University
tw2579@columbia.edu

**Sam Buchanan**
Columbia University
s.buchanan@columbia.edu

**Dar Gilboa**
Harvard University
dar_gilboa@fas.harvard.edu

**John Wright**
Columbia University
jw2966@columbia.edu

## Abstract

Data with low-dimensional nonlinear structure are ubiquitous in engineering and scientific problems. We study a model problem with such structure—a binary classification task that uses a deep fully-connected neural network to classify data drawn from two disjoint smooth curves on the unit sphere. Aside from mild regularity conditions, we place no restrictions on the configuration of the curves. We prove that when (i) the network depth is large relative to certain geometric properties that set the difficulty of the problem and (ii) the network width and number of samples are polynomial in the depth, randomly-initialized gradient descent quickly learns to correctly classify all points on the two curves with high probability. To our knowledge, this is the first generalization guarantee for deep networks with nonlinear data that depends only on intrinsic data properties. Our analysis proceeds by a reduction to dynamics in the neural tangent kernel (NTK) regime, where the network depth plays the role of a fitting resource in solving the classification problem. In particular, via fine-grained control of the decay properties of the NTK, we demonstrate that when the network is sufficiently deep, the NTK can be locally approximated by a translationally invariant operator on the manifolds and stably inverted over smooth functions, which guarantees convergence and generalization.

## 1   Introduction

In applied machine learning, engineering, and the sciences, we are frequently confronted with the problem of identifying low-dimensional structure in high-dimensional data. In certain well-structured data sets, identifying a good low-dimensional model is the principal task: examples include convolutional sparse models in microscopy [43] and neuroscience [10, 16], and low-rank models in collaborative filtering [7, 8]. Even more complicated datasets from problems such as image classification exhibit some form of low-dimensionality: recent experiments estimate the effective dimension of CIFAR-10 as 26 and the effective dimension of ImageNet as 43 [61]. The variability in these datasets can be thought of as comprising two parts: a "probabilistic" variability induced by the distribution of geometries associated with a given class, and a "geometric" variability associated with physical nuisances such as pose and illumination. The former is challenging to model analytically; virtually all progress on this issue has come through the introduction of large datasets and high-capacity learning machines. The latter induces a much cleaner analytical structure: transformations of a given image lie near a low-dimensional submanifold of the image space (Figure 1). The celebrated successes of convolutional neural networks in image classification seem to derive from their ability to simultaneously handle both types of variability. Studying how neural networks compute with data lying near a low-dimensional manifold is an essential step towards understanding how neural

35th Conference on Neural Information Processing Systems (NeurIPS 2021).

networks achieve invariance to continuous transformations of the image domain, and towards the longer term goal of developing a more comprehensive mathematical understanding of how neural networks compute with real data. At the same time, in some scientific and engineering problems, classifying manifold-structured data *is* the goal—one example is in gravitational wave astronomy [22, 30], where the goal is to distinguish true events from noise, and the events are generated by relatively simple physical systems with only a few degrees of freedom.

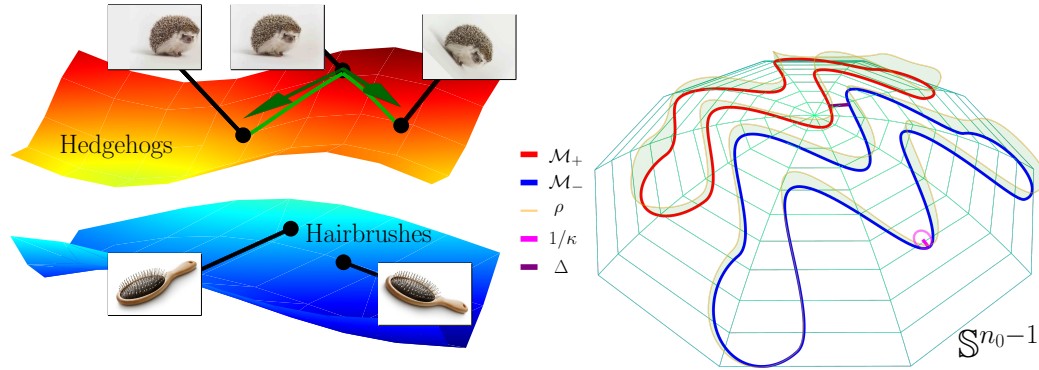

Figure 1: **Low-dimensional structure in image data and the two curves problem.** *Left:* Manifold structure in natural images arises due to invariance of the label to continuous domain transformations such as translations and rotations. *Right:* The two curve problem. We train a neural network to classify points sampled from a density $\rho$ on the submanifolds $\mathcal{M}_+, \mathcal{M}_-$ of the unit sphere. We illustrate the angle injectivity radius $\Delta$ and curvature $1/\kappa$. These parameters help to control the difficulty of the problem: problems with smaller separation and larger curvature are more readily separated with deeper networks.

Motivated by these long term goals, in this paper we study the *multiple manifold problem* (Figure 1), a mathematical model problem in which we are presented with a finite set of labeled samples lying on disjoint low-dimensional submanifolds of a high-dimensional space, and the goal is to correctly classify *every* point on each of the submanifolds—a strong form of generalization. The central mathematical question is how the *structure* of the data (properties of the manifolds such as dimension, curvature, and separation) influences the *resources* (data samples, and network depth and width) required to guarantee generalization. Our main contribution is the first end-to-end analysis of this problem for a nontrivial class of manifolds: one-dimensional smooth curves that are non-intersecting, cusp-free, and without antipodal pairs of points. Subject to these constraints, the curves can be oriented essentially arbitrarily (say, non-linearly-separably, as in Figure 1), and the hypotheses of our results depend only on architectural resources and intrinsic geometric properties of the data. To our knowledge, this is the first generalization result for training a deep nonlinear network to classify structured data that makes no a-priori assumptions about the representation capacity of the network or about properties of the network after training.

Our analysis proceeds in the neural tangent kernel (NTK) regime of training, where the network is wide enough to guarantee that gradient descent can make large changes in the network output while making relatively small changes to the network weights. This approach is inspired by the recent work [57], which reduces the analysis of generalization in the one-dimensional multiple manifold problem to an auxiliary problem called the *certificate problem*. Solving the certificate problem amounts to proving that the target label function lies near the stable range of the NTK. The existence of certificates (and more generally, the conditions under which practically-trained neural networks can fit structured data) is open, except for a few very simple geometries which we will review below—in particular, [57] leaves this question completely open. Our technical contribution is to show that setting the network depth sufficiently large relative to intrinsic properties of the data guarantees the existence of a certificate (Theorem 3.1), resolving the one-dimensional case of the multiple manifold problem for a broad class of curves (Theorem 3.2). This leads in turn to a novel perspective on the role of the network depth as a *fitting resource* in the classification problem, which is inaccessible to shallow networks.

## 1.1 Related Work

**Deep networks and low dimensional structure.** Modern applications of deep neural networks include numerous examples of low-dimensional manifold structure, including pose and illumination variations in image classification [1, 5], as well as detection of structured signals such as electrocardiograms [14, 20], gravitational waves [22, 30], audio signals [13], and solutions to the diffusion equation [48]. Conventionally, to compute with such data one might begin by extracting a low-dimensional representation using nonlinear dimensionality reduction ("manifold learning") algorithms [2–4, 6, 12, 54, 56]. For supervised tasks, there is also theoretical work on kernel regression over manifolds [9, 11, 19, 51]. These results rely on very general Sobolev embedding theorems, which are not precise enough to specify the interplay between regularity of the kernel and properties of the data need to obtain concrete resource tradeoffs in the two curve problem. There is also a literature which studies the resource requirements associated with approximating functions over low-dimensional manifolds [15, 29, 38, 44]: a typical result is that for a sufficiently smooth function there *exists* an approximating network whose complexity is controlled by intrinsic properties such as the dimension. In contrast, we seek *algorithmic* guarantees that prove that we can efficiently train deep neural networks for tasks with low-dimensional structure. This requires us to grapple with how the geometry of the data influences the dynamics of optimization methods.

**Neural networks and structured data—theory?** Spurred by insights in asymptotic infinite width [23, 24] and non-asymptotic [18, 21] settings, there has been a surge of recent theoretical work aimed at establishing guarantees for neural network training and generalization [26–28, 34, 37, 40, 49, 55]. Here, our interest is in end-to-end generalization guarantees, which are scarce in the literature: those that exist pertain to unstructured data with general targets, in the regression setting [32, 36, 46, 59], and those that involve low-dimensional structure consider only linear structure (i.e., spheres) [46]. For less general targets, there exist numerous works that pertain to the teacher-student setting, where the target is implemented by a neural network of suitable architecture with unstructured inputs [17, 33, 40, 49, 63]. Although adding this extra structure to the target function allows one to establish interesting separations in terms of e.g. sample complexity [31, 39, 49, 62] relative to the preceding analyses, which proceed in the "kernel regime", we leverage kernel regime techniques in our present work because they allow us to study the interactions between deep networks and data with nonlinear low-dimensional structure, which is not possible with existing teacher-student tools. Relaxing slightly from results with end-to-end guarantees, there exist 'conditional' guarantees which require the existence of an efficient representation of the target mapping in terms of a certain RKHS associated to the neural network [34, 53, 57, 58]. In contrast, our present work obtains unconditional, end-to-end generalization guarantees for a nontrivial class of low-dimensional data geometries.

## 2 Problem Formulation

**Notation.** We use bold notation $\boldsymbol{x}$, $\boldsymbol{A}$ for vectors and matrices/operators (respectively). We write $\|\boldsymbol{x}\|_p = (\sum_{i=1}^n |x_i|^p)^{1/p}$ for the $\ell^p$ norm of $\boldsymbol{x}$, $\langle \boldsymbol{x}, \boldsymbol{y} \rangle = \sum_{i=1}^n x_i y_i$ for the euclidean inner product, and for a measure space $(X, \mu)$, $\|g\|_{L_\mu^p} = (\int_X |g(x)|^p \, \mathrm{d}\mu(x))^{1/p}$ denotes the $L_\mu^p$ norm of a function $g : X \to \mathbb{R}$. The unit sphere in $\mathbb{R}^n$ is denoted $\mathbb{S}^{n-1}$, and $\angle(\boldsymbol{x}, \boldsymbol{y}) = \cos^{-1}(\langle \boldsymbol{x}, \boldsymbol{y} \rangle)$ denotes the angle between unit vectors. For a kernel $K : X \times X \to \mathbb{R}$, we write $\boldsymbol{K}_\mu[g](x) = \int_X K(x, x') g(x') \, \mathrm{d}\mu(x')$ for the action of the associated Fredholm integral operator; an omitted subscript denotes Lebesgue measure. We write $\boldsymbol{P}_S$ to denote the orthogonal projection operator onto a (closed) subspace $S$. Full notation is provided in Appendix B.

### 2.1 The Two Curve Problem[1]

A natural model problem for the tasks discussed in Section 1 is the classification of low-dimensional submanifolds using a neural network. In this work, we study the one-dimensional, two-class case of this problem, which we refer to as the *two curve problem*. To fix ideas, let $n_0 \geq 3$ denote the ambient dimension, and let $\mathcal{M}_+$ and $\mathcal{M}_-$ be two disjoint smooth regular simple closed curves taking values in $\mathbb{S}^{n_0-1}$, which represent the two classes (Figure 1). In addition, we require that

---

[1]The content of this section follows the presentation of [57]; we reproduce it here for self-containedness. We omit some nonessential definitions and derivations for concision; see Appendix C.1 for these details.

the curves lie in a spherical cap of radius $\pi/2$: for example, the intersection of the sphere and the nonnegative orthant $\{\boldsymbol{x} \in \mathbb{R}^{n_0} \mid \boldsymbol{x} \geq 0\}$.[2] Given $N$ i.i.d. samples $\{\boldsymbol{x}_i\}_{i=1}^N$ from a density $\rho$ supported on $\mathcal{M} = \mathcal{M}_+ \cup \mathcal{M}_-$, which is bounded above and below by positive constants $\rho_{\max}$ and $\rho_{\min}$ and has associated measure $\mu$, as well as their corresponding $\pm 1$ labels, we train a feedforward neural network $f_{\boldsymbol{\theta}} : \mathbb{R}^{n_0} \to \mathbb{R}$ with ReLU nonlinearities, uniform width $n$, and depth $L$ (and parameters $\boldsymbol{\theta}$) by minimizing the empirical mean squared error using randomly-initialized gradient descent. Our goal is to prove that this procedure yields a separator for the geometry given sufficient resources $n$, $L$, and $N$—i.e., that $\mathrm{sign}(f_{\boldsymbol{\theta}_k}) = 1$ on $\mathcal{M}_+$ and $-1$ on $\mathcal{M}_-$ at some iteration $k$ of gradient descent.

To achieve this, we need an understanding of the progress of gradient descent. Let $f_\star : \mathcal{M} \to \{\pm 1\}$ denote the classification function for $\mathcal{M}_+$ and $\mathcal{M}_-$ that generates our labels, write $\zeta_{\boldsymbol{\theta}}(\boldsymbol{x}) = f_{\boldsymbol{\theta}}(\boldsymbol{x}) - f_\star(\boldsymbol{x})$ for the network's prediction error, and let $\boldsymbol{\theta}_{k+1} = \boldsymbol{\theta}_k - (\tau/N) \sum_{i=1}^N \zeta_{\boldsymbol{\theta}_k}(\boldsymbol{x}_i) \nabla_{\boldsymbol{\theta}} f_{\boldsymbol{\theta}_k}(\boldsymbol{x}_i)$ denote the gradient descent parameter sequence, where $\tau > 0$ is the step size and $\boldsymbol{\theta}_0$ represents our Gaussian initialization. Elementary calculus then implies the error dynamics equation $\zeta_{\boldsymbol{\theta}_{k+1}} = \zeta_{\boldsymbol{\theta}_k} - (\tau/N) \sum_{i=1}^N \Theta_k^N(\,\cdot\,, \boldsymbol{x}_i) \zeta_{\boldsymbol{\theta}_k}(\boldsymbol{x}_i)$ for $k = 0, 1, \ldots$, where $\Theta_k^N : \mathcal{M} \times \mathcal{M} \to \mathbb{R}$ is a certain kernel. The precise expression for this kernel is not important for our purposes: what matters is that (i) making the width $n$ large relative to the depth $L$ guarantees that $\Theta_k^N$ remains close throughout training to its 'initial value' $\Theta^{\mathrm{NTK}}(\boldsymbol{x}, \boldsymbol{x}') = \langle \nabla_{\boldsymbol{\theta}} f_{\boldsymbol{\theta}_0}(\boldsymbol{x}), \nabla_{\boldsymbol{\theta}} f_{\boldsymbol{\theta}_0}(\boldsymbol{x}') \rangle$, the *neural tangent kernel*; and (ii) taking the sample size $N$ to be sufficiently large relative to the depth $L$ implies that a nominal error evolution defined as $\zeta_{k+1} = \zeta_k - \tau \boldsymbol{\Theta}_\mu^{\mathrm{NTK}}[\zeta_k]$ with $\zeta_0 = \zeta_{\boldsymbol{\theta}_0}$ uniformly approximates the actual error $\zeta_{\boldsymbol{\theta}_k}$ throughout training. In other words: to prove that gradient descent yields a neural network classifier that separates the two manifolds, it suffices to overparameterize, sample densely, and show that the norm of $\zeta_k$ decays sufficiently rapidly with $k$. This constitutes the "NTK regime" approach to gradient descent dynamics for neural network training [23].

The evolution of $\zeta_k$ is relatively straightforward: we have $\zeta_{k+1} = (\mathrm{Id} - \tau \boldsymbol{\Theta}_\mu^{\mathrm{NTK}})^k[\zeta_0]$, and $\boldsymbol{\Theta}_\mu^{\mathrm{NTK}}$ is a positive, compact operator, so there exist an orthonormal basis of $L_\mu^2$ functions $v_i$ and eigenvalues $\lambda_1 \geq \lambda_2 \geq \cdots \geq 0$ such that $\zeta_{k+1} = \sum_{i=1}^\infty (1 - \tau \lambda_i)^k \langle \zeta_0, v_i \rangle_{L_\mu^2} v_i$. In particular, with bounded step size $\tau < \lambda_1^{-1}$, gradient descent leads to rapid decrease of the error if and only if the initial error $\zeta_0$ is well-aligned with the eigenvectors of $\boldsymbol{\Theta}_\mu^{\mathrm{NTK}}$ corresponding to large eigenvalues. Arguing about this alignment explicitly is a challenging problem in geometry: although closed-form expressions for the functions $v_i$ exist in cases where $\mathcal{M}$ and $\mu$ are particularly well-structured, *no such expression is available for general nonlinear geometries*, even in the one-dimensional case we study here. However, this alignment can be guaranteed *implicitly* if one can show there exists a function $g : \mathcal{M} \to \mathbb{R}$ of small $L_\mu^2$ norm such that $\boldsymbol{\Theta}_\mu^{\mathrm{NTK}}[g] \approx \zeta_0$—in this situation, most of the energy of $\zeta_0$ must be concentrated on directions corresponding to large eigenvalues. We call the construction of such a function the certificate problem [57, Eqn. (2.3)]:

**Certificate Problem.** *Given a two curves problem instance $(\mathcal{M}, \rho)$, find conditions on the architectural hyperparameters $(n, L)$ so that there exists $g : \mathcal{M} \to \mathbb{R}$ satisfying $\|\boldsymbol{\Theta}_\mu^{\mathrm{NTK}}[g] - \zeta_0\|_{L_\mu^2} \lesssim 1/L$ and $\|g\|_{L_\mu^2} \lesssim 1/n$, with constants depending on the density $\rho$ and logarithmic factors suppressed.*

The construction of certificates demands a fine-grained understanding of the integral operator $\boldsymbol{\Theta}_\mu^{\mathrm{NTK}}$ and its interactions with the geometry $\mathcal{M}$. We therefore proceed by identifying those intrinsic properties of $\mathcal{M}$ that will play a role in our analysis and results.

## 2.2 Key Geometric Properties

In the NTK regime described in Section 2.1, gradient descent makes rapid progress if there exists a small certificate $g$ satisfying $\boldsymbol{\Theta}_\mu^{\mathrm{NTK}}[g] \approx \zeta_0$. The NTK is a function of the network width $n$ and depth $L$—in particular, we will see that the depth $L$ serves as a fitting resource, enabling the network to accommodate more complicated geometries. Our main analytical task is to establish relationships between these architectural resources and the intrinsic geometric properties of the manifolds that guarantee existence of a certificate.

---

[2]The specific value $\pi/2$ is immaterial to our arguments: this constraint is only to avoid technical issues that arise when antipodal points are present in $\mathcal{M}$, so any constant less than $\pi$ would work just as well. This choice allows for some extra technical expediency, and connects with natural modeling assumptions (e.g. data corresponding to image manifolds, with nonnegative pixel intensities).

Intuitively, one would expect it to be harder to separate curves that are close together or oscillate wildly. In this section, we formalize these intuitions in terms of the curves' *curvature*, and quantities which we term the *angle injectivity radius* and ⌘-*number*, which control the separation between the curves and their tendency to self-intersect. Given that the curves are regular, we may parameterize the two curves at unit speed with respect to arc length: for $\sigma \in \{\pm\}$, we write $\text{len}(\mathcal{M}_\sigma)$ to denote the length of each curve, and use $\boldsymbol{x}_\sigma(s) : [0, \text{len}(\mathcal{M}_\sigma)] \to \mathbb{S}^{n_0 - 1}$ to represent these parameterizations. We let $\boldsymbol{x}_\sigma^{(i)}(s)$ denote the $i$-th derivative of $\boldsymbol{x}_\sigma$ with respect to arc length. Because our parameterization is unit speed, $\|\boldsymbol{x}_\sigma^{(1)}(s)\|_2 = 1$ for all $\boldsymbol{x}_\sigma(s) \in \mathcal{M}$. We provide full details regarding this parameterization in Appendix C.2.

**Curvature and Manifold Derivatives.** Our curves $\mathcal{M}_\sigma$ are submanifolds of the sphere $\mathbb{S}^{n_0-1}$. The curvature of $\mathcal{M}_\sigma$ at a point $\boldsymbol{x}_\sigma(s)$ is the norm $\|\boldsymbol{P}_{\boldsymbol{x}_\sigma(s)^\perp} \boldsymbol{x}_\sigma^{(2)}(s)\|_2$ of the component $\boldsymbol{P}_{\boldsymbol{x}_\sigma(s)^\perp} \boldsymbol{x}_\sigma^{(2)}(s)$ of the second derivative of $\boldsymbol{x}_\sigma(s)$ that lies tangent to the sphere $\mathbb{S}^{n_0-1}$ at $\boldsymbol{x}_\sigma(s)$. Geometrically, this measures the extent to which the curve $\boldsymbol{x}_\sigma(s)$ deviates from a geodesic (great circle) on the sphere. Our technical results are phrased in terms of the *maximum curvature* $\kappa = \sup_{\sigma,s}\{\|\boldsymbol{P}_{\boldsymbol{x}_\sigma(s)^\perp} \boldsymbol{x}_\sigma^{(2)}(s)\|_2\}$. In stating results, we also use $\hat{\kappa} = \max\{\kappa, \frac{2}{\pi}\}$ to simplify various dependencies on $\kappa$. When $\kappa$ is large, $\mathcal{M}_\sigma$ is highly curved, and we will require a larger network depth $L$. In addition to the maximum curvature $\kappa$, our technical arguments require $\boldsymbol{x}_\sigma(s)$ to be five times continuously differentiable, and use bounds $M_i = \sup_{\sigma,s}\{\|\boldsymbol{x}_\sigma^{(i)}(s)\|_2\}$ on their higher order derivatives.

**Angle Injectivity Radius.** Another key geometric quantity that determines the hardness of the problem is the separation between manifolds: the problem is more difficult when $\mathcal{M}_+$ and $\mathcal{M}_-$ are close together. We measure closeness through the extrinsic distance (angle) $\angle(\boldsymbol{x}, \boldsymbol{x}') = \cos^{-1}\langle \boldsymbol{x}, \boldsymbol{x}' \rangle$ between $\boldsymbol{x}$ and $\boldsymbol{x}'$ over the sphere. In contrast, we use $d_\mathcal{M}(\boldsymbol{x}, \boldsymbol{x}')$ to denote the intrinsic distance between $\boldsymbol{x}$ and $\boldsymbol{x}'$ on $\mathcal{M}$, setting $d_\mathcal{M}(\boldsymbol{x}, \boldsymbol{x}') = \infty$ if $\boldsymbol{x}$ and $\boldsymbol{x}'$ reside on different components $\mathcal{M}_+$ and $\mathcal{M}_-$. We set

$$\Delta = \inf_{\boldsymbol{x}, \boldsymbol{x}' \in \mathcal{M}}\{\angle(\boldsymbol{x}, \boldsymbol{x}') \mid d_\mathcal{M}(\boldsymbol{x}, \boldsymbol{x}') \geq \tau_1\}, \tag{2.1}$$

where $\tau_1 = \frac{1}{\sqrt{20\hat{\kappa}}}$, and call this quantity the *angle injectivity radius*. In words, the angle injectivity radius is the minimum angle between two points whose intrinsic distance exceeds $\tau_1$. The angle injectivity radius $\Delta$ (i) lower bounds the distance between different components $\mathcal{M}_+$ and $\mathcal{M}_-$, *and* (ii) accounts for the possibility that a component will "loop back," exhibiting points with large intrinsic distance but small angle. This phenomenon is important to account for: the certificate problem is harder when one or both components of $\mathcal{M}$ nearly self-intersect. At an intuitive level, this increases the difficulty of the certificate problem because it introduces nonlocal correlations across the operator $\boldsymbol{\Theta}_\mu^{\text{NTK}}$, hurting its conditioning. As we will see in Section 4, increasing depth $L$ makes $\Theta^{\text{NTK}}$ better localized; setting $L$ sufficiently large relative to $\Delta^{-1}$ compensates for these correlations.

⌘-**number** The conditioning of $\boldsymbol{\Theta}_\mu^{\text{NTK}}$ depends not only on how near $\mathcal{M}$ comes to intersecting itself, which is captured by $\Delta$, but also on *the number of times* that $\mathcal{M}$ can "loop back" to a particular point. If $\mathcal{M}$ "loops back" many times, $\boldsymbol{\Theta}_\mu^{\text{NTK}}$ can be highly correlated, leading to a hard certificate problem. The ⌘-*number* (verbally, "clover number") reflects the number of near self-intersections:

$$⌘(\mathcal{M}) = \sup_{\boldsymbol{x} \in \mathcal{M}}\left\{ N_\mathcal{M}\left(\{\boldsymbol{x}' \mid d_\mathcal{M}(\boldsymbol{x}, \boldsymbol{x}') \geq \tau_1, \angle(\boldsymbol{x}, \boldsymbol{x}') \leq \tau_2\}, \frac{1}{\sqrt{1+\kappa^2}}\right)\right\} \tag{2.2}$$

with $\tau_2 = \frac{19}{20\sqrt{20\hat{\kappa}}}$. The set $\{\boldsymbol{x}' \mid d_\mathcal{M}(\boldsymbol{x}, \boldsymbol{x}') \geq \tau_1, \angle(\boldsymbol{x}, \boldsymbol{x}') \leq \tau_2\}$ is the union of looping pieces, namely points that are close to $\boldsymbol{x}$ in extrinsic distance but far in intrinsic distance. $N_\mathcal{M}(T, \delta)$ is the cardinality of a minimal $\delta$ covering of $T \subset \mathcal{M}$ in the intrinsic distance on the manifold, serving as a way to count the number of disjoint looping pieces. The ⌘-number accounts for the maximal volume of the curve where the angle injectivity radius $\Delta$ is active. It will generally be large if the manifolds nearly intersect multiple times, as illustrated in Fig. 2. The ⌘-number is typically small, but can be large when the data are generated in a way that induces certain near symmetries, as in the right panel of Fig. 2.

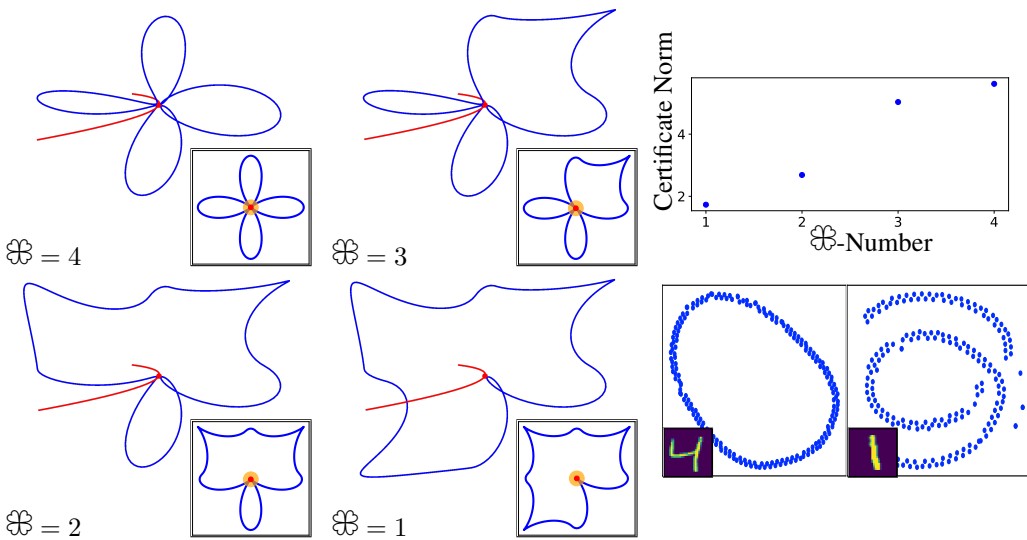

Figure 2: **The �֎-number—theory and practice.** *Left:* We generate a parametric family of space curves with fixed maximum curvature and length, but decreasing ✷-number, by reflecting 'petals' of a clover about a circumscribing square. We set $\mathcal{M}_+$ to be a fixed circle with large radius that crosses the center of the configurations, then rescale and project the entire geometry onto the sphere to create a two curve problem instance. In the insets, we show a two-dimensional projection of each of the blue $\mathcal{M}_-$ curves as well as a base point $\boldsymbol{x} \in \mathcal{M}_+$ at the center (also highlighted in the three-dimensional plots). The intersection of $\mathcal{M}_-$ with the neighborhood of $\boldsymbol{x}$ denoted in orange represents the set whose covering number gives the ✷-number of the configuration (see (2.2)). *Top right:* We numerically generate a certificate for each of the four geometries at left and plot its norm as a function of ✷-number. The trend demonstrates that increasing ✷-number correlates with increasing classification difficulty, measured through the certificate problem: this is in line with the intuition we have discussed. *Bottom right:* t-SNE projection of MNIST images (top: a "four" digit; bottom: a "one" digit) subject to rotations. Due to the approximate symmetry of the one digit under rotation by an angle $\pi$, the projection appears to nearly intersect itself. This may lead to a higher ✷-number compared to the embedding of the less-symmetric four digit. For experimental details for all panels, see Appendix A.

## 3 Main Results

Our main theorem establishes a set of sufficient resource requirements for the certificate problem under the class of geometries we consider here—by the reductions detailed in Section 2.1, this implies that gradient descent rapidly separates the two classes given a neural network of sufficient depth and width. First, we note a convenient aspect of the certificate problem, which is its amenability to approximate solutions: that is, if we have a kernel $\Theta$ that approximates $\Theta^{\mathrm{NTK}}$ in the sense that $\|\boldsymbol{\Theta}_\mu - \boldsymbol{\Theta}_\mu^{\mathrm{NTK}}\|_{L_\mu^2 \to L_\mu^2} \lesssim n/L$, and a function $\zeta$ such that $\|\zeta - \zeta_0\|_{L_\mu^2} \lesssim 1/L$, then by the triangle inequality and the Schwarz inequality, it suffices to solve the equation $\boldsymbol{\Theta}_\mu[g] \approx \zeta$ instead. In our arguments, we will exploit the fact that the random kernel $\Theta^{\mathrm{NTK}}$ concentrates well for wide networks with $n \gtrsim L$, choosing $\Theta$ as

$$\Theta(\boldsymbol{x}, \boldsymbol{x}') = (n/2) \sum_{\ell=0}^{L-1} \prod_{\ell'=\ell}^{L-1} \left(1 - (1/\pi)\varphi^{[\ell']}(\angle(\boldsymbol{x}, \boldsymbol{x}'))\right), \tag{3.1}$$

where $\varphi(t) = \cos^{-1}((1 - t/\pi)\cos t + (1/\pi)\sin t)$ and $\varphi^{[\ell']}$ denotes $\ell'$-fold composition of $\varphi$; as well as the fact that for wide networks with $n \gtrsim L^5$, depth 'smooths out' the initial error $\zeta_0$, choosing $\zeta$ as the piecewise-constant function $\zeta(\boldsymbol{x}) = -f_\star(\boldsymbol{x}) + \int_{\mathcal{M}} f_{\boldsymbol{\theta}_0}(\boldsymbol{x}') \, \mathrm{d}\mu(\boldsymbol{x}')$. We reproduce

high-probability concentration guarantees from the literature that justify these approximations in Appendix G.

**Theorem 3.1** (Approximate Certificates for Curves). *Let $\mathcal{M}$ be two disjoint smooth, regular, simple closed curves, satisfying $\angle(\boldsymbol{x}, \boldsymbol{x}') \leq \pi/2$ for all $\boldsymbol{x}, \boldsymbol{x}' \in \mathcal{M}$. There exist absolute constants $C, C', C'', C'''$ and a polynomial $P = \mathrm{poly}(M_3, M_4, M_5, \mathrm{len}(\mathcal{M}), \Delta^{-1})$ of degree at most 36, with degree at most 12 in $(M_3, M_4, M_5, \mathrm{len}(\mathcal{M}))$ and degree at most 24 in $\Delta^{-1}$, such that when*

$$L \geq \max\left\{ \exp(C' \operatorname{len}(\mathcal{M}) \hat{\kappa}), \left(\Delta\sqrt{1+\kappa^2}\right)^{-C'' \maltese(\mathcal{M})}, C''' \hat{\kappa}^{10}, P, \rho_{\max}^{12} \right\},$$

*there exists a certificate $g$ with $\|g\|_{L_\mu^2} \leq \frac{C\|\zeta\|_{L_\mu^2}}{\rho_{\min} n \log L}$ such that $\|\boldsymbol{\Theta}_\mu[g] - \zeta\|_{L_\mu^2} \leq \frac{\|\zeta\|_{L^\infty}}{L}$.*

Theorem 3.1 is our main technical contribution: it provides a sufficient condition on the network depth $L$ to resolve the approximate certificate problem for the class of geometries we consider, with the required resources depending only on the geometric properties we introduce in Section 2.2. Given the connection between certificates and gradient descent, Theorem 3.1 demonstrates that *deeper networks fit more complex geometries*, which shows that the network depth plays the role of a fitting resource in classifying the two curves. We provide a numerical corroboration of the interaction between the network depth, the geometry, and the size of the certificate in Figure 3. For any family of geometries with bounded $\maltese$-number, Theorem 3.1 implies a polynomial dependence of the depth on the angle injectivity radius $\Delta$, whereas we are unable to avoid an exponential dependence of the depth on the curvature $\kappa$. Nevertheless, these dependences may seem overly pessimistic in light of the existence of 'easy' two curve problem instances—say, linearly-separable classes, each of which is a highly nonlinear manifold—for which one would expect gradient descent to succeed without needing an unduly large depth. In fact, such geometries *will not* admit a small certificate norm in general unless the depth is sufficiently large: intuitively, this is a consequence of the operator $\boldsymbol{\Theta}_\mu$ being ill-conditioned for such geometries.[3]

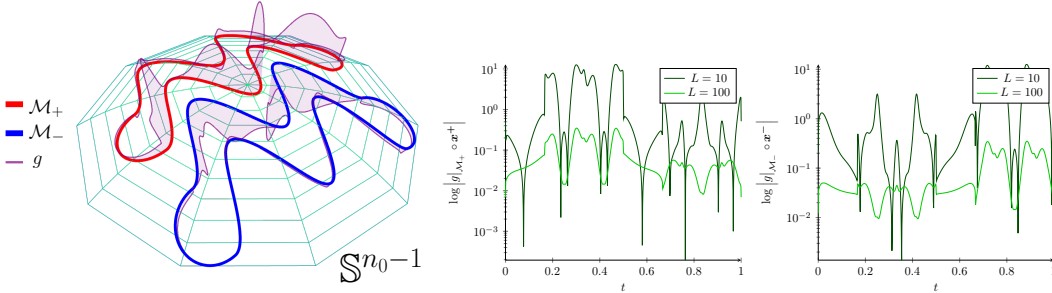

Figure 3: **The effect of geometry and depth on the certificate.** *Left:* The certificate $g$ computed numerically from the kernel $\Theta$ for depth $L = 50$ (defined in (3.1)) and the geometry from Figure 1 with a uniform density, graphed over the manifolds. Control of the norm of the certificate implies rapid progress of gradient descent, as reflected in Theorem 3.2. Comparing to Section 1, we note that the certificate has large magnitude near the point of minimum distance between the two curves—this is suggestive of the way the geometry sets the difficulty of the fitting problem. *Right:* To visualize the certificate norm more precisely, we graph the log-magnitude of the certificate for kernels $\Theta$ of varying depth $L$, viewing them through the arc-length parameterizations $\boldsymbol{x}_\sigma$ for the curves (left: $\mathcal{M}_+$; right: $\mathcal{M}_-$). At a coarse scale, the maximum magnitude decreases as the depth increases; at a finer scale, curvature-associated defects are 'smoothed out'. This indicates the role of depth as a fitting resource. See Appendix A for further experimental details.

The proof of Theorem 3.1 is novel, both in the context of kernel regression on manifolds and in the context of NTK-regime neural network training. We detail the key intuitions for the proof in

---

[3]Again, the equivalence between the difficulty of the certificate problem and the progress of gradient descent on decreasing the error is a consequence of our analysis proceeding in the kernel regime with the square loss—using alternate techniques to analyze the dynamics can allow one to prove that neural networks continue to fit such 'easy' classification problems efficiently (e.g. [34]).

Section 4. As suggested above, applying Theorem 3.1 to construct a certificate is straightforward: given a suitable setting of $L$ for a two curve problem instance, we obtain an approximate certificate $g$ via Theorem 3.1. Then with the triangle inequality and the Schwarz inequality, we can bound

$$\|\boldsymbol{\Theta}_\mu^{\mathrm{NTK}}[g] - \zeta_0\|_{L_\mu^2} \leq \|\boldsymbol{\Theta}_\mu^{\mathrm{NTK}} - \boldsymbol{\Theta}_\mu\|_{L_\mu^2 \to L_\mu^2}\|g\|_{L_\mu^2} + \|\zeta_0 - \zeta\|_{L_\mu^2} + \|\boldsymbol{\Theta}_\mu[g] - \zeta\|_{L_\mu^2},$$

and leveraging suitable probabilistic control (see Appendix G) of the approximation errors in the previous expression, as well as on $\|\zeta\|_{L_\mu^2}$, then yields bounds for the certificate problem. Applying the reductions from gradient descent dynamics in the NTK regime to certificates discussed in Section 2.1, we then obtain an end-to-end guarantee for the two curve problem.

**Theorem 3.2** (Generalization)**.** *Let $\mathcal{M}$ be two disjoint smooth, regular, simple closed curves, satisfying $\angle(\boldsymbol{x}, \boldsymbol{x}') \leq \pi/2$ for all $\boldsymbol{x}, \boldsymbol{x}' \in \mathcal{M}$. For any $0 < \delta \leq 1/e$, choose $L$ so that*

$$L \geq K \max\left\{ \frac{1}{\left(\Delta\sqrt{1+\kappa^2}\right)^{C\circledast(\mathcal{M})}}, C_\mu \log^9(\tfrac{1}{\delta}) \log^{24}(C_\mu n_0 \log(\tfrac{1}{\delta})), e^{C' \max\{\mathrm{len}(\mathcal{M})\hat{\kappa}, \log(\hat{\kappa})\}}, P \right\}$$

$$n = K' L^{99} \log^9(1/\delta) \log^{18}(Ln_0)$$

$$N \geq L^{10},$$

*and fix $\tau > 0$ such that $\frac{C''}{nL^2} \leq \tau \leq \frac{c}{nL}$. Then with probability at least $1 - \delta$, the parameters obtained at iteration $\lfloor L^{39/44}/(n\tau) \rfloor$ of gradient descent on the finite sample loss yield a classifier that separates the two manifolds.*

*The constants $c, C, C', C'', K, K' > 0$ are absolute, and $C_\mu$ equals to $\frac{\max\{\rho_{\min}^{19}, \rho_{\min}^{-19}\}(1+\rho_{\max})^{12}}{(\min\{\mu(\mathcal{M}_+), \mu(\mathcal{M}_-)\})^{11/2}}$ is a constant only depends on $\mu$. $P$ is a polynomial $\mathrm{poly}\{M_3, M_4, M_5, \mathrm{len}(\mathcal{M}), \Delta^{-1}\}$ of degree at most 36, with degree at most 12 when viewed as a polynomial in $M_3, M_4, M_5$ and $\mathrm{len}(\mathcal{M})$, and of degree at most 24 as a polynomial in $\Delta^{-1}$.*

Theorem 3.2 represents the first end-to-end guarantee for training a deep neural network to classify a nontrivial class of low-dimensional nonlinear manifolds. We call attention to the fact that the hypotheses of Theorem 3.2 are completely self-contained, making reference only to intrinsic properties of the data and the architectural hyperparameters of the neural network (as well as $\mathrm{poly}(\log n_0)$), and that the result is algorithmic, as it applies to training the network via constant-stepping gradient descent on the empirical square loss and guarantees generalization within $L^2$ iterations. Furthermore, Theorem 3.2 can be readily extended to the more general setting of regression on curves, given that we have focused on training with the square loss.

## 4 Proof Sketch

In this section, we provide an overview of the key elements of the proof of Theorem 3.1, where we show that the equation $\boldsymbol{\Theta}_\mu[g] \approx \zeta$ admits a solution $g$ (the certificate) of small norm. To solve the certificate problem for $\mathcal{M}$, we require a fine-grained understanding of the kernel $\Theta$. The most natural approach is to formally set $g = \sum_{i=1}^\infty \lambda_i^{-1}\langle\zeta, v_i\rangle_{L_\mu^2} v_i$ using the eigendecomposition of $\boldsymbol{\Theta}_\mu$ (just as constructed in Section 2.1 for $\boldsymbol{\Theta}_\mu^{\mathrm{NTK}}$), and then argue that this formal expression converges by studying the rate of decay of $\lambda_i$ and the alignment of $\zeta$ with eigenvectors of $\boldsymbol{\Theta}_\mu$; this is the standard approach in the literature [46, 53]. However, as discussed in Section 2.1, the nonlinear structure of $\mathcal{M}$ makes obtaining a full diagonalization for $\boldsymbol{\Theta}_\mu$ intractable, and simple asymptotic characterizations of its spectrum are insufficient to prove that the solution $g$ has small norm. Our approach will therefore be more direct: we will study the 'spatial' properties of the kernel $\Theta$ itself, in particular its rate of decay away from $\boldsymbol{x} = \boldsymbol{x}'$, and thereby use the network depth $L$ as a resource to reduce the study of the operator $\boldsymbol{\Theta}_\mu$ to a simpler, localized operator whose invertibility can be proved using harmonic analysis. We will then use differentiability properties of $\Theta$ to transfer the solution obtained by inverting this auxiliary operator back to the operator $\boldsymbol{\Theta}_\mu$. We refer readers to Appendix E for the full proof.

We simplify the proceedings using two basic reductions. First, with a small amount of auxiliary argumentation, we can reduce from the study of the operator-with-density $\boldsymbol{\Theta}_\mu$ to the density-free operator

$\Theta$. Second, the kernel $\Theta(\boldsymbol{x}, \boldsymbol{x}')$ is a function of the angle $\angle(\boldsymbol{x}, \boldsymbol{x}')$, and hence is rotationally invariant. This kernel is maximized at $\angle(\boldsymbol{x}, \boldsymbol{x}') = 0$ and decreases monotonically as the angle increases, reaching its minimum value at $\angle(\boldsymbol{x}, \boldsymbol{x}') = \pi$. If we subtract this minimum value, it should not affect our ability to fit functions, and we obtain a rotationally invariant kernel $\Theta^\circ(\boldsymbol{x}, \boldsymbol{x}') = \psi^\circ(\angle(\boldsymbol{x}, \boldsymbol{x}'))$ that is concentrated around angle 0. In the following, we focus on certificate construction for the kernel $\Theta^\circ$. Both simplifications are justified in Appendix E.3.

## 4.1 The Importance of Depth: Localization of the Neural Tangent Kernel

The first problem one encounters when attempting to directly establish (a property like) invertibility of the operator $\Theta^\circ$ is its action across connected components of $\mathcal{M}$: the operator $\Theta^\circ$ acts by integrating against functions defined on $\mathcal{M} = \mathcal{M}_+ \cup \mathcal{M}_-$, and although it is intuitive that most of its image's values on each component will be due to integration of the input over the same component, there will always be some 'cross-talk' corresponding to integration over the opposite component that interferes with our ability to apply harmonic analysis tools. To work around this basic issue (as well as others we will see below), our argument proceeds via a localization approach: we will exploit the fact that as the depth $L$ increases, the kernel $\Theta^\circ$ sharpens and concentrates around its value at $\boldsymbol{x} = \boldsymbol{x}'$, to the extent that we can neglect its action across components of $\mathcal{M}$ and even pass to the analysis of an auxiliary localized operator. This reduction is enabled by new sharp estimates for the decay of the angle function $\psi^\circ$ that we establish in Appendix F.3. Moreover, the perspective of using the network depth as a resource to localize the kernel $\Theta^\circ$ and exploiting this to solve the classification problem appears to be new: this localization is typically presented as a deficiency in the literature (e.g. [47]).

At a more formal level, when the network is deep enough compared to geometric properties of the curves, for each point $\boldsymbol{x}$, the majority of the mass of the kernel $\Theta^\circ(\boldsymbol{x}, \boldsymbol{x}')$ is taken within a small neighborhood $d_\mathcal{M}(\boldsymbol{x}, \boldsymbol{x}') \le r$ of $\boldsymbol{x}$. When $d_\mathcal{M}(\boldsymbol{x}, \boldsymbol{x}')$ is small relative to $\kappa$, we have $d_\mathcal{M}(\boldsymbol{x}, \boldsymbol{x}') \approx \angle(\boldsymbol{x}, \boldsymbol{x}')$. This allows us to approximate the local component by the following invariant operator:

$$\widehat{\boldsymbol{M}}[f](\boldsymbol{x}_\sigma(s)) = \int_{s'=s-r}^{s+r} \psi^\circ(|s - s'|) f(\boldsymbol{x}_\sigma(s')) ds'. \tag{4.1}$$

This approximation has two main benefits: (i) the operator $\widehat{\boldsymbol{M}}$ is defined by intrinsic distance $s' - s$, and (ii) it is highly localized. In fact, (4.1) takes the form of a convolution over the arc length parameter $s$. This implies that $\widehat{\boldsymbol{M}}$ diagonalizes in the Fourier basis, giving an explicit characterization of its eigenvalues and eigenvectors. Moreover, because $\widehat{\boldsymbol{M}}$ is localized, the eigenvalues corresponding to slowly oscillating Fourier basis functions are large, and $\widehat{\boldsymbol{M}}$ is stably invertible over such functions. Both of these benefits can be seen as consequences of depth: depth leads to localization, which facilitates approximation by $\widehat{\boldsymbol{M}}$, *and* renders that approximation invertible over low-frequency functions. In our proofs, we will work with a subspace $S$ spanned by low-frequency basis functions that are nearly constant over a length $2r$ interval (this subspace ends up having dimension proportional to $1/r$; see Appendix C.3 for a formal definition), and use Fourier arguments to prove invertibility of $\widehat{\boldsymbol{M}}$ over $S$ (see Lemma E.6).

## 4.2 Stable Inversion over Smooth Functions

Our remaining task is to leverage the invertibility of $\widehat{\boldsymbol{M}}$ over $S$ to argue that $\Theta$ is also invertible. In doing so, we need to account for the residual $\Theta - \widehat{\boldsymbol{M}}$. We accomplish this directly, using a Neumann series argument: when setting $r \lesssim L^{-1/2}$ and the dimension of the subspace $S$ proportional to $1/r$, the minimum eigenvalue of $\widehat{\boldsymbol{M}}$ over $S$ exceeds the norm of the residual operator $\Theta^\circ - \widehat{\boldsymbol{M}}$ (Lemma E.2). This argument leverages a decomposition of the domain into "near", "far" and "winding" pieces, whose contribution to $\Theta^\circ$ is controlled using the curvature, angle injectivity radius and ⌘-number (Lemma E.8, Lemma E.9, Lemma E.10). This guarantees the strict invertibility of $\Theta^\circ$ over the subspace $S$, and yields a unique solution $g_S$ to the *restricted* equation $\boldsymbol{P}_S \Theta^\circ[g_S] = \zeta$ (Theorem E.1).

This does not yet solve the certificate problem, which demands near solutions to the *unrestricted* equation $\Theta^\circ[g] = \zeta$. To complete the argument, we set $g = g_S$ and use harmonic analysis considerations to show that $\Theta^\circ[g]$ is very close to $S$. The subspace $S$ contains functions that do not oscillate

rapidly, and hence whose derivatives are small relative to their norm (Lemma E.23). We prove that $\boldsymbol{\Theta}^\circ[g]$ is close to $S$ by controlling the first three derivatives of $\boldsymbol{\Theta}^\circ[g]$, which introduces dependencies on $M_1, \cdots, M_5$ in the final statement of our results (Lemma E.27). In controlling these derivatives, we leverage the assumption that $\sup_{\boldsymbol{x}, \boldsymbol{x}' \in \mathcal{M}} \angle(\boldsymbol{x}, \boldsymbol{x}') \leq \pi/2$ to avoid issues that arise at antipodal points—we believe the removal of this constraint is purely technical, given our sharp characterization of the decay of $\psi^\circ$ and its derivatives. Finally, we move from $\boldsymbol{\Theta}^\circ$ back to $\boldsymbol{\Theta}$ by combining near solutions to $\boldsymbol{\Theta}^\circ[g] = \zeta$ and $\boldsymbol{\Theta}^\circ[g_1] = 1$, and iterating the construction to reduce the approximation error to an acceptable level (Appendix E.3).

## 5 Discussion

**A role for depth.** In the setting of fitting functions on the sphere $\mathbb{S}^{n_0-1}$ in the NTK regime with unstructured (e.g., uniformly random) data, it is well-known that there is very little marginal benefit to using a deeper network: for example, [32, 46, 59] show that the risk lower bound for RKHS methods is nearly met by kernel regression with a 2-layer network's NTK in an asymptotic ($n_0 \to \infty$) setting, and results for fitting degree-1 functions in the nonasymptotic setting [52] are suggestive of a similar phenomenon. In a similar vein, fitting in the NTK regime with a deeper network does not change the kernel's RKHS [41, 42, 45], and in a certain "infinite-depth" limit, the corresponding NTK for networks with ReLU activations, as we consider here, is a spike, guaranteeing that it fails to generalize [47, 50]. Our results are certainly not in contradiction to these facts—we consider a setting where the data are highly structured, and our proofs only show that an appropriate choice of the depth relative to this structure is *sufficient* to guarantee generalization, not necessary—but they nonetheless highlight an important role for the network depth in the NTK regime that has not been explored in the existing literature. In particular, the localization phenomenon exhibited by the deep NTK is completely inaccessible by fixed-depth networks, and simultaneously essential to our arguments to proving Theorem 3.2, as we have described in Section 4. It is an interesting open problem to determine whether there exist low-dimensional geometries that cannot be efficiently separated without a deep NTK, or whether the essential sufficiency of the depth-two NTK persists.

**Closing the gap to real networks and data.** Theorem 3.2 represents an initial step towards understanding the interaction between neural networks and data with low-dimensional structure, and identifying network resource requirements sufficient to guarantee generalization. There are several important avenues for future work. First, although the resource requirements in Theorem 3.1, and by extension Theorem 3.2, reflect only intrinsic properties of the data, the rates are far from optimal—improvements here will demand a more refined harmonic analysis argument beyond the localization approach we take in Section 4.1. A more fundamental advance would consist of extending the analysis to the setting of a model for image data, such as cartoon articulation manifolds, and the NTK of a convolutional neural network with architectural settings that impose translation invariance [25, 35]—recent results show asymptotic statistical efficiency guarantees with the NTK of a simple convolutional architecture, but only in the context of generic data [60]. The approach to certificate construction we develop in Theorem 3.1 will be of use in establishing guarantees analogous to Theorem 3.2 here, as our approach does not require an explicit diagonalization of the NTK.

In addition, extending our certificate construction approach to smooth manifolds of dimension larger than one is a natural next step. We believe our localization argument generalizes to this setting: as our bounds for the kernel $\psi$ are sharp with respect to depth and independent of the manifold dimension, one could seek to prove guarantees analogous to Theorem 3.1 with a similar subspace-restriction argument for sufficiently regular manifolds, such as manifolds diffeomorphic to spheres, where the geometric parameters of Section 2.2 have natural extensions. Such a generalization would incur at best an exponential dependence of the network on the manifold dimension for localization in high dimensions.

More broadly, the localization phenomena at the core of our argument appear to be relevant beyond the regime in which the hypotheses of Theorem 3.2 hold: we provide a preliminary numerical experiment to this end in Appendix A.3. Training fully-connected networks with gradient descent on a simple manifold classification task, low training error appears to be easily achievable only when the decay scale of the kernel is small relative to the inter-manifold distance even at moderate depth and width, and this decay scale is controlled by the depth of the network.

## Funding Transparency Statement and Acknowledgements

This work was supported by a Swartz fellowship (DG), by a fellowship award (SB) through the National Defense Science and Engineering Graduate (NDSEG) Fellowship Program, sponsored by the Air Force Research Laboratory (AFRL), the Office of Naval Research (ONR) and the Army Research Office (ARO), and by the National Science Foundation through grants NSF 1733857, NSF 1838061, NSF 1740833, and NSF 174039. We thank Alberto Bietti for bringing to our attention relevant prior art on kernel regression on manifolds.

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
