# Deep Networks Provably Classify Data on Curves Supplemental

**Tingran Wang**
Columbia University
tw2579@columbia.edu

**Sam Buchanan**
Columbia University
s.buchanan@columbia.edu

**Dar Gilboa**
Harvard University
dar_gilboa@fas.harvard.edu

**John Wright**
Columbia University
jw2966@columbia.edu

## Contents

35th Conference on Neural Information Processing Systems (NeurIPS 2021).

# A   Details of Figures

## A.1   Figure 2

✿-**number experiment.**    In each panel, the two curves are projection of curves $\boldsymbol{x}_+ : [0, 2\pi] \rightarrow \mathbb{S}^3$ and $\boldsymbol{x}_- : [0, 2\pi] \rightarrow \mathbb{S}^3$. We actually generate the curves as shown in the figure (i.e., in a three-dimensional space), then map them to the sphere using the map $(u, v, w) \mapsto (u, v, w, \sqrt{1 - u^2 - v^2 - w^2})$. In this three-dimensional space, the top left panel's blue curve (denoted $\boldsymbol{x}_-$ henceforth) and each panel's red curve (denoted $\boldsymbol{x}_+$ henceforth, and which is the same for all panels) are defined by the parametric equations

$$\begin{pmatrix} x_{-,1}(t) \\ x_{-,2}(t) \\ x_{-,3}(t) \end{pmatrix} = \begin{pmatrix} \cos(4t) \\ \cos\left(\frac{\pi}{8}\right)\cos(t)\left(\sin(4t) + 1 + \delta\right) + \sin\left(\frac{\pi}{8}\right)\sin(t)\left(\sin(4t) + 1 + \delta\right) \\ -\sin\left(\frac{\pi}{8}\right)\cos(t)\left(\sin(4t) + 1 + \delta\right) + \cos\left(\frac{\pi}{8}\right)\sin(t)\left(\sin(4t) + 1 + \delta\right) \end{pmatrix}$$
$$\begin{pmatrix} x_{+,1}(t) \\ x_{+,2}(t) \\ x_{+,3}(t) \end{pmatrix} = \begin{pmatrix} 4\sin(t) \\ 4\left(\cos(t) - 1\right) \\ 0 \end{pmatrix},$$

where $\delta$ sets the separation between the manifolds and is set here to $\delta = 0.05$. We then rescale both curves by a factor .01: the scale of the curves is chosen such that the curvature of the sphere has a negligible effect on the curvature of the manifolds (since the chart mapping we use here distorts the curves more nearer to the boundary of the unit disk $\{(u, v, w) \mid u^2 + v^2 + w^2 \leq 1\}$).[1]

From here, we use an "unfolding" process to obtain the blue curves in the other three panels from $\boldsymbol{x}_-$. To do this, points where $\left|\frac{\mathrm{d}x_{-,2}}{\mathrm{d}t}\right| = \left|\frac{\mathrm{d}x_{-,3}}{\mathrm{d}t}\right|$ are found numerically. There are 8 such points in total, and parts of the curve between pairs of these points are reflected across the line defined by such a pair in the $(x_2, x_3)$ plane. This can be done for any number of pairs between 1 and 4, generating the curves shown. This procedure ensures that aside from the set of 8 points, the curvature at every point along the curve is preserved and there is no discontinuity in the first derivative, while making the geometries loop back to the common center point more. For an additional visualization of the geometry, see Figure 1.[2]

Given these geometries, in order to compute the certificate norm for the experiment in the top-right panel, we evaluate the resulting curves at 200 points each, chosen by picking equally spaced points in $[0, 2\pi]$ and evaluating the parametric equations. The certificate itself is evaluated numerically as in Appendix A.2.

**Rotated MNIST digits.**    We rotate an MNIST image around its center by $i * \pi/100$ for integer $i$ between 0 and 199. We then apply t-SNE [2] using the scikit-learn package with perplexity 20 to generate the embeddings.

## A.2   Figure 3

We give full implementation details for this figure here, mixed with conceptual ideas that underlie the implementation. The manifolds $\mathcal{M}_+$ and $\mathcal{M}_-$ are defined by parametric equations $\boldsymbol{x}_+ : [0, 1] \rightarrow \mathbb{S}^2$ and $\boldsymbol{x}_- : [0, 1] \rightarrow \mathbb{S}^2$; it is not practical to obtain unit-speed parameterizations of general curves, so we also have parametric equations for their derivatives $\dot{\boldsymbol{x}}_\sigma : [0, 1] \rightarrow \mathbb{R}^2$. These are important in

---

[1]Although this adds a minor confounding effect to our experiments with certificate norm in the top-right panel, it is suppressed by setting the scale sufficiently small, and it can be removed in principle by using an isometric chart for the upper hemisphere instead of the map given above.

[2]For a three-dimensional interactive visualization, see https://colab.research.google.com/drive/1xmpYeLK6O6DtXOkJEt_apAniEB9fARRv?usp=sharing.

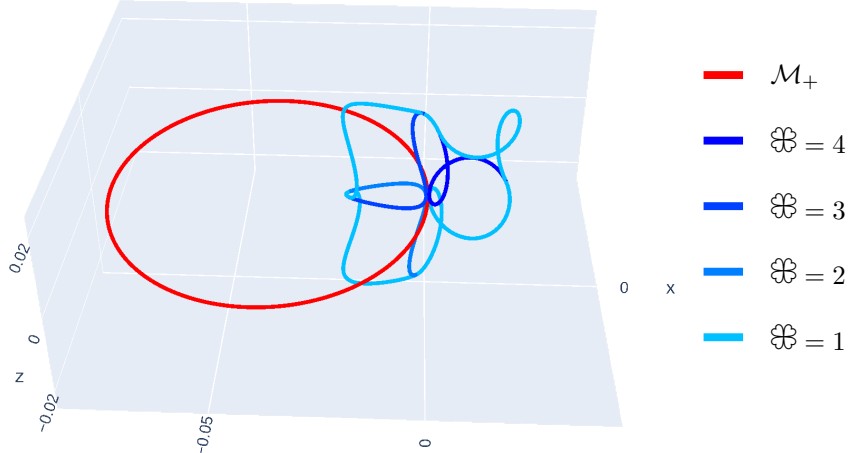

Figure 1: The two curve geometry described in Appendix A.1. The different choices of $\mathcal{M}_-$ that lead to different �-number are overlapping. The legend indicates the ✽-number of the two curves problem obtained by considering the same $\mathcal{M}_+$ but a different $\mathcal{M}_-$ as indicated by the color.

our setting since for non-unit-speed curves, the chain rule gives for the integral of a function (say) $f : \mathcal{M}_+ \to \mathbb{R}$

$$\int_{\mathcal{M}_+} f(\boldsymbol{x})\, \mathrm{d}\boldsymbol{x} = \int_{[0,1]} (f \circ \boldsymbol{x}_+(t)) \|\dot{\boldsymbol{x}}_+(t)\|_2\, \mathrm{d}t.$$

In particular, in our experiments, we want to work with a uniform density $\rho = (\rho_+, \rho_-)$ on the manifolds, where the classes are balanced. To achieve this, use the previous equation to get that we require

$$1 = \int_{\mathcal{M}_+} \rho_+(\boldsymbol{x})\, \mathrm{d}\boldsymbol{x} + \int_{\mathcal{M}_-} \rho_-(\boldsymbol{x})\, \mathrm{d}\boldsymbol{x}$$

$$= \int_{\mathcal{M}_+} (\rho_+ \circ \boldsymbol{x}_+)(t) \|\dot{\boldsymbol{x}}_+(t)\|_2\, \mathrm{d}t + \int_{\mathcal{M}_-} (\rho_- \circ \boldsymbol{x}_-)(t) \|\dot{\boldsymbol{x}}_-(t)\|_2\, \mathrm{d}t.$$

A uniform density on $\mathcal{M}$ is not a constant value—rather, it is characterized by being translation-invariant. It follows that $\rho_\sigma$ should be defined by

$$\rho_\sigma \circ \boldsymbol{x}_\sigma(t) = \frac{1}{2\|\dot{\boldsymbol{x}}_\sigma(t)\|_2}.$$

For the experiment, we solve a discretization of the certificate problem, for which the above ideas will be useful. Consider $\Theta$ in (3.1) for a fixed depth $L$ (and $n = 2$, since width is essentially irrelevant here). By the above discussion, the certificate problem in this setting is to solve for the certificate $g = (g_+, g_-)$

$$f_\star = \frac{1}{2} \left( \int_{[0,1]} \Theta(\,\cdot\,, \boldsymbol{x}_+(t)) g_+ \circ \boldsymbol{x}_+(t)\, \mathrm{d}t + \int_{[0,1]} \Theta(\,\cdot\,, \boldsymbol{x}_-(t)) g_- \circ \boldsymbol{x}_-(t)\, \mathrm{d}t \right).$$

Here, we have eliminated the initial random neural network output $f_{\boldsymbol{\theta}_0}$ from the RHS. Aside from making computation easier, this is motivated by fact that the network output is approximately piecewise constant for large depth $L$, and we therefore expect it not to play much of a role here. Let $M \in \mathbb{N}$ denote the discretization size. Then a finite-dimensional approximation of the previous integral equation is given by the linear system

$$f_\star \circ \boldsymbol{x}_\sigma(t_i) = \frac{1}{2M} \left( \sum_{j=1}^{M} \Theta(\boldsymbol{x}_\sigma(t_i), \boldsymbol{x}_+(t_j)) g_+ \circ \boldsymbol{x}_+(t_j) + \sum_{j=1}^{M} \Theta(\boldsymbol{x}_\sigma(t_i), \boldsymbol{x}_-(t_j)) g_- \circ \boldsymbol{x}_-(t_j) \right)$$

(A.1)

for all $i \in [M]$ and $\sigma \in \{\pm 1\}$, and where $t_i = (i-1)/M$. Of course, $f_\star \circ \boldsymbol{x}_\sigma(t) = \sigma$, so the equation simplifies further, and because the kernel $\Theta$ and this target $f_\star$ are smooth, there is a convergence

of the data in this linear system in a precise sense to the data in the original integral equation as $M \to \infty$. In particular, define a matrix $\boldsymbol{T}^+$ by $T_{ij}^+ = \Theta(\boldsymbol{x}_+(t_i), \boldsymbol{x}_+(t_j))$, define a matrix $\boldsymbol{T}^-$ by $T_{ij}^- = \Theta(\boldsymbol{x}_-(t_i), \boldsymbol{x}_-(t_j))$, and define a matrix $\boldsymbol{T}^\pm$ by $T_{ij}^\pm = \Theta(\boldsymbol{x}_+(t_i), \boldsymbol{x}_-(t_j))$, all of size $M \times M$. Then the $2M \times 2M$ linear system

$$\begin{bmatrix} \mathbf{1} \\ -\mathbf{1} \end{bmatrix} = \frac{1}{2M} \begin{bmatrix} \boldsymbol{T}^+ & \boldsymbol{T}^\pm \\ (\boldsymbol{T}^\pm)^* & \boldsymbol{T}^- \end{bmatrix} \begin{bmatrix} \boldsymbol{g}_+ \\ \boldsymbol{g}_- \end{bmatrix} \tag{A.2}$$

is equivalent to the discretization in (A.1). We implement and solve the system in (A.2) using the definitions we have given above, using the pseudoinverse of the $2M \times 2M$ matrix appearing in this expression to obtain $[\boldsymbol{g}_+, \boldsymbol{g}_-]^*$, and plot the results in Figure 3, in particular interpreting $(\boldsymbol{g}_\sigma)_i$ as the sampled point $g_\sigma \circ \boldsymbol{x}_\sigma(t_i)$ as in (A.1) when we plot in the left panel of Figure 3. Evidently, it would be immediate to modify the experiment to replace the LHS of (A.1) by the error $f_{\boldsymbol{\theta}_0} - f_\star$: the same protocol given above would work, but there would be an element of randomness added to the experiments.

Specifically, in Figure 3 we set $M = 900$. When plotting the solution to (A.2), i.e. the vector $[\boldsymbol{g}_+, \boldsymbol{g}_-]^*$, we moreover scale the vector by a factor of 0.3 to facilitate visualization.

## A.3 Kernel Decay Scale and Trainability of Realisting Networks: Empirical Evidence

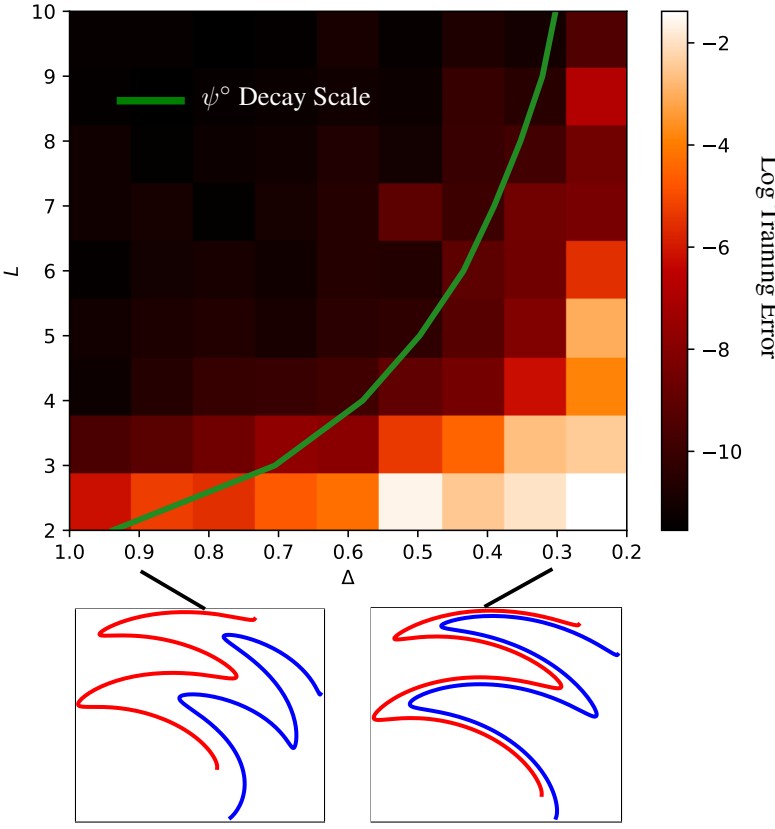

Figure 2: **The decay properties of the NTK are predictive of trainability on a toy dataset.** We plot the log training error of networks of varying depth that are trained to classify two curves with varying separation. The insets show a projection of the geometry onto the plane for separation values 0.3 and 0.9. For each depth $L$, the characteristic decay scale of the DC-subtracted NTK ($\psi^\circ$) is computed numerically and plotted in green. We find that small training loss is only achievable if the decay scale of the kernel is small compared to the inter-manifold distance, hence the decay scale is predictive of trainability.

One of the main insights into the manifold classification problem that is utilized to obtain Theorem 3.2 is that (roughly speaking) the depth of a fully-connected network controls the decay properties of the

network's NTK, and that fitting can be guaranteed once the decay occurs on a spatial scale that is small relative to certain geometric properties of the data. Here we provide empirical evidence that this phenomenon holds beyond the regime in which our main theorems hold, and in fact is relevant for networks of moderate width and depth as well.

We draw 400 samples each from a uniform distribution over a union of two curves that are related by a rotation by a geodesic angle that is varied from 0.2 to 1.0 in increments of 0.1. The curves are not linearly separable even for large angle (see insets in Fig. 2). These curves are embedded in $\mathbb{S}^{n_0 - 1}$ for $n_0 = 128$ and subjected to a rotation drawn uniformly from the Haar measure. We then train a fully-connected network to classify the curves using $\ell^2$ loss. The network has width $n = 256$ and we vary the depth from $L = 2$ to $L = 10$, and train using full-batch gradient descent for $10^5$ iterations with learning rate $\tau = 1/(4nL)$ (so that the total effective "training time" is independent of depth). We plot the log training error after training as a function of depth and the inter-manifold distance. For each depth $L$, we estimate an effective "decay scale" of the DC-subtracted skeleton $\psi^\circ$ by determining the point $s^\star$ such that $\psi^\circ(s^\star) = \frac{\psi^\circ(0)}{2}$.

The results are presented in Fig. 2. We observe that the network convergences to small training loss only when the depth is large comparable to the inverse of the manifold separation. As the depth represents the decay rate of the NTK, this indicates that a deeper network generates a localized NTK, allowing faster decay of the training error and making the classification problem easy. Notice that since the geometry of the dataset and network architecture do not satisfy all the assumptions of Theorem 3.2, the experiment provides evidence that the underlying phenomena regarding the role of the depth hold in greater generality. This preliminary result also suggests that the connection between the network architecture and the data geometry, as expressed through the decay properties of the NTK, can have a dramatic effect on the training process even for fully-connected networks.

# B   Notation

We use bold lowercase $\boldsymbol{x}$ for vectors and uppercase $\boldsymbol{A}$ for matrices and operators. We generally use non-bold notation to represent scalars and scalar-valued functions. $\mathbb{R}, \mathbb{C}, \mathbb{Z}$ are used for the real numbers, complex numbers and integers, respectively. $\mathbb{N}_0$ represents non-negative integers, and $\mathbb{N}$ represents the natural numbers. $\mathbb{R}^n$ represents $n$-dimensional Euclidean space, $\mathbb{C}^n$ represents the space of complex $n$-tuples (as a $n$-dimensional vector space over $\mathbb{C}$) and $\mathbb{S}^{n-1} \subset \mathbb{R}^n$ represents the $n - 1$ dimensional sphere centered at zero with unit radius. For a complex number $z = x + iy$ (or a complex-valued function), $|z| = \sqrt{x^2 + y^2}$ denotes the complex modulus, and $\bar{z} = x - iy$ denotes the complex conjugate. For $\boldsymbol{x}, \boldsymbol{y} \in \mathbb{C}^n$, we denote $\|\boldsymbol{x}\|_p = \left(\sum_{i=1}^n |x_i|^p\right)^{1/p}$ as the $p$-norm and $\langle \boldsymbol{x}, \boldsymbol{y} \rangle = \sum_{i=1}^n \bar{x}_i y_i$ as the standard (second-argument-linear) inner product. We use $\boldsymbol{x}^*$ and $\boldsymbol{A}^*$ to represent the conjugate transpose of vectors or matrices of complex numbers (so e.g. $\boldsymbol{x}^* \boldsymbol{y} = \langle \boldsymbol{x}, \boldsymbol{y} \rangle$). We use $\boldsymbol{P}_S$ to represent the orthogonal projection operator onto a closed subspace $S$ of a normed vector space (typically a Hilbert space).

For a Borel measure space $(X, \mu)$ and any measurable function $f : X \to \mathbb{C}$, we use $\|f\|_{L_\mu^p} = \left(\int_{\boldsymbol{x} \in X} |f(\boldsymbol{x})|^p d\mu(\boldsymbol{x})\right)^{1/p}$ to represent the $L^p$ norm of $f$ for $0 < p < \infty$. We omit the measure from the notation when it is clear from context. For $p = \infty$, we use $\|f\|_{L_\mu^\infty} = \inf\{C \geq 0 \mid |f(\boldsymbol{x})| \leq C$ for $\mu$-almost every $\boldsymbol{x}\}$ to represent its essential supremum. We denote the $L^p$ space of $(X, \mu)$ by $L_\mu^p(X)$ (or simply $L_\mu^p$ when the space is clear from context), which is formed by all complex-valued measurable functions with finite $L_\mu^p$ norm. For another space $(Y, \nu)$ and a (linear) operator $\boldsymbol{T} : L_\mu^p(X) \to L_\nu^q(Y)$, we represent its $L_\mu^p \to L_\nu^q$ operator norm as $\|\boldsymbol{T}\|_{L_\mu^p \to L_\nu^q} = \sup_{\|f\|_{L_\mu^p}=1} \|\boldsymbol{T}[f]\|_{L_\nu^q}$. When $X = Y$, $\mu = \nu$, and $p = q = 2$ (and $(X, \mu)$ is sufficiently regular), we have a Hilbert space; we write $\langle f, g \rangle_{L_\mu^2} = \int_X \bar{f}(\boldsymbol{x}) g(\boldsymbol{x}) \, \mathrm{d}\mu(\boldsymbol{x})$ for the inner product, and $\boldsymbol{T}^*$ to denote the associated adjoint of an operator $\boldsymbol{T}$ (so e.g. $f^* = \langle f, \cdot \rangle$ denotes the corresponding dual element of a function $f$). We use $\mathrm{Id} : L_\mu^p(X) \to L_\mu^p(X)$ to denote the identity operator, i.e. $\mathrm{Id}[f] = f$ for every $f \in L_\mu^p$. For $S \subset X$, we use $\mathbb{1}_S$ to represent the indicator function $\mathbb{1}_S(\boldsymbol{x}) = 1, \forall \boldsymbol{x} \in S$ and 0 otherwise; we will write $\mathbb{1}$ to denote $\mathbb{1}_X$. For a map $\varphi : X \to X$ and $i \in N$, we use $\varphi^{[i]}$ to denote its $i$-th fold iterated composition of itself, i.e. $\varphi^{[i]}(\boldsymbol{x}) = \varphi\left(\varphi^{(i-1)}(\boldsymbol{x})\right)$. For $i \in N$, $f^{(i)}$ is normally used to represent a function of a real variable $f$'s $i$-th order derivatives. For example, when

the space is a two curve problem instance $\mathcal{M}$, if $\boldsymbol{h} : \mathcal{M} \to \mathbb{C}^n$, we define its derivatives $\boldsymbol{h}^{(i)}$ in (C.5); for a kernel $\Theta : \mathcal{M} \times \mathcal{M} \to \mathbb{R}$, we define its derivatives along the curve in Definition E.11.

For a Borel measure space $(X, \mu)$, a kernel $K$ is a mapping $K : X \times X \to \mathbb{R}$. We use $\boldsymbol{K}$ for its associated Fredholm integral operator. In other words, for measurable function $f$ we have $\boldsymbol{K}_\mu[g](\boldsymbol{x}) = \int_{\boldsymbol{x}' \in X} K(\boldsymbol{x}, \boldsymbol{x}') f(\boldsymbol{x}') \, \mathrm{d}\mu(\boldsymbol{x}')$. When $X$ is a Riemannian manifold, an omitted subscript/measure will always denote the Riemannian measure.

We use both lowercase and uppercase letters $c, C$ for absolute constants whose value are independent of all parameters and $c_\tau, C_\tau$ for numbers whose value only depend on some parameter $\tau$. Throughout the text, $c$ is used to represent numbers whose value should be small while $C$ is for those whose value should be large. We use $C_1, C_2, \ldots$ for constants whose values are fixed within a proof while values of $C, C', C'', \ldots$ may change from line to line.

## C   Key Definitions

### C.1   Problem Formulation

The contents of this section will mirror Section 2.1, but provide additional technical details that were omitted there for the sake of concision and clarity of exposition. In this sense, we will focus on a rigorous formulation of the problem here, rather than on intuition: we encourage the reader to consult Section 2.1 for a more conceptually-oriented problem formulation. As in Section 2.1, we acknowledge that much of this material follows the technical exposition of [6].

Adopting the model proposed in [6], we let $\mathcal{M}_+, \mathcal{M}_-$, denote two class manifolds, each a smooth, regular, simple closed curve in $\mathbb{S}^{n_0 - 1}$, with ambient dimension $n_0 \geq 3$. We further assume $\mathcal{M}$ precludes antipodal points by asking

$$\angle(\boldsymbol{x}, \boldsymbol{x}') \leq \pi/2, \quad \forall \boldsymbol{x}, \boldsymbol{x}' \in \mathcal{M}. \tag{C.1}$$

We denote $\mathcal{M} = \mathcal{M}_+ \cup \mathcal{M}_-$, and the data measure supported on $\mathcal{M}$ as $\mu$. We assume that $\mu$ admits a density $\rho$ with respect to the Riemannian measure on $\mathcal{M}$, and that this density is bounded from below by some $\rho_{\min} > 0$. We will also write $\rho_{\max} = \sup_{\boldsymbol{x} \in \mathcal{M}} \rho(\boldsymbol{x})$. For background on curves and manifolds, we refer the reader to to [3, 4].

Given $N$ i.i.d. samples $(\boldsymbol{x}_1, \cdots, \boldsymbol{x}_N)$ from $\mu$ and their labels, given by the labeling function $f_\star : \mathcal{M} \to \{\pm 1\}$ defined by

$$f_\star(\boldsymbol{x}) = \begin{cases} +1 & \boldsymbol{x} \in \mathcal{M}_+ \\ -1 & \boldsymbol{x} \in \mathcal{M}_-, \end{cases}$$

we train a fully-connected network with ReLU activations and $L$ hidden layers of width $n$ and scalar output. We will write $\boldsymbol{\theta} = (\boldsymbol{W}^1, \ldots, \boldsymbol{W}^{L+1})$ to denote an abstract set of admissible parameters for such a network; concretely, the features at layer $\ell \in \{1, 2, \ldots, L\}$ with parameters $\boldsymbol{\theta}$ and input $\boldsymbol{x}$ are written as $\boldsymbol{\alpha}_{\boldsymbol{\theta}}^\ell(\boldsymbol{x}) = \left[ \boldsymbol{W}^\ell \boldsymbol{\alpha}_{\boldsymbol{\theta}}^{\ell-1}(\boldsymbol{x}) \right]_+$, where $[x]_+ = \max\{x, 0\}$ denotes the ReLU (and we adopt in general the convention of writing $[\boldsymbol{x}]_+$ to denote application of the scalar function $[\,\cdot\,]_+$ to each entry of the vector $\boldsymbol{x}$), with boundary condition $\boldsymbol{\alpha}_{\boldsymbol{\theta}}^0(\boldsymbol{x}) = \boldsymbol{x}$, and the network output on an input $\boldsymbol{x}$ is written $f_{\boldsymbol{\theta}}(\boldsymbol{x}) = \boldsymbol{W}^{L+1} \boldsymbol{\alpha}_{\boldsymbol{\theta}}^L(\boldsymbol{x})$. We will also write $\zeta_{\boldsymbol{\theta}}(\boldsymbol{x}) = f_{\boldsymbol{\theta}}(\boldsymbol{x}) - f_\star(\boldsymbol{x})$ to denote the fitting error. We use Gaussian initialization: if $\ell \in \{1, 2, \ldots, L\}$, the weights are initialized as $W_{ij}^\ell \sim_{\text{i.i.d.}} \mathcal{N}(0, \frac{2}{n})$, and the top level weights are initialized as $W_i^{L+1} \sim_{\text{i.i.d.}} \mathcal{N}(0, 1)$ in order to preserve the expected feature norm.[3] In the sequel, we will write $\boldsymbol{\theta}_0$ to denote the collection of these initial random parameters, and therefore $f_{\boldsymbol{\theta}_0}$ to denote the initial random network.

We will employ a convenient "empirical measure" notation to concisely represent finite-sample and population quantities in the analysis. Let $\mu^N = \frac{1}{N} \sum_{i=1}^N \delta_{\{\boldsymbol{x}_i\}}$ denote the empirical measure associated to our i.i.d. random sample from the population measure $\mu$, where $\delta_{\boldsymbol{p}}$ denotes a Dirac measure at a point $\boldsymbol{p}$. We train on the square loss $\mathcal{L}_{\mu^N}(\boldsymbol{\theta}) = (1/2) \int_{\mathcal{M}} (\zeta_{\boldsymbol{\theta}}(\boldsymbol{x}))^2 \, \mathrm{d}\mu^N(\boldsymbol{x})$ (of course

---

[3] This initialization style is common in practice (it might be referred to as "fan-out initialization" in that context), but less common in the theoretical literature on kernel regime training of deep neural networks, where a less-natural "NTK parameterization" is typically employed. A detailed discussion of these differences, and how to translate results for one parameterization into those for another, can be found (for example) in [6, §A.3].

one simply has $\mathcal{L}_{\mu^N}(\boldsymbol{\theta}) = 1/(2N) \sum_{i=1}^{N} (\zeta_{\boldsymbol{\theta}}(\boldsymbol{x}_i))^2)$, which we minimize using randomly-initialized "gradient descent" starting at $\boldsymbol{\theta}_0$ with constant step size $\tau > 0$. We put gradient descent in quotations here because the loss $\mathcal{L}_{\mu^N}$ is only almost-everywhere differentiable, due to the nondifferentiability of the ReLU activation $[\,\cdot\,]_+$: in this sense our algorithm for minimization is 'gradient-like', in that it corresponds to a gradient descent iteration at almost all values of the parameters. Concretely, we define

$$\boldsymbol{\beta}_{\boldsymbol{\theta}}^\ell(\boldsymbol{x}) = \left(\boldsymbol{W}^{L+1}\boldsymbol{P}_{I_L(\boldsymbol{x})}\boldsymbol{W}^L\boldsymbol{P}_{I_{L-1}(\boldsymbol{x})}\cdots\boldsymbol{W}^{\ell+2}\boldsymbol{P}_{I_{\ell+1}(\boldsymbol{x})}\right)^*$$

for $\ell = 0, 1, \ldots, L-1$, where

$$I_\ell(\boldsymbol{x}) = \left\{i \in [n] \,\middle|\, \left\langle \boldsymbol{e}_i, \boldsymbol{\alpha}_{\boldsymbol{\theta}}^\ell(\boldsymbol{x})\right\rangle > 0\right\}, \qquad \boldsymbol{P}_{I_\ell(\boldsymbol{x})} = \sum_{i \in I_\ell(\boldsymbol{x})} \boldsymbol{e}_i \boldsymbol{e}_i^*$$

denotes the orthogonal projection onto the set of coordinates where the $\ell$-th activation at input $\boldsymbol{x}$ is positive (above, $\boldsymbol{e}_i$ denotes the $i$-th canonical basis vector, having its $j$-th entry equal to 1 if $j = i$ and 0 otherwise). Then we define 'formal gradients' of the network output with respect to the parameters (denoted by an operator $\widetilde{\nabla}$) by

$$\widetilde{\nabla}_{\boldsymbol{W}^\ell} f_{\boldsymbol{\theta}}(\boldsymbol{x}) = \boldsymbol{\beta}_{\boldsymbol{\theta}}^{\ell-1}(\boldsymbol{x})\boldsymbol{\alpha}_{\boldsymbol{\theta}}^{\ell-1}(\boldsymbol{x})^*$$

for $\ell \in [L]$, and

$$\widetilde{\nabla}_{\boldsymbol{W}^{L+1}} f_{\boldsymbol{\theta}}(\boldsymbol{x}) = \boldsymbol{\alpha}_{\boldsymbol{\theta}}^L(\boldsymbol{x})^*.$$

As stated above, these expressions agree with the actual gradients at points of differentiability (to see this, apply the chain rule). We then define a formal gradient of $\mathcal{L}_{\mu^N}$ by

$$\widetilde{\nabla}\mathcal{L}_{\mu^N}(\boldsymbol{\theta}) = \int_{\mathcal{M}} \widetilde{\nabla} f_{\boldsymbol{\theta}}(\boldsymbol{x})\zeta_{\boldsymbol{\theta}}(\boldsymbol{x})\,\mathrm{d}\mu^N(\boldsymbol{x}).$$

Thus, our gradient-like algorithm we study here is given by the sequence of parameters $\boldsymbol{\theta}_{k+1} = \boldsymbol{\theta}_k - \tau\widetilde{\nabla}\mathcal{L}_{\mu^N}(\boldsymbol{\theta}_k)$, with $\boldsymbol{\theta}_0$ given by the Gaussian initialization we describe above.

Our study of this gradient-like iteration is facilitated by using kernel regime techniques, which we will describe now. Formally, the gradient descent iteration implies the following "error dynamics" equation:

$$\zeta_{\boldsymbol{\theta}_{k+1}^N}(\boldsymbol{x}) = \zeta_{\boldsymbol{\theta}_k^N}(\boldsymbol{x}) - \tau \int_{\mathcal{M}} \Theta_k^N(\boldsymbol{x}, \boldsymbol{x}')\zeta_{\boldsymbol{\theta}_k^N}(\boldsymbol{x}')\,\mathrm{d}\mu^N(\boldsymbol{x}'),$$

where $\Theta_k^N(\boldsymbol{x}, \boldsymbol{x}') = \int_0^1 \langle\widetilde{\nabla} f_{\boldsymbol{\theta}_k^N}(\boldsymbol{x}'), \widetilde{\nabla} f_{\boldsymbol{\theta}_k^N - t\tau\widetilde{\nabla}\mathcal{L}_{\mu^N}(\boldsymbol{\theta}_k^N)}(\boldsymbol{x})\rangle\,\mathrm{d}t$. For a proof of this claim, see [6, Lemma B.8]. As we describe in Section 2.1, under suitable conditions on the network width, depth, and the number of samples, this error dynamics update is well-approximated by a "nominal dynamics" update equation defined by $\zeta_{k+1} = \left(\mathrm{Id} - \tau\Theta_\mu^{\mathrm{NTK}}\right)[\zeta_k]$ with boundary condition $\zeta_0 = \zeta_{\boldsymbol{\theta}_0}$, where $\Theta^{\mathrm{NTK}}(\boldsymbol{x}, \boldsymbol{x}') = \langle\widetilde{\nabla} f_{\boldsymbol{\theta}_0}(\boldsymbol{x}), \widetilde{\nabla} f_{\boldsymbol{\theta}_0}(\boldsymbol{x}')\rangle$ is the "neural tangent kernel". The analysis of this nominal evolution leads us to the certificate problem that we have posed in Section 2.1, and which we resolve for the two curve problem in this work.

In the remainder of this section, we introduce several notations for quantities related to the certificate problem which we will refer to throughout these appendices. We let $\Theta$ denote the following approximation to the neural tangent kernel:

$$\Theta(\boldsymbol{x}, \boldsymbol{x}') = \frac{n}{2}\sum_{\ell=0}^{L-1}\prod_{\ell'=\ell}^{L-1}\left(1 - \frac{\varphi^{[\ell']}(\angle(\boldsymbol{x}, \boldsymbol{x}'))}{\pi}\right), \tag{C.2}$$

where $\varphi^{[\ell]}$ denotes the $\ell$-fold composition of the angle evolution function $\varphi(t) = \cos^{-1}\left((1 - \frac{t}{\pi})\cos t + \frac{\sin t}{\pi}\right)$. We let $\zeta$ denote the following piecewise constant approximation to $\zeta_0$:

$$\zeta(\boldsymbol{x}) = -f_\star(\boldsymbol{x}) + \int_{\mathcal{M}} f_{\boldsymbol{\theta}_0}(\boldsymbol{x}')\,\mathrm{d}\mu(\boldsymbol{x}'). \tag{C.3}$$

We also use the notation

$$\xi_\ell(t) = \prod_{\ell'=\ell}^{L-1}\left(1 - \frac{\varphi^{[\ell']}(t)}{\pi}\right)$$

$$\psi(t) = \frac{n}{2} \sum_{\ell=0}^{L-1} \xi_\ell(t)$$

for convenience. We find it convenient in our analysis to consider $\psi$ and its "DC component", i.e., its value at $\pi$, separately. To this end, we write $\psi^\circ = \psi - \psi(\pi)$. We also write the subtracted approximate NTK as $\Theta^\circ(\boldsymbol{x}, \boldsymbol{x}') = \psi^\circ(\angle(\boldsymbol{x}, \boldsymbol{x}'))$. As a consequence, we have

$$\psi^\circ(\angle(\boldsymbol{x}, \boldsymbol{x}')) = \Theta^\circ(\boldsymbol{x}, \boldsymbol{x}') = \Theta(\boldsymbol{x}, \boldsymbol{x}') - \psi(\pi). \tag{C.4}$$

We use $\boldsymbol{\Theta}_\mu$ to represent the integral operator with

$$\boldsymbol{\Theta}_\mu[g](x) = \int_{\mathcal{M}} \Theta(\boldsymbol{x}, \boldsymbol{x}') g(\boldsymbol{x}') \, \mathrm{d}\mu(\boldsymbol{x}'),$$

and similarly for $\boldsymbol{\Theta}_\mu^\circ$. An omitted subscript/measure will denote the Riemannian measure on $\mathcal{M}$.

## C.2 Geometric Properties

We assume our data manifold $\mathcal{M} = \mathcal{M}_+ \cup \mathcal{M}_-$, where $\mathcal{M}_+$ and $\mathcal{M}_-$ each is a smooth, regular, simple closed curve on the unit sphere $\mathbb{S}^{n_0-1}$. Because the curves are regular, it is without loss of generality to assume they are unit-speed and parameterized with respect to arc length $s$, giving parameterizations as maps from $[0, \mathrm{len}(\mathcal{M}_\sigma)]$ to $\mathbb{S}^{n_0-1}$, as we have defined them in Section 2.2 of the main body. Throughout the appendices, we will find it convenient to consider periodic extensions of these arc-length parameterizations, which are smooth and well-defined by the fact that our manifolds are smooth, closed curves: for $\sigma \in \{\pm\}$, we use $\boldsymbol{x}_\sigma(s) : \mathbb{R} \to \mathbb{S}^{n_0-1}$ to represent these parameterizations of the two manifolds.[4] We require that the two curves are disjoint. Notice that as the two curves do not self intersect, we have $\boldsymbol{x}_\sigma(s) = \boldsymbol{x}_{\sigma'}(s')$ if and only if $\sigma = \sigma'$ and $s' = s + k\,\mathrm{len}(\mathcal{M}_\sigma)$ for some $k \in \mathbb{Z}$. Precisely, our arguments will require our curves to have 'five orders' of smoothness, in other words $\boldsymbol{x}_\sigma(s)$ must be five times continuously differentiable for $\sigma \in \{+, -\}$.

For a differentiable function $\boldsymbol{h} : \mathcal{M} \to \mathbb{C}^p$ with $p \in \mathbb{N}$, we define its derivative $\frac{d}{ds}\boldsymbol{h}$ as

$$\frac{d}{ds}\boldsymbol{h}(\boldsymbol{x}) = \left[ \frac{d}{dt}\bigg|_s \boldsymbol{h}\big(\boldsymbol{x}_\sigma(t)\big) \right] \bigg|_{\boldsymbol{x}_\sigma(s) = \boldsymbol{x}}$$

$$= \left[ \lim_{t \to 0} \frac{1}{t}\big(\boldsymbol{h}(\boldsymbol{x}_\sigma(s+t)) - \boldsymbol{h}(\boldsymbol{x}_\sigma(s))\big) \right] \bigg|_{\boldsymbol{x}_\sigma(s) = \boldsymbol{x}}. \tag{C.5}$$

We call attention to the "restriction" bar used in this notation: it should be read as "let $s$ and $\sigma$ be such that $\boldsymbol{x}_\sigma(s) = \boldsymbol{x}$" in the definition's context. This leads to a valid definition in (C.5) because our curves are simple and disjoint, so for any choice $s, s'$ with $\boldsymbol{x}_\sigma(s) = \boldsymbol{x}_\sigma(s') = \boldsymbol{x}$, we have $\boldsymbol{x}_\sigma(s+t) = \boldsymbol{x}_\sigma(s'+t)$ for all $t$. We will use this notation systematically throughout these appendices. We further denote its $i$-th order derivative by $\boldsymbol{h}^{(i)}(\boldsymbol{x})$. For $i \in \mathbb{N}$, we use $\mathcal{C}^i(\mathcal{M})$ to represent the collection of real-valued functions $h : \mathcal{M} \to \mathbb{R}$ whose derivatives $h^{(1)}, \ldots, h^{(i)}$ exist and are continuous.

In particular, consider the inclusion map $\boldsymbol{\iota} : \mathcal{M} \to \mathbb{R}^{n_0}$, which is the identification $\boldsymbol{\iota}(\boldsymbol{x}) = \boldsymbol{x}$. Following the definition as above, we have

$$\boldsymbol{\iota}^{(i+1)}(\boldsymbol{x}) = \left[ \lim_{t \to 0} \frac{1}{t}\big(\boldsymbol{\iota}^{(i)}(\boldsymbol{x}_\sigma(s+t)) - \boldsymbol{\iota}^{(i)}(\boldsymbol{x}_\sigma(s))\big) \right] \bigg|_{\boldsymbol{x}_\sigma(s) = \boldsymbol{x}}. \tag{C.6}$$

In the sequel, with abuse of notation we will use $\boldsymbol{x}^{(i)}$ to represent $\boldsymbol{\iota}^{(i)}(\boldsymbol{x})$. For example, we will write expressions such as $\sup_{\boldsymbol{x} \in \mathcal{M}} \|\boldsymbol{x}^{(2)}\|_2$ to denote the quantity $\sup_{\boldsymbol{x} \in \mathcal{M}} \|\boldsymbol{\iota}^{(2)}(\boldsymbol{x})\|_2$. This notation will

---

[4]We clarify an abuse of notation we will commit with these parameterizations throughout the analysis, which stems from the fact that the curves are closed (i.e. topologically circles). That is, there is no preferred basepoint (i.e. the points $\boldsymbol{x}_\sigma(0)$) for the arc length parameterizations (the curves are only defined up to translation): because our primary use for these parameterizations is in the analysis of extrinsic distances between points on the curves, the basepoint will be irrelevant.

enable increased concision, and it is benign, in the sense that it is essentially an identification. We call attention to it specifically to note a possible conflict with our notation for the parameterizations and their derivatives $\boldsymbol{x}_\sigma^{(i)}$, which are maps from $\mathbb{R}$ to $\mathbb{R}^{n_0}$ (say), rather than maps defined on $\mathcal{M}$. In this context, we also use $\dot{\boldsymbol{x}}$ and $\ddot{\boldsymbol{x}}$ to represent first and second derivatives $\boldsymbol{x}^{(1)}$ and $\boldsymbol{x}^{(2)}$ for brevity. We have $\|\boldsymbol{x}\|_2 = \|\dot{\boldsymbol{x}}\|_2 = 1$ from the fact that $\mathcal{M} \subset \mathbb{S}^{n_0-1}$ and that we have a unit-speed parameterization. This and associated facts are collected in Lemma E.3.

For any real or complex-valued function $h$, the integral operator over manifold can be written as

$$\int_{\boldsymbol{x}\in\mathcal{M}} h(\boldsymbol{x})d\mu(\boldsymbol{x}) = \sum_{\sigma=\pm} \int_{s=0}^{\mathrm{len}(\mathcal{M}_\sigma)} h(\boldsymbol{x}_\sigma(s))\rho(\boldsymbol{x}_\sigma(s))ds,$$

$$\int_{\boldsymbol{x}\in\mathcal{M}} h(\boldsymbol{x})d\boldsymbol{x} = \sum_{\sigma=\pm} \int_{s=0}^{\mathrm{len}(\mathcal{M}_\sigma)} h(\boldsymbol{x}_\sigma(s))ds.$$

We have defined key geometric properties in the main body, in Section 2.2. Our arguments will require slightly more technical definitions of these quantities, however. In the remainder of this section, we introduce the same definition of angle injectivity radius and �֎-number with a variable scale, which helps us in proofs in Appendix E.

First, we give a precise definition for the intrinsic distance $d_\mathcal{M}$ on the curves. To separate the notions of "close over the sphere" and "close over the manifold", we use the extrinsic distance (angle) $\angle(\boldsymbol{x},\boldsymbol{x}') = \cos^{-1}\langle\boldsymbol{x},\boldsymbol{x}'\rangle$ to measures closeness between two points $\boldsymbol{x}$, $\boldsymbol{x}'$ over the sphere. The distance over the manifold is measured through the intrinsic distance $d_\mathcal{M}(\boldsymbol{x},\boldsymbol{x}')$, which takes $\infty$ when $\boldsymbol{x}$ and $\boldsymbol{x}'$ reside on different components $\mathcal{M}_+$ and $\mathcal{M}_-$ and the length of the shortest curve on the manifold connecting the two points when they belong to the same component. More formally, we have

$$d_\mathcal{M}(\boldsymbol{x},\boldsymbol{x}') = \begin{cases} \inf\{|s-s'| \ : \ \boldsymbol{x}_\sigma(s)=\boldsymbol{x}, \ \boldsymbol{x}_\sigma(s')=\boldsymbol{x}'\} & f_\star(\boldsymbol{x})=f_\star(\boldsymbol{x}'), \\ +\infty & \text{otherwise}, \end{cases} \tag{C.7}$$

where the infimum is taken over all valid $\sigma \in \{+,-\}$ and $(s,s') \in \mathbb{R}^2$. Notice that as the curves $\mathcal{M}_\sigma$ do not intersect themselves, one has $x_\sigma(s_1) = x_\sigma(s_2)$ if and only if $s_1 = s_2 + k\,\mathrm{len}(\mathcal{M}_\sigma)$ for some $k \in \mathbb{Z}$. Thus for any two points $\boldsymbol{x},\boldsymbol{x}'$ that belong to the same component $\mathcal{M}_\sigma$, the above infimum is attained: there exist $s,s'$ such that $\boldsymbol{x}_\sigma(s)=\boldsymbol{x}, \boldsymbol{x}_\sigma(s')=\boldsymbol{x}'$, and $d_\mathcal{M}(\boldsymbol{x},\boldsymbol{x}')=|s-s'|$.

**Angle Injectivity Radius**    For $\varepsilon \in (0,1)$ we define the angle injectivity radius of scale $\varepsilon$ as

$$\Delta_\varepsilon = \min\left\{\frac{\sqrt{\varepsilon}}{\hat{\kappa}}, \inf_{\boldsymbol{x},\boldsymbol{x}'\in\mathcal{M}} \left\{\angle(\boldsymbol{x},\boldsymbol{x}') \,\middle|\, d_\mathcal{M}(\boldsymbol{x},\boldsymbol{x}') \geq \frac{\sqrt{\varepsilon}}{\hat{\kappa}}\right\}\right\}, \tag{C.8}$$

which is the smallest extrinsic distance between two points whose intrinsic distance exceeds $\frac{\sqrt{\varepsilon}}{\hat{\kappa}}$ with

$$\hat{\kappa} = \max\left\{\kappa, \frac{2}{\pi}\right\}. \tag{C.9}$$

Observe that for any scale $\varepsilon$, $\Delta_\varepsilon$ is smaller than inter manifold separation $\min_{\boldsymbol{x}\in\mathcal{M}_+,\boldsymbol{x}'\in\mathcal{M}_-} \angle(\boldsymbol{x},\boldsymbol{x}')$.

**�֎-number**    For $\varepsilon \in (0,1), \delta \in (0,(1-\varepsilon)]$, we define ✖-number of scale $\varepsilon,\delta$ as

$$\varoplus_{\varepsilon,\delta}(\mathcal{M}) = \sup_{\boldsymbol{x}\in\mathcal{M}} N_\mathcal{M}\left(\left\{\boldsymbol{x}' \,\middle|\, d_\mathcal{M}(\boldsymbol{x},\boldsymbol{x}') \geq \frac{\sqrt{\varepsilon}}{\hat{\kappa}} \text{ and } \angle(\boldsymbol{x},\boldsymbol{x}') \leq \frac{\delta\sqrt{\varepsilon}}{\hat{\kappa}}\right\}, \frac{1}{\sqrt{1+\kappa^2}}\right). \tag{C.10}$$

Here, $N_\mathcal{M}(T,\varepsilon)$ is the size of a minimal $\varepsilon$ covering of $T$ in the intrinsic distance on the manifold. We call the set $\left\{\boldsymbol{x}' \,\middle|\, d_\mathcal{M}(\boldsymbol{x},\boldsymbol{x}') \geq \frac{\sqrt{\varepsilon}}{\hat{\kappa}}, \ \angle(\boldsymbol{x},\boldsymbol{x}') \leq \frac{\delta\sqrt{\varepsilon}}{\hat{\kappa}}\right\}$ appearing in this definition the winding piece of scale $\varepsilon$ and $\delta$: it contains points that are far away in intrinsic distance but close in extrinsic distance. We will give it a formal definition in (E.6), where it will play a key role in our arguments.

In the sequel, we denote $\Delta$, ✖$(\mathcal{M})$ to be the angle injectivity radius and ✖-number with the specific instantiations $\varepsilon = \frac{1}{20}$ and $\delta = 1-\varepsilon$. These are key geometric features used in Theorem 3.1 and Theorem 3.2.

## C.3 Subspace of Smooth Functions and Kernel Derivatives

As the behavior of the kernel and its approximation is easier to understand when constrained in a low frequency subspace, we first introduce the notion of low-frequency subspace formed by the Fourier basis on the two curves.

**Fourier Basis and Subspace of Smooth Functions**   We define a Fourier basis of functions over the manifold as

$$\phi_{\sigma,k}(\boldsymbol{x}_{\sigma'}(s)) = \begin{cases} \frac{1}{\sqrt{\mathrm{len}(\mathcal{M}_\sigma)}} \exp\left(\frac{\mathrm{i}2\pi ks}{\mathrm{len}(\mathcal{M}_\sigma)}\right), & \sigma' = \sigma \\ 0, & \sigma' \neq \sigma \end{cases} \tag{C.11}$$

for each $k = 0, 1, \ldots$, and further define a subspace of low frequency functions

$$S_{K_+,K_-} = \mathrm{span}_{\mathbb{C}}\{\phi_{+,0}, \phi_{+,-1}, \phi_{+,1}, \ldots, \phi_{+,-K_+}, \phi_{+,K_+}, \phi_{-,0}, \ldots, \phi_{-,K_-}\} \tag{C.12}$$

for $K_+, K_- \geq 0$. Using the fact that our curves are unit-speed, one can see that indeed (C.11) defines an orthonormal basis for $L^2$ functions on $\mathcal{M}$.

# D   Main Results

**Theorem D.1** (Generalization). *Let $\mathcal{M}$ be two disjoint smooth, regular, simple closed curves, satisfying $\angle(\boldsymbol{x}, \boldsymbol{x}') \leq \pi/2$ for all $\boldsymbol{x}, \boldsymbol{x}' \in \mathcal{M}$. For any $0 < \delta \leq 1/e$, choose $L$ so that*

$$L \geq K \max\left\{ \frac{1}{(\Delta(1+\kappa^2))^{C\circledast(\mathcal{M})}}, C_\mu \log^9(\tfrac{1}{\delta}) \log^{24}(C_\mu n_0 \log(\tfrac{1}{\delta})), e^{C' \max\{\mathrm{len}(\mathcal{M})\hat{\kappa}, \log(\hat{\kappa})\}}, P \right\}$$

$$n = K' L^{99} \log^9(1/\delta) \log^{18}(Ln_0)$$

$$N \geq L^{10},$$

*and fix $\tau > 0$ such that $\frac{C''}{nL^2} \leq \tau \leq \frac{c}{nL}$. Then with probability at least $1 - \delta$, the parameters obtained at iteration $\lfloor L^{39/44}/(n\tau) \rfloor$ of gradient descent on the finite sample loss yield a classifier that separates the two manifolds.*

*The constants $c, C, C', C'', K, K' > 0$ are absolute, and the constant $C_\mu$ is equal to $\frac{\max\{\rho_{\min}^{19}, \rho_{\min}^{-19}\}(1+\rho_{\max})^{12}}{(\min\{\mu(\mathcal{M}_+), \mu(\mathcal{M}_-)\})^{11/2}}$. $P$ is a polynomial $\mathrm{poly}\{M_3, M_4, M_5, \mathrm{len}(\mathcal{M}), \Delta^{-1}\}$ of degree at most 36, with degree at most 12 when viewed as a polynomial in $M_3, M_4, M_5$ and $\mathrm{len}(\mathcal{M})$, and of degree at most 24 as a polynomial in $\Delta^{-1}$.*

**Proof.**   The proof is an application of Theorem G.1; we note that the conditions on $n$, $L$, $\delta$, $N$, and $\tau$ imply all hypotheses of this theorem, except for the certificate condition. We will complete the proof by showing that the certificate condition is also satisfied, under the additional hypotheses on $L$ and with a suitable choice of $q_{\mathrm{cert}}$.

First, we navigate a difference in the formulation of the two curves' regularity properties between our work and [6], from which Theorem G.1 is drawn. Theorem G.1 includes a condition $L \geq C\kappa_{\mathrm{ext}}^2 C_\lambda$ for some absolute constant $C$, where $\kappa_{\mathrm{ext}}^2 = \sup_{\boldsymbol{x} \in \mathcal{M}} \|\ddot{\boldsymbol{x}}\|_2^2$ is a bound on the extrinsic curvature (we will discuss $C_\lambda$ momentarily). In our context, we have $M_2 = \kappa_{\mathrm{ext}}$, and following Lemma E.3 (using that our curves are unit-speed spherical curves), we get that it suffices to require $L \gtrsim (1 + \kappa^2)C_\lambda$ instead. In turn, we can pass to $\hat{\kappa}$: since this constant is lower-bounded by a positive number and is larger than $\kappa$, it suffices to require $L \gtrsim \hat{\kappa}^2 C_\lambda$. As for $C_\lambda$, this is a constant related to the angle injectivity radius $\Delta$, and is defined by $C_\lambda = K_\lambda^2/c_\lambda^2$, where these two constants satisfy

$$\forall s \in (0, c_\lambda/\kappa_{\mathrm{ext}}], (\boldsymbol{x}, \boldsymbol{x}') \in \mathcal{M}_\star \times \mathcal{M}_\star, \star \in \{+, -\} \quad : \quad \angle(\boldsymbol{x}, \boldsymbol{x}') \leq s \Rightarrow d_{\mathcal{M}}(\boldsymbol{x}, \boldsymbol{x}') \leq K_\lambda s.$$

We will relate this constant to constants in our formulation. Consider any $\boldsymbol{x}, \boldsymbol{x}' \in \mathcal{M}$. If $\angle(\boldsymbol{x}, \boldsymbol{x}') \leq \frac{\Delta}{2}$ then from the definition of $\Delta$ we have $d_{\mathcal{M}}(\boldsymbol{x}, \boldsymbol{x}') \leq \frac{\sqrt{\varepsilon}}{\hat{\kappa}}$ and hence by (E.31) we find $d_{\mathcal{M}}(\boldsymbol{x}, \boldsymbol{x}') \leq \angle(\boldsymbol{x}, \boldsymbol{x}')$. If on the other hand $\angle(\boldsymbol{x}, \boldsymbol{x}') > \frac{\Delta}{2}$, then a trivial bound gives

$$d_{\mathcal{M}}(\boldsymbol{x}, \boldsymbol{x}') \leq \mathrm{len}(\mathcal{M}) = \frac{2\mathrm{len}(\mathcal{M})}{\Delta} \frac{\Delta}{2} < \frac{2\mathrm{len}(\mathcal{M})}{\Delta} \angle(\boldsymbol{x}, \boldsymbol{x}'). \tag{D.1}$$

We can thus choose $c_\lambda = 1, K_\lambda = \max\left\{1, \frac{2\mathrm{len}(\mathcal{M})}{\Delta}\right\}$ to satisfy (D.1), giving $C_\lambda = \max\left\{1, \frac{4\mathrm{len}^2(\mathcal{M})}{\Delta^2}\right\}$. Thus the requirement $L > C\kappa_{\mathrm{ext}}^2 C_\lambda$ of Theorem G.1 is automatically satisfied if $L \gtrsim \max\{P, e^{C\,\mathrm{len}(\mathcal{M})\hat{\kappa}}\}$ for a suitable exponent $C$, where $P$ is the polynomial in the hypotheses of our result, and so our hypotheses imply this condition.

Next, we establish the certificate claim. The proof will follow closely the argument of [6, Proposition B.4]. Write $\Theta^{\mathrm{NTK}}$ for the network's neural tangent kernel, as defined in Appendix C.1, and $\Theta_\mu^{\mathrm{NTK}}$ for the associated Fredholm integral operator on $L_\mu^2$. In addition, write $\zeta_0 = f_{\boldsymbol{\theta}_0} - f_\star$ for the initial random network error. Because we have modified some exponents in the constant $C_\mu$, and added conditions on $L$, all hypotheses of Theorem D.2 are satisfied: invoking it, we have that there exists $g : \mathcal{M} \to \mathbb{R}$ satisfying

$$\|g\|_{L_\mu^2} \leq C\frac{\|\zeta\|_{L_\mu^2}}{\rho_{\min}n}$$

and

$$\|\boldsymbol{\Theta}_\mu[g] - \zeta\|_{L_\mu^2} \leq \frac{\|\zeta\|_{L^\infty}}{L}.$$

By these bounds, the triangle inequality, the Minkowski inequality, and the fact that $\mu$ is a probability measure, we have

$$\left\|\boldsymbol{\Theta}_\mu^{\mathrm{NTK}}[g] - \zeta_0\right\|_{L_\mu^2} \leq \|\Theta - \Theta^{\mathrm{NTK}}\|_{L^\infty(\mathcal{M}\times\mathcal{M})}\|g\|_{L_\mu^2} + \|\boldsymbol{\Theta}_\mu[g] - \zeta\|_{L_\mu^2} + \|\zeta - \zeta_0\|_{L_\mu^2}$$

$$\leq C\|\Theta - \Theta_{\mathrm{NTK}}\|_{L^\infty(\mathcal{M}\times\mathcal{M})}\frac{\|\zeta\|_{L^\infty(\mathcal{M})}}{n\rho_{\min}} + \frac{\|\zeta\|_{L^\infty}}{L} + \|\zeta - \zeta_0\|_{L^\infty(\mathcal{M})}.$$
(D.2)

An application of Theorem G.2 gives that on an event of probability at least $1 - e^{-cd}$

$$\|\Theta - \Theta^{\mathrm{NTK}}\|_{L^\infty(\mathcal{M}\times\mathcal{M})} \leq Cn/L$$

if $d \geq K\log(nn_0\,\mathrm{len}(\mathcal{M}))$ and $n \geq K'd^4L^5$. In translating this result from [6], we use that in the context of the two curve problem, the covering constant $C_\mathcal{M}$ appearing in [6, Theorem B.2] is bounded by a constant multiple of $\mathrm{len}(\mathcal{M})$ (this is how we obtain Theorem G.2 and some other results in Appendix G). An application of Lemma G.3 gives

$$\mathbb{P}\left[\|\zeta_0 - \zeta\|_{L^\infty(\mathcal{M})} \leq \frac{\sqrt{2d}}{L}\right] \geq 1 - e^{-cd}$$

and

$$\mathbb{P}\left[\|\zeta_0\|_{L^\infty(\mathcal{M})} \leq \sqrt{d}\right] \geq 1 - e^{-cd}$$

as long as $n \geq Kd^4L^5$ and $d \geq K'\log(nn_0\,\mathrm{len}(\mathcal{M}))$, where we use these conditions to simplify the residual that appears in Lemma G.3. In particular, combining the previous two bounds with the triangle inequality and a union bound and then rescaling $d$, which worsens the constant $c$ and the absolute constants in the preceding conditions, gives

$$\mathbb{P}\left[\|\zeta\|_{L^\infty(\mathcal{M})} \leq \sqrt{d}\right] \geq 1 - 2e^{-cd}.$$

Combining these bounds using a union bound and substituting into (D.2), we get that under the preceding conditions, on an event of probability at least $1 - 3e^{-cd}$ we have

$$\left\|\boldsymbol{\Theta}_\mu^{\mathrm{NTK}}[g] - \zeta_0\right\|_{L_\mu^2} \leq \frac{C\sqrt{d}}{L}\left(2 + \frac{1}{\rho_{\min}}\right)$$

$$\leq \frac{C\sqrt{d}}{L}\max\{\rho_{\min}, \rho_{\min}^{-1}\},$$
(D.3)

where we worst-case the density constant in the second line, and in addition, on the same event, we have by the norm bound on the certificate $g$

$$\|g\|_{L_\mu^2} \leq C\frac{\sqrt{d}}{n\rho_{\min}}.$$
(D.4)

To conclude, we simplify the preceding conditions on $n$ and turn the parameter $d$ into a parameter $\delta > 0$ in order to obtain the form of the result necessary to apply Theorem G.1. We have in this one-dimensional setting

$$\operatorname{len}(\mathcal{M}) \leq \frac{\operatorname{len}(\mathcal{M}_+)}{\mu(\mathcal{M}_+)} + \frac{\operatorname{len}(\mathcal{M}_-)}{\mu(\mathcal{M}_-)} \leq \frac{2}{\rho_{\min}} \leq 2\max\{\rho_{\min}, \rho_{\min}^{-1}\},$$

where the second inequality here uses simply

$$\mu(\mathcal{M}_+) = \int_{\mathcal{M}_+} \rho_+(\boldsymbol{x})\,\mathrm{d}\boldsymbol{x} \geq \operatorname{len}(\mathcal{M}_+)\rho_{\min}$$

(say). Because $n \geq 1$ and $n_0 \geq 3$ and $\max\{\rho_{\min}, \rho_{\min}^{-1}\} \geq 1$, it therefore suffices to instead enforce the condition on $d$ as $d \geq K \log(nn_0 C_\mu)$, where $C_\mu$ is the constant defined in the lemma statement. But note from our hypotheses here that we have $n \geq L$ and $L \geq C_\mu$; so in particular it suffices to enforce $d \geq K \log(nn_0)$ for an adjusted absolute constant. Choosing $d \geq (1/c)\log(1/\delta)$, we obtain that the previous two bounds (D.3) and (D.4) hold on an event of probability at least $1 - 3\delta$. When $\delta \leq 1/e$, given that $n_0 \geq 3$ we have $nn_0 \geq e$ and $\max\{\log(1/\delta), \log(nn_0)\} \leq \log(1/\delta)\log(nn_0)$, so that it suffices to enforce the requirement $d \geq K \log(1/\delta)\log(nn_0)$ for a certain absolute constant $K > 0$. We can then substitute this lower bound on $d$ into the two certificate bounds above to obtain the form claimed in (G.1) in Theorem G.1 with the instantiation $q_{\mathrm{cert}} = 1$, and this setting of $q_{\mathrm{cert}}$ matches the choice of $C_\mu$ that we have enforced in our hypotheses here. For the hypothesis on $n$, we substitute this lower bound on $d$ into the condition on $n$ to obtain the sufficient condition $n \geq K'L^5 \log^4(1/\delta)\log^4(nn_0)$. Using a standard log-factor reduction (e.g. [6, Lemma B.15]) and possibly worsening absolute constants, we then get that it suffices to enforce $n \geq K'L^5 \log^4(1/\delta)\log^4(Ln_0 \log(1/\delta))$, which is redundant with the (much larger) condition on $n$ that we have enforced here. This completes the proof. ∎

**Theorem D.2** (Certificates). *Let $\mathcal{M}$ be two disjoint smooth, regular, simple closed curves, satisfying $\angle(\boldsymbol{x}, \boldsymbol{x}') \leq \pi/2$ for all $\boldsymbol{x}, \boldsymbol{x}' \in \mathcal{M}$. There exist constants $C, C', C'', C'''$ and a polynomial $P = \operatorname{poly}(M_3, M_4, M_5, \operatorname{len}(\mathcal{M}), \Delta^{-1})$ of degree at most 36, with degree at most 12 in $(M_3, M_4, M_5, \operatorname{len}(\mathcal{M}))$ and degree at most 24 in $\Delta^{-1}$, such that when*

$$L \geq \max\left\{\exp(C'\operatorname{len}(\mathcal{M})\hat{\kappa}), \left(\frac{1}{\Delta\sqrt{1+\kappa^2}}\right)^{C''\text{⌘}(\mathcal{M})}, C'''\hat{\kappa}^{10}, P, \rho_{\max}^{12}\right\},$$

*then for $\zeta$ defined in (C.3), there exists a certificate $g : \mathcal{M} \to \mathbb{R}$ with*

$$\|g\|_{L_\mu^2} \leq \frac{C\|\zeta\|_{L_\mu^2}}{\rho_{\min} n \log L}$$

*such that*

$$\|\boldsymbol{\Theta}_\mu[g] - \zeta\|_{L_\mu^2} \leq \|\zeta\|_{L^\infty} L^{-1}.$$

**Proof.** This is a direct consequence of Lemma E.33. Notice that $\zeta$ in (C.3) is a real, piecewise constant function over the manifolds, and therefore has its higher order derivatives vanish. This makes it directly belong to $\Phi(\|\zeta\|_{L^2}, \frac{1}{20})$ defined in Definition E.24 and satisfy the condition in Lemma E.33 with $K = 1$. ∎

# E   Proof for the Certificate Problem

The goal of this section is to prove Lemma E.33, a generalized version of Theorem D.2. Instead of showing the certificate exists for the particular piecewise constant function $\zeta$ defined in (C.3), as claimed in Theorem D.2, Lemma E.33 claims that for *any* reasonably $\zeta$ with bounded higher order derivatives, there exists a small norm certificate $g$ such that $\boldsymbol{\Theta}_\mu[g] \approx \zeta$. There are two main technical difficulties in establishing this result. First, $\boldsymbol{\Theta}$ contains a very large constant term: $\boldsymbol{\Theta} = \boldsymbol{\Theta}^\circ + \psi(\pi)\mathbb{1}\mathbb{1}^*$. This renders the operator $\boldsymbol{\Theta}$ somewhat ill-conditioned. Second, the eigenvalues of $\boldsymbol{\Theta}^\circ$ are not bounded away from zero: because the kernel is sufficiently regular, it is possible to demonstrate high-frequency functions $h$ for which $\|\boldsymbol{\Theta}^\circ[h]\|_{L^2} \ll \|h\|_{L^2}$.

Our proof handles these technical challenges sequentially: in Appendix E.1, we restrict attention to the DC subtracted kernel $\Theta^\circ$ and a subspace $S$ containing low-frequency functions, and show that the restriction $\boldsymbol{P}_S\Theta^\circ\boldsymbol{P}_S$ to $S$ is stably invertible over $S$. In Appendix E.2, we argue that the solution $g$ to $\boldsymbol{P}_S\Theta^\circ[g] = \zeta$ is regularized enough that $\Theta^\circ[g] \approx \zeta$, i.e., the restriction to $S$ can be dropped. Finally, in Appendix E.3 we move from the DC subtracted kernel $\Theta^\circ$ without density to the full kernel $\Theta_\mu$. This move entails additional technical complexity; to maintain accuracy of approximation, we develop an iterative construction that successively applies the results of Appendix E.1–Appendix E.2 to whittle away approximation errors, yielding a complete proof of Lemma E.33.

## E.1 Invertibility Over a Subspace of Smooth Functions

**Proof Sketch and Organization.** In this section, we solve a restricted version of the certificate problem for DC subtracted kernel $\Theta^\circ$, over a subspace $S$ of low-frequency functions defined in (C.12). Namely, for $\zeta \in S$, we demonstrate the existence of a small norm solution $g \in S$ to the equation

$$\boldsymbol{P}_S\Theta^\circ[g] = \zeta. \tag{E.1}$$

This equation involves the integral operator $\Theta^\circ$, which acts via

$$\Theta^\circ[g](\boldsymbol{x}) = \int_{\boldsymbol{x}' \in \mathcal{M}} \Theta^\circ(\boldsymbol{x}, \boldsymbol{x}')g(\boldsymbol{x}')d\boldsymbol{x}'. \tag{E.2}$$

We argue that this operator is invertible over $S$, by decomposing this integral into four pieces, which we call the **Local**, **Near**, **Far**, and **Winding** components. The formal definitions of these four components follow: for parameters $0 < \varepsilon < 1$, $r > 0$, and $\delta > 0$, we define

$$\textbf{[Local]}: \qquad L_r(\boldsymbol{x}) = \left\{\boldsymbol{x}' \in \mathcal{M} \mid d_\mathcal{M}(\boldsymbol{x}, \boldsymbol{x}') < r\right\}, \tag{E.3}$$

$$\textbf{[Near]}: \qquad N_{r,\varepsilon}(\boldsymbol{x}) = \left\{\boldsymbol{x}' \in \mathcal{M} \,\middle|\, r \le d_\mathcal{M}(\boldsymbol{x}, \boldsymbol{x}') \le \frac{\sqrt{\varepsilon}}{\hat{\kappa}}\right\}, \tag{E.4}$$

$$\textbf{[Far]}: \qquad F_{\varepsilon,\delta}(\boldsymbol{x}) = \left\{\boldsymbol{x}' \in \mathcal{M} \,\middle|\, d_\mathcal{M}(\boldsymbol{x}, \boldsymbol{x}') \ge \frac{\sqrt{\varepsilon}}{\hat{\kappa}}, \ \angle(\boldsymbol{x}, \boldsymbol{x}') > \frac{\delta\sqrt{\varepsilon}}{\hat{\kappa}}\right\}, \tag{E.5}$$

$$\textbf{[Winding]}: \qquad W_{\varepsilon,\delta}(\boldsymbol{x}) = \left\{\boldsymbol{x}' \in \mathcal{M} \,\middle|\, d_\mathcal{M}(\boldsymbol{x}, \boldsymbol{x}') \ge \frac{\sqrt{\varepsilon}}{\hat{\kappa}}, \ \angle(\boldsymbol{x}, \boldsymbol{x}') \le \frac{\delta\sqrt{\varepsilon}}{\hat{\kappa}}\right\}. \tag{E.6}$$

It is easy to verify that for any choice of these parameters and any $\boldsymbol{x} \in \mathcal{M}$, these four pieces cover $\mathcal{M}$: i.e., $L_r(\boldsymbol{x}) \cup N_{r,\varepsilon}(\boldsymbol{x}) \cup F_{\varepsilon,\delta}(\boldsymbol{x}) \cup W_{\varepsilon,\delta}(\boldsymbol{x}) = \mathcal{M}$. Intuitively, the **Local** and **Near** pieces contain points that are close to $\boldsymbol{x}$, in the intrinsic distance on $\mathcal{M}$. The **Far** component contains points that are far from $\boldsymbol{x}$ in intrinsic distance, *and* far in the extrinsic distance (angle). The **Winding** component contains portions of $\mathcal{M}$ that are far in intrinsic distance, but close in extrinsic distance. Intuitively, this component captures parts of $\mathcal{M}$ that "loop back" into the vicinity of $\boldsymbol{x}$.

**Parameter choice.** The specific parameters $r, \varepsilon, \delta$ will be chosen with an eye towards the properties of both $\mathcal{M}$ and $\Theta^\circ$. The parameter $\varepsilon \in (0, \frac{3}{4})$ is a scale parameter, which controls $r = r_\varepsilon$ such that

1. $r$ is large enough to enable the local component $L_r(\boldsymbol{x})$ to dominate the kernel's behavior;

2. $r$ is not too large, so the kernel stays sharp and localized over the local component $L_r(\boldsymbol{x})$.

Specifically, we choose

$$a_\varepsilon = (1 - \varepsilon)^3(1 - \varepsilon/12), \tag{E.7}$$

$$r_\varepsilon = 6\pi L^{-\frac{a_\varepsilon}{a_\varepsilon + 1}}. \tag{E.8}$$

Notice that when $\varepsilon \searrow 0$, we have $r_\varepsilon \approx L^{-1/2}$. So with a smaller choice of $\varepsilon$ we may get a larger local component with the price of a larger constant dependence.

We further choose $\delta$ to ensure that the **Near** and **Far** components overlap. To see that this is possible, note that at the boundary of the **Near** component, $d_\mathcal{M}(\boldsymbol{x}, \boldsymbol{x}') = \sqrt{\varepsilon}/\hat{\kappa}$; from Lemma E.4, we have

$$\angle(\boldsymbol{x}, \boldsymbol{x}') \ge d_\mathcal{M}(\boldsymbol{x}, \boldsymbol{x}') - \hat{\kappa}^2 d_\mathcal{M}^3(\boldsymbol{x}, \boldsymbol{x}'), \tag{E.9}$$

so at this point $\angle(\boldsymbol{x}, \boldsymbol{x}') \ge (1 - \varepsilon)\sqrt{\varepsilon}/\hat{\kappa}$. Thus as long as $\delta < 1 - \varepsilon$, **Near** and **Far** overlap.

**Kernel as main and residual.** The kernel $\Theta^\circ(\boldsymbol{x}, \boldsymbol{x}')$ is a decreasing function of $\angle(\boldsymbol{x}, \boldsymbol{x}')$: $\Theta^\circ$ is largest over the **Local** component, smaller over the **Near** and **Winding** components, and smallest over the **Far** component. By choosing the scale parameter $r_\varepsilon$ as in (E.8), we define an operator $\boldsymbol{M}_\varepsilon$ which captures the contribution of the **Local** component to the kernel:

$$\boldsymbol{M}_\varepsilon[f](\boldsymbol{x}) = \int_{\boldsymbol{x}' \in L_{r_\varepsilon}(\boldsymbol{x})} \psi^\circ(\angle(\boldsymbol{x}, \boldsymbol{x}')) f(\boldsymbol{x}') d\boldsymbol{x}'. \tag{E.10}$$

Because $\angle(\boldsymbol{x}, \boldsymbol{x}')$ is small over $L_{r_\varepsilon}(\boldsymbol{x})$ when $r_\varepsilon$ is chosen to be small compared to inverse curvature $1/\hat{\kappa}$, on this component, $d_{\mathcal{M}}(\boldsymbol{x}, \boldsymbol{x}') \approx \angle(\boldsymbol{x}, \boldsymbol{x}')$ (which we formalize in Lemma E.4). We will use this property to argue that $\boldsymbol{M}_\varepsilon$ can be approximated by a self-adjoint convolution operator, defined as

$$\widehat{\boldsymbol{M}}_\varepsilon[f](\boldsymbol{x}) = \left. \int_{s'=s-r_\varepsilon}^{s+r_\varepsilon} \psi^\circ(|s-s'|) f(\boldsymbol{x}_\sigma(s')) ds' \right|_{\boldsymbol{x}_\sigma(s)=\boldsymbol{x}}. \tag{E.11}$$

The restriction is valid because for any choice of $\sigma$ and $s$ such that $\boldsymbol{x}_\sigma(s) = \boldsymbol{x}$, the RHS has the same value. On the other hand, given that we require $0 < \varepsilon < \frac{3}{4}$, (E.7) and (E.8) show that when $L$ is chosen larger than a certain absolute constant, we have $r_\epsilon \leq \pi$, assuring $|s'-s|$ falls in the domain of $\psi^\circ$, which makes this operator well-defined. We will always assume such a choice has been made in the sequel, and in particular include it as a hypothesis in our results.

Notice that $\widehat{\boldsymbol{M}}_\varepsilon$ is an *invariant operator*: it commutes with the natural translation action on $\mathcal{M}$. As a result, it diagonalizes in the Fourier basis defined in (C.11) (i.e., each of these functions is an eigenfunction of $\widehat{\boldsymbol{M}}_\varepsilon$). See Lemma E.6 and its proof for the precise formulation of these properties. This enables us to study its spectrum on the subspace of smooth functions defined in (C.12) at the specific scale $\varepsilon$, defined as

$$S_\varepsilon = S_{K_{\varepsilon,+}, K_{\varepsilon,-}} \tag{E.12}$$

with $K_{\varepsilon,\sigma} = \left\lfloor \frac{\varepsilon^{1/2} \mathrm{len}(\mathcal{M}_\sigma)}{2\pi r_\varepsilon} \right\rfloor$ for $\sigma \in \{+,-\}$.[5] In this way, we will establish that $\widehat{\boldsymbol{M}}_\varepsilon$ is stably invertible on $S_\varepsilon$.

In the remainder of the section, we show the diagonalizability and restricted invertibility of $\widehat{\boldsymbol{M}}_\varepsilon$ in Lemma E.6, and control the $L^2$ to $L^2$ operator norm of all four components of $\Theta^\circ$ in Lemma E.7, Lemma E.8, Lemma E.9 and Lemma E.10. Then we show $\Theta^\circ$ is stably invertible using these results by a Neumann series construction (Lemma E.2) and finally prove the main theorem for this section in Theorem E.1.

**Theorem E.1.** *For any $\varepsilon \in (0, \frac{3}{4})$, $\delta \in (0, 1-\varepsilon]$, there exist an absolute constant $C$ and constants $C_\varepsilon, C'_{\varepsilon,\delta}, C''_\varepsilon$ depending only on the subscripted parameters such that if*

$$L \geq \max\left\{ \exp\left(C'_{\varepsilon,\delta} \mathrm{len}(\mathcal{M})\hat{\kappa}\right), \left(1 + \frac{1}{\Delta_\varepsilon \sqrt{1+\kappa^2}}\right)^{C''_\varepsilon \mathfrak{B}_{\varepsilon,\delta}(\mathcal{M})}, \left(\varepsilon^{-1/2} 12\pi\hat{\kappa}\right)^{\frac{a_\varepsilon+1}{a_\varepsilon}}, C_\varepsilon \right\},$$

*where $a_\varepsilon$, $r_\varepsilon$ as in (E.7) and (E.8) and we set subspace $S_\varepsilon$ and the invariant operator $\widehat{\boldsymbol{M}}_\varepsilon$ as in (E.12) and (E.11), we have $\boldsymbol{P}_{S_\varepsilon} \widehat{\boldsymbol{M}}_\varepsilon \boldsymbol{P}_{S_\varepsilon}$ is invertible over $S_\varepsilon$, and*

$$\left\| \left(\boldsymbol{P}_{S_\varepsilon} \widehat{\boldsymbol{M}}_\varepsilon \boldsymbol{P}_{S_\varepsilon}\right)^{-1} \boldsymbol{P}_{S_\varepsilon} \left(\Theta^\circ - \widehat{\boldsymbol{M}}_\varepsilon\right) \boldsymbol{P}_{S_\varepsilon} \right\|_{L^2 \to L^2} \leq 1 - \varepsilon.$$

*Moreover, for any $\zeta \in S_\varepsilon$, the equation $\boldsymbol{P}_{S_\varepsilon} \Theta^\circ[g] = \zeta$ has a unique solution $g_\varepsilon[\zeta] \in S_\varepsilon$ given by the convergent Neumann series*

$$g_\varepsilon[\zeta] = \sum_{\ell=0}^{\infty} (-1)^\ell \left( \left(\boldsymbol{P}_{S_\varepsilon} \widehat{\boldsymbol{M}}_\varepsilon \boldsymbol{P}_{S_\varepsilon}\right)^{-1} \boldsymbol{P}_{S_\varepsilon} (\Theta^\circ - \widehat{\boldsymbol{M}}_\varepsilon) \boldsymbol{P}_{S_\varepsilon} \right)^\ell \left(\boldsymbol{P}_{S_\varepsilon} \widehat{\boldsymbol{M}}_\varepsilon \boldsymbol{P}_{S_\varepsilon}\right)^{-1} \zeta, \tag{E.13}$$

---

[5]Notice that although $\Theta$ and $\zeta$ are real objects, our subspace $S_\varepsilon$ contains complex-valued functions. In the remainder of Appendix E, we will work with complex objects for convenience, which means our constructed certificate candidates can be complex-valued. This will not affect our result because (intuitively) the fact that $\Theta$ and $\zeta$ are real makes the imaginary component of the certificate is redundant, and removing it with a projection onto the subspace of real-valued functions will give us the same norm and residual guarantees for the certificate problem. We make this claim rigorous and guarantee the existence of a real certificate in Lemma E.33, which is invoked in the proof of our main result on certificates, Theorem D.2.

*which satisfies*

$$\|g_\varepsilon[\zeta]\|_{L^2} \leq \frac{C\|\zeta\|_{L^2}}{\varepsilon n \log L}. \tag{E.14}$$

**Proof.** We construct $g \in S_\varepsilon$ satisfying $\boldsymbol{P}_{S_\varepsilon}\boldsymbol{\Theta}^\circ[g] = \zeta$ by equivalently writing

$$\boldsymbol{P}_{S_\varepsilon}\boldsymbol{\Theta}^\circ[g] = \left(\boldsymbol{P}_{S_\varepsilon}\widehat{\boldsymbol{M}}_\varepsilon\boldsymbol{P}_{S_\varepsilon} + \boldsymbol{P}_{S_\varepsilon}\left(\boldsymbol{\Theta}^\circ - \widehat{\boldsymbol{M}}_\varepsilon\right)\boldsymbol{P}_{S_\varepsilon}\right)[g].$$

Under our hypotheses, Lemma E.2 implies the invertibility of $\boldsymbol{P}_{S_\varepsilon}\widehat{\boldsymbol{M}}_\varepsilon\boldsymbol{P}_{S_\varepsilon}$ with

$$\lambda_{\min}\left(\boldsymbol{P}_{S_\varepsilon}\widehat{\boldsymbol{M}}_\varepsilon\boldsymbol{P}_{S_\varepsilon}\right) \geq \frac{1}{1-\varepsilon}\left\|\boldsymbol{\Theta}^\circ - \widehat{\boldsymbol{M}}_\varepsilon\right\|_{L^2 \to L^2}, \tag{E.15}$$

where $\lambda_{\min}\left(\boldsymbol{P}_{S_\varepsilon}\widehat{\boldsymbol{M}}_\varepsilon\boldsymbol{P}_{S_\varepsilon}\right)$ is the minimum eigenvalue of the self-adjoint operator $\boldsymbol{P}_{S_\varepsilon}\widehat{\boldsymbol{M}}_\varepsilon\boldsymbol{P}_{S_\varepsilon}$ : $S_\varepsilon \to S_\varepsilon$ as shown in Lemma E.6. In particular, $\boldsymbol{P}_{S_\varepsilon}\widehat{\boldsymbol{M}}_\varepsilon\boldsymbol{P}_{S_\varepsilon}$ is invertible, and the system we seek to solve can be written equivalently as

$$\left(\boldsymbol{P}_{S_\varepsilon}\widehat{\boldsymbol{M}}_\varepsilon\boldsymbol{P}_{S_\varepsilon}\right)^{-1}\zeta = \left(\mathrm{Id}_{S_\varepsilon} + \left(\boldsymbol{P}_{S_\varepsilon}\widehat{\boldsymbol{M}}_\varepsilon\boldsymbol{P}_{S_\varepsilon}\right)^{-1}\boldsymbol{P}_{S_\varepsilon}\left(\boldsymbol{\Theta}^\circ - \widehat{\boldsymbol{M}}_\varepsilon\right)\boldsymbol{P}_{S_\varepsilon}\right)[g],$$

where the LHS of the last system is in $S_\varepsilon$. Next, we argue that the operator that remains on the RHS of the last equation is invertible. Noting that

$$\left\|\left(\boldsymbol{P}_{S_\varepsilon}\widehat{\boldsymbol{M}}_\varepsilon\boldsymbol{P}_{S_\varepsilon}\right)^{-1}\boldsymbol{P}_{S_\varepsilon}\left(\boldsymbol{\Theta}^\circ - \widehat{\boldsymbol{M}}_\varepsilon\right)\boldsymbol{P}_{S_\varepsilon}\right\|_{L^2 \to L^2}$$

$$\leq \quad \left\|\left(\boldsymbol{P}_{S_\varepsilon}\widehat{\boldsymbol{M}}_\varepsilon\boldsymbol{P}_{S_\varepsilon}\right)^{-1}\right\|_{L^2 \to L^2}\left\|\boldsymbol{P}_{S_\varepsilon}\left(\boldsymbol{\Theta}^\circ - \widehat{\boldsymbol{M}}_\varepsilon\right)\boldsymbol{P}_{S_\varepsilon}\right\|_{L^2 \to L^2}$$

$$\leq \quad \lambda_{\min}\left(\boldsymbol{P}_{S_\varepsilon}\widehat{\boldsymbol{M}}_\varepsilon\boldsymbol{P}_{S_\varepsilon}\right)^{-1}\left\|\boldsymbol{\Theta}^\circ - \widehat{\boldsymbol{M}}_\varepsilon\right\|_{L^2 \to L^2}$$

$$\leq \quad 1 - \varepsilon \tag{E.16}$$

using both Lemma E.6 and (E.15), we have by the Neumann series that

$$\left(\mathrm{Id}_{S_\varepsilon} + \left(\boldsymbol{P}_{S_\varepsilon}\widehat{\boldsymbol{M}}_\varepsilon\boldsymbol{P}_{S_\varepsilon}\right)^{-1}\boldsymbol{P}_{S_\varepsilon}\left(\boldsymbol{\Theta}^\circ - \widehat{\boldsymbol{M}}_\varepsilon\right)\boldsymbol{P}_{S_\varepsilon}\right)^{-1}$$

$$= \sum_{i=0}^\infty (-1)^i \left(\left(\boldsymbol{P}_{S_\varepsilon}\widehat{\boldsymbol{M}}_\varepsilon\boldsymbol{P}_{S_\varepsilon}\right)^{-1}\boldsymbol{P}_{S_\varepsilon}\left(\boldsymbol{\Theta}^\circ - \widehat{\boldsymbol{M}}_\varepsilon\right)\boldsymbol{P}_{S_\varepsilon}\right)^i.$$

Thus we know $g_\varepsilon[\zeta]$ in (E.13) serves as the solution to the equation $\boldsymbol{P}_{S_\varepsilon}\boldsymbol{\Theta}^\circ[g] = \zeta$.

Furthermore, from Lemma E.6 when $L \geq C_\varepsilon$, we have

$$\left\|\left(\boldsymbol{P}_{S_\varepsilon}\widehat{\boldsymbol{M}}_\varepsilon\boldsymbol{P}_{S_\varepsilon}\right)^{-1}\zeta\right\|_{L^2} \leq \lambda_{\min}\left(\boldsymbol{P}_{S_\varepsilon}\widehat{\boldsymbol{M}}_\varepsilon\boldsymbol{P}_{S_\varepsilon}\right)^{-1}\|\zeta\|_{L^2}$$

$$\leq \frac{1}{cn \log L}\|\zeta\|_{L^2}.$$

Combining this bound with (E.16) and the triangle inequality in the series representation (E.13), we obtain the claimed norm bound in (E.14):

$$\|g_\varepsilon[\zeta]\|_{L^2} \leq \sum_{\ell=0}^\infty (1-\varepsilon)^\ell \left\|\left(\boldsymbol{P}_{S_\varepsilon}\widehat{\boldsymbol{M}}_\varepsilon\boldsymbol{P}_{S_\varepsilon}\right)^{-1}\zeta\right\|_{L^2}$$

$$\leq \frac{C\|\zeta\|_{L^2}}{\varepsilon n \log L}.$$

∎

**Lemma E.2.** *Let $\varepsilon \in (0, \frac{3}{4})$, $\delta \in (0, 1 - \varepsilon]$, and let $a_\varepsilon$, $\widehat{\boldsymbol{M}}_\varepsilon$ and $S_\varepsilon$ be as in (E.7), (E.11) and (E.12). There are constants $C_\varepsilon, C''_{\varepsilon,\delta}, C''_\varepsilon$ depending only on the subscripted parameters such that if*

$$
L \geq \max\left\{ \exp\Big(C'_{\varepsilon,\delta}\operatorname{len}(\mathcal{M})\hat{\kappa}\Big), \left(1 + \frac{1}{\Delta_\varepsilon\sqrt{1+\kappa^2}}\right)^{C''_\varepsilon \circledast_{\varepsilon,\delta}(\mathcal{M})}, \left(\varepsilon^{-1/2}12\pi\hat{\kappa}\right)^{\frac{a_\varepsilon+1}{a_\varepsilon}}, C_\varepsilon \right\},
$$

*we have $\boldsymbol{P}_{S_\varepsilon}\widehat{\boldsymbol{M}}_\varepsilon\boldsymbol{P}_{S_\varepsilon}$ is invertible over $S_\varepsilon$ with*

$$
\lambda_{\min}\left(\boldsymbol{P}_{S_\varepsilon}\widehat{\boldsymbol{M}}_\varepsilon\boldsymbol{P}_{S_\varepsilon}\right) \geq \frac{1}{1-\varepsilon}\left\|\boldsymbol{\Theta}^\circ - \widehat{\boldsymbol{M}}_\varepsilon\right\|_{L^2 \to L^2}
$$

*where $\lambda_{\min}\left(\boldsymbol{P}_{S_\varepsilon}\widehat{\boldsymbol{M}}_\varepsilon\boldsymbol{P}_{S_\varepsilon}\right)$ is defined in Lemma E.6.*

**Proof.** From triangle inequality for the $L^2 \to L^2$ operator norm, we have

$$
\left\|\boldsymbol{\Theta}^\circ - \widehat{\boldsymbol{M}}_\varepsilon\right\|_{L^2 \to L^2} \leq \|\boldsymbol{\Theta}^\circ - \boldsymbol{M}_\varepsilon\|_{L^2 \to L^2} + \left\|\boldsymbol{M}_\varepsilon - \widehat{\boldsymbol{M}}_\varepsilon\right\|_{L^2 \to L^2}.
$$

To bound the first term, we define

$$
M_\varepsilon(\boldsymbol{x}, \boldsymbol{x}') = \mathbb{1}_{d_{\mathcal{M}}(\boldsymbol{x}, \boldsymbol{x}') < r_\varepsilon}\psi^\circ(\angle(\boldsymbol{x}, \boldsymbol{x}')).
$$

Then it is a bounded symmetric kernel $\mathcal{M} \times \mathcal{M} \to \mathbb{R}$, and following (E.10), $\boldsymbol{M}_\varepsilon$ is its associated Fredholm integral operator. We can thus apply Lemma E.5 and get

$$
\|\boldsymbol{\Theta}^\circ - \boldsymbol{M}_\varepsilon\|_{L^2 \to L^2} \leq \sup_{\boldsymbol{x} \in \mathcal{M}}\int_{\boldsymbol{x}' \in \mathcal{M}}|\Theta^\circ(\boldsymbol{x}, \boldsymbol{x}') - M_\varepsilon(\boldsymbol{x}, \boldsymbol{x}')|d\boldsymbol{x}'
$$

$$
= \sup_{\boldsymbol{x} \in \mathcal{M}}\int_{\boldsymbol{x}' \in \mathcal{M} \setminus L_{r_\varepsilon}(\boldsymbol{x})}|\Theta^\circ(\boldsymbol{x}, \boldsymbol{x}')|d\boldsymbol{x}'.
$$

Because the **Near**, **Far** and **Winding** pieces cover $\mathcal{M} \setminus L_{r_\varepsilon}(\boldsymbol{x})$, we have

$$
\int_{\boldsymbol{x}' \in \mathcal{M} \setminus L_{r_\varepsilon}(\boldsymbol{x})}|\Theta^\circ(\boldsymbol{x}, \boldsymbol{x}')|d\boldsymbol{x}' \leq \int_{\boldsymbol{x}' \in N_{r_\varepsilon, \varepsilon}(\boldsymbol{x})}|\Theta^\circ(\boldsymbol{x}, \boldsymbol{x}')|d\boldsymbol{x}' + \int_{\boldsymbol{x}' \in W_{\varepsilon, \delta}(\boldsymbol{x})}|\Theta^\circ(\boldsymbol{x}, \boldsymbol{x}')|d\boldsymbol{x}'
$$

$$
+ \int_{\boldsymbol{x}' \in F_{\varepsilon, \delta}(\boldsymbol{x})}|\Theta^\circ(\boldsymbol{x}, \boldsymbol{x}')|d\boldsymbol{x}'
$$

From Lemma E.7, Lemma E.8, Lemma E.9 and Lemma E.10, we know that there exist constants $C_2, C_3, C_4$ and for any $\varepsilon'' \leq 1$ exist numbers $C_{\varepsilon''}, C'_{\varepsilon''}$ such that when $L \geq C_{\varepsilon''}$ and $L \geq \left(\varepsilon^{-1/2}12\pi\hat{\kappa}\right)^{\frac{a_\varepsilon+1}{a_\varepsilon}}$, we have

$$
\left\|\boldsymbol{\Theta}^\circ - \widehat{\boldsymbol{M}}_\varepsilon\right\|_{L^2 \to L^2} \leq \sup_{\boldsymbol{x} \in \mathcal{M}}\int_{\boldsymbol{x}' \in N_{r_\varepsilon, \varepsilon}(\boldsymbol{x})}|\Theta^\circ(\boldsymbol{x}, \boldsymbol{x}')|d\boldsymbol{x}' + \sup_{\boldsymbol{x} \in \mathcal{M}}\int_{\boldsymbol{x}' \in W_{\varepsilon, \delta}(\boldsymbol{x})}|\Theta^\circ(\boldsymbol{x}, \boldsymbol{x}')|d\boldsymbol{x}'
$$

$$
+ \sup_{\boldsymbol{x} \in \mathcal{M}}\int_{\boldsymbol{x}' \in F_{\varepsilon, \delta}(\boldsymbol{x})}|\Theta^\circ(\boldsymbol{x}, \boldsymbol{x}')|d\boldsymbol{x}' + \left\|\boldsymbol{M}_\varepsilon - \widehat{\boldsymbol{M}}_\varepsilon\right\|_{L^2 \to L^2}
$$

$$
\leq \frac{3\pi n}{4(1-\varepsilon)}(1 + \varepsilon'')\log\left(\frac{\sqrt{\varepsilon}}{\hat{\kappa}r_\varepsilon}\right) + C'_{\varepsilon''}n
$$

$$
+ C_2\operatorname{len}(\mathcal{M})\,n\frac{\hat{\kappa}}{\delta\sqrt{\varepsilon}}
$$

$$
+ C_3\circledast_{\varepsilon,\delta}(\mathcal{M})\,n\log\left(1 + \frac{\frac{1}{\sqrt{1+\kappa^2}}}{\Delta_\varepsilon}\right)
$$

$$
+ C_4\,(1-\varepsilon)^{-2}\hat{\kappa}^2nr_\varepsilon^2. \tag{E.17}
$$

Meanwhile, from Lemma E.6 there exists constant $C_\varepsilon, C_1$ such that when $L \geq C_\varepsilon$

$$
(1-\varepsilon)\lambda_{\min}\left(\boldsymbol{P}_{S_\varepsilon}\widehat{\boldsymbol{M}}_\varepsilon\boldsymbol{P}_{S_\varepsilon}\right) \geq (1-\varepsilon)^2\frac{3\pi n}{4}\log\left(1 + \frac{L-2}{3\pi}r_\varepsilon\right) - C_1(1-\varepsilon)nr_\varepsilon\log^2 L. \tag{E.18}
$$

We will treat all named constants appearing in the previous two equations as fixed for the remainder of the proof. We argue that the first term in this expression is large enough to dominate each of the terms in (E.17) and the residual term in (E.18).

Set $\varepsilon' = \frac{\varepsilon}{24}$. We will choose $\varepsilon'' = \frac{\varepsilon'}{1-2\varepsilon'} < 1$, so that both $\varepsilon'$ and $\varepsilon''$ depend only on $\varepsilon$. Then, since $r_\varepsilon = 6\pi L^{-\frac{a_\varepsilon}{a_\varepsilon+1}}$, when $L > 4$, we have

$$\frac{L-2}{3\pi}r_\varepsilon = 2(L-2)L^{-\frac{a_\varepsilon}{a_\varepsilon+1}} > L^{\frac{1}{a_\varepsilon+1}}. \tag{E.19}$$

Since moreover $a_\varepsilon = (1-\varepsilon)^3(1-\frac{\varepsilon}{12}) = (1-\varepsilon)^3(1-2\varepsilon')$, we have $a_\varepsilon\varepsilon'' = \varepsilon'(1-\varepsilon)^3$, and therefore $(1+\varepsilon'')a_\varepsilon = (1-\varepsilon')(1-\varepsilon)^3$. Thus

$$
\begin{aligned}
(1-\varepsilon')(1-\varepsilon)^2\frac{3\pi n}{4}\log\left(1+\frac{L-2}{3\pi}r_\varepsilon\right) &= \frac{3\pi n}{4(1-\varepsilon)}(1+\varepsilon'')\,a_\varepsilon\log\left(1+\frac{L-2}{3\pi}r_\varepsilon\right) \\
&\geq \frac{3\pi n}{4(1-\varepsilon)}(1+\varepsilon'')\log\left(L^{\frac{a_\varepsilon}{a_\varepsilon+1}}\right) \\
&\geq \frac{3\pi n}{4(1-\varepsilon)}(1+\varepsilon'')\log\left(\frac{\sqrt{\varepsilon}}{\hat\kappa r_\varepsilon}\right),
\end{aligned} \tag{E.20}
$$

where in the last bound we use $\sqrt{\varepsilon}\hat\kappa^{-1} \leq 6\pi$, given that $\varepsilon < 1$ and $\hat\kappa \leq \pi/2$. The RHS at the end of this chain of inequalities is the first term of the RHS of the last bound in (E.17). Since the LHS has a leading coefficient of $(1-\varepsilon')$, we can conclude provided we can split the remaining $\varepsilon'$ across the remaining terms.

Next, we will cover the negative term in (E.18) and the second and fifth terms in (E.17). Using (E.19), we have

$$\frac{\varepsilon'}{3}(1-\varepsilon)^2\frac{3\pi n}{4}\log\left(1+\frac{L-2}{3\pi}r_\varepsilon\right) \geq \frac{\varepsilon'}{3}(1-\varepsilon)^2\frac{3\pi n}{4}\frac{1}{a_\varepsilon+1}\log(L). \tag{E.21}$$

There exists a constant $C_\varepsilon$ such that when $L \geq C_\varepsilon$, we have for the RHS

$$\frac{\varepsilon'}{3}(1-\varepsilon)^2\frac{3\pi n}{4}\frac{1}{a_\varepsilon+1}\log(L) \geq (C_1 + C_4 + C'_{\varepsilon''})n.$$

In particular, we can take

$$C_\varepsilon \geq \exp\left(\frac{C_1 + C_4 + C'_{\varepsilon''}}{\frac{\varepsilon'}{3}(1-\varepsilon)^2\frac{3\pi}{4}\frac{1}{a_\varepsilon+1}}\right).$$

Next, there exists another constant $C_\varepsilon > 0$ such that when $L \geq C_\varepsilon$, we have $r_\varepsilon\log^2 L \leq 1$, whence by the previous bound

$$\frac{\varepsilon'}{3}(1-\varepsilon)^2\frac{3\pi n}{4}\frac{1}{a_\varepsilon+1}\log(L) \geq (C_1 r_\varepsilon\log^2 L + C_4 + C'_{\varepsilon''})n.$$

Finally, notice that when $L \geq \left(\varepsilon^{-1/2}12\pi\hat\kappa\right)^{\frac{a_\varepsilon+1}{a_\varepsilon}}$, we have

$$r_\varepsilon = 6\pi L^{-\frac{a_\varepsilon}{a_\varepsilon+1}} \leq \frac{\sqrt{\varepsilon}}{2\hat\kappa},$$

so $r_\varepsilon\hat\kappa \leq \sqrt{\varepsilon}/2$, and since $\varepsilon \in (0, 3/4)$, we have

$$(1-\varepsilon)C_1 nr_\varepsilon\log^2 L + C_4(1-\varepsilon)^{-2}\hat\kappa^2 nr_\varepsilon^2 + C'_{\varepsilon''}n \leq \left(C_1 r_\varepsilon\log^2 L + 3C_4 + C'_{\varepsilon''}\right)n,$$

where we used that $\varepsilon \mapsto \varepsilon(1-\varepsilon)^{-2}$ is increasing. Combining our previous bounds, this gives

$$\frac{\varepsilon'}{3}(1-\varepsilon)^2\frac{3\pi n}{4}\log\left(1+\frac{L-2}{3\pi}r_\varepsilon\right) \geq (1-\varepsilon)C_1 nr_\varepsilon\log^2 L + C_4(1-\varepsilon)^{-2}\hat\kappa^2 nr_\varepsilon^2 + C'_{\varepsilon''}n, \tag{E.22}$$

as desired.

For the remaining two terms, define

$$C'_{\varepsilon,\delta} = \frac{(a_\varepsilon+1)C_2}{(1-\varepsilon)^2\frac{\varepsilon'}{3}\frac{3\pi}{4}\delta\sqrt{\varepsilon}}, \qquad C''_\varepsilon = \frac{(a_\varepsilon+1)C_3}{(1-\varepsilon)^2\frac{\varepsilon'}{3}\frac{3\pi}{4}}.$$

We will use the estimate (E.21) as our base. Then when

$$L \geq \max\left\{ \exp\left(C'_{\varepsilon,\delta}\operatorname{len}(\mathcal{M})\hat{\kappa}\right), \left(1 + \frac{1}{\Delta_\varepsilon\sqrt{1+\kappa^2}}\right)^{C''_\varepsilon \mathfrak{B}_{\varepsilon,\delta}(\mathcal{M})} \right\},$$

we have

$$\frac{\varepsilon'}{3}(1-\varepsilon)^2 \frac{3\pi n}{4}\log\left(1 + \frac{L-2}{3\pi}r_\varepsilon\right) \geq C_2\operatorname{len}(\mathcal{M})\, n\frac{\hat{\kappa}}{\delta\sqrt{\varepsilon}}, \tag{E.23}$$

$$\frac{\varepsilon'}{3}(1-\varepsilon)^2 \frac{3\pi n}{4}\log\left(1 + \frac{L-2}{3\pi}r_\varepsilon\right) \geq C_3\,\mathfrak{B}_{\varepsilon,\delta}(\mathcal{M})\, n\log\left(1 + \frac{1}{\Delta_\varepsilon\sqrt{1+\kappa^2}}\right). \tag{E.24}$$

Combining (E.20), (E.22), (E.23), (E.24) completes the proof. ∎

**Lemma E.3.** *For any $\boldsymbol{x} \in \mathcal{M}$, we have*

$$\langle \boldsymbol{x}, \dot{\boldsymbol{x}}\rangle = \langle \dot{\boldsymbol{x}}, \ddot{\boldsymbol{x}}\rangle = \left\langle \boldsymbol{x}, \boldsymbol{x}^{(3)}\right\rangle = 0, \tag{E.25}$$

$$\langle \boldsymbol{x}, \ddot{\boldsymbol{x}}\rangle = -1, \tag{E.26}$$

$$\left\langle \boldsymbol{x}, \boldsymbol{x}^{(4)}\right\rangle = -\left\langle \dot{\boldsymbol{x}}, \boldsymbol{x}^{(3)}\right\rangle = \|\ddot{\boldsymbol{x}}\|_2^2,$$

$$\left\langle \ddot{\boldsymbol{x}}, \boldsymbol{x}^{(3)}\right\rangle = -\frac{1}{3}\left\langle \dot{\boldsymbol{x}}, \boldsymbol{x}^{(4)}\right\rangle,$$

$$\|\boldsymbol{P}_{\boldsymbol{x}^\perp}\ddot{\boldsymbol{x}}\|_2^2 = \|\ddot{\boldsymbol{x}}\|_2^2 - 1,$$

$$M_2 = \sqrt{1+\kappa^2} \leq M_4, \tag{E.27}$$

$$M_2 < 2\hat{\kappa}, \tag{E.28}$$

$$\frac{1}{\hat{\kappa}} \leq \min\{\operatorname{len}(\mathcal{M}_-), \operatorname{len}(\mathcal{M}_+)\}, \tag{E.29}$$

*where we use above the notation introduced near (C.6).*

**Proof.** As our curve is defined over sphere and has unit speed, we have

$$\|\boldsymbol{x}\|_2^2 = \|\dot{\boldsymbol{x}}\|_2^2 = 1.$$

Taking derivatives on both sides, we get

$$\langle \boldsymbol{x}, \dot{\boldsymbol{x}}\rangle = \langle \dot{\boldsymbol{x}}, \ddot{\boldsymbol{x}}\rangle = 0.$$

Continuing to take higher derivatives, we get the following relationships:

$$\|\dot{\boldsymbol{x}}\|_2^2 + \langle \boldsymbol{x}, \ddot{\boldsymbol{x}}\rangle = 0,$$

$$3\langle \dot{\boldsymbol{x}}, \ddot{\boldsymbol{x}}\rangle + \left\langle \boldsymbol{x}, \boldsymbol{x}^{(3)}\right\rangle = 0,$$

$$3\|\ddot{\boldsymbol{x}}\|_2^2 + 4\left\langle \dot{\boldsymbol{x}}, \boldsymbol{x}^{(3)}\right\rangle + \left\langle \boldsymbol{x}, \boldsymbol{x}^{(4)}\right\rangle = 0,$$

$$\|\ddot{\boldsymbol{x}}\|_2^2 + \left\langle \dot{\boldsymbol{x}}, \boldsymbol{x}^{(3)}\right\rangle = 0,$$

$$3\left\langle \ddot{\boldsymbol{x}}, \boldsymbol{x}^{(3)}\right\rangle + \left\langle \dot{\boldsymbol{x}}, \boldsymbol{x}^{(4)}\right\rangle = 0.$$

which gives us by plugging in the previous constraints

$$\langle \boldsymbol{x}, \ddot{\boldsymbol{x}}\rangle = -1,$$

$$\left\langle \boldsymbol{x}, \boldsymbol{x}^{(3)}\right\rangle = 0,$$

$$\left\langle \dot{\boldsymbol{x}}, \boldsymbol{x}^{(3)}\right\rangle = -\|\ddot{\boldsymbol{x}}\|_2^2,$$

$$\left\langle \boldsymbol{x}, \boldsymbol{x}^{(4)}\right\rangle = \|\ddot{\boldsymbol{x}}\|_2^2,$$

$$\left\langle \ddot{\boldsymbol{x}}, \boldsymbol{x}^{(3)}\right\rangle = -\frac{1}{3}\left\langle \dot{\boldsymbol{x}}, \boldsymbol{x}^{(4)}\right\rangle.$$

As a consequence, the intrinsic curvature $\|\boldsymbol{P}_{\boldsymbol{x}^\perp}\ddot{\boldsymbol{x}}\|_2$ and extrinsic curvature $\|\ddot{\boldsymbol{x}}\|_2$ are related by

$$\|\boldsymbol{P}_{\boldsymbol{x}^\perp}\ddot{\boldsymbol{x}}\|_2^2 = \left\|\left(\boldsymbol{I} - \boldsymbol{x}\boldsymbol{x}^*\right)\ddot{\boldsymbol{x}}\right\|_2^2$$
$$= \langle\boldsymbol{x}, \ddot{\boldsymbol{x}}\rangle^2 + \langle\ddot{\boldsymbol{x}}, \ddot{\boldsymbol{x}}\rangle - 2\langle\boldsymbol{x}, \ddot{\boldsymbol{x}}\rangle^2$$
$$= \|\ddot{\boldsymbol{x}}\|_2^2 - 1.$$

Thus we know

$$M_2 = \sup_{\boldsymbol{x}\in\mathcal{M}} \|\ddot{\boldsymbol{x}}\|_2$$
$$= \sup_{\boldsymbol{x}\in\mathcal{M}} \sqrt{1 + \|\boldsymbol{P}_{\boldsymbol{x}^\perp}\ddot{\boldsymbol{x}}\|_2^2}$$
$$= \sqrt{1 + \sup_{\boldsymbol{x}\in\mathcal{M}} \{\|\boldsymbol{P}_{\boldsymbol{x}^\perp}\ddot{\boldsymbol{x}}\|_2\}^2}$$
$$= \sqrt{1 + \kappa^2}$$
$$\leq \sqrt{\left(\frac{\pi}{2}\hat{\kappa}\right)^2 + \kappa^2}$$
$$< 2\hat{\kappa}.$$

Furthermore, the above shows that $M_2 \geq 1$, so we have

$$M_2 \leq M_2^2 = \sup_{\boldsymbol{x}\in\mathcal{M}} \|\ddot{\boldsymbol{x}}\|_2^2$$
$$= \sup_{\boldsymbol{x}\in\mathcal{M}} \left\langle\boldsymbol{x}, \boldsymbol{x}^{(4)}\right\rangle$$
$$\leq M_4,$$

using one of our previously-derived relationships in the second line and Cauchy-Schwarz in the third. Finally, for any point $\boldsymbol{x} = \boldsymbol{x}_\sigma(s)$, as $\boldsymbol{x}_\sigma(s + \text{len}(\mathcal{M}_\sigma)) = \boldsymbol{x}_\sigma(s)$, we have

$$\boldsymbol{0} = \boldsymbol{x}_\sigma(s + \text{len}(\mathcal{M}_\sigma)) - \boldsymbol{x}_\sigma(s) = \int_{s'=s}^{s+\text{len}(\mathcal{M}_\sigma)} \dot{\boldsymbol{x}}_\sigma(s')ds'$$
$$= \text{len}(\mathcal{M}_\sigma)\dot{\boldsymbol{x}}_\sigma(s) + \int_{s'=s}^{s+\text{len}(\mathcal{M}_\sigma)} \int_{s''=s}^{s'} \ddot{\boldsymbol{x}}_\sigma(s'')ds''ds'$$

which leads to

$$\text{len}(\mathcal{M}_\sigma) = \|\text{len}(\mathcal{M}_\sigma)\dot{\boldsymbol{x}}_\sigma(s)\|_2$$
$$= \left\|\int_{s'=s}^{s+\text{len}(\mathcal{M}_\sigma)} \int_{s''=s}^{s'} \ddot{\boldsymbol{x}}_\sigma(s'')ds''ds'\right\|_2$$
$$\leq \int_{s'=s}^{s+\text{len}(\mathcal{M}_\sigma)} \int_{s''=s}^{s'} M_2 ds''ds'$$
$$= \frac{\text{len}(\mathcal{M}_\sigma)^2}{2}M_2 < \text{len}(\mathcal{M}_\sigma)^2\hat{\kappa},$$

completing the proof, where the first line uses the unit-speed property, the second uses the previous relation, the third uses Jensen's inequality (given that $\|\cdot\|_2$ is convex and 1-homogeneous), and the last line comes from (E.28). $\blacksquare$

**Lemma E.4.** *Let* $\hat{\kappa} = \max\left\{\kappa, \frac{2}{\pi}\right\}$. *For* $\sigma \in \{\pm\}$ *and* $|s' - s| \leq \frac{1}{\hat{\kappa}}$, *we have*

$$|s - s'| - \hat{\kappa}^2|s - s'|^3 \leq \angle\left(\boldsymbol{x}_\sigma(s), \boldsymbol{x}_\sigma(s')\right) \leq |s - s'|. \tag{E.30}$$

*As a consequence, for* $|s - s'| \leq \frac{\sqrt{\varepsilon}}{\hat{\kappa}}$,

$$(1 - \varepsilon)|s - s'| \leq \angle\left(\boldsymbol{x}_\sigma(s), \boldsymbol{x}_\sigma(s')\right) \leq |s - s'|. \tag{E.31}$$

*In particular, for any two points* $\boldsymbol{x}, \boldsymbol{x}' \in \mathcal{M}_\sigma$, *choosing* $s, s'$ *such that* $\boldsymbol{x}_\sigma(s) = \boldsymbol{x}$, $\boldsymbol{x}_\sigma(s') = \boldsymbol{x}'$, *and* $|s - s'| = d_\mathcal{M}(\boldsymbol{x}, \boldsymbol{x}')$, *we have when* $d_\mathcal{M}(\boldsymbol{x}, \boldsymbol{x}') \leq \frac{\sqrt{\varepsilon}}{\hat{\kappa}}$

$$(1 - \varepsilon)d_\mathcal{M}(\boldsymbol{x}, \boldsymbol{x}') \leq \angle\left(\boldsymbol{x}, \boldsymbol{x}'\right) \leq d_\mathcal{M}(\boldsymbol{x}, \boldsymbol{x}').$$

**Proof.** We prove (E.30) first.

The upper bound is direct from the fact that $\mathcal{M}$ is a pair of paths in the sphere and $\angle(\boldsymbol{x}, \boldsymbol{x}')$ is the length of a path in the sphere of minimum distance between points $\boldsymbol{x}$, $\boldsymbol{x}'$, and then using the fact that the distance $|s' - s| \geq d_{\mathcal{M}}(\boldsymbol{x}_\sigma(s), \boldsymbol{x}_\sigma(s'))$ from (C.7).

The lower bound requires some additional estimates. We fix $s, s'$ satisfying our assumptions; as both $|s - s'|$ and $\angle(\boldsymbol{x}_\sigma(s), \boldsymbol{x}_\sigma(s'))$ are symmetric functions of $(s, s')$, it suffices to assume that $s' \geq s$. Define $t = s' - s$, then by assumption we have $0 \leq t \leq \frac{1}{\hat{\kappa}} \leq \frac{\pi}{2}$. As $\cos^{-1}$ is strictly decreasing on $[-1, 1]$, we only need to show that

$$\langle \boldsymbol{x}_\sigma(s), \boldsymbol{x}_\sigma(s + t) \rangle \leq \cos(t - \hat{\kappa}^2 t^3). \tag{E.32}$$

Using the second order Taylor expansion at $s$, we have

$$\boldsymbol{x}_\sigma(s + t) = \boldsymbol{x}_\sigma(s) + \dot{\boldsymbol{x}}_\sigma(s) + \int_{a=s}^{s+t} \int_{b=s}^{a} \ddot{\boldsymbol{x}}_\sigma(b) db\, da$$

and so

$$\langle \boldsymbol{x}_\sigma(s), \boldsymbol{x}_\sigma(s + t) \rangle = \left\langle \boldsymbol{x}_\sigma(s), \boldsymbol{x}_\sigma(s) + \dot{\boldsymbol{x}}_\sigma(s) + \int_{a=s}^{s+t} \int_{b=s}^{a} \ddot{\boldsymbol{x}}_\sigma(b) db\, da \right\rangle$$

$$= \|\boldsymbol{x}_\sigma(s)\|_2^2 + \langle \boldsymbol{x}_\sigma(s), \dot{\boldsymbol{x}}_\sigma(s) \rangle + \left\langle \boldsymbol{x}_\sigma(s), \int_{a=s}^{s+t} \int_{b=s}^{a} \ddot{\boldsymbol{x}}_\sigma(b) db\, da \right\rangle$$

$$= 1 + \int_{a=s}^{s+t} \int_{b=s}^{a} \langle \boldsymbol{x}_\sigma(s), \ddot{\boldsymbol{x}}_\sigma(b) \rangle\, db\, da \tag{E.33}$$

where we use properties established in Lemma E.3, in particular (E.25) in the last line. Take second order Taylor expansion at $b$ for $\boldsymbol{x}_\sigma(s)$, we have similarly

$$\boldsymbol{x}_\sigma(s) = \boldsymbol{x}_\sigma(b) + \dot{\boldsymbol{x}}_\sigma(b) + \int_{c=s}^{b} \int_{d=c}^{b} \ddot{\boldsymbol{x}}_\sigma(d) dd\, dc.$$

From (E.25) and (E.26), we have $\langle \boldsymbol{x}_\sigma(b), \ddot{\boldsymbol{x}}_\sigma(b) \rangle = -1$ and $\langle \dot{\boldsymbol{x}}_\sigma(b), \ddot{\boldsymbol{x}}_\sigma(b) \rangle = 0$. Thus uniformly for $b \in [s, s + t]$

$$\langle \boldsymbol{x}_\sigma(s), \ddot{\boldsymbol{x}}_\sigma(b) \rangle = -1 + \left\langle \int_{c=s}^{b} \int_{d=c}^{b} \ddot{\boldsymbol{x}}_\sigma(d) dd\, dc, \ddot{\boldsymbol{x}}_\sigma(b) \right\rangle$$

$$= -1 + \int_{c=s}^{b} \int_{d=c}^{b} \langle \ddot{\boldsymbol{x}}_\sigma(d), \ddot{\boldsymbol{x}}_\sigma(b) \rangle\, dd\, dc$$

$$\leq -1 + \int_{c=s}^{b} \int_{d=c}^{b} \|\ddot{\boldsymbol{x}}_\sigma(d)\|_2 \|\ddot{\boldsymbol{x}}_\sigma(b)\|_2 dd\, dc$$

$$\leq -1 + \int_{c=s}^{b} \int_{d=c}^{b} M_2^2 dd\, dc$$

$$\leq -1 + \frac{M_2^2}{2} (b - s)^2,$$

where in the third line we use Cauchy-Schwarz. Plugging this last bound into (E.33), it follows

$$\langle \boldsymbol{x}_\sigma(s), \boldsymbol{x}_\sigma(s + t) \rangle \leq 1 + \int_{a=s}^{s+t} \int_{b=s}^{a} (-1 + (M_2^2/2)(b - s)^2) db\, da$$

$$= 1 - \frac{t^2}{2} + \frac{M_2^2}{2} \int_{a=s}^{s+t} \int_{b=s}^{a} (b - s)^2 db\, da$$

$$= 1 - \frac{t^2}{2} + \frac{M_2^2}{4!} t^4$$

$$= 1 - \frac{t^2}{2} + \frac{1 + \kappa^2}{4!} t^4, \tag{E.34}$$

with an application of Lemma E.3 in the final equality. To conclude, we derive a suitable estimate for $\cos(t - \hat{\kappa}^2 t^3)$. Because $0 \leq t \leq \hat{\kappa}^{-1}$, we have that $t^{-1}(t - \hat{\kappa}^2 t^3) \in [0, 1]$, and because $t \leq \hat{\kappa}^{-1} \leq \pi/2$, we can apply concavity of $\cos$ on $[0, \pi/2]$ to obtain

$$\cos(t - \hat{\kappa}^2 t^3) \geq \frac{t - \hat{\kappa}^2 t^3}{t} \cos(t) + \left(1 - \frac{t - \hat{\kappa}^2 t^3}{t}\right) \cos(0).$$

Next, the estimate $\cos(x) \geq 1 - \frac{x^2}{2} + \frac{x^4}{4!} - \frac{x^6}{6!}$ for all $x$, a consequence of Taylor expansion, gives

$$(1 - \hat{\kappa}^2 t^2) \cos(t) + \hat{\kappa}^2 t^2 \geq (1 - \hat{\kappa}^2 t^2) \left(1 - \frac{t^2}{2} + \frac{t^4}{4!} - \frac{t^6}{6!}\right) + \hat{\kappa}^2 t^2$$

$$= 1 - \frac{t^2}{2} + \frac{t^4}{4!} + \frac{\hat{\kappa}^2 t^4}{2} - \frac{t^6}{6!} - \frac{\hat{\kappa}^2 t^6}{4!} + \frac{\hat{\kappa}^2 t^8}{6!}$$

after distributing. Because $\hat{\kappa} \geq \kappa$, we can split terms and write

$$t^4/4! + \hat{\kappa}^2 t^4/2 \geq \frac{1 + \kappa^2}{4!} t^4 + \hat{\kappa}^2 t^4/4,$$

and then grouping terms in the preceding estimates gives

$$\cos(t - \hat{\kappa}^2 t^3) \geq 1 - \frac{t^2}{2} + \frac{1 + \kappa^2}{4!} t^4 + \hat{\kappa}^2 t^4 \left(\frac{1}{4} - \frac{t^2}{4!} + \frac{t^4}{6!} - \frac{t^2}{6!\hat{\kappa}^2}\right).$$

By way of (E.34) and (E.32), we will therefore be done if we can show that

$$\frac{1}{4} - \left(\frac{1}{4!} + \frac{1}{6!\hat{\kappa}^2}\right) t^2 + \frac{t^4}{6!} \geq 0.$$

This is not hard to obtain: for example, we can prove the weaker but sufficient bound

$$1 - \frac{1}{3!} \left(1 + \frac{1}{30\hat{\kappa}^2}\right) t^2 \geq 0$$

by noticing that because $t \leq \hat{\kappa}^{-1}$, it suffices to show

$$\frac{1}{\hat{\kappa}^2} \left(1 + \frac{1}{30\hat{\kappa}^2}\right) \leq 6,$$

and because the LHS of the previous line is an increasing function of $\hat{\kappa}^{-1}$ and moreover $\hat{\kappa}^{-1} \leq \pi/2$, this bound follows by verifying that indeed $(\pi/2)^2(1 + (1/30)(\pi/2)^2) \leq 6$. Because $s, s'$ were arbitrary we have thus proved (E.30).

For the remaining claims, (E.31) follows naturally from the fact that when $|s - s'| \leq \frac{\sqrt{\varepsilon}}{\hat{\kappa}}$, we have $|s - s'| - \hat{\kappa}^2 |s - s'|^3 \geq (1 - \varepsilon)|s - s'|$. The final claim is a restatement of (E.31) under the additional stated hypotheses. ∎

### Invertibility of $\widehat{M}$ over $S$.

**Lemma E.5** (Young's inequality for Fredholm operators). *Let $K : \mathcal{M} \times \mathcal{M} \to \mathbb{R}$ satisfy $K(\boldsymbol{x}, \boldsymbol{x}') = K(\boldsymbol{x}', \boldsymbol{x})$ for all $(\boldsymbol{x}, \boldsymbol{x}') \in \mathcal{M} \times \mathcal{M}$ and $\sup_{(\boldsymbol{x}, \boldsymbol{x}') \in \mathcal{M} \times \mathcal{M}} |K(\boldsymbol{x}, \boldsymbol{x}')| < +\infty$, and let $\boldsymbol{K}$ denote its Fredholm integral operator (defined as $g \mapsto \boldsymbol{K}[g] = \int_{\mathcal{M}} K(\,\cdot\,, \boldsymbol{x}') g(\boldsymbol{x}') d\boldsymbol{x}'$). For any $1 \leq p \leq +\infty$, we have*

$$\|\boldsymbol{K}\|_{L^p \to L^p} \leq \sup_{\boldsymbol{x} \in \mathcal{M}} \int_{\boldsymbol{x}' \in \mathcal{M}} |K(\boldsymbol{x}, \boldsymbol{x}')| d\boldsymbol{x}'.$$

**Proof.** The proof uses the M. Riesz convexity theorem for interpolation of operators [1, §V, Theorem 1.3], which we need here in the form of a special case: it states that for all $1 \leq p \leq +\infty$, one has

$$\|\boldsymbol{K}\|_{L^p \to L^p} \leq \|\boldsymbol{K}\|_{L^\infty \to L^\infty}^{1/p} \|\boldsymbol{K}\|_{L^1 \to L^1}^{1 - 1/p}. \tag{E.35}$$

To proceed, we will bound the two operator norm terms on the RHS. We have

$$\|\boldsymbol{K}\|_{L^1 \to L^1} = \sup_{\|g\|_{L^1} = 1} \int_{\boldsymbol{x} \in \mathcal{M}} \left| \int_{\boldsymbol{x}' \in \mathcal{M}} K(\boldsymbol{x}, \boldsymbol{x}') g(\boldsymbol{x}') d\boldsymbol{x}' \right| d\boldsymbol{x}$$

$$\leq \sup_{\|g\|_{L^1}=1} \int_{\boldsymbol{x}\in\mathcal{M}} \int_{\boldsymbol{x}'\in\mathcal{M}} |K(\boldsymbol{x},\boldsymbol{x}')||g(\boldsymbol{x}')|\, d\boldsymbol{x}'d\boldsymbol{x}$$

$$= \sup_{\|g\|_{L^1}=1} \int_{\boldsymbol{x}'\in\mathcal{M}} \left( \int_{\boldsymbol{x}\in\mathcal{M}} |K(\boldsymbol{x},\boldsymbol{x}')|\, d\boldsymbol{x} \right) |g(\boldsymbol{x}')|\, d\boldsymbol{x}'$$

$$\leq \sup_{\|g\|_{L^1}=1} \left( \|g\|_{L^1} \sup_{\boldsymbol{x}'\in\mathcal{M}} \left| \int_{\boldsymbol{x}\in\mathcal{M}} |K(\boldsymbol{x},\boldsymbol{x}')|\, \mathrm{d}\boldsymbol{x} \right| \right)$$

$$= \sup_{\boldsymbol{x}\in\mathcal{M}} \int_{\boldsymbol{x}'\in\mathcal{M}} |K(\boldsymbol{x},\boldsymbol{x}')|\, d\boldsymbol{x}'. \tag{E.36}$$

The first inequality above uses the triangle inequality for the integral. In the third line, we rearrange the order of integration using Fubini's theorem, given that $g$ is integrable and $K$ is bounded on $\mathcal{M} \times \mathcal{M}$. In the fourth line, we use $L^1$-$L^\infty$ control of the integrand (i.e., Hölder's inequality), and in the final line we use that $\|g\|_{L^1} = 1$ along with symmetry of $K$ and nonnegativity of the integrand to to re-index and remove the outer absolute value. On the other hand, $L^1$-$L^\infty$ control and the triangle inequality give immediately

$$\|\boldsymbol{K}\|_{L^\infty\to L^\infty} = \sup_{\boldsymbol{x}\in\mathcal{M},\, \|g\|_{L^\infty}=1} \left| \int_{\boldsymbol{x}'\in\mathcal{M}} K(\boldsymbol{x},\boldsymbol{x}')g(\boldsymbol{x}')d\boldsymbol{x}' \right|$$

$$\leq \sup_{\boldsymbol{x}\in\mathcal{M}} \int_{\boldsymbol{x}'\in\mathcal{M}} |K(\boldsymbol{x},\boldsymbol{x}')|d\boldsymbol{x}'.$$

These two bounds are equal; plugging them into (E.35) thus proves the claim. ∎

**Lemma E.6.** *Let $\varepsilon \in (0, \frac{3}{4})$, $r_\varepsilon, S_\varepsilon$ and $\widehat{\boldsymbol{M}}_\varepsilon$ be as defined in* (E.8), (C.12) *and* (E.11). *Then $\widehat{\boldsymbol{M}}_\varepsilon$ diagonalizes in the Fourier orthonormal basis* (C.11). *Write $\lambda_{\min}\left(\boldsymbol{P}_{S_\varepsilon}\widehat{\boldsymbol{M}}_\varepsilon\boldsymbol{P}_{S_\varepsilon}\right)$ for the minimum eigenvalue of the operator $\boldsymbol{P}_{S_\varepsilon}\widehat{\boldsymbol{M}}_\varepsilon\boldsymbol{P}_{S_\varepsilon} : S_\varepsilon \to S_\varepsilon$. Then there exist constants $c, C$ and a constant $C_\varepsilon$ such that when $L \geq C_\varepsilon$, we have*

$$\lambda_{\min}\left(\boldsymbol{P}_{S_\varepsilon}\widehat{\boldsymbol{M}}_\varepsilon\boldsymbol{P}_{S_\varepsilon}\right) \geq (1-\varepsilon)\frac{3\pi n}{4}\log\left(1 + \frac{L-2}{3\pi}r_\varepsilon\right) - Cnr_\varepsilon \log^2 L,$$

$$\geq cn\log L.$$

*As a consequence, $\boldsymbol{P}_{S_\varepsilon}\widehat{\boldsymbol{M}}_\varepsilon\boldsymbol{P}_{S_\varepsilon}$ is invertible over $S_\varepsilon$, and*

$$\left\| \left(\boldsymbol{P}_{S_\varepsilon}\widehat{\boldsymbol{M}}_\varepsilon\boldsymbol{P}_{S_\varepsilon}\right)^{-1} \right\|_{L^2\to L^2} = \lambda_{\min}^{-1}\left(\boldsymbol{P}_{S_\varepsilon}\widehat{\boldsymbol{M}}_\varepsilon\boldsymbol{P}_{S_\varepsilon}\right).$$

**Proof.** Choose $L \gtrsim 1$ to guarantee that $\widehat{\boldsymbol{M}}_\varepsilon$ is well-defined. We use $\psi^\circ$ to denote the DC subtracted skeleton, as defined in (C.4), and $(\phi_{\sigma,k})_{\sigma,k}$ the (intrinsic) Fourier basis on $\mathcal{M}$, as defined in (C.11). For any Fourier basis function $\phi_{\sigma,k}$, we have

$$\widehat{\boldsymbol{M}}_\varepsilon[\phi_{\sigma,k}](\boldsymbol{x}_\sigma(s)) = \int_{s'=s-r_\varepsilon}^{s+r_\varepsilon} \psi^\circ(|s-s'|)\phi_{\sigma,k}\left(\boldsymbol{x}_\sigma(s')\right)ds'$$

$$= \int_{s'=-r_\varepsilon}^{r_\varepsilon} \psi^\circ(|s'|)\exp\left(\frac{i2\pi ks'}{\mathrm{len}(\mathcal{M}_\sigma)}\right)ds'\,\phi_{\sigma,k}(\boldsymbol{x}_\sigma(s))$$

$$= \phi_{\sigma,k}(\boldsymbol{x}_\sigma(s))\int_{s'=-r_\varepsilon}^{r_\varepsilon}\psi^\circ(|s'|)\cos\left(\frac{2\pi ks'}{\mathrm{len}(\mathcal{M}_\sigma)}\right)ds',$$

which shows that each Fourier basis function is an eigenfunction of $\widehat{\boldsymbol{M}}_\varepsilon$; because these functions form an orthonormal basis for $L^2(\mathcal{M})$ (by classical results from Fourier analysis on the circle), $\widehat{\boldsymbol{M}}_\varepsilon$ diagonalizes in this basis. Moreover, because $S_\varepsilon$ is the span of Fourier basis functions, $\boldsymbol{P}_{S_\varepsilon}$ also diagonalizes in this basis, and hence so does $\boldsymbol{P}_{S_\varepsilon}\widehat{\boldsymbol{M}}_\varepsilon\boldsymbol{P}_{S_\varepsilon}$. Because $\widehat{\boldsymbol{M}}_\varepsilon$ is self-adjoint and $\boldsymbol{P}_{S_\varepsilon}$ is an orthogonal projection, $\boldsymbol{P}_{S_\varepsilon}\widehat{\boldsymbol{M}}_\varepsilon\boldsymbol{P}_{S_\varepsilon}$ is self-adjoint; and because $\dim(S_\varepsilon) < +\infty$, the operator $\boldsymbol{P}_{S_\varepsilon}\widehat{\boldsymbol{M}}_\varepsilon\boldsymbol{P}_{S_\varepsilon}$ has finite rank, and therefore has a well-defined minimum eigenvalue, which we denote

as in the statement of the lemma. As $K_{\varepsilon,\sigma} = \lfloor \frac{\varepsilon^{1/2}\operatorname{len}(\mathcal{M}_\sigma)}{2\pi r_\varepsilon} \rfloor$, we have for any $|k_\sigma| \le K_{\varepsilon,\sigma}$ and any $|s'| \le r_\varepsilon$,

$$1 \ge \cos\left(\frac{2\pi k_\sigma s'}{\operatorname{len}(\mathcal{M}_\sigma)}\right) \ge 1 - \left(\frac{2\pi k_\sigma s'}{\operatorname{len}(\mathcal{M}_\sigma)}\right)^2 \ge 1 - \varepsilon.$$

Then for $\sigma \in \{+,-\}$ and $|k| \le K_{\varepsilon,\pm}$,

$$\widehat{\boldsymbol{M}}_\varepsilon[\phi_{\sigma,k}](\boldsymbol{x}_\sigma(s)) = \phi_{\sigma,k}(\boldsymbol{x}_\sigma(s)) \int_{s'=-r_\varepsilon}^{r_\varepsilon} \psi^\circ(|s'|) \cos\left(\frac{2\pi k s'}{\operatorname{len}(\mathcal{M}_\sigma)}\right) ds'$$

$$\ge (1-\varepsilon)\phi_{\sigma,k}(\boldsymbol{x}_\sigma(s)) \int_{s'=-r_\varepsilon}^{r_\varepsilon} \psi^\circ(|s'|) ds',$$

and so

$$\lambda_{\min}\left(\boldsymbol{P}_{S_\varepsilon} \widehat{\boldsymbol{M}}_\varepsilon \boldsymbol{P}_{S_\varepsilon}\right) \ge 2(1-\varepsilon) \int_0^{r_\varepsilon} \psi^\circ(s)\, ds.$$

From Lemma F.7, we have if $L \gtrsim 1$

$$2 \int_{s=0}^{r_\varepsilon} \psi^\circ(s) ds \ge \frac{3\pi n}{4} \log\left(1 + \frac{L-2}{3\pi} r_\varepsilon\right) - C n r_\varepsilon \log^2 L.$$

In particular, as $r_\varepsilon = 6\pi L^{-\frac{a_\varepsilon}{a_\varepsilon+1}}$, there exists a constant $C_\varepsilon$ such that when $L \ge C'_\varepsilon$, we have

$$C n r_\varepsilon \log^2 L \le \frac{\varepsilon}{4} \frac{3\pi n}{4} \log\left(1 + \frac{L-2}{3\pi} r_\varepsilon\right),$$

and thus

$$\begin{aligned}
\lambda_{\min}\left(\boldsymbol{P}_{S_\varepsilon} \widehat{\boldsymbol{M}}_\varepsilon \boldsymbol{P}_{S_\varepsilon}\right) &\ge (1-\varepsilon)\left(1-\frac{\varepsilon}{4}\right)\frac{3\pi n}{4}\log\left(1+\frac{L-2}{3\pi}r_\varepsilon\right) \\
&\ge \left(1-\frac{5\varepsilon}{4}\right)\frac{3\pi n}{4}\log\left(L^{1-\frac{a_\varepsilon}{a_\varepsilon+1}}\right) \\
&= \left(1-\frac{5\varepsilon}{4}\right)\frac{3\pi n}{4}\frac{1}{a_\varepsilon+1}\log L \\
&\ge \left(1-\frac{5}{4}\cdot\frac{3}{4}\right)\frac{3\pi n}{4}\cdot\frac{1}{2}\log L \\
&\ge cn\log L \\
&> 0,
\end{aligned}$$

where we used $L \gtrsim 1$ in the second inequality, and $\varepsilon < 3/4$ and $a_\varepsilon \le 1$ in the third inequality. So $\boldsymbol{P}_{S_\varepsilon}\widehat{\boldsymbol{M}}_\varepsilon\boldsymbol{P}_{S_\varepsilon}$ is invertible over $S_\varepsilon$, with

$$\left(\boldsymbol{P}_{S_\varepsilon}\widehat{\boldsymbol{M}}_\varepsilon\boldsymbol{P}_{S_\varepsilon}\right)^{-1}[h] = \sum_{\sigma=\pm}\sum_{k=0}^{K_{\varepsilon,\sigma}}\left(\int_{s=-r_\varepsilon}^{r_\varepsilon}\psi^\circ(|s|)\cos\left(\frac{2\pi ks}{\operatorname{len}(\mathcal{M}_\sigma)}\right)ds\right)^{-1}\phi_{\sigma,k}\phi_{\sigma,k}^* h. \quad \text{(E.37)}$$

The final claim is a consequence of the fact that $\boldsymbol{P}_{S_\varepsilon}\widehat{\boldsymbol{M}}_\varepsilon\boldsymbol{P}_{S_\varepsilon}$ is self-adjoint and finite-rank.

∎

**Lemma E.7.** *Let* $\varepsilon \in (0, \frac{3}{4})$, $a_\varepsilon, r_\varepsilon, \boldsymbol{M}_\varepsilon$ *and* $\widehat{\boldsymbol{M}}_\varepsilon$ *be as defined in* (E.7), (E.8), (E.10) *and* (E.11). *There exist constants* $C, C'$, *such that when* $L \ge C$ *and* $L \ge \left(\varepsilon^{-1/2}12\pi\hat\kappa\right)^{\frac{a_\varepsilon+1}{a_\varepsilon}}$, *we have*

$$\left\|\boldsymbol{M}_\varepsilon - \widehat{\boldsymbol{M}}_\varepsilon\right\|_{L^2\to L^2} \le (1-\varepsilon)^{-2}C'\hat\kappa^2 nr_\varepsilon^2.$$

**Proof.** We choose $L \gtrsim 1$ to guarantee that $\widehat{\boldsymbol{M}}_\varepsilon$ is well-defined for all $0 < \varepsilon < 3/4$. We would like to use Lemma E.5 to bound $\|\boldsymbol{M}_\varepsilon - \widehat{\boldsymbol{M}}_\varepsilon\|_{L^2\to L^2}$, and thus we define two (suggestively-named) bounded symmetric kernels $\mathcal{M} \times \mathcal{M} \to \mathbb{R}$:

$$M_\varepsilon(\boldsymbol{x}, \boldsymbol{x}') = \mathbb{1}_{d_\mathcal{M}(\boldsymbol{x},\boldsymbol{x}')<r_\varepsilon}\psi^\circ(\angle(\boldsymbol{x},\boldsymbol{x}'))$$

and

$$\widehat{M_\varepsilon}(\boldsymbol{x}, \boldsymbol{x}') = \mathbb{1}_{d_{\mathcal{M}}(\boldsymbol{x},\boldsymbol{x}')<r_\varepsilon}\psi^\circ(d_{\mathcal{M}}(\boldsymbol{x}, \boldsymbol{x}')).$$

From (E.10), $\boldsymbol{M}_\varepsilon$ is indeed $M_\varepsilon$'s associated Fredholm integral operator. To show that under our constraints for $L$, $\widehat{\boldsymbol{M}_\varepsilon}$ is also $\widehat{M_\varepsilon}$'s associated integral operator, we first notice that following (C.7), for any $\boldsymbol{x}, \boldsymbol{x}' \in \mathcal{M}$, $d_{\mathcal{M}}(\boldsymbol{x}, \boldsymbol{x}') < r_\varepsilon$ if and only if there exist $\sigma$, $s$ and $s'$ such that $\boldsymbol{x} = \boldsymbol{x}_\sigma(s)$, $\boldsymbol{x}' = \boldsymbol{x}_\sigma(s')$ and $|s' - s| < r_\varepsilon$. This means for any fixed $\boldsymbol{x}$, if we let $\sigma$ and $s$ be chosen such that $\boldsymbol{x} = \boldsymbol{x}_\sigma(s)$, then $L_{r_\varepsilon}(\boldsymbol{x}) = \{\boldsymbol{x}_\sigma(s') | |s' - s| < r_\varepsilon\}$. Furthermore, as $L \geq \left(\varepsilon^{-1/2}12\pi\hat\kappa\right)^{\frac{a_\varepsilon+1}{a_\varepsilon}}$, by (E.29) in Lemma E.3 we have $r_\varepsilon \leq \frac{\sqrt\varepsilon}{2\hat\kappa} < \min\{\text{len}(\mathcal{M}_+), \text{len}(\mathcal{M}_-)\}/2$. Under this condition, we can unambiguously express the intrinsic distance $d_{\mathcal{M}}$ in terms of arc length at the local scale: for any $\boldsymbol{x}' \in L_{r_\varepsilon}(\boldsymbol{x})$, there is a *unique* $s'$ such that $|s' - s| \leq r_\varepsilon$. To see this, note that for any other parameter choice that attains the infimum in (C.7) $s'' = s' + k\,\text{len}(\mathcal{M}_\sigma)$ with integer $k \neq 0$, the triangle inequality implies $|s'' - s| \geq |r_\epsilon - k\,\text{len}(\mathcal{M}_\sigma)|$, and one has $|r_\epsilon - k\,\text{len}(\mathcal{M}_\sigma)| > r_\epsilon$ for every $k \neq 0$ if $0 < r_\epsilon < \text{len}(\mathcal{M}_\sigma)/2$. Then for $\boldsymbol{x}' \in L_{r_\varepsilon}(\boldsymbol{x})$ and any $s' \in [s - r_\varepsilon, s + r_\varepsilon]$ such that $\boldsymbol{x}_\sigma(s') = \boldsymbol{x}'$, we have $d_{\mathcal{M}}(\boldsymbol{x}, \boldsymbol{x}') = |s - s'|$. Combining all these points, $\widehat{M_\varepsilon}$'s associated Fredholm integral operator $\boldsymbol{H}$ can be written as:

$$\begin{aligned}
\boldsymbol{H}[f](\boldsymbol{x}_\sigma(s)) &= \int_{d_{\mathcal{M}}(\boldsymbol{x}_\sigma(s),\boldsymbol{x}')<r_\varepsilon} \psi^\circ(d_{\mathcal{M}}(\boldsymbol{x}_\sigma(s), \boldsymbol{x}'))f(\boldsymbol{x}')d\boldsymbol{x}' \\
&= \int_{\boldsymbol{x}'\in\{\boldsymbol{x}_\sigma(s)||s'-s|<r_\varepsilon\}} \psi^\circ(d_{\mathcal{M}}(\boldsymbol{x}_\sigma(s), \boldsymbol{x}'))f(\boldsymbol{x}')d\boldsymbol{x}' \\
&= \int_{s'=s-r_\varepsilon}^{s+r_\varepsilon} \psi^\circ(d_{\mathcal{M}}(\boldsymbol{x}_\sigma(s), \boldsymbol{x}_\sigma(s')))f(\boldsymbol{x}_\sigma(s'))ds' \\
&= \int_{s'=s-r_\varepsilon}^{s+r_\varepsilon} \psi^\circ(|s - s'|)f(\boldsymbol{x}_\sigma(s'))ds' \\
&= \widehat{\boldsymbol{M}_\varepsilon}[f](\boldsymbol{x}_\sigma(s)),
\end{aligned}$$

which means $\widehat{\boldsymbol{M}_\varepsilon}$ is indeed $\widehat{M_\epsilon}$'s associated integral kernel.

We can now apply Lemma E.5 and and get

$$\begin{aligned}
\|\boldsymbol{M}_\varepsilon - \widehat{\boldsymbol{M}_\varepsilon}\|_{L^2\to L^2} &\leq \sup_{\boldsymbol{x}\in\mathcal{M}} \int_{\boldsymbol{x}'\in\mathcal{M}} |M_\varepsilon(\boldsymbol{x}, \boldsymbol{x}') - \widehat{M_\varepsilon}(\boldsymbol{x}, \boldsymbol{x}')|d\boldsymbol{x}' \\
&= \sup_{\boldsymbol{x}\in\mathcal{M}} \int_{\boldsymbol{x}'\in L_{r_\varepsilon}(\boldsymbol{x})} |M_\varepsilon(\boldsymbol{x}, \boldsymbol{x}') - \widehat{M_\varepsilon}(\boldsymbol{x}, \boldsymbol{x}')|d\boldsymbol{x}' \\
&= \sup_{s,\sigma} \int_{s'=s-r_\varepsilon}^{s+r_\varepsilon} \left|\psi^\circ\Big(\angle(\boldsymbol{x}_\sigma(s), \boldsymbol{x}_\sigma(s'))\Big) - \psi^\circ(|s - s'|)\right| ds'. \quad \text{(E.38)}
\end{aligned}$$

Here, we recall that $r_\epsilon < \pi/4$ (because $\hat\kappa \leq \pi/2$), so there is no issue with these evaluations and the domain of $\psi^\circ$ being $[0, \pi]$. Note that from (E.30), when $|s - s'| \leq r_\varepsilon \leq \frac{\sqrt\varepsilon}{\hat\kappa}$, we have

$$\begin{aligned}
\angle(\boldsymbol{x}_\sigma(s), \boldsymbol{x}_\sigma(s')) &\geq |s - s'| - \hat\kappa^2|s - s'|^3 \\
&\geq (1 - \varepsilon)|s - s'|. \quad \text{(E.39)}
\end{aligned}$$

As $\psi^\circ$ is nonnegtive, strictly decreasing and convex by Lemma G.5, we know both $\psi^\circ$ and $|\dot\psi^\circ|$ are decreasing. Also, by the upper bound in Lemma E.4, we have that $\psi^\circ(\angle(\boldsymbol{x}_\sigma(s), \boldsymbol{x}_\sigma(s'))) - \psi^\circ(|s - s'|) \geq 0$, so we can essentially ignore the absolute value in the integrand in (E.38). We can then calculate

$$\begin{aligned}
\int_{s'=s-r_\varepsilon}^{s+r_\varepsilon} &\psi^\circ\Big(\angle(\boldsymbol{x}_\sigma(s), \boldsymbol{x}_\sigma(s'))\Big) - \psi^\circ(|s - s'|)\, ds' \\
&\leq \int_{s'=s-r_\varepsilon}^{s+r_\varepsilon} \psi^\circ\Big(|s - s'| - \hat\kappa^2|s - s'|^3\Big) - \psi^\circ(|s - s'|)\, ds' \\
&= \int_{t=-r_\varepsilon}^{r_\varepsilon} \psi^\circ\big(|t| - \hat\kappa^2|t|^3\big) - \psi^\circ(|t|)\, dt
\end{aligned}$$

$$= \int_{t=-r_\varepsilon}^{r_\varepsilon} \int_{a=|t|-\hat{\kappa}^2|t|^3}^{|t|} \left| \dot{\psi}^\circ(a) \right| da \, dt$$

$$\leq \hat{\kappa}^2 \int_{t=-r_\varepsilon}^{r_\varepsilon} |t|^3 \left| \dot{\psi}^\circ(|t| - \hat{\kappa}^2|t|^3) \right| dt$$

$$\leq \hat{\kappa}^2 \int_{t=-r_\varepsilon}^{r_\varepsilon} |t|^3 \left| \dot{\psi}^\circ((1-\varepsilon)|t|) \right| dt$$

$$= 2(1-\varepsilon)^{-4} \hat{\kappa}^2 \int_{t=0}^{(1-\varepsilon)r_\varepsilon} t^3 |\dot{\psi}^\circ(t)| \, dt.$$

Above, the first line comes from (E.39) and the the fact that $\psi^\circ$ is strictly decreasing, the fourth and fifth line comes from the fact that $|\dot{\psi}^\circ|$ is decreasing and (E.39). The last line uses symmetry and a linear transformation. Note that from (E.39) we always have $|t| - \hat{\kappa}^2|t^3|$ nonnegative when $|t| \leq r_\varepsilon$ and thus all above formulas are well defined. From Lemma F.10, we know that there exists $C, C'$ such that when $L \geq C$, we have

$$\int_0^{(1-\varepsilon)r_\varepsilon} t^3 |\dot{\psi}^\circ(t)| dt \leq C'n(1-\varepsilon)^2 r_\varepsilon^2,$$

and plugging all bounds back to (E.38) we get

$$\left\| M_\varepsilon - \widehat{M_\varepsilon} \right\|_{L^2 \to L^2} \leq (1-\varepsilon)^{-2} C' \hat{\kappa}^2 n r_\varepsilon^2$$

as claimed. $\blacksquare$

**Lemma E.8.** *Let $\varepsilon \in (0, \frac{3}{4})$, $r_\varepsilon$ and $N_{r_\varepsilon,\varepsilon}$ as defined in (E.8) and (E.4). For any $0 < \varepsilon'' \leq 1$, there exist numbers $C_{\varepsilon''}, C'_{\varepsilon''}$ such that when $L \geq C_{\varepsilon''}$, we have*

$$\sup_{x \in \mathcal{M}} \int_{x' \in N_{r_\varepsilon,\varepsilon}(x)} \psi^\circ \left( \angle(x, x') \right) ds' \leq \frac{3\pi n}{4(1-\varepsilon)} (1 + \varepsilon'') \log \left( \frac{\sqrt{\varepsilon}}{\hat{\kappa} r_\varepsilon} \right) + C'_{\varepsilon''} n.$$

**Proof.** For $x \in \mathcal{M}$, assume the parameters are chosen such that the corresponding near piece is nonempty, for otherwise the claim is immediate. Recalling (E.4), for any $x' \in N_{r_\varepsilon,\varepsilon}(x)$, we have $d_\mathcal{M}(x, x') \leq \sqrt{\varepsilon}/\hat{\kappa}$. From Lemma E.4, this implies $\angle(x, x') \geq (1-\varepsilon)d_\mathcal{M}(x, x')$. Let $\sigma, s$ be such that $x_\sigma(s) = x$. Notice by the discussion following the definition of the intrinsic distance in (C.7) that the near component $N_{r_\varepsilon,\varepsilon}(x)$ is contained in the set $\{x_\sigma(s') \mid |s' - s| \in [r_\varepsilon, \sqrt{\varepsilon}/\hat{\kappa}]\}$. And from Lemma G.5, $\psi^\circ$ is strictly decreasing, thus we have

$$\int_{x' \in N_{r_\varepsilon,\varepsilon}(x)} \psi^\circ \left( \angle(x, x') \right) ds' \leq \int_{s'=s+r_\varepsilon}^{s+\frac{\sqrt{\varepsilon}}{\hat{\kappa}}} \psi^\circ \left( \angle(x_\sigma(s), x_\sigma(s')) \right) ds'$$

$$+ \int_{s'=s-\frac{\sqrt{\varepsilon}}{\hat{\kappa}}}^{s-r_\varepsilon} \psi^\circ \left( \angle(x_\sigma(s), x_\sigma(s')) \right) ds'$$

$$\leq \int_{s'=s+r_\varepsilon}^{s+\frac{\sqrt{\varepsilon}}{\hat{\kappa}}} \psi^\circ \left( (1-\varepsilon)|s' - s| \right) ds'$$

$$+ \int_{s'=s-\frac{\sqrt{\varepsilon}}{\hat{\kappa}}}^{s-r_\varepsilon} \psi^\circ \left( (1-\varepsilon)|s' - s| \right) ds'$$

$$= 2 \int_{t=r_\varepsilon}^{\frac{\sqrt{\varepsilon}}{\hat{\kappa}}} \psi^\circ \left( (1-\varepsilon)t \right) dt$$

$$= \frac{2}{1-\varepsilon} \int_{t=(1-\varepsilon)r_\varepsilon}^{\frac{(1-\varepsilon)\sqrt{\varepsilon}}{\hat{\kappa}}} \psi^\circ (t) \, dt,$$

where in the last line we apply a linear change of variables. We also note that in the above integrals $|s' - s| \leq \hat{\kappa}^{-1} \leq \pi/2$, so there are no issues above with the domain of $\psi^\circ$ being $[0, \pi]$. From

Lemma F.9, for any $0 < \varepsilon'' \leq 1$, there exist numbers $C_{\varepsilon''}, C'_{\varepsilon''}$ such that if $L \geq C_{\varepsilon''}$, then $r_\varepsilon$ satisfies the condition in (F.12) and we have

$$
\sup_{\boldsymbol{x} \in \mathcal{M}} \int_{\boldsymbol{x}' \in N_{r_\varepsilon, \varepsilon}(\boldsymbol{x})} \psi^\circ \Big( \angle(\boldsymbol{x}, \boldsymbol{x}') \Big) ds' \leq \frac{2}{1-\varepsilon} \int_{t=(1-\varepsilon)r_\varepsilon}^{\frac{(1-\varepsilon)\sqrt{\varepsilon}}{\hat{\kappa}}} \psi^\circ \left( |t| \right) dt
$$

$$
\leq \frac{2}{1-\varepsilon}(1+\varepsilon'')\frac{3\pi n}{8} \log \left( \frac{1+(L-3)\frac{(1-\varepsilon)\sqrt{\varepsilon}/(3\pi)}{\hat{\kappa}}}{1+(L-3)(1-\varepsilon)r_\varepsilon/(3\pi)} \right)
$$

$$
+ \quad C'_{\varepsilon''} n
$$

$$
\leq \frac{3\pi n}{4(1-\varepsilon)}(1+\varepsilon'') \log \left( \frac{\sqrt{\varepsilon}}{\hat{\kappa} r_\varepsilon} \right) + C'_{\varepsilon''} n.
$$

■

**Lemma E.9.** *Let $\varepsilon \in (0, \frac{3}{4})$, $\delta \in (0, 1-\varepsilon]$. Let $W_{\varepsilon, \delta}$ as in* (E.6)*. There exist constants $C, C'$ such that when $L \geq C$, for any $\boldsymbol{x} \in \mathcal{M}$,*

$$
\int_{\boldsymbol{x}' \in W_{\varepsilon, \delta}(\boldsymbol{x})} \psi^\circ \Big( \angle(\boldsymbol{x}, \boldsymbol{x}') \Big) ds' \leq \text{⌗}_{\varepsilon, \delta}(\mathcal{M}) C' n \log \left( 1 + \frac{\frac{1}{\sqrt{1+\kappa^2}}}{\Delta_\varepsilon} \right).
$$

**Proof.** To bound the integral, we rely on the observation that for each 'curve segment' inside the winding component, the angle $\angle(\boldsymbol{x}, \boldsymbol{x}')$ cannot stay small for the whole segment, and thus we can avoid worst case control for the angle as we have employed for the far component in Lemma E.10.[6] We will begin by constructing a specific finite cover of curve segments for the winding component, then we will bound the integral over each curve segment by providing a lower bound for the angle function.

As $\mathcal{M}$ is compact with bounded length, from the definition in (C.10) we know $\text{⌗}_{\varepsilon, \delta}(\mathcal{M})$ is a finite number for any choice of $\varepsilon, \delta$. From the definition of the winding component (E.6), for any point $\boldsymbol{x} \in \mathcal{M}$, we can cover $W_{\varepsilon, \delta}(\boldsymbol{x})$ by at most $\text{⌗}_{\varepsilon, \delta}(\mathcal{M})$ closed balls in the intrinsic distance on the manifold with radii no larger than $1/\sqrt{1+\kappa^2}$. Topologically, each ball in the intrinsic distance of radii $r$ is a curve segment of length $2r$; thus, $W_{\varepsilon, \delta}(\boldsymbol{x})$ can be covered by at most $2\text{⌗}_{\varepsilon, \delta}(\mathcal{M})$ curve segments, each with length no larger than $1/\sqrt{1+\kappa^2}$. Formally, this implies that for each $\boldsymbol{x} \in \mathcal{M}$, there exists a number $N(\boldsymbol{x}) \leq 2\text{⌗}_{\varepsilon, \delta}(\mathcal{M})$ and for each $i \in \{1, \cdots, N(\boldsymbol{x})\}$, there exist a sign $\sigma_i(\boldsymbol{x}) \in \{\pm\}$ and a nonempty interval $I_i(\boldsymbol{x}) = [s_{1,i}(\boldsymbol{x}), s_{2,i}(\boldsymbol{x})]$ with length no greater than $\frac{1}{\sqrt{1+\kappa^2}}$ and strictly less than $\text{len}(\mathcal{M}_{\sigma_i(\boldsymbol{x})})$ such that

$$
W_{\varepsilon, \delta}(\boldsymbol{x}) \subseteq \bigcup_{i=1}^{N(\boldsymbol{x})} X_i(\boldsymbol{x})
$$

where $X_i(\boldsymbol{x}) = \{\boldsymbol{x}_{\sigma_i(\boldsymbol{x})}(s) \mid s \in I_i(\boldsymbol{x})\} \subset \mathcal{M}$ with $X_i(\boldsymbol{x}) \cap W_{\varepsilon, \delta}(\boldsymbol{x}) \neq \varnothing$. For the purpose of minimum coverage, we can further assume without loss of generality that for each $\boldsymbol{x}$ and each $i$, the boundary points $\boldsymbol{x}_{\sigma_i(\boldsymbol{x})}(s_{1,i}(\boldsymbol{x}))$ and $\boldsymbol{x}_{\sigma_i(\boldsymbol{x})}(s_{2,i}(\boldsymbol{x}))$ belong to $W_{\varepsilon, \delta}(\boldsymbol{x})$: we can always set $p_{1,i}(\boldsymbol{x}) = \inf\{s \mid s \in [s_{1,i}(\boldsymbol{x}), s_{2,i}(\boldsymbol{x})], \boldsymbol{x}_{\sigma_i(\boldsymbol{x})}(s) \in W_{\varepsilon, \delta}(\boldsymbol{x})\}$ and $p_{2,i}(\boldsymbol{x}) = \sup\{s | s \in [s_{1,i}(\boldsymbol{x}), s_{2,i}(\boldsymbol{x})], \boldsymbol{x}_{\sigma_i(\boldsymbol{x})}(s) \in W_{\varepsilon, \delta}(\boldsymbol{x})\}$, then the curve segment associated with $\sigma_i(\boldsymbol{x})$ and interval $[p_{1,i}(\boldsymbol{x}), p_{2,i}(\boldsymbol{x})]$ still covers $X_i(\boldsymbol{x}) \cap W_{\varepsilon, \delta}(\boldsymbol{x})$. As $W_{\varepsilon, \delta}(\boldsymbol{x})$ is closed, we have the boundary points $\boldsymbol{x}_{\sigma_i(\boldsymbol{x})}(p_{1,i}(\boldsymbol{x})), \boldsymbol{x}_{\sigma_i(\boldsymbol{x})}(p_{2,i}(\boldsymbol{x})) \in W_{\varepsilon, \delta}(\boldsymbol{x})$ and as $X_i(\boldsymbol{x})$ intersect with $W_{\varepsilon, \delta}(\boldsymbol{x})$, the definition above is well defined.

We will next increase the number of sets in these coverings, so that they are guaranteed not to fall into any of the "local pieces" at $\boldsymbol{x}$: although by the definitions (E.3) and (E.6) the local and winding pieces at any $\boldsymbol{x}$ are disjoint, it may be the case that when we pass to the covering sets $(X_i(\boldsymbol{x}))_{i \in [N(\boldsymbol{x})]}$, we overlap with the local piece. In particular, consider a "local piece" $L_{\sqrt{\varepsilon}/\hat{\kappa}}(\boldsymbol{x})$ defined as in (E.3), which from the definition does not intersect with $W_{\varepsilon, \delta}(\boldsymbol{x})$. For each $i$, as the boundary points of

---

[6]Within the lemma, a curve segment means $\{\boldsymbol{x}_\sigma(s) | s \in [s_1, s_2]\} \subseteq \mathcal{M}_\sigma$ for certain $\sigma$, $s_1$ and $s_2$ with $|s_1 - s_2| < \text{len}(\mathcal{M}_\sigma)$, and we call $|s_1 - s_2|$ the length of the curve segment.

$X_i(\boldsymbol{x})$ fall in $W_{\varepsilon,\delta}(\boldsymbol{x})$, these boundary points do not belong to $L_{\sqrt{\varepsilon}/\hat{\kappa}}(\boldsymbol{x})$. And as $L_{\sqrt{\varepsilon}/\hat{\kappa}}(\boldsymbol{x})$ is topologically connected and one dimensional, if $X_i(\boldsymbol{x})$ intersects with $L_{\sqrt{\varepsilon}/\hat{\kappa}}(\boldsymbol{x})$, it must contains the whole local piece. As $X_i(\boldsymbol{x})$ itself is a curve segment and one dimensional, and $L_{\sqrt{\varepsilon}/\hat{\kappa}}(\boldsymbol{x})$ is open, removing $L_{\sqrt{\varepsilon}/\hat{\kappa}}(\boldsymbol{x})$ would leave two curve segments with smaller length. Then these two curve segments lie in $\mathcal{M} \setminus L_{\sqrt{\varepsilon}/\hat{\kappa}}(\boldsymbol{x})$, and cover $X_i(\boldsymbol{x}) \setminus L_{\sqrt{\varepsilon}/\hat{\kappa}}(\boldsymbol{x})$. In other words, for any $\boldsymbol{x} \in \mathcal{M}$, there exists $N'(\boldsymbol{x}) \leq 4 \aleph_{\varepsilon,\delta}(\mathcal{M})$ and for $i \in \{1, \cdots, N'(\boldsymbol{x})\}$, there exist signs $\sigma_i'(\boldsymbol{x}) \in \{\pm\}$ and intervals $I_i'(\boldsymbol{x}) = [s_{1,i}'(\boldsymbol{x}), s_{2,i}'(\boldsymbol{x})]$ with length no greater than $\frac{1}{\sqrt{1+\kappa^2}}$ such that

$$W_{\varepsilon,\delta}(\boldsymbol{x}) \subseteq \bigcup_{i=1}^{N'(\boldsymbol{x})} X_i'(\boldsymbol{x}),$$

where $X_i'(\boldsymbol{x}) = \{\boldsymbol{x}_{\sigma_i'(\boldsymbol{x})}(s) \mid s \in I_i'(\boldsymbol{x})\} \subset \mathcal{M} \setminus L_{\sqrt{\varepsilon}/\hat{\kappa}}(\boldsymbol{x})$ with $X_i'(\boldsymbol{x}) \cap W_{\varepsilon,\delta}(\boldsymbol{x}) \neq \varnothing$. We therefore have

$$\int_{\boldsymbol{x}' \in W_{\varepsilon,\delta}(\boldsymbol{x})} \psi^\circ\Big(\angle\big(\boldsymbol{x}, \boldsymbol{x}'\big)\Big) ds' \leq \sum_{i=1}^{N'(\boldsymbol{x})} \int_{s \in I_i'(\boldsymbol{x})} \psi^\circ\Big(\angle\big(\boldsymbol{x}, \boldsymbol{x}_{\sigma_i'(\boldsymbol{x})}(s)\big)\Big) ds. \tag{E.40}$$

We next derive additional properties of the pieces $X_i'(\boldsymbol{x})$ that will allow us to obtain suitable estimates for the integrals on the RHS of (E.40). As each $X_i'(\boldsymbol{x})$ is a compact set, we let

$$s_i^*(\boldsymbol{x}) \in \arg \min_{s \in I_i'(\boldsymbol{x})} \angle\big(\boldsymbol{x}, \boldsymbol{x}_{\sigma_i'(\boldsymbol{x})}(s)\big)$$

and denote $\boldsymbol{x}_i^*(\boldsymbol{x}) = \boldsymbol{x}_{\sigma_i(\boldsymbol{x})}(s_i^*(\boldsymbol{x}))$. Below we will abbreviate $\boldsymbol{x}_i^*(\boldsymbol{x}), s_i^*(\boldsymbol{x})$ and $\sigma_i'(\boldsymbol{x})$ as $\boldsymbol{x}_i^*, s_i^*$ and $\sigma_i'$ when the base point $\boldsymbol{x}$ is clear. We further abbreviate $\dot{\boldsymbol{x}}_i^* = \dot{\boldsymbol{x}}_{\sigma_i(\boldsymbol{x})}(s_i^*(\boldsymbol{x}))$. As $X_i'(\boldsymbol{x})$ intersects with the winding component, we have $\angle\big(\boldsymbol{x}, \boldsymbol{x}_i'\big) \leq \frac{\delta\sqrt{\varepsilon}}{\hat{\kappa}} < \frac{\pi}{2}$. And as $X_i'(\boldsymbol{x}) \cap L_{\sqrt{\varepsilon}/\hat{\kappa}}(\boldsymbol{x}) = \varnothing$, we have $d_{\mathcal{M}}(\boldsymbol{x}, \boldsymbol{x}_i^*) \geq \sqrt{\varepsilon}/\hat{\kappa}$. This means $\boldsymbol{x}_i^* \in W_{\varepsilon,\delta}(\boldsymbol{x})$ from (E.6). As $\cos$ is strictly decreasing from $0$ to $\pi$ and $s_i^*$ minimizes $\angle\big(\boldsymbol{x}, \boldsymbol{x}_{\sigma_i'}(s)\big)$, it also maximizes $\langle \boldsymbol{x}, \boldsymbol{x}_{\sigma_i'}(s)\rangle$. For any $s \in I_i'(\boldsymbol{x})$, from the second order Taylor expansion of $\boldsymbol{x}_{\sigma_i'}(s)$ around $\boldsymbol{x}_i^*$ we have

$$\langle \boldsymbol{x}, \boldsymbol{x}_i^* \rangle \geq \langle \boldsymbol{x}, \boldsymbol{x}_{\sigma_i'}(s)\rangle$$

$$= \langle \boldsymbol{x}, \boldsymbol{x}_i^*\rangle + (s - s_i^*)\langle \boldsymbol{x}, \dot{\boldsymbol{x}}_i^*\rangle + \Big\langle \boldsymbol{x}, \int_{a=s_i^*}^{s} \int_{b=s_i^*}^{a} \boldsymbol{x}_{\sigma_i'}^{(2)}(b)\, db\, da \Big\rangle$$

$$\geq \langle \boldsymbol{x}, \boldsymbol{x}_i^*\rangle + (s - s_i^*)\langle \boldsymbol{x}, \dot{\boldsymbol{x}}_i^*\rangle - \frac{(s - s_i^*)^2}{2} M_2,$$

with the last line following from Cauchy-Schwarz. In the previous equations, we are of course using the convention that for a real-valued function $f$ and numbers $a < b$, the notation $\int_b^a f(x)\, dx$ denotes the integral $-\int_a^b f(x)\, dx$. We are going to use this bound to reprove a classical first-order optimality condition for interval-constrained problems. We split into cases depending on where the point $s_i^*$ lies: if $s_i^*$ is not the right end point $s_{2,i}'$, by taking $s$ approaching $s_i^*$ from above, we would have $\langle \boldsymbol{x}, \dot{\boldsymbol{x}}_i^* \rangle \leq 0$. Similarly, if $s_i^*$ is not the left end point $s_{1,i}'$, by taking $s$ approaching $s_i^*$ from below, we would have $\langle \boldsymbol{x}, \dot{\boldsymbol{x}}_i^* \rangle \geq 0$. This gives

$$\langle \boldsymbol{x}, \dot{\boldsymbol{x}}_i^* \rangle \text{ is } \begin{cases} \leq 0 & s_i^* = s_{2,i} \\ \geq 0 & s_i^* = s_{1,i} \\ = 0 & o.w. \end{cases}$$

which implies

$$(s - s_i^*)\langle \boldsymbol{x}, \dot{\boldsymbol{x}}_i^* \rangle \leq 0, \qquad \forall s \in I_i'(\boldsymbol{x}). \tag{E.41}$$

We use again the Taylor expansion at $s_i^*$ and get

$$\big\| \boldsymbol{x}_{\sigma_i'}(s) - \boldsymbol{x}_i^* - (s - s_i^*)\dot{\boldsymbol{x}}_i^* \big\|_2 = \Big\| \int_{a=s_i^*}^{s} \int_{b=s_i^*}^{a} \boldsymbol{x}_{\sigma_i'}^{(2)}(b)\, db\, da \Big\|_2$$

$$\leq \frac{(s - s_i^*)^2}{2} M_2$$
$$= \frac{1}{2}(1 + \kappa^2)^{1/2}(s - s_i^*)^2 \tag{E.42}$$

with an application of (E.27) in the last line. Moreover, we have

$$\|\boldsymbol{x} - \boldsymbol{x}_i^*\|_2 = 2\sin\left(\frac{\angle(\boldsymbol{x}, \boldsymbol{x}_i^*)}{2}\right)$$
$$\geq \frac{4}{\pi}\sin\left(\frac{\pi}{4}\right)\angle(\boldsymbol{x}, \boldsymbol{x}_i^*)$$
$$= \frac{2\sqrt{2}}{\pi}\angle(\boldsymbol{x}, \boldsymbol{x}_i^*)$$
$$\geq \frac{2\sqrt{2}}{\pi}\Delta_\varepsilon, \tag{E.43}$$

where the first line is a trigonometric identity, the first inequality uses $\angle(\boldsymbol{x}, \boldsymbol{x}_i^*) < \pi/2$ together with the fact that $\sin$ function is concave from $0$ to $\pi$ and thus $\sin(at) \geq a\sin(t)$ for $a \in [0, 1]$ and $t \in [0, \pi]$ (applied to $a = \angle(\boldsymbol{x}, \boldsymbol{x}_i^*)/(\pi/2)$ and $t = \pi/4$), and the last line follows directly from the definition of $\Delta_\varepsilon$ in (C.8). Making use of the preceding estimates, for any $s \in I_i'(\boldsymbol{x})$ we can finally calculate

$$\|\boldsymbol{x}_{\sigma_i'}(s) - \boldsymbol{x}\|_2^2 = \|\boldsymbol{x}_i^* - \boldsymbol{x} + (s - s_i^*)\dot{\boldsymbol{x}}_i^* + (\boldsymbol{x}_{\sigma_i'}(s) - \boldsymbol{x}_i^* - (s - s_i^*)\dot{\boldsymbol{x}}_i^*)\|_2^2$$
$$\geq \|\boldsymbol{x}_i^* - \boldsymbol{x} + (s - s_i^*)\dot{\boldsymbol{x}}_i^*\|_2^2 - \|\boldsymbol{x}_{\sigma_i'}(s) - \boldsymbol{x}_i^* - (s - s_i^*)\dot{\boldsymbol{x}}_i^*\|_2^2$$
$$\geq \|\boldsymbol{x}_i^* - \boldsymbol{x}\|_2^2 + \|(s - s_i^*)\dot{\boldsymbol{x}}_i^*\|_2^2 - 2\langle\boldsymbol{x}, (s - s_i^*)\dot{\boldsymbol{x}}_i^*\rangle$$
$$- \left(\frac{1}{2}(1 + \kappa^2)^{1/2}(s - s_i^*)^2\right)^2$$
$$\geq \left(\frac{2\sqrt{2}}{\pi}\Delta_\varepsilon\right)^2 + (s - s_i^*)^2 - \frac{1}{4}(1 + \kappa^2)(s - s_i^*)^4$$
$$\geq \left(\frac{2\sqrt{2}}{\pi}\Delta_\varepsilon\right)^2 + \frac{3}{4}(s - s_i^*)^2$$
$$\geq \left(\frac{2}{\pi}\Delta_\varepsilon + \frac{\sqrt{3}}{2\sqrt{2}}|s - s_i^*|\right)^2. \tag{E.44}$$

Above, the second line uses the triangle inequality, the third line uses the parallelogram identity plus Lemma E.3 (first term) and (E.42) (second term), the fourth line comes from (E.43) and (E.41), and the fifth line comes from our construction that the length of each interval $I_i'(\boldsymbol{x})$ is no greater than $1/\sqrt{1 + \kappa^2}$ and therefore the same is true of $|s - s_i^*|$. The last line is an application of inequality of arithmetic and geometric means. Additionally, for any $\boldsymbol{x}, \boldsymbol{x}'$ of unit norm, one has

$$\angle(\boldsymbol{x}, \boldsymbol{x}') \geq 2\sin\left(\frac{\angle(\boldsymbol{x}, \boldsymbol{x}')}{2}\right)$$
$$= \|\boldsymbol{x} - \boldsymbol{x}'\|_2.$$

Combining this and (E.44), for all $s \in I_i'(\boldsymbol{x})$ we have

$$\angle(\boldsymbol{x}_{\sigma_i'}(s), \boldsymbol{x}) \geq \|\boldsymbol{x}_{\sigma_i'}(s) - \boldsymbol{x}\|_2 \geq \frac{2}{\pi}\Delta_\varepsilon + \frac{\sqrt{3}}{2\sqrt{2}}|s - s_i^*|$$
$$\geq \frac{1}{\sqrt{3}}\Delta_\varepsilon + \frac{1}{\sqrt{3}}|s - s_i^*|,$$

where the last line just worst-cases constants for simplicity. From Lemma G.5, $\psi^\circ$ is nonnegative and strictly decreasing, so

$$\int_{s \in I_i'(\boldsymbol{x})} \psi^\circ\Big(\angle\big(\boldsymbol{x}, \boldsymbol{x}_{\sigma_i'}(s)\big)\Big) ds$$

$$= \int_{s=s_i^*}^{s_{2,i}'(\boldsymbol{x})} \psi^\circ\Big(\angle\big(\boldsymbol{x}, \boldsymbol{x}_{\sigma_i'}(s)\big)\Big)ds + \int_{s=s_{1,i}'(\boldsymbol{x})}^{s_i^*} \psi^\circ\Big(\angle\big(\boldsymbol{x}, \boldsymbol{x}_{\sigma_i'}(s)\big)\Big)ds$$

$$\leq \int_{s=s_i^*}^{s_{2,i}'(\boldsymbol{x})} \psi^\circ\left(\frac{1}{\sqrt{3}}\Delta_\varepsilon + \frac{1}{\sqrt{3}}|s - s_i^*|\right)ds$$

$$+ \int_{s=s_{1,i}'(\boldsymbol{x})}^{s_i^*} \psi^\circ\left(\frac{1}{\sqrt{3}}\Delta_\varepsilon + \frac{1}{\sqrt{3}}|s - s_i^*|\right)ds$$

$$\leq 2\int_{s=0}^{\frac{1}{\sqrt{1+\kappa^2}}} \psi^\circ\left(\frac{1}{\sqrt{3}}\Delta_\varepsilon + \frac{1}{\sqrt{3}}s\right)ds$$

$$= 2\sqrt{3}\int_{t=\frac{1}{\sqrt{3}}\Delta_\varepsilon}^{\frac{1}{\sqrt{3}}\Delta_\varepsilon + \frac{1}{\sqrt{3}\sqrt{1+\kappa^2}}} \psi^\circ(t)dt$$

where again, the second to third line comes from the fact that our intervals has length at most $1/\sqrt{1+\kappa^2}$. From (F.11) in Lemma F.9 and a summation over all $N'(\boldsymbol{x}) \leq 4\aleph_{\varepsilon,\delta}(\mathcal{M})$ segments in the covering, there exists constant $C'$ such that when $L \geq C$,

$$\sum_{i=1}^{N'(\boldsymbol{x})} \int_{s\in I_i'(\boldsymbol{x})} \psi^\circ\Big(\angle\big(\boldsymbol{x}, \boldsymbol{x}_{\sigma_i'}(s)\big)\Big)ds$$

$$\leq \aleph_{\varepsilon,\delta}(\mathcal{M})C'n\log\left(\frac{1 + (L-3)\left(\frac{1}{\sqrt{3}}\Delta_\varepsilon + \frac{1}{\sqrt{3}\sqrt{1+\kappa^2}}\right)/(3\pi)}{1 + (L-3)\frac{1}{\sqrt{3}}\Delta_\varepsilon/(3\pi)}\right)$$

$$\leq \aleph_{\varepsilon,\delta}(\mathcal{M})C'n\log\left(1 + \frac{\frac{1}{\sqrt{1+\kappa^2}}}{\Delta_\varepsilon}\right).$$

Recalling our bound (E.40), we can thus take a supremum over $\boldsymbol{x} \in \mathcal{M}$ and conclude. ∎

**Lemma E.10.** *Let $\varepsilon \in (0,1)$, $\delta \in (0, 1-\varepsilon]$. Let $F_{\varepsilon,\delta}$ as in (E.5). There exist constants $C, C'$ such that when $L \geq C$, we have for any $\boldsymbol{x} \in \mathcal{M}$,*

$$\int_{\boldsymbol{x}'\in F_{\varepsilon,\delta}(\boldsymbol{x})} \psi^\circ\Big(\angle\big(\boldsymbol{x}, \boldsymbol{x}'\big)\Big)ds' \leq C'\mathrm{len}(\mathcal{M})n\frac{\hat{\kappa}}{\delta\sqrt{\varepsilon}}.$$

**Proof.** We have the simple bound from Lemma F.8 and decreasingness of $\psi^\circ$ from Lemma G.5, that there exists constant $C'$, with

$$\int_{\boldsymbol{x}'\in F_{\varepsilon,\delta}(\boldsymbol{x})} \psi^\circ\Big(\angle\big(\boldsymbol{x}, \boldsymbol{x}'\big)\Big)ds' \leq \mathrm{len}(\mathcal{M})\psi^\circ\left(\frac{\delta\sqrt{\varepsilon}}{\hat{\kappa}}\right)$$

$$\leq \mathrm{len}(\mathcal{M})C'n\frac{L-3}{1 + (L-3)\frac{\delta\sqrt{\varepsilon}}{\hat{\kappa}}/(3\pi)}$$

$$\leq \mathrm{len}(\mathcal{M})C'n\frac{\hat{\kappa}}{\delta\sqrt{\varepsilon}},$$

as claimed.

∎

## E.2 Certificates for the DC-Subtracted Kernel

**Proof Sketch and Organization** In Appendix E.1, we constructed a certificate for the DC subtracted kernel $\Theta^\circ$ over the subspace $S_\varepsilon$. In this section, we show that the certificate $g = g_\varepsilon[\zeta]$ defined in Theorem E.1 can also be viewed as the certificate without subspace constraints, satisfying

$$\Theta^\circ[g_\varepsilon[\zeta]] \approx \zeta.$$

As $\boldsymbol{P}_{S_\varepsilon}\boldsymbol{\Theta}^\circ[g_\varepsilon[\zeta]] = \zeta$, we only need $\boldsymbol{P}_{S_\varepsilon^\perp}\boldsymbol{\Theta}^\circ[g_\varepsilon[\zeta]]$ to be small. The subspace $S_\varepsilon$ is formed by all Fourier basis with low frequency, and thus contains functions that do not oscillate rapidly, in the sense that for any function $h$ and integer $k$

$$\|\boldsymbol{P}_{S_\varepsilon^\perp}h\|_{L^2} \lesssim \frac{\|\frac{d^k}{ds^k}h\|_{L^2}}{\dim(S_\varepsilon)^k}.$$

This argument is made rigorous in Lemma E.23; by choosing $k = 3$ and extracting the dimension of the subspace from (E.12), we obtain the estimate we are looking for. This leaves us to show the derivatives of $\boldsymbol{\Theta}^\circ[g_\varepsilon[\zeta]]$ are small compared to its norm.

The remainder of this subsection is organized as follows. We define a relevant notion of derivatives for the kernel $\Theta^\circ$ in Definition E.11. These derivatives can be represented as a function of the higher order derivatives of $\psi$ and that of the angle function (Lemma E.13). We bound the derivatives of the angle by higher order curvatures in Lemmas E.15 to E.17, and borrow results in Lemmas F.10 to F.12 that $\psi$'s higher order derivatives decrease rapidly since $\psi$ is localized when the network is deep enough. These bounds together allow us to control the $L^2$ to $L^2$ operator norm of operators corresponding to the $i$-th order derivatives of $\Theta^\circ$ in Lemmas E.18 to E.20 by geometric parameters of the manifold $\mathcal{M}$, including higher order regularity constants $M_i$ and the angle injectivity radius $\Delta_\varepsilon$. In Lemma E.22, we show that the projection operator $\boldsymbol{P}_{S_\varepsilon}$ and main invariant operator $\widehat{\boldsymbol{M}}_\varepsilon$ commute with differential operators on functions on $\mathcal{M}$, and thus the "low oscillation" property of the target function $\zeta$ can be transferred to the "low oscillation" of $g_\varepsilon[\zeta]$ and further down to that of $\boldsymbol{\Theta}[g_\varepsilon[\zeta]]$. To simplify the language, we introduce Definition E.24 to represent the required regularity property, and prove that $g_\varepsilon[\zeta]$ and $\boldsymbol{\Theta}^\circ[g_\varepsilon[\zeta]]$ satisfy such regularity in Lemmas E.25 and E.27. Finally, we get control of $\boldsymbol{P}_{S_\varepsilon^\perp}\boldsymbol{\Theta}^\circ[g_\varepsilon[\zeta]]$ in Lemma E.28.

**Definition E.11.** *For any $\boldsymbol{x}, \boldsymbol{x}' \in \mathcal{M}$, let $\sigma, \sigma' \in \{\pm 1\}$ denote the class memberships of $\boldsymbol{x}$ and $\boldsymbol{x}'$, let $s, s' \in \mathbb{R}$ be such that $\boldsymbol{x}_\sigma(s) = \boldsymbol{x}$, $\boldsymbol{x}_{\sigma'}(s') = \boldsymbol{x}'$, and write $\Theta^{(0)}(\boldsymbol{x}, \boldsymbol{x}') = \Theta^\circ(\boldsymbol{x}, \boldsymbol{x}') = \psi^\circ(\angle(\boldsymbol{x}, \boldsymbol{x}'))$. We consider higher order derivatives of the kernel with respect to a "simultaneous advance". For $i = 1, 2, 3$, define inductively*

$$\Theta^{(i)}(\boldsymbol{x}, \boldsymbol{x}') = \left[\frac{d}{dt}\Big|_0 \Theta^{(i-1)}\big(\boldsymbol{x}_\sigma(s+t), \boldsymbol{x}_{\sigma'}(s'+t)\big)\right]\Bigg|_{\boldsymbol{x}_\sigma(s)=\boldsymbol{x}, \boldsymbol{x}_{\sigma'}(s')=\boldsymbol{x}'}.$$

*Let $\boldsymbol{\Theta}^{(i)}$ denote the Fredholm integral operator associated to $\Theta^{(i)}$:*

$$\boldsymbol{\Theta}^{(i)}[h](\boldsymbol{x}) = \int_{\boldsymbol{x}' \in \mathcal{M}} \Theta^{(i)}(\boldsymbol{x}, \boldsymbol{x}')h(\boldsymbol{x}')d\boldsymbol{x}'.$$

*It is clear that these definitions do not depend on the choice of $s, s' \in \mathbb{R}$ among 'equivalent' points (c.f. (C.5) and surrounding discussion).*

**Remark E.12.** *For the moment, we have elided the issue that due to differentiability issues with the angle function $(\boldsymbol{x}, \boldsymbol{x}') \mapsto \angle(\boldsymbol{x}, \boldsymbol{x}')$, the kernels $\Theta^{(i)}$ defined in Definition E.11 may not be well-defined on all of $\mathcal{M} \times \mathcal{M}$. This issue is resolved in Lemma E.13.*

**Lemma E.13.** *Let*

$$\lambda_0(\boldsymbol{x}, \boldsymbol{x}') = \angle(\boldsymbol{x}, \boldsymbol{x}')$$

$$\lambda_{i+1}(\boldsymbol{x}, \boldsymbol{x}') = \left[\frac{d}{dt}\Big|_0 \lambda_i(\boldsymbol{x}_\sigma(s+t), \boldsymbol{x}_{\sigma'}(s'+t))\right]\Bigg|_{\boldsymbol{x}_\sigma(s)=\boldsymbol{x}, \boldsymbol{x}_{\sigma'}(s')=\boldsymbol{x}'}$$

$$= \left[\left(\frac{\partial}{\partial s} + \frac{\partial}{\partial s'}\right)\lambda_i(\boldsymbol{x}_\sigma(s), \boldsymbol{x}_{\sigma'}(s'))\right]\Bigg|_{\boldsymbol{x}_\sigma(s)=\boldsymbol{x}, \boldsymbol{x}_{\sigma'}(s')=\boldsymbol{x}'}, \quad i = 0, 1, 2.$$

*denote derivatives of the angle function with respect to a "simultaneous advance". Then when the parameterizations $\boldsymbol{x}_\sigma$ are five times continuously differentiable (as required in Appendix C.1), these functions are well-defined on $\mathcal{M} \times \mathcal{M}$.*

*In addition, the kernels $\Theta^{(i)}$ defined in Definition E.11 are well-defined on $\mathcal{M} \times \mathcal{M}$ and can be expressed in terms of the derivatives of $\psi$ and the functions $\lambda_i$ as*

$$\Theta^{(0)}(\boldsymbol{x}, \boldsymbol{x}') = \psi^\circ(\angle(\boldsymbol{x}, \boldsymbol{x}'))$$

$$\Theta^{(1)}(\boldsymbol{x}, \boldsymbol{x}') = \dot{\psi}(\angle(\boldsymbol{x}, \boldsymbol{x}'))\lambda_1(\boldsymbol{x}, \boldsymbol{x}')$$
$$\Theta^{(2)}(\boldsymbol{x}, \boldsymbol{x}') = \ddot{\psi}(\angle(\boldsymbol{x}, \boldsymbol{x}'))\lambda_1^2(\boldsymbol{x}, \boldsymbol{x}') + \dot{\psi}(\angle(\boldsymbol{x}, \boldsymbol{x}'))\lambda_2(\boldsymbol{x}, \boldsymbol{x}')$$
$$\Theta^{(3)}(\boldsymbol{x}, \boldsymbol{x}') = \dddot{\psi}(\angle(\boldsymbol{x}, \boldsymbol{x}'))\lambda_1^3(\boldsymbol{x}, \boldsymbol{x}') + 3\ddot{\psi}(\angle(\boldsymbol{x}, \boldsymbol{x}'))\lambda_2(\boldsymbol{x}, \boldsymbol{x}')\lambda_1(\boldsymbol{x}, \boldsymbol{x}')$$
$$+ \dot{\psi}(\angle(\boldsymbol{x}, \boldsymbol{x}'))\lambda_3(\boldsymbol{x}, \boldsymbol{x}'),$$

*where $\dot{\psi}, \ddot{\psi}, \dddot{\psi}$ denote the first three derivatives of $\psi^\circ$.*

**Proof.** Because the function $t \mapsto \cos^{-1}(t)$ is infinitely differentiable except at $\{-1, 1\} \subset [-1, +1]$ and $\psi$ is 3 times continuously differentiable on $[0, \pi]$ (Lemma G.5), and given the differentiability assumption on the curves and the fact that (C.1) precludes $\mathcal{M}$ from containing any antipodal points, the claim follows immediately by the chain rule except on the diagonal $\{(\boldsymbol{x}, \boldsymbol{x}) \mid \boldsymbol{x} \in \mathcal{M}\}$. Here, suppose $s, s'$ are such that $\boldsymbol{x}_\sigma(s) = \boldsymbol{x}_\sigma(s')$. Then we have $\boldsymbol{x}_\sigma(s + t) = \boldsymbol{x}_\sigma(s' + t)$ for every $t \in \mathbb{R}$. In particular, $\angle(\boldsymbol{x}_\sigma(s + t), \boldsymbol{x}_\sigma(s' + t)) = 0$ for all $t \in \mathbb{R}$, which implies that $\lambda_i(\boldsymbol{x}, \boldsymbol{x}) = 0$ for all $i$. A similar argument implies well-definedness of $\Theta^{(i)}(\boldsymbol{x}, \boldsymbol{x})$ for all $i$, which establishes the claimed formulas on all of $\mathcal{M} \times \mathcal{M}$. $\blacksquare$

**Lemma E.14.** *For points $\boldsymbol{x}, \boldsymbol{x}' \in \mathcal{M}$ and $\varepsilon \in (0, 1)$ we have*

$$\sqrt{1 - (\boldsymbol{x}^*\boldsymbol{x}')^2} \geq \begin{cases} \frac{d_\mathcal{M}(\boldsymbol{x}, \boldsymbol{x}')}{3} & d_\mathcal{M}(\boldsymbol{x}, \boldsymbol{x}') \leq \frac{1}{\hat{\kappa}} \\ \frac{2}{\pi}\Delta_\varepsilon & d_\mathcal{M}(\boldsymbol{x}, \boldsymbol{x}') \geq \frac{\sqrt{\varepsilon}}{\hat{\kappa}} \end{cases}$$

**Proof.** When $d_\mathcal{M}(\boldsymbol{x}, \boldsymbol{x}') \geq \frac{\sqrt{\varepsilon}}{\hat{\kappa}}$, from definition of the angle injectivity radius in (C.8) we have $\angle(\boldsymbol{x}, \boldsymbol{x}') \geq \Delta_\varepsilon$. From (C.1) we also have $\angle(\boldsymbol{x}, \boldsymbol{x}') \leq \pi/2$, then

$$\begin{aligned} \sqrt{1 - (\boldsymbol{x}^*\boldsymbol{x}')^2} &= \sin(\angle(\boldsymbol{x}, \boldsymbol{x}')) \\ &\geq \sin(\Delta_\varepsilon) \\ &\geq \frac{2}{\pi}\Delta_\varepsilon, \end{aligned} \tag{E.45}$$

where the first inequality comes from the monotonicity of $\sin(t)$ from $0$ to $\pi/2$. The second inequality uses concavity of $\sin$ to get $\sin(t) \geq (2/\pi)t$ for $0 \leq t \leq \pi/2$, and the fact that $\varepsilon < 1$ and hence $\Delta_\varepsilon \leq \pi/2$.

When $d_\mathcal{M}(\boldsymbol{x}, \boldsymbol{x}') \leq \frac{1}{\hat{\kappa}} \leq \frac{\pi}{2}$, assume $\boldsymbol{x}, \boldsymbol{x}'$ are parameterized by $\boldsymbol{x}_\sigma(s), \boldsymbol{x}_\sigma(s')$ separately with $|s - s'| = d_\mathcal{M}(\boldsymbol{x}, \boldsymbol{x}')$, then $|s' - s| \leq \frac{1}{\hat{\kappa}}$. Assuming without loss of generality that $s' \geq s$, using a second-order Taylor expansion and properties from Lemma E.3 gives

$$\begin{aligned} \boldsymbol{x}_\sigma(s)^*\boldsymbol{x}_\sigma(s') &= \boldsymbol{x}_\sigma(s)^* \left( \boldsymbol{x}_\sigma(s) + (s' - s)\dot{\boldsymbol{x}}_\sigma(s) + \int_{a=s}^{s'} \int_{b=s}^{a} \ddot{\boldsymbol{x}}_\sigma(b) \, db \, da \right) \\ &= 1 + \int_{a=s}^{s'} \int_{b=s}^{a} \langle \boldsymbol{x}_\sigma(s), \ddot{\boldsymbol{x}}_\sigma(b) \rangle \, db \, da \\ &= 1 + \int_{a=s}^{s'} \int_{b=s}^{a} \left\langle \boldsymbol{x}_\sigma(b) + (s - b)\dot{\boldsymbol{x}}_\sigma(b) + \int_{c=b}^{s} \int_{d=b}^{c} \ddot{\boldsymbol{x}}_\sigma(d) \, dd \, dc, \ddot{\boldsymbol{x}}_\sigma(b) \right\rangle \, db \, da \\ &= 1 - \frac{(s' - s)^2}{2} + \int_{a=s}^{s'} \int_{b=s}^{a} \int_{c=b}^{s} \int_{d=b}^{c} \ddot{\boldsymbol{x}}_\sigma(d)^* \ddot{\boldsymbol{x}}_\sigma(b) \, dd \, dc \, db \, da, \end{aligned}$$

with a Taylor expansion at $b$ used in the third line, and using the convention that for a real-valued function $f$ and numbers $a < b$, the notation $\int_b^a f(x) \, dx$ denotes the integral $-\int_a^b f(x) \, dx$. As $\hat{\kappa} = \max\{\kappa, \frac{2}{\pi}\}$, we can use the previous expression (with a bound of the integrand in the last line by $M_2$, and Lemma E.3 again) to obtain after an integration

$$|\boldsymbol{x}_\sigma(s)^*\boldsymbol{x}_\sigma(s') - 1 + \tfrac{1}{2}(s' - s)^2| \leq \frac{(s' - s)^4}{4!}(1 + \kappa^2)$$

$$\leq \frac{(s'-s)^2}{4!} \frac{1+\kappa^2}{\hat{\kappa}^2}$$

$$\leq \frac{(s'-s)^2}{4!} \left(\frac{\pi^2}{4}+1\right)$$

$$< \frac{(s'-s)^2}{6},$$

and thus

$$\sqrt{1-(\boldsymbol{x}_\sigma(s)^*\boldsymbol{x}_\sigma(s'))^2}$$

$$= \sqrt{1+\boldsymbol{x}_\sigma(s)^*\boldsymbol{x}_\sigma(s')}\sqrt{1-\boldsymbol{x}_\sigma(s)^*\boldsymbol{x}_\sigma(s')}$$

$$\geq \sqrt{\left(1+\left(1-\frac{1}{2}(s'-s)^2-\frac{1}{6}(s'-s)^2\right)\right)\left(\frac{1}{2}(s'-s)^2-\frac{1}{6}(s'-s)^2\right)}$$

$$= \sqrt{\left(2-\frac{2}{3}(s'-s)^2\right)\frac{1}{3}(s'-s)^2}$$

$$\geq \sqrt{\left(2-\frac{2}{3}\left(\frac{\pi}{2}\right)^2\right)\frac{(s'-s)^2}{3}}$$

$$> \frac{|s'-s|}{3}.$$

∎

**Lemma E.15.** *For any $\boldsymbol{x}, \boldsymbol{x}' \in \mathcal{M}$, we have*

$$|\lambda_1(\boldsymbol{x},\boldsymbol{x}')| \leq \begin{cases} \frac{7d_{\mathcal{M}}(\boldsymbol{x},\boldsymbol{x}')^3}{12}M_4 & d_{\mathcal{M}}(\boldsymbol{x},\boldsymbol{x}') \leq \frac{1}{\hat{\kappa}} \\ 2 & \forall \boldsymbol{x}, \boldsymbol{x}' \in \mathcal{M}. \end{cases}$$

**Proof.** Let $s, s'$ be such that $\boldsymbol{x} = \boldsymbol{x}_\sigma(s)$ and $\boldsymbol{x}' = \boldsymbol{x}_{\sigma'}(s')$, with $|s-s'| = d_{\mathcal{M}}(\boldsymbol{x},\boldsymbol{x}')$ when in addition $\sigma = \sigma'$. As $\angle(\boldsymbol{x},\boldsymbol{x}') = \cos^{-1}(\boldsymbol{x}^*\boldsymbol{x}')$,

$$\lambda_1(\boldsymbol{x},\boldsymbol{x}') = \frac{\partial}{\partial s}\angle(\boldsymbol{x}_\sigma(s),\boldsymbol{x}_{\sigma'}(s')) + \frac{\partial}{\partial s'}\angle(\boldsymbol{x}_\sigma(s),\boldsymbol{x}_{\sigma'}(s'))$$

$$= -\frac{\dot{\boldsymbol{x}}^*\boldsymbol{x}' + \dot{\boldsymbol{x}}'^*\boldsymbol{x}}{\sqrt{1-(\boldsymbol{x}^*\boldsymbol{x}')^2}}. \tag{E.46}$$

Notice that $\sqrt{1-(\boldsymbol{x}^*\boldsymbol{x}')^2} = \|(\boldsymbol{I}-\boldsymbol{x}\boldsymbol{x}^*)\boldsymbol{x}'\|_2$, and therefore by Lemma E.3 and Cauchy-Schwarz

$$\left|\frac{\dot{\boldsymbol{x}}^*\boldsymbol{x}'}{\sqrt{1-(\boldsymbol{x}^*\boldsymbol{x}')^2}}\right| = \left|\left\langle \dot{\boldsymbol{x}}, \frac{(\boldsymbol{I}-\boldsymbol{x}\boldsymbol{x}^*)\boldsymbol{x}'}{\|(\boldsymbol{I}-\boldsymbol{x}\boldsymbol{x}^*)\boldsymbol{x}'\|_2}\right\rangle\right| \leq 1,$$

and thus $|\lambda_1(\boldsymbol{x},\boldsymbol{x}')| \leq 2$ by symmetry.

When $d_{\mathcal{M}}(\boldsymbol{x},\boldsymbol{x}') \leq \frac{1}{\hat{\kappa}}$, we have $\sigma = \sigma'$, (as above) $|s'-s| \leq \frac{1}{\hat{\kappa}}$. By symmetry, we may assume $s' \geq s$. From Lemma E.3, we have $\dot{\boldsymbol{x}}^*\boldsymbol{x} = \dot{\boldsymbol{x}}^*\ddot{\boldsymbol{x}} = 0$, $\dot{\boldsymbol{x}}^*\dot{\boldsymbol{x}} = 1$, $\ddot{\boldsymbol{x}}^*\boldsymbol{x}^{(3)} = -\frac{1}{3}\dot{\boldsymbol{x}}^*\boldsymbol{x}^{(4)}$. In the remainder of the proof, with an abuse of notation we will write $\dot{\boldsymbol{x}} = \dot{\boldsymbol{x}}_\sigma(s)$, $\dot{\boldsymbol{x}}' = \dot{\boldsymbol{x}}_\sigma(s')$, and so on for the higher derivatives to represent the *specific* points of interest concisely. Thus by a fourth-order Taylor expansion (respectively, of $\boldsymbol{x}' = \boldsymbol{x}_\sigma(s')$ at $s$, and of $\boldsymbol{x} = \boldsymbol{x}_\sigma(s)$ at $s'$)

$$|\dot{\boldsymbol{x}}^*\boldsymbol{x}' + \dot{\boldsymbol{x}}'^*\boldsymbol{x}|$$

$$= \left|\dot{\boldsymbol{x}}^*\left(\boldsymbol{x} + (s'-s)\dot{\boldsymbol{x}} + \frac{(s'-s)^2}{2}\ddot{\boldsymbol{x}} + \frac{(s'-s)^3}{3!}\boldsymbol{x}^{(3)}\right.\right.$$

$$\left.\left. + \int_{a=s}^{s'}\int_{b=s}^{a}\int_{c=s}^{b}\int_{d=s}^{c}\boldsymbol{x}_\sigma^{(4)}(d)\,dd\,dc\,db\,da\right)\right.$$

$$+ \dot{\boldsymbol{x}}'^* \left( \boldsymbol{x}' - (s'-s)\dot{\boldsymbol{x}}' + \frac{(s'-s)^2}{2}\ddot{\boldsymbol{x}}' - \frac{(s'-s)^3}{3!}\boldsymbol{x}'^{(3)} \right.$$

$$\left. + \int_{a=s}^{s'}\int_{b=a}^{s'}\int_{c=b}^{s'}\int_{d=c}^{s'} \boldsymbol{x}_\sigma^{(4)}(d)\, dd\, dc\, db\, da \right) \Bigg|$$

$$= \left| s' - s + \dot{\boldsymbol{x}}^* \left( \frac{(s'-s)^3}{3!}\boldsymbol{x}^{(3)} + \int_{a=s}^{s'}\int_{b=s}^{a}\int_{c=s}^{b}\int_{d=s}^{c} \boldsymbol{x}_\sigma^{(4)}(d)\, dd\, dc\, db\, da \right) \right.$$

$$\left. - (s'-s) + \dot{\boldsymbol{x}}'^* \left( -\frac{(s'-s)^3}{3!}\boldsymbol{x}'^{(3)} + \int_{a=s}^{s'}\int_{b=a}^{s'}\int_{c=b}^{s'}\int_{d=c}^{s'} \boldsymbol{x}_\sigma^{(4)}(d)\, dd\, dc\, db\, da \right) \right|$$

$$\le \frac{(s'-s)^3}{3!} |\dot{\boldsymbol{x}}'^*\boldsymbol{x}'^{(3)} - \dot{\boldsymbol{x}}^*\boldsymbol{x}^{(3)}| + \frac{2(s'-s)^4}{4!}M_4$$

$$\le \frac{(s'-s)^3}{3!} \int_{a=s}^{s'} |\dot{\boldsymbol{x}}_\sigma(a)^*\boldsymbol{x}_\sigma^{(4)}(a) + \ddot{\boldsymbol{x}}_\sigma(a)^*\boldsymbol{x}_\sigma^{(3)}(a)|\, da + \frac{2(s'-s)^4}{4!}M_4$$

$$= \frac{(s'-s)^3}{3!} \int_{a=s}^{s'} \left| \frac{2}{3}\dot{\boldsymbol{x}}_\sigma(a)^*\boldsymbol{x}_\sigma^{(4)}(a) \right|\, da + \frac{2(s'-s)^4}{4!}M_4$$

$$\le \frac{7}{36}M_4(s'-s)^4. \tag{E.47}$$

Above, the first inequality uses the triangle inequality and Cauchy-Schwarz; the second inequality Taylor expands the first term in the difference at $s$ (which leads to a cancellation with the second term) and uses the triangle inequality to move the absolute value inside the integral; the following line rewrites using Lemma E.3; and then the final line uses Cauchy-Schwarz, integrates and collects constants. Using Lemma E.14, we obtain that when $d_{\mathcal{M}}(\boldsymbol{x}, \boldsymbol{x}') \le \frac{1}{\hat{\kappa}}$,

$$|\lambda_1(\boldsymbol{x}, \boldsymbol{x}')| \le \frac{7d_{\mathcal{M}}^3(\boldsymbol{x}, \boldsymbol{x}')}{12}M_4.$$

∎

**Lemma E.16.** *There exists an absolute constant $C$ such that for any $\varepsilon \in (0,1)$ and $\boldsymbol{x}, \boldsymbol{x}' \in \mathcal{M}$ we have*

$$|\lambda_2(\boldsymbol{x}, \boldsymbol{x}')| \le \begin{cases} C(M_4^2 + M_5)d_{\mathcal{M}}(\boldsymbol{x}, \boldsymbol{x}')^3, & d_{\mathcal{M}}(\boldsymbol{x}, \boldsymbol{x}') \le \frac{1}{\hat{\kappa}} \\ \frac{\pi}{4}\Delta_\varepsilon^{-1} + 2M_2, & d_{\mathcal{M}}(\boldsymbol{x}, \boldsymbol{x}') > \frac{\sqrt{\varepsilon}}{\hat{\kappa}} \end{cases}. \tag{E.48}$$

**Proof.** Let $s, s'$ be such that $\boldsymbol{x} = \boldsymbol{x}_\sigma(s)$ and $\boldsymbol{x}' = \boldsymbol{x}_{\sigma'}(s')$, with $|s - s'| = d_{\mathcal{M}}(\boldsymbol{x}, \boldsymbol{x}')$ when in addition $\sigma = \sigma'$. From (E.46),

$$\lambda_2(\boldsymbol{x}, \boldsymbol{x}') = \frac{d}{dt}\bigg|_0 \lambda_1(\boldsymbol{x}_\sigma(s+t), \boldsymbol{x}_{\sigma'}(s'+t))$$

$$= -\frac{(\dot{\boldsymbol{x}}^*\boldsymbol{x}' + \dot{\boldsymbol{x}}'^*\boldsymbol{x})^2 \boldsymbol{x}^*\boldsymbol{x}'}{(1 - (\boldsymbol{x}^*\boldsymbol{x}')^2)^{3/2}} - \frac{\dot{\boldsymbol{x}}^*\dot{\boldsymbol{x}}' + \ddot{\boldsymbol{x}}^*\boldsymbol{x}' + \ddot{\boldsymbol{x}}'^*\boldsymbol{x} + \dot{\boldsymbol{x}}'^*\dot{\boldsymbol{x}}}{\sqrt{1 - (\boldsymbol{x}^*\boldsymbol{x}')^2}}. \tag{E.49}$$

First consider the case where $d_{\mathcal{M}}(\boldsymbol{x}, \boldsymbol{x}') > \frac{\sqrt{\varepsilon}}{\hat{\kappa}}$. As $\sqrt{1 - (\boldsymbol{x}^*\boldsymbol{x}')^2} = \|(\boldsymbol{I} - \boldsymbol{x}\boldsymbol{x}^*)\boldsymbol{x}'\|_2$, we can write

$$\frac{\ddot{\boldsymbol{x}}^*\boldsymbol{x}'}{\sqrt{1 - (\boldsymbol{x}^*\boldsymbol{x}')^2}} = \left\langle \ddot{\boldsymbol{x}}, \frac{(\boldsymbol{I} - \boldsymbol{x}\boldsymbol{x}^*)\boldsymbol{x}'}{\|(\boldsymbol{I} - \boldsymbol{x}\boldsymbol{x}^*)\boldsymbol{x}'\|_2} \right\rangle + \frac{\ddot{\boldsymbol{x}}^*\boldsymbol{x}\boldsymbol{x}^*\boldsymbol{x}'}{\|(\boldsymbol{I} - \boldsymbol{x}\boldsymbol{x}^*)\boldsymbol{x}'\|_2}$$

$$= \left\langle \ddot{\boldsymbol{x}}, \frac{(\boldsymbol{I} - \boldsymbol{x}\boldsymbol{x}^*)\boldsymbol{x}'}{\|(\boldsymbol{I} - \boldsymbol{x}\boldsymbol{x}^*)\boldsymbol{x}'\|_2} \right\rangle - \frac{\boldsymbol{x}^*\boldsymbol{x}'}{\|(\boldsymbol{I} - \boldsymbol{x}\boldsymbol{x}^*)\boldsymbol{x}'\|_2}$$

using Lemma E.3. Thus following (E.46) and (E.45) and Lemmas E.3, E.14 and E.15,

$$|\lambda_2(\boldsymbol{x}, \boldsymbol{x}')| \le \left| \lambda_1(\boldsymbol{x}, \boldsymbol{x}')^2 \frac{\boldsymbol{x}^*\boldsymbol{x}'}{\sqrt{1 - (\boldsymbol{x}^*\boldsymbol{x}')^2}} \right| + \left| \frac{2\dot{\boldsymbol{x}}^*\dot{\boldsymbol{x}}' - 2\boldsymbol{x}^*\boldsymbol{x}'}{\sqrt{1 - (\boldsymbol{x}^*\boldsymbol{x}')^2}} \right|$$

$$+\left|\left\langle \ddot{\boldsymbol{x}}, \frac{(\boldsymbol{I}-\boldsymbol{x}\boldsymbol{x}^*)\boldsymbol{x}'}{\|(\boldsymbol{I}-\boldsymbol{x}\boldsymbol{x}^*)\boldsymbol{x}'\|_2}\right\rangle\right| + \left|\left\langle \ddot{\boldsymbol{x}}', \frac{(\boldsymbol{I}-\boldsymbol{x}'\boldsymbol{x}'^*)\boldsymbol{x}}{\|(\boldsymbol{I}-\boldsymbol{x}'\boldsymbol{x}'^*)\boldsymbol{x}\|_2}\right\rangle\right|$$

$$\leq (4+4)\left(\frac{2}{\pi}\Delta_\varepsilon\right)^{-1} + 2M_2$$

$$= \frac{\pi}{4}\Delta_\varepsilon^{-1} + 2M_2.$$

When $d_{\mathcal{M}}(\boldsymbol{x},\boldsymbol{x}') \leq \frac{1}{\tilde{\kappa}}$, we have $\sigma = \sigma'$ and $|s'-s| \leq \frac{1}{\tilde{\kappa}}$. By symmetry, we may assume $s' \geq s$. Following Lemma E.3, we have $\dot{\boldsymbol{x}}^*\ddot{\boldsymbol{x}} = 0$, $\boldsymbol{x}^*\ddot{\boldsymbol{x}} = -1$. In the remainder of the proof, with an abuse of notation we will write $\dot{\boldsymbol{x}} = \dot{\boldsymbol{x}}_\sigma(s)$, $\dot{\boldsymbol{x}}' = \dot{\boldsymbol{x}}_\sigma(s')$, and so on for the higher derivatives to represent the *specific* points of interest concisely. We can calculate by Taylor expansion and Lemma E.3

$$\ddot{\boldsymbol{x}}^*\boldsymbol{x}' = \ddot{\boldsymbol{x}}^*\left(\boldsymbol{x} + (s'-s)\dot{\boldsymbol{x}} + \frac{(s'-s)^2}{2}\ddot{\boldsymbol{x}} + \frac{(s'-s)^3}{6}\boldsymbol{x}^{(3)}\right.$$

$$\left. + \int_{a=s}^{s'}\int_{b=s}^{a}\int_{c=s}^{b}\int_{d=s}^{c} \boldsymbol{x}_\sigma^{(4)}(d)\,dd\,dc\,db\,da\right)$$

$$= -1 + \ddot{\boldsymbol{x}}^*\left(\frac{(s'-s)^2}{2}\ddot{\boldsymbol{x}} + \frac{(s'-s)^3}{6}\boldsymbol{x}^{(3)} + \int_{a=s}^{s'}\int_{b=s}^{a}\int_{c=s}^{b}\int_{d=s}^{c} \boldsymbol{x}_\sigma^{(4)}(d)\,dd\,dc\,db\,da\right), \quad \text{(E.50)}$$

$$\ddot{\boldsymbol{x}}'^*\boldsymbol{x} = \ddot{\boldsymbol{x}}'^*\left(\boldsymbol{x}' - (s'-s)\dot{\boldsymbol{x}}' + \frac{(s'-s)^2}{2}\ddot{\boldsymbol{x}}' - \frac{(s'-s)^3}{6}\boldsymbol{x}'^{(3)}\right.$$

$$\left. + \int_{a=s}^{s'}\int_{b=a}^{s'}\int_{c=b}^{s'}\int_{d=c}^{s'} \boldsymbol{x}_\sigma^{(4)}(d)\,dd\,dc\,db\,da\right)$$

$$= -1 + \ddot{\boldsymbol{x}}^*\left(\frac{(s'-s)^2}{2}\ddot{\boldsymbol{x}}' - \frac{(s'-s)^3}{6}\boldsymbol{x}'^{(3)} + \int_{a=s}^{s'}\int_{b=a}^{s'}\int_{c=b}^{s'}\int_{d=c}^{s'} \boldsymbol{x}_\sigma^{(4)}(d)\,dd\,dc\,db\,da\right), \quad \text{(E.51)}$$

$$\dot{\boldsymbol{x}}^*\dot{\boldsymbol{x}}' = \dot{\boldsymbol{x}}^*\left(\dot{\boldsymbol{x}} + (s'-s)\ddot{\boldsymbol{x}} + \frac{(s'-s)^2}{2}\boldsymbol{x}^{(3)} + \frac{(s'-s)^3}{6}\boldsymbol{x}^{(4)}\right.$$

$$\left. + \int_{a=s}^{s'}\int_{b=s}^{a}\int_{c=s}^{b}\int_{d=s}^{c} \boldsymbol{x}_\sigma^{(5)}(d)\,dd\,dc\,db\,da\right)$$

$$= 1 + \dot{\boldsymbol{x}}^*\left(\frac{(s'-s)^2}{2}\boldsymbol{x}^{(3)} + \frac{(s'-s)^3}{6}\boldsymbol{x}^{(4)} + \int_{a=s}^{s'}\int_{b=s}^{a}\int_{c=s}^{b}\int_{d=s}^{c} \boldsymbol{x}_\sigma^{(5)}(d)\,dd\,dc\,db\,da\right) \quad \text{(E.52)}$$

$$= 1 + \dot{\boldsymbol{x}}'^*\left(\frac{(s'-s)^2}{2}\boldsymbol{x}'^{(3)} - \frac{(s'-s)^3}{6}\boldsymbol{x}'^{(4)} + \int_{a=s}^{s'}\int_{b=a}^{s'}\int_{c=b}^{s'}\int_{d=c}^{s'} \boldsymbol{x}_\sigma^{(5)}(d)\,dd\,dc\,db\,da\right). \quad \text{(E.53)}$$

In addition

$$|\dot{\boldsymbol{x}}^*\boldsymbol{x}^{(4)} - \dot{\boldsymbol{x}}'^*\boldsymbol{x}'^{(4)}| = \left|-\int_{a=s}^{s'} \ddot{\boldsymbol{x}}(a)^*\boldsymbol{x}^{(4)}(a) + \dot{\boldsymbol{x}}(a)^*\boldsymbol{x}^{(5)}(a)\,da\right|,$$

$$\leq |s'-s|(M_2M_4 + M_5) \quad \text{(E.54)}$$

by Taylor expansion of the first term in the difference on the LHS at $s'$. From Lemma E.3, $\ddot{\boldsymbol{x}}^*\boldsymbol{x}^{(3)} = -\frac{1}{3}\dot{\boldsymbol{x}}^*\boldsymbol{x}^{(4)}$, $\ddot{\boldsymbol{x}}^*\ddot{\boldsymbol{x}} = -\dot{\boldsymbol{x}}^*\boldsymbol{x}^{(3)}$. Whence adding (E.50), (E.51), (E.52), (E.53) and applying (E.54) we get

$$|\ddot{\boldsymbol{x}}^*\boldsymbol{x}' + \boldsymbol{x}^*\ddot{\boldsymbol{x}}' + 2\dot{\boldsymbol{x}}^*\dot{\boldsymbol{x}}'| \leq \left|\frac{(s'-s)^2}{2}\left(\ddot{\boldsymbol{x}}^*\ddot{\boldsymbol{x}} + \dot{\boldsymbol{x}}^*\boldsymbol{x}^{(3)}\right) + \frac{(s'-s)^2}{2}\left(\ddot{\boldsymbol{x}}'^*\ddot{\boldsymbol{x}}' + \dot{\boldsymbol{x}}'\boldsymbol{x}'^{(3)}\right)\right|$$

$$+ \frac{(s'-s)^3}{6}\left(1 - \frac{1}{3}\right)|\dot{\boldsymbol{x}}^*\boldsymbol{x}^{(4)} - \dot{\boldsymbol{x}}'^*\boldsymbol{x}'^{(4)}|$$

$$+ \frac{2(s'-s)^4}{4!}(M_2 M_4 + M_5)$$

$$\leq \frac{(s'-s)^4}{9}(M_2 M_4 + M_5) + \frac{(s'-s)^4}{12}(M_2 M_4 + M_5)$$

$$= \frac{7(s'-s)^4}{36}(M_2 M_4 + M_5). \tag{E.55}$$

From Lemma E.3, $M_2 \leq M_4$. Plugging (E.55) and (E.47) into the bound (E.49) and using Lemma E.14, we obtain that when $d_{\mathcal{M}}(\boldsymbol{x}, \boldsymbol{x}') \leq \frac{1}{\hat{\kappa}} \leq \frac{\pi}{2}$

$$|\lambda_2(\boldsymbol{x}, \boldsymbol{x}')| \leq \left(\frac{3}{|s'-s|}\right)^3 \left(\frac{7 M_4 |s'-s|^4}{36}\right)^2 + \frac{3}{|s'-s|}\frac{7(M_2 M_4 + M_5)}{36}|s'-s|^4$$

$$= \frac{49}{48}M_4^2|s'-s|^5 + \frac{7}{12}(M_2 M_4 + M_5)|s'-s|^3$$

$$\leq C(M_4^2 + M_5)d_{\mathcal{M}}^3(\boldsymbol{x}, \boldsymbol{x}')$$

for some absolute constant $C > 0$. ∎

**Lemma E.17.** *There exists an absolute constant $C > 0$ such that for any $\varepsilon \in (0,1)$ and $\boldsymbol{x}, \boldsymbol{x}' \in \mathcal{M}$ we have*

$$|\lambda_3(\boldsymbol{x}, \boldsymbol{x}')| \leq \begin{cases} C(M_4^3 + M_4 M_5 + M_3 M_4)d_{\mathcal{M}}(\boldsymbol{x}, \boldsymbol{x}')^3, & d_{\mathcal{M}}(\boldsymbol{x}, \boldsymbol{x}') \leq \frac{1}{\hat{\kappa}} \\ \frac{3\pi^2}{4}\Delta_\varepsilon^{-2} + 9\pi M_2 \Delta_\varepsilon^{-1} + 2M_3 + 8, & d_{\mathcal{M}}(\boldsymbol{x}, \boldsymbol{x}') > \frac{\sqrt{\varepsilon}}{\hat{\kappa}} \end{cases}.$$

**Proof.**

Let $s, s'$ be such that $\boldsymbol{x} = \boldsymbol{x}_\sigma(s)$ and $\boldsymbol{x}' = \boldsymbol{x}_{\sigma'}(s')$, with $|s - s'| = d_{\mathcal{M}}(\boldsymbol{x}, \boldsymbol{x}')$ when in addition $\sigma = \sigma'$. Then from (E.49),

$$\lambda_3(\boldsymbol{x}, \boldsymbol{x}') = \frac{d}{dt}\bigg|_0 \lambda^2(\boldsymbol{x}_\sigma(s+t), \boldsymbol{x}_{\sigma'}(s'+t))$$

$$= -\frac{(\dot{\boldsymbol{x}}^*\boldsymbol{x}' + \dot{\boldsymbol{x}}'^*\boldsymbol{x})^3}{(1 - (\boldsymbol{x}^*\boldsymbol{x}')^2)^{3/2}} - 3\frac{(\dot{\boldsymbol{x}}^*\boldsymbol{x}' + \dot{\boldsymbol{x}}'^*\boldsymbol{x})^3 (\boldsymbol{x}^*\boldsymbol{x}')^2}{(1 - (\boldsymbol{x}^*\boldsymbol{x}')^2)^{5/2}}$$

$$- 3\frac{(\dot{\boldsymbol{x}}^*\boldsymbol{x}' + \dot{\boldsymbol{x}}'^*\boldsymbol{x})\,\boldsymbol{x}^*\boldsymbol{x}'\,(2\dot{\boldsymbol{x}}^*\dot{\boldsymbol{x}}' + \ddot{\boldsymbol{x}}^*\boldsymbol{x}' + \ddot{\boldsymbol{x}}'^*\boldsymbol{x})}{(1 - (\boldsymbol{x}^*\boldsymbol{x}')^2)^{3/2}} \tag{E.56}$$

$$- \frac{3\dot{\boldsymbol{x}}^*\ddot{\boldsymbol{x}}' + 3\ddot{\boldsymbol{x}}^*\dot{\boldsymbol{x}}' + \boldsymbol{x}^{(3)*}\boldsymbol{x}' + \boldsymbol{x}'^{(3)*}\boldsymbol{x}}{\sqrt{1 - (\boldsymbol{x}^*\boldsymbol{x}')^2}}.$$

When $d_{\mathcal{M}}(\boldsymbol{x}, \boldsymbol{x}') > \frac{\sqrt{\varepsilon}}{\hat{\kappa}}$, as $\sqrt{1 - (\boldsymbol{x}^*\boldsymbol{x}')^2} = \|(\boldsymbol{I} - \boldsymbol{x}\boldsymbol{x}^*)\boldsymbol{x}'\|_2$ and from Lemma E.3 $\boldsymbol{x}^{(3)*}\boldsymbol{x} = 0$,

$$\frac{\boldsymbol{x}^{(3)*}\boldsymbol{x}'}{\sqrt{1 - (\boldsymbol{x}^*\boldsymbol{x}')^2}} = \left\langle \boldsymbol{x}^{(3)}, \frac{(\boldsymbol{I} - \boldsymbol{x}\boldsymbol{x}^*)\boldsymbol{x}'}{\|(\boldsymbol{I} - \boldsymbol{x}\boldsymbol{x}^*)\boldsymbol{x}'\|_2} \right\rangle.$$

Thus from (E.46), (E.49), (E.45), Lemma E.14, Lemma E.15 and Lemma E.16,

$$|\lambda_3(\boldsymbol{x}, \boldsymbol{x}')| \leq \left|\lambda_1(\boldsymbol{x}, \boldsymbol{x}')^3\right| + \left|3\lambda_1(\boldsymbol{x}, \boldsymbol{x}')\lambda_2(\boldsymbol{x}, \boldsymbol{x}')\frac{\boldsymbol{x}^*\boldsymbol{x}'}{\sqrt{1 - (\boldsymbol{x}^*\boldsymbol{x}')^2}}\right| + \left|\frac{3\dot{\boldsymbol{x}}^*\ddot{\boldsymbol{x}}' + 3\ddot{\boldsymbol{x}}^*\dot{\boldsymbol{x}}'}{\sqrt{1 - (\boldsymbol{x}^*\boldsymbol{x}')^2}}\right| \quad \text{(E.57)}$$

$$+ \left|\left\langle \boldsymbol{x}^{(3)}, \frac{(\boldsymbol{I} - \boldsymbol{x}\boldsymbol{x}^*)\boldsymbol{x}'}{\|(\boldsymbol{I} - \boldsymbol{x}\boldsymbol{x}^*)\boldsymbol{x}'\|_2} \right\rangle\right| + \left|\left\langle \boldsymbol{x}'^{(3)}, \frac{(\boldsymbol{I} - \boldsymbol{x}'\boldsymbol{x}'^*)\boldsymbol{x}}{\|(\boldsymbol{I} - \boldsymbol{x}'\boldsymbol{x}'^*)\boldsymbol{x}\|_2} \right\rangle\right|$$

$$\leq 8 + 6\left(\frac{\pi}{4}\Delta_\varepsilon^{-1} + 2M_2\right)\left(\frac{2}{\pi}\Delta_\varepsilon\right)^{-1} + 6M_2\left(\frac{2}{\pi}\Delta_\varepsilon\right)^{-1} + 2M_3$$

$$\leq \frac{3\pi^2}{4}\Delta_\varepsilon^{-2} + 9\pi M_2 \Delta_\varepsilon^{-1} + 2M_3 + 8.$$

When $d_{\mathcal{M}}(\boldsymbol{x}, \boldsymbol{x}') \le \frac{1}{\bar{\kappa}}$, we have $\sigma = \sigma'$ and $|s' - s| \le \frac{1}{\bar{\kappa}}$. By symmetry, we may assume $s' \ge s$. Following Lemma E.3, $\dot{\boldsymbol{x}}^* \ddot{\boldsymbol{x}} = 0$ and $\ddot{\boldsymbol{x}}^* \ddot{\boldsymbol{x}} = -\dot{\boldsymbol{x}}^* \boldsymbol{x}^{(3)}$. In the remainder of the proof, with an abuse of notation we will write $\dot{\boldsymbol{x}} = \dot{\boldsymbol{x}}_\sigma(s)$, $\dot{\boldsymbol{x}}' = \dot{\boldsymbol{x}}_\sigma(s')$, and so on for the higher derivatives to represent the *specific* points of interest concisely. Because we can reuse bounds for lower-order $\lambda_i$ terms to bound the first three terms in (E.56), we will focus on controlling the last term. We can calculate by Taylor expansion

$$
3\dot{\boldsymbol{x}}'^* \ddot{\boldsymbol{x}} + 3\ddot{\boldsymbol{x}}'^* \dot{\boldsymbol{x}} + \boldsymbol{x}'^{(3)*} \boldsymbol{x} + \boldsymbol{x}^{(3)*} \boldsymbol{x}'
$$

$$
= \; 3\ddot{\boldsymbol{x}}^* \left( \dot{\boldsymbol{x}} + (s'-s)\ddot{\boldsymbol{x}} + \frac{(s'-s)^2}{2} \boldsymbol{x}^{(3)} + \frac{(s'-s)^3}{6} \boldsymbol{x}^{(4)} \right.
$$
$$
\left. + \int_{a=s}^{s'} \int_{b=s}^{a} \int_{c=s}^{b} \int_{d=s}^{c} \boldsymbol{x}_\sigma^{(5)}(d)\, dd\, dc\, db\, da \right)
$$

$$
+ 3\ddot{\boldsymbol{x}}'^* \left( \dot{\boldsymbol{x}}' - (s'-s)\ddot{\boldsymbol{x}}' + \frac{(s'-s)^2}{2} \boldsymbol{x}'^{(3)} - \frac{(s'-s)^3}{6} \boldsymbol{x}'^{(4)} \right.
$$
$$
\left. + \int_{a=s}^{s'} \int_{b=a}^{s'} \int_{c=b}^{s'} \int_{d=c}^{s'} \boldsymbol{x}_\sigma^{(5)}(d)\, dd\, dc\, db\, da \right)
$$

$$
+ \boldsymbol{x}^{(3)*} \left( \boldsymbol{x} + (s'-s)\dot{\boldsymbol{x}} + \frac{(s'-s)^2}{2} \ddot{\boldsymbol{x}} + \frac{(s'-s)^3}{6} \boldsymbol{x}^{(3)} \right.
$$
$$
\left. + \int_{a=s}^{s'} \int_{b=s}^{a} \int_{c=s}^{b} \int_{d=s}^{c} \boldsymbol{x}_\sigma^{(4)}(d)\, dd\, dc\, db\, da \right)
$$

$$
+ \boldsymbol{x}'^{(3)*} \left( \boldsymbol{x}' - (s'-s)\dot{\boldsymbol{x}}' + \frac{(s'-s)^2}{2} \ddot{\boldsymbol{x}}' - \frac{(s'-s)^3}{6} \boldsymbol{x}'^{(3)} \right.
$$
$$
\left. + \int_{a=s}^{s'} \int_{b=a}^{s'} \int_{c=b}^{s'} \int_{d=c}^{s'} \boldsymbol{x}_\sigma^{(4)}(d)\, dd\, dc\, db\, da \right)
$$

$$
= 2(s'-s)\left(\ddot{\boldsymbol{x}}^* \ddot{\boldsymbol{x}} - \ddot{\boldsymbol{x}}'^* \ddot{\boldsymbol{x}}'\right) + 2(s'-s)^2 \left( \ddot{\boldsymbol{x}}^* \boldsymbol{x}^{(3)} + \ddot{\boldsymbol{x}}'^* \boldsymbol{x}'^{(3)} \right)
$$

$$
+ \frac{(s'-s)^3}{6} \left( 3\ddot{\boldsymbol{x}}^* \boldsymbol{x}^{(4)} - 3\ddot{\boldsymbol{x}}'^* \boldsymbol{x}'^{(4)} + \boldsymbol{x}^{(3)*} \boldsymbol{x}^{(3)} - \boldsymbol{x}'^{(3)*} \boldsymbol{x}'^{(3)} \right)
$$

$$
+ 3\ddot{\boldsymbol{x}}^* \int_{a=s}^{s'} \int_{b=s}^{a} \int_{c=s}^{b} \int_{d=s}^{c} \boldsymbol{x}_\sigma^{(5)}(d)\, dd\, dc\, db\, da
$$

$$
+ 3\ddot{\boldsymbol{x}}'^* \int_{a=s}^{s'} \int_{b=a}^{s'} \int_{c=b}^{s'} \int_{d=c}^{s'} \boldsymbol{x}_\sigma^{(5)}(d)\, dd\, dc\, db\, da
$$

$$
+ \boldsymbol{x}^{(3)*} \int_{a=s}^{s'} \int_{b=s}^{a} \int_{c=s}^{b} \int_{d=s}^{c} \boldsymbol{x}_\sigma^{(4)}(d)\, dd\, dc\, db\, da
$$

$$
+ \boldsymbol{x}'^{(3)*} \int_{a=s}^{s'} \int_{b=a}^{s'} \int_{c=b}^{s'} \int_{d=c}^{s'} \boldsymbol{x}_\sigma^{(4)}(d)\, dd\, dc\, db\, da. \tag{E.58}
$$

We expand the first term by successive Taylor expansion

$$
\left| \left( \ddot{\boldsymbol{x}}^* \ddot{\boldsymbol{x}} - \ddot{\boldsymbol{x}}'^* \ddot{\boldsymbol{x}}' \right) + (s'-s)\left( \ddot{\boldsymbol{x}}^* \boldsymbol{x}^{(3)} + \ddot{\boldsymbol{x}}'^* \boldsymbol{x}'^{(3)} \right) \right|
$$

$$
= \left| -\int_{a=s}^{s'} 2\ddot{\boldsymbol{x}}_\sigma(a)^* \boldsymbol{x}_\sigma^{(3)}(a)\, da + (s'-s)\left( \ddot{\boldsymbol{x}}^* \boldsymbol{x}^{(3)} + \ddot{\boldsymbol{x}}'^* \boldsymbol{x}'^{(3)} \right) \right|
$$

$$
= \left| \int_{a=s}^{s'} \left[ \left( \ddot{\boldsymbol{x}}'^* \boldsymbol{x}'^{(3)} - \ddot{\boldsymbol{x}}_\sigma(a)^* \boldsymbol{x}_\sigma^{(3)}(a) \right) - \left( \ddot{\boldsymbol{x}}_\sigma(a)^* \boldsymbol{x}_\sigma^{(3)}(a) - \ddot{\boldsymbol{x}}^* \boldsymbol{x}^{(3)} \right) \right] da \right|
$$

$$
= \left| \int_{a=s}^{s'} \int_{b=a}^{s'} \left( \ddot{\boldsymbol{x}}_\sigma(b)^* \boldsymbol{x}_\sigma^{(4)}(b) + \boldsymbol{x}_\sigma^{(3)}(b)^* \boldsymbol{x}_\sigma^{(3)}(b) \right) db\, da \right|
$$

$$- \int_{a=s}^{s'} \int_{b=s}^{a} \left( \ddot{\boldsymbol{x}}_\sigma(b)^* \boldsymbol{x}_\sigma^{(4)}(b) + \boldsymbol{x}_\sigma^{(3)}(b)^* \boldsymbol{x}_\sigma^{(3)}(b) \right) \, db \, da \Bigg|.$$

$$= \left| \int_{b=s}^{s'} \int_{a=s}^{b} \left( \ddot{\boldsymbol{x}}_\sigma(b)^* \boldsymbol{x}_\sigma^{(4)}(b) + \boldsymbol{x}_\sigma^{(3)}(b)^* \boldsymbol{x}_\sigma^{(3)}(b) \right) \, da \, db \right.$$

$$\left. - \int_{b=s}^{s'} \int_{a=b}^{s'} \left( \ddot{\boldsymbol{x}}_\sigma(b)^* \boldsymbol{x}_\sigma^{(4)}(b) + \boldsymbol{x}_\sigma^{(3)}(b)^* \boldsymbol{x}_\sigma^{(3)}(b) \right) \, da \, db \right|$$

$$= \left| \int_{b=s}^{s'} ((b-s) - (s'-b)) \left( \ddot{\boldsymbol{x}}_\sigma(b)^* \boldsymbol{x}_\sigma^{(4)}(b) + \boldsymbol{x}_\sigma^{(3)}(b)^* \boldsymbol{x}_\sigma^{(3)}(b) \right) \, db \right|.$$

Above, in the fourth equality we rewrite the preceding integrals by switching the limits of integration; the fifth equality then just integrates over $a$. As $2b - s - s'$ stays positive when $b > (s + s')/2$ and negative otherwise, we divide the integral into two parts, change variables using $b' = s + s' - b$ and get

$$\left| (\ddot{\boldsymbol{x}}^* \ddot{\boldsymbol{x}} - \ddot{\boldsymbol{x}}'^* \ddot{\boldsymbol{x}}') + (s' - s) \left( \ddot{\boldsymbol{x}}^* \boldsymbol{x}^{(3)} + \ddot{\boldsymbol{x}}'^* \boldsymbol{x}'^{(3)} \right) \right|$$

$$= \left| \int_{b=(s+s')/2}^{s'} (2b - s - s') \left( \ddot{\boldsymbol{x}}_\sigma(b)^* \boldsymbol{x}_\sigma^{(4)}(b) + \boldsymbol{x}_\sigma^{(3)}(b)^* \boldsymbol{x}_\sigma^{(3)}(b) \right) \, db \right.$$

$$\left. - \int_{b=s}^{(s+s')/2} (s + s' - 2b) \left( \ddot{\boldsymbol{x}}_\sigma(b)^* \boldsymbol{x}_\sigma^{(4)}(b) + \boldsymbol{x}_\sigma^{(3)}(b)^* \boldsymbol{x}_\sigma^{(3)}(b) \right) \, db \right|$$

$$= \left| \int_{b'=s}^{(s+s')/2} (s + s' - 2b') \left( \ddot{\boldsymbol{x}}_\sigma(s + s' - b')^* \boldsymbol{x}_\sigma^{(4)}(s + s' - b') \right. \right.$$

$$\left. + \boldsymbol{x}_\sigma^{(3)}(s + s' - b')^* \boldsymbol{x}_\sigma^{(3)}(s + s' - b') \right) db'$$

$$\left. - \int_{b=s}^{(s+s')/2} (s + s' - 2b) \left( \ddot{\boldsymbol{x}}_\sigma(b)^* \boldsymbol{x}_\sigma^{(4)}(b) + \boldsymbol{x}_\sigma^{(3)}(b)^* \boldsymbol{x}_\sigma^{(3)}(b) \right) \, db \right|$$

$$= \left| \int_{b=s}^{(s+s')/2} (s + s' - 2b) \int_{c=b}^{s+s'-b} \left( \ddot{\boldsymbol{x}}_\sigma(c)^* \boldsymbol{x}_\sigma^{(5)}(c) + 3 \boldsymbol{x}_\sigma^{(3)}(c)^* \boldsymbol{x}_\sigma^{(4)}(c) \right) \, dc \, db \right|$$

$$\leq \left| \int_{b=s}^{(s+s')/2} (s + s' - 2b)^2 \left( M_2 M_5 + 3 M_3 M_4 \right) \, db \, da \right|$$

$$= - \left( M_2 M_5 + 3 M_3 M_4 \right) \frac{(s + s' - 2b)^3}{6} \Bigg|_{b=s}^{(s+s')/2}$$

$$= \frac{(s' - s)^3}{6} \left( M_2 M_5 + 3 M_3 M_4 \right) \tag{E.59}$$

We use Taylor expansion again for the second term of (E.58)

$$3 \ddot{\boldsymbol{x}}^* \boldsymbol{x}^{(4)} - 3 \ddot{\boldsymbol{x}}'^* \boldsymbol{x}'^{(4)} + \boldsymbol{x}^{(3)*} \boldsymbol{x}^{(3)} - \boldsymbol{x}'^{(3)*} \boldsymbol{x}'^{(3)} = - \int_{a=s}^{s'} \left[ 3 \ddot{\boldsymbol{x}}_\sigma(a)^* \boldsymbol{x}_\sigma^{(5)}(a) + 5 \boldsymbol{x}_\sigma^{(3)}(a) \boldsymbol{x}_\sigma^{(4)}(a) \right] \, da. \tag{E.60}$$

Plug (E.59) and (E.60) back to (E.58) and we conclude that

$$\left| 3 \dot{\boldsymbol{x}}'^* \ddot{\boldsymbol{x}} + 3 \ddot{\boldsymbol{x}}'^* \dot{\boldsymbol{x}} + \boldsymbol{x}'^{(3)*} \boldsymbol{x} + \boldsymbol{x}^{(3)*} \boldsymbol{x}' \right|$$

$$\leq \frac{(s' - s)^4}{3} \left( M_2 M_5 + 3 M_3 M_4 \right) + \frac{(s' - s)^4}{6} \left( 3 M_2 M_5 + 5 M_3 M_4 \right)$$

$$+ \frac{(s' - s)^4}{4!} \left( 6 M_2 M_5 + 2 M_3 M_4 \right)$$

$$= \frac{(s' - s)^4}{12} \left( 13 M_2 M_5 + 23 M_3 M_4 \right). \tag{E.61}$$

As from Lemma E.3 $M_2 \le M_4$, when $|s' - s| \le \frac{1}{\hat{\kappa}} \le \frac{\pi}{2}$, plugging (E.47), (E.55), (E.61), and Lemma E.14 into (E.56), we have

$$\lambda_3(\boldsymbol{x}, \boldsymbol{x}') \le \left(\frac{|s'-s|}{3}\right)^{-3}\left(\frac{7}{36}M_4|s'-s|^4\right)^3 + 3\left(\frac{|s'-s|}{3}\right)^{-5}\left(\frac{7}{36}M_4|s'-s|^4\right)^3$$

$$+ 3\left(\frac{|s'-s|}{3}\right)^{-3}\left(\frac{7}{36}M_4|s'-s|^4\right)\left(\frac{7|s'-s|^4}{36}(M_2M_4+M_5)\right)$$

$$+ \left(\frac{|s'-s|}{3}\right)^{-1}\frac{|s'-s|^4}{12}(13M_2M_5 + 23M_3M_4)$$

$$= (\frac{343}{1728}|s'-s|^9 + \frac{343}{64}|s'-s|^7)M_4^3 + \frac{49}{16}M_4(M_2M_4+M_5)|s'-s|^5$$

$$+ \frac{1}{4}|s'-s|^3(13M_2M_5 + 23M_3M_4)$$

$$\lesssim (M_4^3 + M_4M_5 + M_3M_4)d_{\mathcal{M}}^3(\boldsymbol{x},\boldsymbol{x}'),$$

where the last line uses the fact that we can adjust constants to keep only the lowest-order term involving the distance, given that the distance is bounded. ∎

**Lemma E.18.** *For $\varepsilon \in (0, \frac{3}{4})$, there exist positive constants $C, C_1$ such that when $L \ge C$, we have*

$$\|\boldsymbol{\Theta}^{(1)}\|_{L^2 \to L^2} \le P_1(M_4, \mathrm{len}(\mathcal{M}), \Delta_\varepsilon^{-1})\, n,$$

*where $P_1(M_4, \mathrm{len}(\mathcal{M}), \Delta_\varepsilon^{-1}) = C_1\left(M_4 + \mathrm{len}(\mathcal{M})\Delta_\varepsilon^{-2}\right)$ is a polynomial in $M_4, \mathrm{len}(\mathcal{M})$ and $\Delta_\varepsilon^{-1}$.*

**Proof.** From Lemma E.5 and Lemma E.13 we have

$$\|\boldsymbol{\Theta}^{(1)}\|_{L^2 \to L^2} \le \sup_{\boldsymbol{x} \in \mathcal{M}} \int_{\boldsymbol{x}' \in \mathcal{M}} |\Theta^{(1)}(\boldsymbol{x}, \boldsymbol{x}')| d\boldsymbol{x}'$$

$$= \sup_{\boldsymbol{x} \in \mathcal{M}} \int_{\boldsymbol{x}' \in \mathcal{M}} |\dot{\psi}(\angle(\boldsymbol{x}, \boldsymbol{x}'))\lambda_1(\boldsymbol{x}, \boldsymbol{x}')| d\boldsymbol{x}'$$

$$\le \sup_{\boldsymbol{x} \in \mathcal{M}} \int_{\boldsymbol{x}' \in \mathcal{M}} |\dot{\psi}(\angle(\boldsymbol{x}, \boldsymbol{x}'))||\lambda_1(\boldsymbol{x}, \boldsymbol{x}')| d\boldsymbol{x}'$$

Lemmas E.15 and F.10 provide us the control for $\dot{\psi}$ and $\lambda_1$. From Lemma F.10, there exist constants $C, C_1$, such that when $L > C$, we have

$$\max_{t \ge r} |\dot{\psi}(t)| \le \frac{C_1 n}{r^2}.$$

From Lemma E.15, we have

$$|\lambda_1(\boldsymbol{x}, \boldsymbol{x}')| \le \begin{cases} \frac{7d_{\mathcal{M}}(\boldsymbol{x},\boldsymbol{x}')^3}{12}M_4 & d_{\mathcal{M}}(\boldsymbol{x}, \boldsymbol{x}') \le \frac{1}{\hat{\kappa}} \\ 2 & \forall \boldsymbol{x}, \boldsymbol{x}' \in \mathcal{M}. \end{cases}$$

In order to get a lower bound for the angle $\angle(\boldsymbol{x}, \boldsymbol{x}')$, for a fixed point $\boldsymbol{x} \in \mathcal{M}$, we decompose the integral into nearby piece $\mathcal{N}(\boldsymbol{x}) = \{\boldsymbol{x}' \in \mathcal{M} \,|\, d_{\mathcal{M}}(\boldsymbol{x}, \boldsymbol{x}') \le \frac{\sqrt{\varepsilon}}{\hat{\kappa}}\}$ and faraway piece $\mathcal{F}(\boldsymbol{x}) = \{\boldsymbol{x}' \in \mathcal{M} \,|\, d_{\mathcal{M}}(\boldsymbol{x}, \boldsymbol{x}') > \frac{\sqrt{\varepsilon}}{\hat{\kappa}}\}$ and have

$$\|\boldsymbol{\Theta}^{(1)}\|_{L^2 \to L^2} \le \sup_{\boldsymbol{x} \in \mathcal{M}} \left( \int_{\boldsymbol{x}' \in \mathcal{N}(\boldsymbol{x})} |\dot{\psi}(\angle(\boldsymbol{x}, \boldsymbol{x}'))||\lambda_1(\boldsymbol{x}, \boldsymbol{x}')| d\boldsymbol{x}' \right.$$

$$\left. + \int_{\boldsymbol{x}' \in \mathcal{F}(\boldsymbol{x})} |\dot{\psi}(\angle(\boldsymbol{x}, \boldsymbol{x}'))||\lambda_1(\boldsymbol{x}, \boldsymbol{x}')| d\boldsymbol{x}' \right). \tag{E.62}$$

Then for any point $\boldsymbol{x}'$ in faraway piece $\mathcal{F}(\boldsymbol{x})$, $d_{\mathcal{M}}(\boldsymbol{x}, \boldsymbol{x}') > \frac{\sqrt{\varepsilon}}{\hat{\kappa}}$, we have $\angle(\boldsymbol{x}, \boldsymbol{x}') \ge \Delta_\varepsilon$ from (C.8) with $\Delta_\varepsilon \le \frac{\sqrt{\varepsilon}}{\hat{\kappa}} \le \frac{\pi}{2}$. From Lemmas E.15 and F.10 we get

$$\int_{\boldsymbol{x}' \in \mathcal{F}(\boldsymbol{x})} |\dot{\psi}(\angle(\boldsymbol{x}, \boldsymbol{x}'))||\lambda_1(\boldsymbol{x}, \boldsymbol{x}')| d\boldsymbol{x}' \le \int_{\boldsymbol{x}' \in \mathcal{F}(\boldsymbol{x})} \frac{C_1 n}{\Delta_\varepsilon^2} |\lambda_1(\boldsymbol{x}, \boldsymbol{x}')| d\boldsymbol{x}'$$

$$\leq \operatorname{len}(\mathcal{M})\left(2\frac{C_1 n}{\Delta_\varepsilon^2}\right)$$

$$\leq C'\Delta_\varepsilon^{-2}\operatorname{len}(\mathcal{M})n \tag{E.63}$$

for some constant $C'$.

For the integral over nearby piece, let $s, \sigma$ be such that $\boldsymbol{x} = \boldsymbol{x}_\sigma(s)$. Follow Lemma E.4 we have $\angle(\boldsymbol{x}_\sigma(s), \boldsymbol{x}_\sigma(s')) \geq (1-\varepsilon)|s-s'|$ when $|s-s'| \leq \frac{\sqrt{\varepsilon}}{\hat{\kappa}}$. As $d(\boldsymbol{x}, \boldsymbol{x}_\sigma(s')) \leq |s-s'|$, from Lemmas E.15 and F.10 we get

$$
\begin{aligned}
\int_{\boldsymbol{x}'\in\mathcal{N}(\boldsymbol{x})}|\dot\psi(\angle(\boldsymbol{x},\boldsymbol{x}'))||\lambda_1(\boldsymbol{x},\boldsymbol{x}')|d\boldsymbol{x}' &\leq \int_{s'=s-\frac{\sqrt{\varepsilon}}{\hat\kappa}}^{s-\frac{\sqrt{\varepsilon}}{\hat\kappa}}|\dot\psi(\angle(\boldsymbol{x},\boldsymbol{x}_\sigma(s')))||\lambda_1(\boldsymbol{x},\boldsymbol{x}_\sigma(s'))|ds'\\
&\leq \int_{s'=s-\frac{\sqrt{\varepsilon}}{\hat\kappa}}^{s-\frac{\sqrt{\varepsilon}}{\hat\kappa}}\frac{C_1 n}{(1-\varepsilon)^2|s'-s|^2}|\lambda_1(\boldsymbol{x},\boldsymbol{x}_\sigma(s'))|ds'\\
&\leq \int_{s'=s-\frac{\sqrt{\varepsilon}}{\hat\kappa}}^{s-\frac{\sqrt{\varepsilon}}{\hat\kappa}}\frac{C_1 n}{(1-\varepsilon)^2|s'-s|^2}\left|\frac{7|s'-s|^3}{12}M_4\right|ds'\\
&\leq C''M_4 n\int_{s'=s-\frac{\sqrt{\varepsilon}}{\hat\kappa}}^{s-\frac{\sqrt{\varepsilon}}{\hat\kappa}}|s'-s|ds'\\
&\leq C''M_4 n\left(\frac{\sqrt{\varepsilon}}{\hat\kappa}\right)^2\\
&\leq C''M_4 n \tag{E.64}
\end{aligned}
$$

for some constant $C''$, where the fourth line comes from $\varepsilon < \frac{3}{4}$ and the last line comes from $\hat\kappa \geq \frac{2}{\pi}$ from definition. Plugging (E.63) and (E.64) back in (E.62) proves the claim. ∎

**Lemma E.19.** *For $\varepsilon \in (0, \frac{3}{4})$, there exist positive constants $C, C_2$ such that when $L \geq C$, we have*

$$\|\boldsymbol{\Theta}^{(2)}\|_{L^2\to L^2} \leq P_2(M_2, M_4, M_5, \operatorname{len}(\mathcal{M}), \Delta_\varepsilon^{-1})n$$

*where*

$$P_2(M_2, M_4, M_5, \operatorname{len}(\mathcal{M}), \Delta_\varepsilon^{-1}) = C_2\left(M_4^2 + M_5 + \operatorname{len}(\mathcal{M})\left(\Delta_\varepsilon^{-3} + M_2\Delta_\varepsilon^{-2}\right)\right)$$

*is a polynomial in $M_2, M_4, M_5, \operatorname{len}(\mathcal{M})$ and $\Delta_\varepsilon^{-1}$.*

**Proof.** From Lemma E.5 we have

$$\|\boldsymbol{\Theta}^{(2)}\|_{L^2\to L^2} = \sup_{\boldsymbol{x}\in\mathcal{M}}\int_{\boldsymbol{x}'\in\mathcal{M}}|\Theta^{(2)}(\boldsymbol{x},\boldsymbol{x}')|d\boldsymbol{x}'.$$

From Lemma E.13, we know

$$
\begin{aligned}
|\Theta^{(2)}(\boldsymbol{x},\boldsymbol{x}')| &= |\ddot\psi(\angle(\boldsymbol{x},\boldsymbol{x}'))\lambda_1^2(\boldsymbol{x},\boldsymbol{x}') + \dot\psi(\angle(\boldsymbol{x},\boldsymbol{x}'))\lambda_2(\boldsymbol{x},\boldsymbol{x}')|\\
&\leq |\ddot\psi(\angle(\boldsymbol{x},\boldsymbol{x}'))||\lambda_1^2(\boldsymbol{x},\boldsymbol{x}')| + |\dot\psi(\angle(\boldsymbol{x},\boldsymbol{x}'))||\lambda_2(\boldsymbol{x},\boldsymbol{x}')|.
\end{aligned}
$$

Lemmas F.10 and F.11 provide bounds for derivatives of $\psi(t)$: there exist constants $C, C_1, C_2$, such that when $L > C$, we have

$$\max_{t\geq r}|\dot\psi(t)| \leq \frac{C_1 n}{r^2}$$

and

$$\max_{t\geq r}|\ddot\psi(t)| \leq \frac{C_2 n}{r^3}.$$

To utilize the bound above, we need to get a lower bound for the angle $\angle(\boldsymbol{x}, \boldsymbol{x}')$. For a fixed point $\boldsymbol{x} \in \mathcal{M}$, we decompose the integral into nearby piece $\mathcal{N}(\boldsymbol{x}) = \{\boldsymbol{x}' \in \mathcal{M} \,|\, d_\mathcal{M}(\boldsymbol{x}, \boldsymbol{x}') \leq \frac{\sqrt{\varepsilon}}{\hat\kappa}\}$ and

faraway piece $\mathcal{F}(\boldsymbol{x}) = \{\boldsymbol{x}' \in \mathcal{M} \mid d_{\mathcal{M}}(\boldsymbol{x}, \boldsymbol{x}') > \frac{\sqrt{\varepsilon}}{\hat{\kappa}}\}$. Then for $\boldsymbol{x}' \in \mathcal{F}(\boldsymbol{x})$, $d_{\mathcal{M}}(\boldsymbol{x}, \boldsymbol{x}') > \frac{\sqrt{\varepsilon}}{\hat{\kappa}}$, and we have $\angle(\boldsymbol{x}, \boldsymbol{x}') \geq \Delta_\varepsilon$ with $\Delta_\varepsilon \leq \frac{\sqrt{\varepsilon}}{\hat{\kappa}} \leq \frac{\pi}{2}$. From Lemmas E.15 and E.16 we get

$$
\begin{aligned}
\int_{\boldsymbol{x}' \in \mathcal{F}(\boldsymbol{x})} |\Theta^{(2)}(\boldsymbol{x}, \boldsymbol{x}')| d\boldsymbol{x}' &\leq \int_{\boldsymbol{x}' \in \mathcal{F}(\boldsymbol{x})} \frac{C_2 n}{\Delta_\varepsilon^3} |\lambda_1^2(\boldsymbol{x}, \boldsymbol{x}')| + \frac{C_1 n}{\Delta_\varepsilon^2} |\lambda_2(\boldsymbol{x}, \boldsymbol{x}')| d\boldsymbol{x}' \\
&\leq \text{len}(\mathcal{M}) \left( 4\frac{C_2 n}{\Delta_\varepsilon^3} + \frac{C_1 n}{\Delta_\varepsilon^2} \left( \frac{\pi}{4} \Delta_\varepsilon^{-1} + 2M_2 \right) \right) \\
&\leq C' \left( \Delta_\varepsilon^{-3} + M_2 \Delta_\varepsilon^{-2} \right) \text{len}(\mathcal{M}) n
\end{aligned}
\tag{E.65}
$$

for some constant $C'$.

For nearby piece, let $\sigma, s$ be such that $\boldsymbol{x} = \boldsymbol{x}_\sigma(s)$. Follow Lemma E.4 we have $\angle(\boldsymbol{x}, \boldsymbol{x}_\sigma(s')) \geq (1 - \varepsilon)|s - s'|$ when $|s - s'| \leq \frac{\sqrt{\varepsilon}}{\hat{\kappa}}$. From Lemmas E.15, E.16, F.10 and F.11 there exists constant $c, C''$ such that

$$
\begin{aligned}
&\int_{\boldsymbol{x}' \in \mathcal{N}(\boldsymbol{x})} |\Theta^{(2)}(\boldsymbol{x}, \boldsymbol{x}')| d\boldsymbol{x}' \\
&\leq \int_{s' = s - \frac{\sqrt{\varepsilon}}{\hat{\kappa}}}^{s - \frac{\sqrt{\varepsilon}}{\hat{\kappa}}} |\Theta^{(2)}(\boldsymbol{x}, \boldsymbol{x}_\sigma(s'))| ds' \\
&\leq \int_{s' = s - \frac{\sqrt{\varepsilon}}{\hat{\kappa}}}^{s - \frac{\sqrt{\varepsilon}}{\hat{\kappa}}} \frac{C_2 n}{(1 - \varepsilon)^3 |s' - s|^3} |\lambda_1^2(\boldsymbol{x}, \boldsymbol{x}_\sigma(s'))| + \frac{C_1 n}{(1 - \varepsilon)^2 |s' - s|^2} |\lambda_2(\boldsymbol{x}, \boldsymbol{x}_\sigma(s'))| ds' \\
&\leq \int_{s' = s - \frac{\sqrt{\varepsilon}}{\hat{\kappa}}}^{s - \frac{\sqrt{\varepsilon}}{\hat{\kappa}}} \frac{C_2 n}{(1 - \varepsilon)^3 |s' - s|^3} \left| \frac{7|s' - s|^3}{12} M_4 \right|^2 + \frac{C_1 n}{(1 - \varepsilon)^2 |s' - s|^2} \left| c(M_4^2 + M_5)|s' - s|^3 \right| ds' \\
&\leq C'' \left( 1 + \left( \frac{\sqrt{\varepsilon}}{\hat{\kappa}} \right)^4 \right) \left( M_4^2 + M_5 \right) n \\
&\leq C'' \left( M_4^2 + M_5 \right) n.
\end{aligned}
\tag{E.66}
$$

combining (E.65) and (E.66) directly proves the claim. ∎

**Lemma E.20.** *For $\varepsilon \in (0, \frac{3}{4})$, there exist positive constants $C, C_3$ such that when $L \geq C$, we have*

$$
\|\boldsymbol{\Theta}^{(3)}\|_{L^2 \to L^2} \leq P_3(M_2, M_3, M_4, M_5, \text{len}(\mathcal{M}), \Delta_\varepsilon^{-1}) n
$$

*where*

$$
\begin{aligned}
&P_3(M_2, M_3, M_4, M_5, \text{len}(\mathcal{M}), \Delta_\varepsilon^{-1}) \\
&\quad = C_3 \left( M_4^3 + M_3 M_4 + M_4 M_5 + \text{len}(\mathcal{M}) \left( \Delta_\varepsilon^{-4} + M_2 \Delta_\varepsilon^{-3} + M_3 \Delta_\varepsilon^{-2} \right) \right)
\end{aligned}
$$

*is a polynomial in $M_2, M_3, M_4, M_5, \text{len}(\mathcal{M})$ and $\Delta_\varepsilon^{-1}$.*

**Proof.** From Lemma E.5 we have

$$
\|\boldsymbol{\Theta}^{(3)}\|_{L^2 \to L^2} = \sup_{\boldsymbol{x} \in \mathcal{M}} \int_{\boldsymbol{x}' \in \mathcal{M}} |\Theta^{(3)}(\boldsymbol{x}, \boldsymbol{x}')| d\boldsymbol{x}'.
$$

From Lemma E.13, we know

$$
\begin{aligned}
&|\Theta^{(3)}(\boldsymbol{x}, \boldsymbol{x}')| \\
&= |\dddot{\psi}(\angle(\boldsymbol{x}, \boldsymbol{x}')) \lambda_1^3(\boldsymbol{x}, \boldsymbol{x}') + 3\ddot{\psi}(\angle(\boldsymbol{x}, \boldsymbol{x}')) \lambda_2(\boldsymbol{x}, \boldsymbol{x}') \lambda_1(\boldsymbol{x}, \boldsymbol{x}') + \dot{\psi}(\angle(\boldsymbol{x}, \boldsymbol{x}')) \lambda_3(\boldsymbol{x}, \boldsymbol{x}')| \\
&\leq |\dddot{\psi}(\angle(\boldsymbol{x}, \boldsymbol{x}'))| |\lambda_1^3(\boldsymbol{x}, \boldsymbol{x}')| + |3\ddot{\psi}(\angle(\boldsymbol{x}, \boldsymbol{x}'))| |\lambda_2(\boldsymbol{x}, \boldsymbol{x}')| |\lambda_1(\boldsymbol{x}, \boldsymbol{x}')| \\
&\quad + |\dot{\psi}(\angle(\boldsymbol{x}, \boldsymbol{x}'))| |\lambda_3(\boldsymbol{x}, \boldsymbol{x}')|
\end{aligned}
$$

From Lemmas F.10 to F.12, there exist constants $C, C_1, C_2, C_3$, such that when $L > C$, we have

$$
\max_{t \geq r} |\dot{\psi}(t)| \leq \frac{C_1 n}{r^2}
$$

$$\max_{t \geq r} |\ddot{\psi}(t)| \leq \frac{C_2 n}{r^3}$$

$$\max_{t \geq r} |\dddot{\psi}(t)| \leq \frac{C_3 n}{r^4}.$$

To get lower bound for the angle $\angle(\boldsymbol{x}, \boldsymbol{x}')$, for a fixed point $\boldsymbol{x} \in \mathcal{M}$, we decompose the integral into a nearby piece $\mathcal{N}(\boldsymbol{x}) = \{\boldsymbol{x}' \in \mathcal{M} \,|\, d_{\mathcal{M}}(\boldsymbol{x}, \boldsymbol{x}') \leq \frac{\sqrt{\varepsilon}}{\hat{\kappa}}\}$ and a faraway piece $\mathcal{F}(\boldsymbol{x}) = \{\boldsymbol{x}' \in \mathcal{M} \,|\, d_{\mathcal{M}}(\boldsymbol{x}, \boldsymbol{x}') > \frac{\sqrt{\varepsilon}}{\hat{\kappa}}\}$. When $d_{\mathcal{M}}(\boldsymbol{x}, \boldsymbol{x}') > \frac{\sqrt{\varepsilon}}{\hat{\kappa}}$, we have $\angle(\boldsymbol{x}, \boldsymbol{x}') \geq \Delta_\varepsilon$ with $\Delta_\varepsilon \leq \frac{\sqrt{\varepsilon}}{\hat{\kappa}} \leq \frac{\pi}{2}$. From Lemmas E.15 to E.17, there exist constants $c, C'$ such that

$$\int_{\boldsymbol{x}' \in \mathcal{F}(\boldsymbol{x})} |\Theta^{(3)}(\boldsymbol{x}, \boldsymbol{x}')| d\boldsymbol{x}'$$

$$\leq \int_{\boldsymbol{x}' \in \mathcal{F}(\boldsymbol{x})} \frac{C_3 n}{\Delta_\varepsilon^4} |\lambda_1^3(\boldsymbol{x}, \boldsymbol{x}')| + \frac{C_2 n}{\Delta_\varepsilon^3} |\lambda_2(\boldsymbol{x}, \boldsymbol{x}')| |\lambda_1(\boldsymbol{x}, \boldsymbol{x}')| + \frac{C_1 n}{\Delta_\varepsilon^2} |\lambda_3(\boldsymbol{x}, \boldsymbol{x}')| d\boldsymbol{x}'$$

$$\leq \text{len}(\mathcal{M}) \left( 8 \frac{C_3 n}{\Delta_\varepsilon^4} + \frac{2 C_2 n}{\Delta_\varepsilon^3} \left( \frac{\pi}{4} \Delta_\varepsilon^{-1} + 2 M_2 \right) \right.$$

$$\left. + \frac{C_1 n}{\Delta_\varepsilon^2} \left( \frac{3\pi^2}{4} \Delta_\varepsilon^{-2} + 9\pi M_2 \Delta_\varepsilon^{-1} + 2 M_3 + 8 \right) \right)$$

$$\leq C' \left( \Delta_\varepsilon^{-4} + M_2 \Delta_\varepsilon^{-3} + M_3 \Delta_\varepsilon^{-2} \right) \text{len}(\mathcal{M}) n \tag{E.67}$$

for some constant $C'$.

For the nearby piece, let $\sigma, s$ be such that $\boldsymbol{x} = \boldsymbol{x}_\sigma(s)$. From Lemma E.4, when $|s - s'| \leq \frac{\sqrt{\varepsilon}}{\hat{\kappa}}$ we have $\angle(\boldsymbol{x}, \boldsymbol{x}_\sigma(s')) \geq (1 - \varepsilon)|s - s'|$. From Lemmas E.15 to E.17 there exists constant $c', c'', C''$ such that

$$\int_{\boldsymbol{x}' \in \mathcal{N}(\boldsymbol{x})} |\Theta^{(3)}(\boldsymbol{x}, \boldsymbol{x}')| d\boldsymbol{x}' \leq \int_{s' = s - \frac{\sqrt{\varepsilon}}{\hat{\kappa}}}^{s - \frac{\sqrt{\varepsilon}}{\hat{\kappa}}} |\Theta^{(3)}(\boldsymbol{x}, \boldsymbol{x}_\sigma(s'))| ds'$$

$$\leq \int_{s' = s - \frac{\sqrt{\varepsilon}}{\hat{\kappa}}}^{s - \frac{\sqrt{\varepsilon}}{\hat{\kappa}}} \frac{C_3 n}{(1 - \varepsilon)^4 |s' - s|^4} |\lambda_1^3(\boldsymbol{x}, \boldsymbol{x}_\sigma(s'))|$$

$$+ \frac{C_2 n}{(1 - \varepsilon)^3 |s' - s|^3} |\lambda_2(\boldsymbol{x}, \boldsymbol{x}_\sigma(s'))| |\lambda_1(\boldsymbol{x}, \boldsymbol{x}_\sigma(s'))|$$

$$+ \frac{C_1 n}{(1 - \varepsilon)^2 |s' - s|^2} |\lambda_3(\boldsymbol{x}, \boldsymbol{x}_\sigma(s'))| \, ds'$$

$$\leq \int_{s' = s - \frac{\sqrt{\varepsilon}}{\hat{\kappa}}}^{s - \frac{\sqrt{\varepsilon}}{\hat{\kappa}}} \frac{C_2 n}{(1 - \varepsilon)^4 |s' - s|^4} \left| \frac{7 |s' - s|^3}{12} M_4 \right|^3$$

$$+ \frac{C_2 n}{(1 - \varepsilon)^3 |s' - s|^3} |c'(M_4^2 + M_5)|s' - s|^3| \left| \frac{7 |s' - s|^3}{12} M_4 \right|$$

$$+ \frac{C_1 n}{(1 - \varepsilon)^2 |s' - s|^2} c''(M_4^3 + M_4 M_5 + M_3 M_4)|s' - s|^3 ds'$$

$$\leq C'' \left( M_4^3 + M_3 M_4 + M_4 M_5 \right) n. \tag{E.68}$$

where the last line comes from the fact that $\int_{s' = s - \frac{\sqrt{\varepsilon}}{\hat{\kappa}}}^{s - \frac{\sqrt{\varepsilon}}{\hat{\kappa}}} |s - s'|^i ds' < 2\hat{\kappa}^{-i-1} \leq 2(2/\pi)^{-i-1}$. Combining (E.67) and (E.68) directly proves the claim. ∎

**Lemma E.21.** *For $i = 0, 1, 2$ and any differentiable $h : \mathcal{M} \to \mathbb{C}$, we have*

$$\frac{d}{ds} \boldsymbol{\Theta}^{(i)}[h](\boldsymbol{x}) = \boldsymbol{\Theta}^{(i)} \left[ \frac{d}{ds} h \right] (\boldsymbol{x}) + \boldsymbol{\Theta}^{(i+1)}[h](\boldsymbol{x}),$$

*where we recall the notation defined in (C.5).*

**Proof.** Let $s$ be such that $\boldsymbol{x}_\sigma(s) = \boldsymbol{x}$. We have

$$\frac{d}{ds}\boldsymbol{\Theta}^{(i)}[h](\boldsymbol{x}_\sigma(s)) = \frac{\partial}{\partial t}\bigg|_{t=0} \int_{\boldsymbol{x}'\in\mathcal{M}} \Theta^{(i)}(\boldsymbol{x}_\sigma(s+t),\boldsymbol{x}')h(\boldsymbol{x}')d\boldsymbol{x}'$$

$$= \frac{\partial}{\partial t}\bigg|_{t=0} \sum_{\sigma'}\int_{s'} \Theta^{(i)}(\boldsymbol{x}_\sigma(s+t),\boldsymbol{x}_{\sigma'}(s'))h(\boldsymbol{x}_{\sigma'}(s'))ds'$$

$$= \sum_{\sigma'}\lim_{t\to 0}\frac{1}{t}\left[\int_{s'}\Theta^{(i)}(\boldsymbol{x}_\sigma(s+t),\boldsymbol{x}_{\sigma'}(s'))h(\boldsymbol{x}_{\sigma'}(s'))ds'\right.$$

$$\left.-\int_{s'}\Theta^{(i)}(\boldsymbol{x}_\sigma(s),\boldsymbol{x}_{\sigma'}(s'))h(\boldsymbol{x}_{\sigma'}(s'))ds'\right]$$

$$= \sum_{\sigma'}\lim_{t\to 0}\frac{1}{t}\left[\int_{s'}\Theta^{(i)}(\boldsymbol{x}_\sigma(s+t),\boldsymbol{x}_{\sigma'}(s'+t))h(\boldsymbol{x}_{\sigma'}(s'+t))ds'\right.$$

$$\left.-\int_{s'}\Theta^{(i)}(\boldsymbol{x}_\sigma(s),\boldsymbol{x}_{\sigma'}(s'))h(\boldsymbol{x}_{\sigma'}(s'))ds'\right]$$

$$= \sum_{\sigma'}\lim_{t\to 0}\frac{1}{t}\left[\int_{s'}\Theta^{(i)}(\boldsymbol{x}_\sigma(s+t),\boldsymbol{x}_{\sigma'}(s'+t))h(\boldsymbol{x}_{\sigma'}(s'+t))ds'\right.$$

$$-\int_{s'}\Theta^{(i)}(\boldsymbol{x}_\sigma(s),\boldsymbol{x}_{\sigma'}(s'))h(\boldsymbol{x}_{\sigma'}(s'+t))ds'$$

$$+\int_{s'}\Theta^{(i)}(\boldsymbol{x}_\sigma(s),\boldsymbol{x}_{\sigma'}(s'))h(\boldsymbol{x}_{\sigma'}(s'+t))ds'$$

$$\left.-\int_{s'}\Theta^{(i)}(\boldsymbol{x}_\sigma(s),\boldsymbol{x}_{\sigma'}(s'))h(\boldsymbol{x}_{\sigma'}(s'))ds'\right]$$

$$= \sum_{\sigma'}\lim_{t\to 0}\frac{1}{t}\left[\int_{s'}\left[\Theta^{(i)}(\boldsymbol{x}_\sigma(s+t),\boldsymbol{x}_{\sigma'}(s'+t))-\Theta^{(i)}(\boldsymbol{x}_\sigma(s),\boldsymbol{x}_{\sigma'}(s'))\right]h(\boldsymbol{x}_{\sigma'}(s'+t))ds'\right.$$

$$\left.\int_{s'}\Theta^{(i)}(\boldsymbol{x}_\sigma(s),\boldsymbol{x}_\sigma(s'))\left[h(\boldsymbol{x}_{\sigma'}(s'+t))-h(\boldsymbol{x}_{\sigma'}(s'))\right]ds'\right].$$

Above, the domain of each of the $s'$ integrals is a fundamental domain for the circles $\mathbb{R}/(\text{len}(\mathcal{M}_\sigma)\cdot\mathbb{Z})$ (by periodicity of the parameterizations, the specific fundamental domain is irrelevant). For $i=0,1,2$ we have by the mean value theorem

$$\left|\frac{\Theta^{(i)}(\boldsymbol{x}_\sigma(s+t),\boldsymbol{x}_{\sigma'}(s'+t))-\Theta^{(i)}(\boldsymbol{x}_\sigma(s),\boldsymbol{x}_{\sigma'}(s'))}{t}h(\boldsymbol{x}_{\sigma'}(s'+t))\right|$$

$$= \left|\Theta^{(i+1)}(\boldsymbol{x}_\sigma(s+t_0),\boldsymbol{x}_{\sigma'}(s'+t_0))h(\boldsymbol{x}_{\sigma'}(s'+t))\right|$$

$$\leq \sup_{\substack{\boldsymbol{x}_1,\boldsymbol{x}_2\in\mathcal{M}\\d_\mathcal{M}(\boldsymbol{x}_1,\boldsymbol{x}_2)=d_\mathcal{M}(\boldsymbol{x},\boldsymbol{x}')}}\left|\Theta^{(i+1)}(\boldsymbol{x}_1,\boldsymbol{x}_2)\right|\sup_{\boldsymbol{x}_3\in\mathcal{M}}|h(\boldsymbol{x}_3)|$$

for some $|t_0|\leq|t|$. As $\mathcal{M}$ is closed with bounded length and $h$ is differentiable, $\sup_{\boldsymbol{x}_3\in\mathcal{M}}h(\boldsymbol{x}_3)$ is bounded. By the formulas in Lemma E.13, the fact that $\psi$ is $\mathcal{C}^3$ by Lemma G.5, and Lemmas E.15 to E.17, it follows that the former supremum is finite as well. From the dominated convergence theorem, we then have

$$\sum_{\sigma'}\lim_{t\to 0}\frac{1}{t}\int_{s'}\left[\Theta^{(i)}(\boldsymbol{x}_\sigma(s+t),\boldsymbol{x}_{\sigma'}(s'+t))-\Theta^{(i)}(\boldsymbol{x}_\sigma(s),\boldsymbol{x}_{\sigma'}(s'))\right]h(\boldsymbol{x}_{\sigma'}(s'+t))ds'$$

$$= \sum_{\sigma'}\int_{s'}\lim_{t\to 0}\frac{1}{t}\left[\Theta^{(i)}(\boldsymbol{x}_\sigma(s+t),\boldsymbol{x}_{\sigma'}(s'+t))-\Theta^{(i)}(\boldsymbol{x}_\sigma(s),\boldsymbol{x}_{\sigma'}(s'))\right]h(\boldsymbol{x}_{\sigma'}(s'+t))ds'$$

$$= \sum_{\sigma'}\int_{s'}\Theta^{(i+1)}(\boldsymbol{x}_\sigma(s),\boldsymbol{x}_{\sigma'}(s'))h(\boldsymbol{x}_{\sigma'}(s'))ds' \tag{E.69}$$

Similarly, as $\operatorname{ess\,sup}_{\boldsymbol{x}\in\mathcal{M}}\left|\frac{d}{ds}h(\boldsymbol{x})\right|$ is finite and $\mathcal{M}$ is compact, from the dominated convergence theorem we also have

$$
\sum_{\sigma'} \lim_{t\to 0}\frac{1}{t}\int_{s'}\Theta^{(i)}(\boldsymbol{x}_\sigma(s),\boldsymbol{x}_\sigma(s'))\left[h(\boldsymbol{x}_{\sigma'}(s'+t))-h(\boldsymbol{x}_{\sigma'}(s'))\right]ds'
$$
$$
=\sum_{\sigma'}\int_{s'}\lim_{t\to 0}\frac{1}{t}\Theta^{(i)}(\boldsymbol{x}_\sigma(s),\boldsymbol{x}_\sigma(s'))\left[h(\boldsymbol{x}_{\sigma'}(s'+t))-h(\boldsymbol{x}_{\sigma'}(s'))\right]ds'
$$
$$
=\sum_{\sigma'}\int_{s'}\Theta^{(i)}(\boldsymbol{x}_\sigma(s),\boldsymbol{x}_\sigma(s'))\frac{d}{ds}h(\boldsymbol{x}_{\sigma'}(s'))ds' \tag{E.70}
$$

Summing (E.69) and (E.70) shows the claim. ∎

**Lemma E.22.** *There is an absolute constant $C>0$ such that for all $L\geq C$, any differentiable $h:\mathcal{M}\to\mathbb{C}$ and any $\varepsilon\in(0,\frac{3}{4})$, if the operator $\widehat{\boldsymbol{M}}_\varepsilon$ and the subspace $S_\varepsilon$ are as defined in (E.11) and (E.12), then we have*

$$
\frac{d}{ds}\boldsymbol{P}_{S_\varepsilon}[h]=\boldsymbol{P}_{S_\varepsilon}\left[\frac{d}{ds}h\right],
$$
$$
\frac{d}{ds}\widehat{\boldsymbol{M}}_\varepsilon\boldsymbol{P}_{S_\varepsilon}[h]=\widehat{\boldsymbol{M}}_\varepsilon\boldsymbol{P}_{S_\varepsilon}\left[\frac{d}{ds}h\right].
$$

*Also, suppose the hypotheses of Lemma E.6 are satisfied, so that $\boldsymbol{P}_{S_\varepsilon}\widehat{\boldsymbol{M}}_\varepsilon\boldsymbol{P}_{S_\varepsilon}$ is invertible over $S_\varepsilon$. Then one has in particular*

$$
\frac{d}{ds}\left(\boldsymbol{P}_{S_\varepsilon}\widehat{\boldsymbol{M}}_\varepsilon\boldsymbol{P}_{S_\varepsilon}\right)^{-1}[h]=\left(\boldsymbol{P}_{S_\varepsilon}\widehat{\boldsymbol{M}}_\varepsilon\boldsymbol{P}_{S_\varepsilon}\right)^{-1}\left[\frac{d}{ds}h\right].
$$

**Proof.** The condition on $L$ implies that $\widehat{\boldsymbol{M}}_\varepsilon$ is well-defined. For any operator $\boldsymbol{T}$ that diagonalizes in the Fourier basis for $S_\varepsilon$, i.e. for any $h\in L^2(\mathcal{M})$, $\boldsymbol{T}$ satisfies

$$
\boldsymbol{T}[h]=\sum_{\sigma\in\{\pm\}}\sum_{k=-K_{\varepsilon,\sigma}}^{K_{\varepsilon,\sigma}}m_{\sigma,k}\phi_{\sigma,k}\phi_{\sigma,k}^* h \tag{E.71}
$$

for some coefficients $m_{\sigma,k}\in\mathbb{C}$ independent of $h$,[7] we have

$$
\frac{d}{ds}\boldsymbol{T}[h]=\sum_{\sigma\in\{\pm\}}\sum_{k=-K_{\varepsilon,\sigma}}^{K_{\varepsilon,\sigma}}m_{\sigma,k}\left[\frac{d}{ds}\phi_{\sigma,k}\right]\phi_{\sigma,k}^* h
$$
$$
=\sum_{\sigma\in\{\pm\}}\sum_{k=-K_{\varepsilon,\sigma}}^{K_{\varepsilon,\sigma}}m_{\sigma,k}\frac{i2\pi k}{\operatorname{len}(\mathcal{M}_\sigma)}\phi_{\sigma,k}\phi_{\sigma,k}^* h,
$$

where we recall the definition of the Fourier basis functions from (C.11) for the second equality. Now fix $h$ differentiable as in the statement of the lemma. On the other hand, since $\phi_{\sigma,k}^* h$ is simply some complex number, which does not depend on $s$, we have

$$
\left(\frac{d}{ds}\phi_{\sigma,k}\right)^* h+\phi_{\sigma,k}^*\frac{d}{ds}h=0,
$$

and so

$$
\boldsymbol{T}\left[\frac{d}{ds}h\right]=\sum_{\sigma\in\{\pm\}}\sum_{k=-K_{\varepsilon,\sigma}}^{K_{\varepsilon,\sigma}}m_{\sigma,k}\phi_{\sigma,k}\phi_{\sigma,k}^*\frac{d}{ds}h
$$

---

[7]Here and in the sequel, we recall that we are using the notation $\phi_{\sigma,k}^* h=\langle\phi_{\sigma,k}^*,h\rangle=\int_\mathcal{M}\overline{\phi}_{\sigma,k}(\boldsymbol{x})h(\boldsymbol{x})\,d\boldsymbol{x}$ for the standard inner product on complex-valued functions on $\mathcal{M}$.

$$= -\sum_{\sigma\in\{\pm\}}\sum_{k=-K_{\varepsilon,\sigma}}^{K_{\varepsilon,\sigma}} m_{\sigma,k}\phi_{\sigma,k}\left(\frac{d}{ds}\phi_{\sigma,k}\right)^* h$$

$$= -\sum_{\sigma\in\{\pm\}}\sum_{k=-K_{\varepsilon,\sigma}}^{K_{\varepsilon,\sigma}} m_{\sigma,k}\phi_{\sigma,k}\left(\frac{i2\pi k}{\mathrm{len}(\mathcal{M}_\sigma)}\phi_{\sigma,k}\right)^* h$$

$$= \frac{d}{ds}\boldsymbol{T}[h].$$

The operators $\widehat{\boldsymbol{M}}_\varepsilon$ and $\boldsymbol{P}_{S_\varepsilon}$ both diagonalize in the Fourier basis for $S_\varepsilon$, following the arguments in the proof of Lemma E.6. By the same token, $(\boldsymbol{P}_{S_\varepsilon}\widehat{\boldsymbol{M}}_\varepsilon\boldsymbol{P}_{S_\varepsilon})^{-1}$ also diagonalizes in the Fourier basis for $S_\varepsilon$ when it is well defined (recall (E.37)), which concludes the proof. ∎

**Lemma E.23.** *There is an absolute constant $C > 0$ such that if $L \geq C$, and for any $\varepsilon \in (0, \frac{3}{4})$ if $a_\varepsilon, r_\varepsilon, S_\varepsilon$ defined as in (E.7), (E.8), and (E.12), respectively, then when in addition*

$$L \geq \left(\varepsilon^{1/2}\min\{\mathrm{len}(\mathcal{M}_+), \mathrm{len}(\mathcal{M}_-)\}/(12\pi^2)\right)^{-\frac{a_\varepsilon+1}{a_\varepsilon}},$$

*we have for any differentiable function $f : \mathcal{M} \to \mathbb{C}$*

$$\left\|\frac{d}{ds}\boldsymbol{P}_{S_\varepsilon}f\right\|_{L^2} \leq \frac{\sqrt{\varepsilon}}{r_\varepsilon}\|\boldsymbol{P}_{S_\varepsilon}f\|_{L^2},$$

$$\left\|\boldsymbol{P}_{S_\varepsilon^\perp}f\right\|_{L^2} \leq \frac{2r_\varepsilon}{\sqrt{\varepsilon}}\left\|\frac{d}{ds}\boldsymbol{P}_{S_\varepsilon^\perp}f\right\|_{L^2}.$$

**Proof.** The condition on $L$ guarantees that $\widehat{\boldsymbol{M}}_\varepsilon$ is well-defined. From (E.12), $S_\varepsilon = S_{K_{\varepsilon,+},K_{\varepsilon,-}}$ with $K_{\varepsilon,\sigma} = \left\lfloor\frac{\varepsilon^{1/2}\mathrm{len}(\mathcal{M}_\sigma)}{2\pi r_\varepsilon}\right\rfloor$ for $\sigma \in \{+,-\}$, then by orthonormality of the Fourier basis functions (C.11), we have

$$\left\|\frac{d}{ds}\boldsymbol{P}_{S_\varepsilon}f\right\|_{L^2} = \left\|\frac{d}{ds}\sum_{\sigma=\pm}\sum_{k=-K_{\varepsilon,\sigma}}^{K_{\varepsilon,\sigma}}\phi_{\sigma,k}\phi_{\sigma,k}^* f\right\|_{L^2}$$

$$= \left\|\sum_{\sigma=\pm}\sum_{k=-K_{\varepsilon,\sigma}}^{K_{\varepsilon,\sigma}}\frac{i2\pi k}{\mathrm{len}(\mathcal{M}_\sigma)}\phi_{\sigma,k}\phi_{\sigma,k}^* f\right\|_{L^2}$$

$$\leq \frac{\sqrt{\varepsilon}}{r_\varepsilon}\left(\sum_{\sigma=\pm}\sum_{k=-K_{\varepsilon,\sigma}}^{K_{\varepsilon,\sigma}}\|\phi_{\sigma,k}\phi_{\sigma,k}^* f\|_{L^2}^2\right)^{1/2}$$

$$= \frac{\sqrt{\varepsilon}}{r_\varepsilon}\|\boldsymbol{P}_{S_\varepsilon}f\|_{L^2}.$$

Above, the inequality follows because $|k| \leq K_{\varepsilon,\sigma}$ implies $2\pi|k|/\mathrm{len}(\mathcal{M}_\sigma) \leq \sqrt{\varepsilon}/r_\varepsilon$, and because the Fourier basis functions are mutually orthogonal (and $\|f\|_{L^2}^2 = \langle f, f\rangle$). This establishes the first claim.

For the second claim, we have

$$\boldsymbol{P}_{S_\varepsilon^\perp}f = \sum_{\sigma=\pm}\sum_{|k|>K_{\varepsilon,\sigma}}\phi_{\sigma,k}\phi_{\sigma,k}^* f.$$

When $L \geq \left(\varepsilon^{1/2}\min\{\mathrm{len}(\mathcal{M}_+), \mathrm{len}(\mathcal{M}_-)\}/(12\pi^2)\right)^{-\frac{a_\varepsilon+1}{a_\varepsilon}}$, we have $K_{\varepsilon,\pm} = \left\lfloor\frac{\varepsilon^{1/2}\mathrm{len}(\mathcal{M}_\pm)}{12\pi^2}L^{\frac{a_\varepsilon}{a_\varepsilon+1}}\right\rfloor \geq 1$ and thus $K_{\varepsilon,\pm} = \left\lfloor\frac{\varepsilon^{1/2}\mathrm{len}(\mathcal{M}_\pm)}{2\pi r_\varepsilon}\right\rfloor \geq \frac{\varepsilon^{1/2}\mathrm{len}(\mathcal{M}_\pm)}{4\pi r_\varepsilon}$, whence

$$\left\|\frac{d}{ds}\boldsymbol{P}_{S_\varepsilon^\perp}f\right\|_{L^2} = \left\|\sum_{\sigma=\pm}\sum_{|k|>K_{\varepsilon,\sigma}}\left(\frac{d}{ds}\phi_{\sigma,k}\right)\phi_{\sigma,k}^* f\right\|_{L^2}$$

$$= \left\| \sum_{\sigma=\pm} \sum_{|k|>K_{\varepsilon,\sigma}} \frac{i2\pi k}{\text{len}(\mathcal{M}_\sigma)} \phi_{\sigma,k} \phi_{\sigma,k}^* f \right\|_{L^2}$$

$$= \left( \sum_{\sigma=\pm} \sum_{|k|>K_{\varepsilon,\sigma}} \left| \frac{i2\pi k}{\text{len}(\mathcal{M}_\sigma)} \right|^2 (\phi_{\sigma,k}^* f)^2 \right)^{1/2}$$

$$\geq \frac{\sqrt{\varepsilon}}{2r_\varepsilon} \left( \sum_{\sigma=\pm} \sum_{|k|>K_{\varepsilon,\sigma}} (\phi_{\sigma,k}^* f)^2 \right)^{1/2}$$

$$= \frac{\sqrt{\varepsilon}}{2r_\varepsilon} \left\| \boldsymbol{P}_{S_\varepsilon^\perp} f \right\|_{L^2},$$

as claimed. Above, the first equality entails an interchange of limit processes—a formal justification for the validity of this interchange follows from the assumed differentiability of $f$ (which implies that its coefficients $\phi_{\sigma,k}^* f$ have a faster rate of decay $o(|k|^{-3/2})$) and a dominated convergence argument, where the difference quotient involving $\phi_{\sigma,k}$ is bounded by $O(|k|)$, which together with the extra smoothness of $f$ leads to an integrable upper bound. ∎

**Definition E.24.** *For any $\varepsilon \in (0, \frac{3}{4})$, let $P_1 = P_1\left(M_4, \text{len}(\mathcal{M}), \Delta_\varepsilon^{-1}\right), P_2 = P_2\left(M_4, M_5, \text{len}(\mathcal{M}), \Delta_\varepsilon^{-1}\right), P_3 = P_3\left(M_3, M_4, M_5, \text{len}(\mathcal{M}), \Delta_\varepsilon^{-1}\right)$ as defined in Lemmas E.18 to E.20. We let $\Phi(C_\zeta, \varepsilon)$ for some constant $C_\zeta \geq 0$ denote the set of all functions $\zeta \in C^3(\mathcal{M})$ which satisfy*

$$\|\zeta\|_{L^2} \leq C_\zeta$$

$$\|\zeta^{(1)}\|_{L^2} \leq C_\zeta \frac{P_1}{\log L}$$

$$\|\zeta^{(2)}\|_{L^2} \leq C_\zeta \left( \frac{P_2}{\log L} + \frac{P_1^2}{\log^2 L} \right)$$

$$\|\zeta^{(3)}\|_{L^2} \leq C_\zeta \left( \frac{P_3}{\log L} + \frac{P_2 P_1}{\log^2 L} + \frac{P_1^3}{\log^3 L} \right).$$

*Furthermore, for $\varepsilon \geq \varepsilon' > 0$, one has $\Delta_\varepsilon^{-1} \leq \Delta_{\varepsilon'}^{-1}$. As $P_1, P_2, P_3$ have positive coefficients, $\zeta \in \Phi(C_\zeta, \varepsilon)$ implies $\zeta \in \Phi(C_\zeta, \varepsilon')$, i.e., $\Phi(C_\zeta, \varepsilon') \subseteq \Phi(C_\zeta, \varepsilon)$.*

**Lemma E.25.** *For any $\varepsilon \in (0, \frac{3}{4})$, there exist numbers $C_\varepsilon, C_\varepsilon' > 0$ such that when the conditions of Theorem E.1 are in force, for any $\zeta \in S_\varepsilon \cap \Phi(C_\zeta, \varepsilon)$, the certificate $g_\varepsilon[\zeta]$ defined in Theorem E.1 satisfies*

$$g_\varepsilon[\zeta] \in \Phi\left( \frac{C_\varepsilon' C_\zeta}{n \log L}, \varepsilon \right). \tag{E.72}$$

**Proof.** Following Lemma E.22 and Lemma E.21, we have that

$$g_\varepsilon^{(1)}[\zeta]$$

$$= \sum_{\ell=1}^{\infty} (-1)^\ell \sum_{a=0}^{\ell-1} \left( \left( \boldsymbol{P}_{S_\varepsilon} \widehat{\boldsymbol{M}}_\varepsilon \boldsymbol{P}_{S_\varepsilon} \right)^{-1} \boldsymbol{P}_{S_\varepsilon} (\boldsymbol{\Theta}^\circ - \widehat{\boldsymbol{M}}_\varepsilon) \boldsymbol{P}_{S_\varepsilon} \right)^a \left( \boldsymbol{P}_{S_\varepsilon} \widehat{\boldsymbol{M}}_\varepsilon \boldsymbol{P}_{S_\varepsilon} \right)^{-1} \boldsymbol{P}_{S_\varepsilon} \boldsymbol{\Theta}^{(1)} \boldsymbol{P}_{S_\varepsilon}$$

$$\times \left( \left( \boldsymbol{P}_{S_\varepsilon} \widehat{\boldsymbol{M}}_\varepsilon \boldsymbol{P}_{S_\varepsilon} \right)^{-1} \boldsymbol{P}_{S_\varepsilon} (\boldsymbol{\Theta}^\circ - \widehat{\boldsymbol{M}}_\varepsilon) \boldsymbol{P}_{S_\varepsilon} \right)^{\ell-a-1} \left( \boldsymbol{P}_{S_\varepsilon} \widehat{\boldsymbol{M}}_\varepsilon \boldsymbol{P}_{S_\varepsilon} \right)^{-1} \zeta$$

$$+ g_\varepsilon[\zeta^{(1)}].$$

As $\zeta \in S_\varepsilon$, we have $\zeta^{(1)} = \frac{d}{ds}\zeta = \frac{d}{ds}\boldsymbol{P}_{S_\varepsilon}\zeta = \boldsymbol{P}_{S_\varepsilon}\frac{d}{ds}\zeta \in S_\varepsilon$, and thus following Theorem E.1, there exists constant $c$ such that

$$\|g_\varepsilon[\zeta^{(1)}]\|_{L^2} \leq \frac{\|\zeta^{(1)}\|_{L^2}}{\varepsilon c n \log L}.$$

From Lemma E.6, there exists $C, c' > 0$ such that when $L \geq C$,

$$\lambda_{\min}(\boldsymbol{P}_S \widehat{\boldsymbol{M}}_\varepsilon \boldsymbol{P}_{S_\varepsilon}) \geq cn \log L.$$

Under the conditions of Theorem E.1, we have

$$\left\| \left(\boldsymbol{P}_{S_\varepsilon} \widehat{\boldsymbol{M}}_\varepsilon \boldsymbol{P}_{S_\varepsilon}\right)^{-1} \boldsymbol{P}_{S_\varepsilon}(\boldsymbol{\Theta}^\circ - \widehat{\boldsymbol{M}}_\varepsilon)\boldsymbol{P}_{S_\varepsilon} \right\|_{L^2 \to L^2} \leq 1 - \varepsilon.$$

Let $P_1 = P_1\left(M_4, \operatorname{len}(\mathcal{M}), \Delta_\varepsilon^{-1}\right)$, $P_2 = P_2\left(M_4, M_5, \operatorname{len}(\mathcal{M}), \Delta_\varepsilon^{-1}\right)$, and $P_3 = P_3\left(M_3, M_4, M_5, \operatorname{len}(\mathcal{M}), \Delta_\varepsilon^{-1}\right)$ be the polynomials in Lemmas E.18 to E.20. From Lemma E.18 , we have

$$\begin{aligned}
\|g_\varepsilon^{(1)}[\zeta]\|_{L^2} &\leq \sum_{\ell=1}^{\infty} \sum_{a=0}^{\ell-1} (1-\varepsilon)^{\ell-1} \|\boldsymbol{\Theta}^{(1)}\|_{L^2 \to L^2} \frac{\|\zeta\|_{L^2}}{(cn \log L)^2} \\
&\quad + \frac{\|\zeta^{(1)}\|_{L^2}}{\varepsilon cn \log L} \\
&= \frac{\|\boldsymbol{\Theta}^{(1)}\|_{L^2 \to L^2} \|\zeta\|_{L^2}}{(cn \log L)^2} \sum_{\ell=0}^{\infty} \ell(1-\varepsilon)^{\ell-1} + \frac{\|\zeta^{(1)}\|_{L^2}}{\varepsilon cn \log L} \\
&= \frac{\|\boldsymbol{\Theta}^{(1)}\|_{L^2 \to L^2} \|\zeta\|_{L^2}}{\varepsilon^2 (cn \log L)^2} + \frac{\|\zeta^{(1)}\|_{L^2}}{\varepsilon cn \log L} \\
&\leq \frac{P_1 \|\zeta\|_{L^2}}{\varepsilon^2 c^2 n \log^2 L} + \frac{\|\zeta^{(1)}\|_{L^2}}{\varepsilon cn \log L}.
\end{aligned}$$ (E.73)

From the fact that $\zeta \in \Phi(C_\zeta, \varepsilon)$, we further obtain

$$\begin{aligned}
\|g_\varepsilon^{(1)}[\zeta]\|_{L^2} &\leq \frac{P_1 C_\zeta}{\varepsilon^2 c^2 n \log^2 L} + \frac{C_\zeta}{\varepsilon cn \log L} \frac{P_1}{\log L} \\
&\leq C_\varepsilon \frac{P_1 C_\zeta}{n \log^2 L}.
\end{aligned}$$

For the second derivative, we have

$$\begin{aligned}
g_\varepsilon^{(2)}[\zeta] &= \sum_{\ell=1}^{\infty} (-1)^\ell \sum_{a=0}^{\ell-1} \left( \left(\boldsymbol{P}_S \widehat{\boldsymbol{M}}_\varepsilon \boldsymbol{P}_{S_\varepsilon}\right)^{-1} \boldsymbol{P}_{S_\varepsilon}(\boldsymbol{\Theta}^\circ - \widehat{\boldsymbol{M}})\boldsymbol{P}_{S_\varepsilon} \right)^a \\
&\quad \times \left(\boldsymbol{P}_S \widehat{\boldsymbol{M}}_\varepsilon \boldsymbol{P}_{S_\varepsilon}\right)^{-1} \boldsymbol{P}_{S_\varepsilon} \boldsymbol{\Theta}^{(2)} \boldsymbol{P}_{S_\varepsilon} \\
&\quad \times \left( \left(\boldsymbol{P}_S \widehat{\boldsymbol{M}}_\varepsilon \boldsymbol{P}_{S_\varepsilon}\right)^{-1} \boldsymbol{P}_{S_\varepsilon}(\boldsymbol{\Theta}^\circ - \widehat{\boldsymbol{M}})\boldsymbol{P}_{S_\varepsilon} \right)^{\ell-a-1} \left(\boldsymbol{P}_{S_\varepsilon} \widehat{\boldsymbol{M}}_\varepsilon \boldsymbol{P}_{S_\varepsilon}\right)^{-1} \zeta \\
&\quad + 2 \sum_{\ell=1}^{\infty} (-1)^\ell \sum_{a=0}^{\ell-2} \sum_{a'=a}^{\ell-2} \left( \left(\boldsymbol{P}_S \widehat{\boldsymbol{M}}_\varepsilon \boldsymbol{P}_{S_\varepsilon}\right)^{-1} \boldsymbol{P}_{S_\varepsilon}(\boldsymbol{\Theta}^\circ - \widehat{\boldsymbol{M}})\boldsymbol{P}_{S_\varepsilon} \right)^a \\
&\quad \times \left(\boldsymbol{P}_S \widehat{\boldsymbol{M}}_\varepsilon \boldsymbol{P}_{S_\varepsilon}\right)^{-1} \boldsymbol{P}_{S_\varepsilon} \boldsymbol{\Theta}^{(1)} \boldsymbol{P}_{S_\varepsilon} \\
&\quad \times \left( \left(\boldsymbol{P}_S \widehat{\boldsymbol{M}}_\varepsilon \boldsymbol{P}_{S_\varepsilon}\right)^{-1} \boldsymbol{P}_{S_\varepsilon}(\boldsymbol{\Theta}^\circ - \widehat{\boldsymbol{M}})\boldsymbol{P}_{S_\varepsilon} \right)^{a'-a} \\
&\quad \times \left(\boldsymbol{P}_S \widehat{\boldsymbol{M}}_\varepsilon \boldsymbol{P}_{S_\varepsilon}\right)^{-1} \boldsymbol{P}_{S_\varepsilon} \boldsymbol{\Theta}^{(1)} \boldsymbol{P}_{S_\varepsilon} \\
&\quad \times \left( \left(\boldsymbol{P}_S \widehat{\boldsymbol{M}}_\varepsilon \boldsymbol{P}_{S_\varepsilon}\right)^{-1} \boldsymbol{P}_{S_\varepsilon}(\boldsymbol{\Theta}^\circ - \widehat{\boldsymbol{M}})\boldsymbol{P}_{S_\varepsilon} \right)^{\ell-a'-2} \left(\boldsymbol{P}_{S_\varepsilon} \widehat{\boldsymbol{M}}_\varepsilon \boldsymbol{P}_{S_\varepsilon}\right)^{-1} \zeta \\
&\quad + 2 g_\varepsilon^{(1)}[\zeta^{(1)}] - g_\varepsilon[\zeta^{(2)}].
\end{aligned}$$

From (E.73), as $\zeta^{(1)}, \zeta^{(2)} \in S_\varepsilon$ we have

$$\|g_\varepsilon^{(1)}[\zeta^{(1)}]\|_{L^2} \le \frac{P_1\|\zeta^{(1)}\|_{L^2}}{\varepsilon^2 c^2 n \log^2 L} + \frac{\|\zeta^{(2)}\|_{L^2}}{\varepsilon c n \log L},$$

$$\|g_\varepsilon[\zeta^{(2)}]\|_{L^2} \le \frac{\|\zeta^{(2)}\|_{L^2 \to L^2}}{\varepsilon c n \log L}.$$

which leads to

$$\begin{aligned}
\|g_\varepsilon^{(2)}[\zeta]\|_{L^2} &\le \frac{\|\Theta^{(2)}\|_{L^2 \to L^2}\|\zeta\|_{L^2}}{\varepsilon^2 (cn\log L)^2} + \sum_{\ell=1}^\infty \ell(\ell-1)(1-\varepsilon)^{\ell-2}\frac{\|\Theta^{(1)}\|_{L^2 \to L^2}^2\|\zeta\|_{L^2}}{(cn\log L)^3} \\
&\quad + \ 2\|g_\varepsilon^{(1)}[\zeta^{(1)}]\|_{L^2} + \|g_\varepsilon[\zeta^{(2)}]\|_{L^2} \\
&\le \frac{\|\Theta^{(2)}\|_{L^2 \to L^2}\|\zeta\|_{L^2}}{\varepsilon^2 (cn\log L)^2} + \frac{2\|\Theta^{(1)}\|_{L^2 \to L^2}^2\|\zeta\|_{L^2}}{\varepsilon^3 (cn\log L)^3} \\
&\quad + \ 2\left(\frac{P_1\|\zeta^{(1)}\|_{L^2}}{\varepsilon^2 c^2 n \log^2 L} + \frac{\|\zeta^{(2)}\|_{L^2}}{\varepsilon c n \log L}\right) + \frac{\|\zeta^{(2)}\|_{L^2}}{\varepsilon c n \log L} \\
&\le \frac{P_2\|\zeta\|_{L^2}}{\varepsilon^2 c^2 n \log^2 L} + \frac{2P_1^2\|\zeta\|_{L^2}}{\varepsilon^3 c^3 n \log^3 L} + \frac{2P_1\|\zeta^{(1)}\|_{L^2}}{\varepsilon^2 c^2 n \log^2 L} + \frac{3\|\zeta^{(2)}\|_{L^2}}{\varepsilon c n \log L}.
\end{aligned} \tag{E.74}$$

Again as $\zeta \in \Phi(C_\zeta, \varepsilon)$ we have

$$\begin{aligned}
\|g_\varepsilon^{(2)}[\zeta]\|_{L^2} &\le \frac{P_2 C_\zeta}{\varepsilon^2 c^2 n \log^2 L} + \frac{2P_1^2 C_\zeta}{\varepsilon^3 c^3 n \log^3 L} \\
&\quad + \frac{2P_1 C_\zeta}{\varepsilon^2 c^2 n \log^2 L}\frac{P_1}{\log L} + \frac{C_\zeta}{\varepsilon c n \log L}\left(\frac{P_2}{\log L} + \frac{P_1^2}{\log^2 L}\right) \\
&\le C_\varepsilon C_\zeta \left(\frac{P_1^2}{n \log^3 L} + \frac{P_2}{n \log^2 L}\right).
\end{aligned}$$

For third derivative, we have

$$\begin{aligned}
g_\varepsilon^{(3)}[\zeta] &= \sum_{\ell=1}^\infty (-1)^\ell \sum_{a=0}^{\ell-1} \left(\left(P_S\widehat{M}_\varepsilon P_{S_\varepsilon}\right)^{-1} P_{S_\varepsilon}(\Theta^\circ - \widehat{M})P_{S_\varepsilon}\right)^a \\
&\quad \times \ \left(P_S\widehat{M}_\varepsilon P_{S_\varepsilon}\right)^{-1} P_{S_\varepsilon}\Theta^{(3)} P_{S_\varepsilon} \\
&\quad \times \ \left(\left(P_S\widehat{M}_\varepsilon P_{S_\varepsilon}\right)^{-1} P_{S_\varepsilon}(\Theta^\circ - \widehat{M})P_{S_\varepsilon}\right)^{\ell-a-1}\left(P_{S_\varepsilon}\widehat{M}_\varepsilon P_{S_\varepsilon}\right)^{-1}\zeta \\
&\quad + 3\sum_{\ell=1}^\infty (-1)^\ell \sum_{a=0}^{\ell-2}\sum_{a'=a}^{\ell-2}\left(\left(P_S\widehat{M}_\varepsilon P_{S_\varepsilon}\right)^{-1} P_{S_\varepsilon}(\Theta^\circ - \widehat{M})P_{S_\varepsilon}\right)^a \\
&\quad \times \ \left(P_S\widehat{M}_\varepsilon P_{S_\varepsilon}\right)^{-1} P_{S_\varepsilon}\Theta^{(2)} P_{S_\varepsilon} \\
&\quad \times \ \left(\left(P_S\widehat{M}_\varepsilon P_{S_\varepsilon}\right)^{-1} P_{S_\varepsilon}(\Theta^\circ - \widehat{M})P_{S_\varepsilon}\right)^{a'-a} \\
&\quad \times \ \left(P_S\widehat{M}_\varepsilon P_{S_\varepsilon}\right)^{-1} P_{S_\varepsilon}\Theta^{(1)} P_{S_\varepsilon} \\
&\quad \times \ \left(\left(P_S\widehat{M}_\varepsilon P_{S_\varepsilon}\right)^{-1} P_{S_\varepsilon}(\Theta^\circ - \widehat{M})P_{S_\varepsilon}\right)^{\ell-a'-2}\left(P_{S_\varepsilon}\widehat{M}_\varepsilon P_{S_\varepsilon}\right)^{-1}\zeta \\
&\quad + 3\sum_{\ell=1}^\infty (-1)^\ell \sum_{a=0}^{\ell-2}\sum_{a'=a}^{\ell-2}\left(\left(P_S\widehat{M}_\varepsilon P_{S_\varepsilon}\right)^{-1} P_{S_\varepsilon}(\Theta^\circ - \widehat{M})P_{S_\varepsilon}\right)^a \\
&\quad \times \ \left(P_S\widehat{M}_\varepsilon P_{S_\varepsilon}\right)^{-1} P_{S_\varepsilon}\Theta^{(1)} P_{S_\varepsilon}
\end{aligned}$$

$$\times \quad \left(\left(\boldsymbol{P}_S\widehat{\boldsymbol{M}}_\varepsilon\boldsymbol{P}_{S_\varepsilon}\right)^{-1}\boldsymbol{P}_{S_\varepsilon}(\boldsymbol{\Theta}^\circ-\widehat{\boldsymbol{M}})\boldsymbol{P}_{S_\varepsilon}\right)^{a'-a}$$

$$\times \quad \left(\boldsymbol{P}_S\widehat{\boldsymbol{M}}_\varepsilon\boldsymbol{P}_{S_\varepsilon}\right)^{-1}\boldsymbol{P}_{S_\varepsilon}\boldsymbol{\Theta}^{(2)}\boldsymbol{P}_{S_\varepsilon}$$

$$\times \quad \left(\left(\boldsymbol{P}_S\widehat{\boldsymbol{M}}_\varepsilon\boldsymbol{P}_{S_\varepsilon}\right)^{-1}\boldsymbol{P}_{S_\varepsilon}(\boldsymbol{\Theta}^\circ-\widehat{\boldsymbol{M}})\boldsymbol{P}_{S_\varepsilon}\right)^{\ell-a'-2}\left(\boldsymbol{P}_{S_\varepsilon}\widehat{\boldsymbol{M}}_\varepsilon\boldsymbol{P}_{S_\varepsilon}\right)^{-1}\zeta$$

$$+6\sum_{\ell=1}^{\infty}(-1)^\ell\sum_{a=0}^{\ell-2}\sum_{a'=a}^{\ell-2}\sum_{a''=a'}^{\ell-2}\left(\left(\boldsymbol{P}_S\widehat{\boldsymbol{M}}_\varepsilon\boldsymbol{P}_{S_\varepsilon}\right)^{-1}\boldsymbol{P}_{S_\varepsilon}(\boldsymbol{\Theta}^\circ-\widehat{\boldsymbol{M}})\boldsymbol{P}_{S_\varepsilon}\right)^{a}$$

$$\times \quad \left(\boldsymbol{P}_S\widehat{\boldsymbol{M}}_\varepsilon\boldsymbol{P}_{S_\varepsilon}\right)^{-1}\boldsymbol{P}_{S_\varepsilon}\boldsymbol{\Theta}^{(1)}\boldsymbol{P}_{S_\varepsilon}$$

$$\times \quad \left(\left(\boldsymbol{P}_S\widehat{\boldsymbol{M}}_\varepsilon\boldsymbol{P}_{S_\varepsilon}\right)^{-1}\boldsymbol{P}_{S_\varepsilon}(\boldsymbol{\Theta}^\circ-\widehat{\boldsymbol{M}})\boldsymbol{P}_{S_\varepsilon}\right)^{a'-a}$$

$$\times \quad \left(\boldsymbol{P}_S\widehat{\boldsymbol{M}}_\varepsilon\boldsymbol{P}_{S_\varepsilon}\right)^{-1}\boldsymbol{P}_{S_\varepsilon}\boldsymbol{\Theta}^{(1)}\boldsymbol{P}_{S_\varepsilon}$$

$$\times \quad \left(\left(\boldsymbol{P}_S\widehat{\boldsymbol{M}}_\varepsilon\boldsymbol{P}_{S_\varepsilon}\right)^{-1}\boldsymbol{P}_{S_\varepsilon}(\boldsymbol{\Theta}^\circ-\widehat{\boldsymbol{M}})\boldsymbol{P}_{S_\varepsilon}\right)^{a''-a'}$$

$$\times \quad \left(\boldsymbol{P}_S\widehat{\boldsymbol{M}}_\varepsilon\boldsymbol{P}_{S_\varepsilon}\right)^{-1}\boldsymbol{P}_{S_\varepsilon}\boldsymbol{\Theta}^{(1)}\boldsymbol{P}_{S_\varepsilon}$$

$$\times \quad \left(\left(\boldsymbol{P}_S\widehat{\boldsymbol{M}}_\varepsilon\boldsymbol{P}_{S_\varepsilon}\right)^{-1}\boldsymbol{P}_{S_\varepsilon}(\boldsymbol{\Theta}^\circ-\widehat{\boldsymbol{M}})\boldsymbol{P}_{S_\varepsilon}\right)^{\ell-a''-2}\left(\boldsymbol{P}_{S_\varepsilon}\widehat{\boldsymbol{M}}_\varepsilon\boldsymbol{P}_{S_\varepsilon}\right)^{-1}\zeta$$

$$+3g_\varepsilon^{(2)}[\zeta^{(1)}]-3g_\varepsilon^{(1)}[\zeta^{(2)}]+g_\varepsilon[\zeta^{(3)}].$$

Similarly, as $\zeta^{(1)},\zeta^{(2)}$ and $\zeta^{(3)}\in S_\varepsilon$, plug in results in (E.73) and (E.74) we can control

$$
\begin{aligned}
\|g_\varepsilon^{(3)}[\zeta]\|_{L^2} \leq{} & \frac{\|\boldsymbol{\Theta}^{(3)}\|_{L^2\to L^2}\|\zeta\|_{L^2}}{\varepsilon^2\left(cn\log L\right)^2} \\
& +\frac{3\|\boldsymbol{\Theta}^{(2)}\|_{L^2\to L^2}\|\boldsymbol{\Theta}^{(1)}\|_{L^2\to L^2}\|\zeta\|_{L^2}}{\varepsilon^3\left(cn\log L\right)^3} \\
& +\sum_{\ell=1}^{\infty}(\ell+1)(\ell-1)(\ell-1)(1-\varepsilon)^{\ell-3}\frac{\|\boldsymbol{\Theta}^{(1)}\|_{L^2\to L^2}^3\|\zeta\|_{L^2}}{\left(cn\log L\right)^4} \\
& +3\left(\frac{P_2\|\zeta^{(1)}\|_{L^2}}{\varepsilon^2c^2n\log^2 L}+\frac{2P_1^2\|\zeta^{(1)}\|_{L^2}}{\varepsilon^3c^3n\log^3 L}+\frac{2P_1\|\zeta^{(2)}\|_{L^2}}{\varepsilon^2c^2n\log^2 L}+\frac{3\|\zeta^{(3)}\|_{L^2}}{\varepsilon cn\log L}\right) \\
& +3\left(\frac{P_1\|\zeta^{(2)}\|_{L^2}}{\varepsilon^2c^2n\log^2 L}+\frac{\|\zeta^{(3)}\|_{L^2}}{\varepsilon cn\log L}\right)+\frac{\|\zeta^{(3)}\|}{\varepsilon cn\log L} \\
={} & \frac{\|\boldsymbol{\Theta}^{(3)}\|_{L^2\to L^2}\|\zeta\|_{L^2}}{\varepsilon^2\left(cn\log L\right)^2}+\frac{3\|\boldsymbol{\Theta}^{(2)}\|_{L^2\to L^2}\|\boldsymbol{\Theta}^{(1)}\|_{L^2\to L^2}\|\zeta\|_{L^2}}{\varepsilon^3\left(cn\log L\right)^3} \\
& +\frac{6\|\boldsymbol{\Theta}^{(1)}\|_{L^2\to L^2}^3\|\zeta\|_{L^2}}{\varepsilon^4\left(cn\log L\right)^4} \\
& +\frac{\|\zeta^{(1)}\|_{L^2}}{\varepsilon^2c^2n\log^2 L}\left(\frac{6P_1^2}{\log L}+3P_2\right)+\frac{9P_1\|\zeta^{(2)}\|_{L^2}}{\varepsilon^2c^2n\log^2 L}+\frac{\|13\zeta^{(3)}\|}{\varepsilon cn\log L} \\
\leq{} & \frac{P_3\|\zeta\|_{L^2}}{\varepsilon^2c^2n\log^2 L}+\frac{3P_2P_1\|\zeta\|_{L^2}}{\varepsilon^3c^3n\log^3 L}+\frac{6P_1^3\|\zeta\|_{L^2}}{\varepsilon^4c^4n\log^4 L} \\
& +\frac{\|\zeta^{(1)}\|_{L^2}}{\varepsilon^2c^2n\log^2 L}\left(\frac{6P_1^2}{\log L}+3P_2\right)+\frac{9P_1\|\zeta^{(2)}\|_{L^2}}{\varepsilon^2c^2n\log^2 L}+\frac{\|13\zeta^{(3)}\|}{\varepsilon cn\log L}.
\end{aligned}
$$

Plug in bounds for norms of $\zeta^{(1)}, \zeta^{(2)}$ and $\zeta^{(3)}$ we get

$$\|g_\varepsilon^{(3)}[\zeta]\|_{L^2} \le C_\varepsilon C_\zeta \left( \frac{P_1^3}{n \log^2 L} + \frac{P_1 P_2}{n \log^3 L} + \frac{P_3}{n \log^2 L} \right).$$

Combined with zero's order condition of $g_\varepsilon[\zeta]$, which follows directly from Theorem E.1, and we know that there exists $C_\varepsilon$ such that $g \in \Phi(\frac{C_\varepsilon C_\zeta}{n \log L}, \varepsilon)$. ∎

**Lemma E.26.** *For $\varepsilon \in (0, \frac{3}{4})$, when $L$ satisfies conditions in Theorem E.1, there exists positive constant $C$ such that*

$$\|\boldsymbol{\Theta}^\circ\|_{L^2 \to L^2} \le Cn \log(L).$$

**Proof.** As $\widehat{\boldsymbol{M}_\varepsilon}$ in (E.11) is invariant in Fourier basis as shown in Lemma E.6, we have $\|\boldsymbol{P}_{S_\varepsilon} \widehat{\boldsymbol{M}_\varepsilon} \boldsymbol{P}_{S_\varepsilon}\|_{L^2 \to L^2} \le \|\widehat{\boldsymbol{M}_\varepsilon}\|_{L^2 \to L^2}$. From Lemma E.2,

$$\begin{aligned}
\|\boldsymbol{\Theta}^\circ\|_{L^2 \to L^2} &= \|\widehat{\boldsymbol{M}_\varepsilon} + \boldsymbol{\Theta}^\circ - \widehat{\boldsymbol{M}_\varepsilon}\|_{L^2 \to L^2} \\
&\le \|\widehat{\boldsymbol{M}_\varepsilon}\|_{L^2 \to L^2} + \|\boldsymbol{\Theta}^\circ - \widehat{\boldsymbol{M}}\|_{L^2 \to L^2} \\
&\le \|\widehat{\boldsymbol{M}_\varepsilon}\|_{L^2 \to L^2} + (1 - \varepsilon)\lambda_{\min}(\boldsymbol{P}_{S_\varepsilon} \widehat{\boldsymbol{M}_\varepsilon} \boldsymbol{P}_{S_\varepsilon}) \\
&\le 2\|\widehat{\boldsymbol{M}_\varepsilon}\|_{L^2 \to L^2}.
\end{aligned}$$

As when $L \ge (\varepsilon^{-1/2} 6\pi\hat{\kappa})^{\frac{a_\varepsilon + 1}{a_\varepsilon}}$ we have $r_\varepsilon \le \frac{\sqrt{\varepsilon}}{\hat{\kappa}}$, where $a_\varepsilon, r_\varepsilon$ are defined in (E.7) and (E.8), and following Lemma E.4 we have

$$\angle \left( \boldsymbol{x}_\sigma(s), \boldsymbol{x}_\sigma(s') \right) \ge (1 - \varepsilon)|s - s'|$$

for any $\boldsymbol{x}_\sigma(s)$ and $|s - s'| \le r_\varepsilon$. Then follow Lemma E.5 and (F.11)Lemma F.9 and monotonicity of $\psi^\circ$ in Lemma G.5, we have

$$\begin{aligned}
\|\widehat{\boldsymbol{M}_\varepsilon}\|_{L^2 \to L^2} &\le \max_{\boldsymbol{x}_\sigma(s) \in \mathcal{M}} \int_{s' = s - r_\varepsilon}^{s + r_\varepsilon} \psi^\circ(\angle \boldsymbol{x}_\sigma(s), \boldsymbol{x}_\sigma(s')) ds' \\
&\le \max_{\boldsymbol{x}_\sigma(s) \in \mathcal{M}} \int_{s' = s - r_\varepsilon}^{s + r_\varepsilon} \psi^\circ((1 - \varepsilon)|s' - s|) ds' \\
&= (1 - \varepsilon)^{-1} \int_{t = -(1-\varepsilon)r_\varepsilon}^{(1-\varepsilon)t_\varepsilon} \psi^\circ(t) dt \\
&\le (1 - \varepsilon)^{-1} C \log \left( 1 + \frac{(L - 3)(1 - \varepsilon)r_\varepsilon}{3\pi} \right) \\
&\le C' n \log L
\end{aligned}$$

for some constant $C$, which concludes the claim. ∎

**Lemma E.27.** *For any $\varepsilon \in (0, \frac{3}{4})$, there exists constant $C \ge 0$ such that for any $g \in \Phi(C_g, \varepsilon)$, under the conditions of Theorem E.1, we have*

$$\boldsymbol{\Theta}^\circ[g] \in \Phi(CC_g n \log L, \varepsilon)$$

*As a consequence, for $\zeta \in S_\varepsilon \cap \Phi(C_\zeta, \varepsilon)$ letting $g_\varepsilon[\zeta]$ be the certificate in the statement of Theorem E.1, there exists number $C_\varepsilon \ge 0$ such that*

$$\boldsymbol{\Theta}^\circ[g_\varepsilon[\zeta]] - \zeta \in \Phi\left(C_\varepsilon C_\zeta, \varepsilon\right).$$

**Proof.** From Lemma E.26 we have $\|\boldsymbol{\Theta}^\circ\|_{L^2 \to L^2} \le Cn \log(L)$ for some constant $C$. Let $P_1, P_2, P_3$ be the polynomials in Lemmas E.18 to E.20. Following Lemmas E.18 to E.21 and the fact that $g \in \Phi(C_g, \varepsilon)$, we have following control for derivatives of $\boldsymbol{\Theta}^\circ[g]$:

$$\left\| \frac{d}{ds} \boldsymbol{\Theta}^\circ[g] \right\|_{L^2 \to L^2} = \|\boldsymbol{\Theta}^\circ[g^{(1)}] + \boldsymbol{\Theta}^{(1)}[g]\|_{L^2 \to L^2}$$

$$\leq Cn \log L \frac{P_1 C_g}{\log L} + P_1 n C_g$$
$$= (C+1) P_1 n C_g,$$

$$\left\| \frac{d^2}{ds^2} \mathbf{\Theta}^\circ[g] \right\|_{L^2 \to L^2} = \| \mathbf{\Theta}^\circ[g^{(2)}] + 2\mathbf{\Theta}^{(1)}[g^{(1)}] + \mathbf{\Theta}^{(2)}[g] \|_{L^2 \to L^2}$$
$$\leq Cn \log L \left( \frac{P_2 C_g}{\log L} + 2\frac{P_1^2 C_g}{\log^2 L} \right) + P_1 n \frac{P_1 C_g}{\log L} + P_2 n C_g$$
$$= (C+2) \left( P_2 C_g + \frac{P_1^2 C_g}{\log L} \right) n,$$

$$\left\| \frac{d^3}{ds^3} \mathbf{\Theta}^\circ[g] \right\|_{L^2 \to L^2} = \| \mathbf{\Theta}^\circ[g^{(3)}] + 3\mathbf{\Theta}^{(1)}[g^{(2)}] + 3\mathbf{\Theta}^{(2)}[g^{(1)}] + \mathbf{\Theta}^{(3)}[g] \|_{L^2 \to L^2}$$
$$\leq Cn \log L \left( \frac{P_3 C_g}{\log L} \right) + 3P_1 n \left( \frac{P_2 C_g}{\log L} + \frac{P_1^2 C_g}{\log^2 L} \right)$$
$$+ 3P_2 n \frac{P_1 C_g}{\log L} + P_3 n C_g$$
$$= (C+3) \left( P_3 C_g + \frac{P_1 P_2 C_g}{\log L} + \frac{P_1^3 C_g}{\log^2 L} \right) n.$$

which leads to $\mathbf{\Theta}^\circ[g] \in \Phi((C+3)C_g n \log L, \varepsilon)$ and finish the claim. The other part of the claim follows directly from Lemma E.25 as $g_\varepsilon[\zeta] \in \Phi\left( \frac{C_\varepsilon C_\zeta}{n \log L}, \varepsilon \right)$. $\blacksquare$

**Lemma E.28.** *Let* $\varepsilon \in (0, \frac{3}{4})$, $a_\varepsilon$, $r_\varepsilon$, $S_\varepsilon$ *be as in* (E.7), (E.8) *and* (E.12). *There exist numbers* $C_\varepsilon, C'_\varepsilon$ *such that when* $L \geq \left( \varepsilon^{1/2} \min\{\text{len}(\mathcal{M}_+), \text{len}(\mathcal{M}_-)\}/(12\pi^2) \right)^{-\frac{a_\varepsilon+1}{a_\varepsilon}}$ *and the conditions of Theorem E.1 are in force, for any* $w \in \Phi(C_w, \varepsilon)$, *we have*

$$\| \mathbf{P}_{S_\varepsilon^\perp} w \|_{L^2} \leq C_\varepsilon C_w \frac{r_\varepsilon^3}{\log L} \left( P_3 + \frac{P_1 P_2}{\log L} + \frac{P_1^3}{\log^2 L} \right).$$

*where* $P_1, P_2, P_3$ *are the polynomials from Lemmas E.18 to E.20 respectively.*

*As a consequence, for* $\zeta \in S_\varepsilon \cap \Phi(C_\zeta, \varepsilon)$, *letting* $g_\varepsilon[\zeta]$ *be as in Theorem E.1, we have*

$$\| \mathbf{\Theta}[g_\varepsilon[\zeta]] - \zeta \|_{L^2} \leq C'_\varepsilon C_\zeta \frac{r_\varepsilon^3}{\log L} \left( P_3 + \frac{P_1 P_2}{\log L} + \frac{P_1^3}{\log^2 L} \right).$$

*for some* $C'_\varepsilon > 0$.

**Proof.** When $L \geq \left( \varepsilon^{1/2} \min\{\text{len}(\mathcal{M}_+), \text{len}(\mathcal{M}_-)\}/(12\pi^2) \right)^{-\frac{a_\varepsilon+1}{a_\varepsilon}}$, from Lemma E.23 we have

$$\| \mathbf{P}_{S_\varepsilon^\perp} w \| \leq \left( \frac{2r_\varepsilon}{\sqrt{\varepsilon}} \right)^3 \| \mathbf{P}_{S_\varepsilon^\perp} w^{(3)} \|_{L^2}$$
$$\leq \left( \frac{2r_\varepsilon}{\sqrt{\varepsilon}} \right)^3 \| w^{(3)} \|_{L^2}$$
$$\leq C_w \left( \frac{2r_\varepsilon}{\sqrt{\varepsilon}} \right)^3 \left( \frac{P_3}{\log L} + \frac{P_2 P_1}{\log^2 L} + \frac{P_1^3}{\log^3 L} \right).$$

As $\zeta \in \Phi(C_\zeta, \varepsilon)$, from Lemma E.27 we get $\mathbf{\Theta}^\circ[g] - \zeta \in \Phi(C'_\varepsilon C_\zeta, \varepsilon)$ for some $C'_\varepsilon > 0$. The rest follows from the fact that $\mathbf{P}_{S_\varepsilon}[\mathbf{\Theta}^\circ[g] - \zeta] = 0$ and thus $\| \mathbf{\Theta}[g_\varepsilon[\zeta]] - \zeta \|_{L^2} = \| \mathbf{P}_{S_\varepsilon^\perp}[\mathbf{\Theta}^\circ[g] - \zeta] \|_{L^2}$. $\blacksquare$

## E.3 Certificates with Density and DC

**Proof Sketch and Organization.** In this section, we leverage the calculations in the previous sections to prove Theorem D.2, which gives a near solution to the equation

$$\boldsymbol{\Theta}_\mu[g](\boldsymbol{x}) = \int \Theta(\boldsymbol{x}, \boldsymbol{x}') g(\boldsymbol{x}') \rho(\boldsymbol{x}') d\boldsymbol{x}' = \zeta(\boldsymbol{x}).$$

To accomplish this, we need to account for two factors: the presence of a constant (DC) term in $\Theta(\boldsymbol{x}, \boldsymbol{x}') = \Theta^\circ(\boldsymbol{x}, \boldsymbol{x}') + \psi(\pi)$, and the presence of the data density $\rho$ in $\boldsymbol{\Theta}_\mu$.

Our approach is conceptually straightforward: since $\boldsymbol{\Theta} = \boldsymbol{\Theta}^\circ + \psi(\pi)\mathbb{1}\mathbb{1}^*$, we produce near solutions to two equations

$$\boldsymbol{\Theta}^\circ[g](\boldsymbol{x}) = \zeta(\boldsymbol{x}),$$
$$\boldsymbol{\Theta}^\circ[g_1](\boldsymbol{x}) = 1,$$

and then combine them to nearly solve $\boldsymbol{\Theta}[h] = \zeta$, by setting $h = g + \alpha g_1$ for an appropriate choice of $\alpha$,

$$\alpha = -\frac{\psi(\pi)\mathbb{1}[g]}{\psi(\pi)\mathbb{1}[g_1] + 1}. \tag{E.75}$$

Here and in the rest of this section, we write $\mathbb{1}[g]$ to denote $\mathbb{1}^*g$.

The statement of Theorem D.2 makes two demands on $h$: small approximation error $\|\boldsymbol{\Theta}[h] - \zeta\|_{L^2}$ and small size $\|h\|_{L^2}$. These demands introduce a tension, which forces us to work with DC subtracted solutions $g_\epsilon[\zeta]$ defined in (E.13) at multiple scales $\varepsilon$. We will set $g = g_{\varepsilon_0}[\zeta]$ with $\varepsilon_0$ small, which ensures that both $\|\boldsymbol{\Theta}^\circ[g] - \zeta\|_{L^2}$ and $\|g\|_{L^2}$ are small. We would like to similarly set $g_1 = g_{\varepsilon_1}[\zeta_1]$, with $\zeta_1 \equiv 1$. In order to ensure that $h$ is small, we need to ensure that the coefficient $\alpha$ defined in (E.75) is also small, which in turn requires a lower bound on $\mathbb{1}[g_1]$. This is straightforward if $g_1$ is (pointwise) nonnegative, but challenging if $g_1$ can take on arbitrary signs. The function $g_1$ is defined by the Neumann series

$$g_1 = g_{\varepsilon_1}[\zeta_1]$$
$$= \sum_{\ell=0}^\infty (-1)^\ell \left( \left( \boldsymbol{P}_{S_{\varepsilon_1}} \widehat{\boldsymbol{M}}_{\varepsilon_1} \boldsymbol{P}_{S_{\varepsilon_1}} \right)^{-1} \boldsymbol{P}_{S_{\varepsilon_1}} \left( \boldsymbol{\Theta}^\circ - \widehat{\boldsymbol{M}}_{\varepsilon_1} \right) \boldsymbol{P}_{S_{\varepsilon_1}} \right)^\ell \left( \boldsymbol{P}_{S_{\varepsilon_1}} \widehat{\boldsymbol{M}}_{\varepsilon_1} \boldsymbol{P}_{S_{\varepsilon_1}} \right)^{-1} \zeta_1.$$

Although this expression is complicated, the first ($\ell = 0$) summand is *always* nonnegative. If we choose $\varepsilon_1$ large, this expression will be dominated by the first term, providing the necessary control on $\mathbb{1}[g_1]$. So, we will use two different scales, $\varepsilon_0 < \varepsilon_1$ in constructing $g$ and $g_1$, respectively.

The issue introduced by the use of a large scale $\varepsilon_1$ is that the approximation error $\|\boldsymbol{\Theta}^\circ[g_1] - \zeta_1\|_{L^2}$ is not sufficiently small for our purposes. To address this issue, we introduce an iterative construction, which produces a sequence of increasingly accurate solutions $h_{(i)}$, each of which removes some portion of the approximation error in the previous solution. This sequence converges to our promised certificate $h$.

More concretely, we will set $\varepsilon_0 = \frac{1}{20}$ and $\varepsilon_1 = \frac{51}{100}$. For parameters $a_\varepsilon, r_\varepsilon$ defined in (E.7) and (E.8), these choices of $\varepsilon$ ensure that $a_{\varepsilon_0} > \frac{4}{5}$ and $a_{\varepsilon_1} > \frac{1}{9}$, and so

$$r_{\varepsilon_0} < 6\pi L^{-\frac{4}{9}},$$
$$r_{\varepsilon_1} < 6\pi L^{-\frac{1}{10}}.$$

We further choose $\delta_0 = 1 - \varepsilon_0$ and $\delta_1 = \delta_0\sqrt{\varepsilon_0}/\sqrt{\varepsilon_1} < 1 - \varepsilon_1$. This setting satisfies $\delta_0\sqrt{\varepsilon_0} = \delta_1\sqrt{\varepsilon_1}$ and thus allows

$$\circledast(\mathcal{M}) = \circledast_{\varepsilon_0, \delta_0}(\mathcal{M}) \geq \circledast_{\varepsilon_1, \delta_1}(\mathcal{M}). \tag{E.76}$$

In the remainder of this section, we carry out the argument described above. Lemma E.29 constructs the aforementioned certificate $g_1$ for the constant function $\zeta_1$. Lemma E.31 combines this construction with a certificate $g$ for $\zeta$ to give a (loose) approximate certificate, for the kernel $\boldsymbol{\Theta}$. Theorem E.32 amplifies this construction to reduce the approximation error to an appropriate level. Finally, we finish by incorporating the density $\rho(\boldsymbol{x})$ to prove our main result on certificates, Theorem D.2.

**Lemma E.29.** *Let $\zeta_1 \equiv 1$ denote the constant function over $\mathcal{M}$. When $L > C$ and the conditions of Theorem E.1 are satisfied for $\varepsilon = \varepsilon_1 = \frac{51}{100}$, then $g_1 = g_{\varepsilon_1}[\zeta_1]$ satisfies*

$$g_1 \in \Phi\left(\frac{C'}{n \log L}\|\zeta_1\|_{L^2}, \varepsilon_1\right)$$

*and*

$$\Theta^\circ[g_1] - \zeta_1 \in \Phi\left(C''\|\zeta_1\|_{L^2}, \varepsilon_1\right).$$

*We also have*

$$\mathbb{1}[g_1] \geq \left(2 - \frac{1}{\varepsilon_1}\right)\frac{\|\zeta_1\|_{L^1}}{C'''n \log(L)}, \tag{E.77}$$

*where $C, C', C''$ and $C'''$ are positive numerical constants.*

**Proof.** Applying Theorem E.1, as conditions of Theorem E.1 for $\varepsilon = \varepsilon_1$ is satisfied, we know $P_{S_{\varepsilon_1}}\widehat{M}_{\varepsilon_1}P_{S_{\varepsilon_1}}$ is invertible over $S_{\varepsilon_1}$. Noting that $\zeta_1$ is a constant function and thus $\zeta_1 \in S_{\varepsilon_1}$, we set

$$\begin{aligned}
g_1 &= g_{\varepsilon_1}[\zeta_1] \\
&= \sum_{\ell=0}^{\infty}(-1)^\ell \left(\left(P_{S_{\varepsilon_1}}\widehat{M}_{\varepsilon_1}P_{S_{\varepsilon_1}}\right)^{-1}P_{S_{\varepsilon_1}}\left(\Theta^\circ - \widehat{M}_{\varepsilon_1}\right)P_{S_{\varepsilon_1}}\right)^\ell \left(P_{S_{\varepsilon_1}}\widehat{M}_{\varepsilon_1}P_{S_{\varepsilon_1}}\right)^{-1}\zeta_1.
\end{aligned}$$

Since $\zeta_1$ is a constant function, all its derivatives are zero and thus $\zeta_1 \in \Phi(\|\zeta_1\|_{L^2}, \varepsilon_1)$. Applying Lemma E.25, we have that

$$g_1 \in \Phi\left(\frac{C_{\varepsilon_1}}{n \log L}\|\zeta_1\|_{L^2}, \varepsilon_1\right)$$

for certain $C_{\varepsilon_1} > 0$. The condition $\Theta^\circ[g_1] - \zeta_1 \in \Phi(C''\|\zeta_1\|_{L^2}, \varepsilon_1)$ follows from Lemma E.27 directly.

To control $\mathbb{1}[g_1]$, notice that because $P_{S_{\varepsilon_1}}\widehat{M}_{\varepsilon_1}P_{S_{\varepsilon_1}}$ is an invariant operator stably invertible in $S_{\varepsilon_1}$, and $\zeta_1 \in S_{\varepsilon_1}$, we can set

$$\widehat{g}_1 = \left(P_{S_{\varepsilon_1}}\widehat{M}_{\varepsilon_1}P_{S_{\varepsilon_1}}\right)^{-1}\zeta_1 = \left(\int_{s=-r_{\varepsilon_1}}^{r_{\varepsilon_1}}\psi^\circ(|s|)ds\right)^{-1}\zeta_1$$

which is also a positive constant function. We have

$$\begin{aligned}
g_1 &= \sum_{\ell=0}^{\infty}(-1)^\ell \left(\left(P_{S_{\varepsilon_1}}\widehat{M}_{\varepsilon_1}P_{S_{\varepsilon_1}}\right)^{-1}P_{S_{\varepsilon_1}}(\Theta^\circ - \widehat{M}_{\varepsilon_1})P_{S_{\varepsilon_1}}\right)^\ell \widehat{g}_1 \\
&= \widehat{g}_1 + \sum_{\ell=1}^{\infty}(-1)^\ell \left(\left(P_{S_{\varepsilon_1}}\widehat{M}_{\varepsilon_1}P_{S_{\varepsilon_1}}\right)^{-1}P_{S_{\varepsilon_1}}\left(\Theta^\circ - \widehat{M}_{\varepsilon_1}\right)P_{S_{\varepsilon_1}}\right)^\ell \widehat{g}_1,
\end{aligned}$$

and from Lemma E.2, we have

$$\left\|\left(P_{S_{\varepsilon_1}}\widehat{M}_{\varepsilon_1}P_{S_{\varepsilon_1}}\right)^{-1}P_{S_{\varepsilon_1}}\left(\Theta^\circ - \widehat{M}_{\varepsilon_1}\right)P_{S_{\varepsilon_1}}\right\|_{L^2 \to L^2} \leq 1 - \varepsilon_1,$$

and so

$$\begin{aligned}
\|g_1 - \widehat{g}_1\|_{L^1} &\leq \sqrt{\text{len}(\mathcal{M})}\|g_1 - \widehat{g}_1\|_{L^2} \\
&\leq \sqrt{\text{len}(\mathcal{M})}\sum_{\ell=1}^{\infty}(1 - \varepsilon_1)^\ell\|\widehat{g}_1\|_{L^2} \\
&= \sqrt{\text{len}(\mathcal{M})}\left(\frac{1}{\varepsilon_1} - 1\right)\|\widehat{g}_1\|_{L^2} \\
&= \left(\frac{1}{\varepsilon_1} - 1\right)\|\widehat{g}_1\|_{L^1}
\end{aligned}$$

$$= \left( \frac{1}{\varepsilon_1} - 1 \right) \mathbb{1}[\widehat{g}_1].$$

The first inequality comes from the equivalence of norms, and the last two lines come from the fact that $\widehat{g}_1$ is a positive constant function. Thus, following Lemma F.9, there exist constants $C, C'$ such that

$$\begin{aligned}
\mathbb{1}[g_1] &\geq \mathbb{1}[\widehat{g}_1] - \|g_1 - \widehat{g}_1\|_{L^1} \\
&\geq \left( 2 - \frac{1}{\varepsilon_1} \right) \mathbb{1}[\widehat{g}_1] \\
&= \left( 2 - \frac{1}{\varepsilon_1} \right) \left( \int_{s=-r_{\varepsilon_1}}^{r_{\varepsilon_1}} \psi^\circ(|s|) ds \right)^{-1} \|\zeta_1\|_{L^1} \\
&\geq \frac{\left( 2 - \frac{1}{\varepsilon_1} \right) \|\zeta_1\|_{L^1}}{\frac{3\pi n}{8} C' \log \left( 1 + \frac{1}{3\pi}(L-3) r_{\varepsilon_1} \right)} \\
&\geq \frac{\left( 2 - \frac{1}{\varepsilon_1} \right) \|\zeta_1\|_{L^1}}{Cn \log(L)}
\end{aligned}$$

as claimed. ∎

**Lemma E.30.** *Suppose that the conditions of Theorem E.1 are satisfied for both $\varepsilon = \varepsilon_0 = \frac{1}{20}$ and $\varepsilon = \varepsilon_1 = \frac{51}{100}$, and let $g_1, \zeta_1$ be as in Lemma E.29. Let $\zeta \in S_{\varepsilon_0}$, and $g = g_{\varepsilon_0}[\zeta]$. Then*

$$\alpha = -\frac{\psi(\pi) \mathbb{1}[g]}{\psi(\pi) \mathbb{1}[g_1] + 1},$$

*satisfies*

$$|\alpha| \leq \frac{C\|\zeta\|_{L^2}}{\|\zeta_1\|_{L^2}},$$

*where $C$ is a numerical constant.*

**Proof.** Set

$$\widehat{g} = \left( P_{S_{\varepsilon_0}} \widehat{M}_{\varepsilon_0} P_{S_{\varepsilon_0}} \right)^{-1} \zeta$$

again, we have

$$g = \sum_{\ell=0}^{\infty} (-1)^\ell \left( \left( P_{S_{\varepsilon_0}} \widehat{M}_{\varepsilon_0} P_{S_{\varepsilon_0}} \right)^{-1} P_{S_{\varepsilon_0}} \left( \Theta^\circ - \widehat{M}_{\varepsilon_0} \right) P_{S_{\varepsilon_0}} \right)^\ell \widehat{g}.$$

Then there exist constants $c, C > 0$ that

$$\begin{aligned}
\|g\|_{L^1} &\leq \sqrt{\text{len}(\mathcal{M})} \|g\|_{L^2} \\
&\leq \sqrt{\text{len}(\mathcal{M})} \sum_{\ell=0}^{\infty} (1 - \varepsilon_0)^\ell \|\widehat{g}\|_{L^2} \\
&= \sqrt{\text{len}(\mathcal{M})} \frac{1}{\varepsilon_0} \|\widehat{g}\|_{L^2} \\
&\leq \sqrt{\text{len}(\mathcal{M})} \frac{1}{\varepsilon_0 \lambda_{\min} \left( P_{S_{\varepsilon_0}} \widehat{M}_{\varepsilon_0} P_{S_{\varepsilon_0}} \right)} \|\zeta\|_{L^2} \\
&\leq \sqrt{\text{len}(\mathcal{M})} \frac{\|\zeta\|_{L^2}}{\varepsilon_0 cn \log L} \\
&\leq \sqrt{\text{len}(\mathcal{M})} \frac{C\|\zeta\|_{L^2}}{n \log L},
\end{aligned}$$

where in the penultimate inequality, we have used Lemma E.6. Applying the previous lemma, we obtain

$$
\begin{aligned}
|\alpha| &\leq \frac{\psi(\pi)|\mathbb{1}[g]|}{\psi(\pi)\mathbb{1}[g_1]} \\
&\leq \frac{Cn\log(L)\|g\|_{L^1}}{(2-\frac{1}{\varepsilon_1})\|\zeta_1\|_{L^1}} \\
&\leq \frac{C\sqrt{\operatorname{len}(\mathcal{M})}\|\zeta\|_{L^2}}{\|\zeta_1\|_{L^1}} \\
&= \frac{C\|\zeta\|_{L^2}}{\|\zeta_1\|_{L^2}},
\end{aligned}
$$

where in the final equation we have used that the constant function $\zeta_1$ satisfies $\|\zeta_1\|_{L^1} = \sqrt{\operatorname{len}(\mathcal{M})}\|\zeta_1\|_{L^2}$. ∎

**Lemma E.31.** *Suppose that the conditions of Theorem E.1 are satisfied for both $\varepsilon = \varepsilon_0 = \frac{1}{20}$ and $\varepsilon = \varepsilon_1 = \frac{51}{100}$, and $L \geq \left(\varepsilon^{1/2}\min\{\operatorname{len}(\mathcal{M}_+), \operatorname{len}(\mathcal{M}_-)\}/(12\pi^2)\right)^{-\frac{a_\varepsilon+1}{a_\varepsilon}}$ for both $\varepsilon = \varepsilon_0$ and $\varepsilon = \varepsilon_1$. There exist numerical constants $C, C' > 0$, such that for every $\zeta \in S_{\varepsilon_0} \cap \Phi(C_\zeta, \varepsilon_1)$, there exists $h$ such that*

$$
\|h\|_{L^2} \leq \frac{C\|\zeta\|_{L^2}}{n\log L}, \tag{E.78}
$$

$$
\|\mathbf{\Theta}[h] - \zeta\|_{L^2} \leq CC_\zeta \frac{P\left(M_3, M_4, M_5, \operatorname{len}(\mathcal{M}), \Delta_{\varepsilon_0}^{-1}\right) \times r_{\varepsilon_1}^3}{\log L}, \tag{E.79}
$$

$$
\|\boldsymbol{P}_{S_{\varepsilon_0}^\perp}[\mathbf{\Theta}[h] - \zeta]\|_{L^2} \leq CC_\zeta P\left(M_3, M_4, M_5, \operatorname{len}(\mathcal{M}), \Delta_{\varepsilon_0}^{-1}\right) \times L^{-\frac{4}{3}}, \tag{E.80}
$$

*where $P$ is a polynomial $\operatorname{poly}\{M_3, M_4, M_5, \operatorname{len}(\mathcal{M}), \Delta_\varepsilon^{-1}\}$ of degree $\leq 9$, with degree $\leq 3$ in $M_3, M_4, M_5 \operatorname{len}(\mathcal{M})$, and degree $\leq 6$ in $\Delta_\varepsilon^{-1}$. Furthermore, we have*

$$
\boldsymbol{P}_{S_{\varepsilon_0}}[\mathbf{\Theta}[h] - \zeta] \in \Phi\left(C'\|\zeta\|_{L^2}, \varepsilon_0\right). \tag{E.81}
$$

**Proof.** Recall that $\mathbf{\Theta} = \mathbf{\Theta}^\circ + \psi(\pi)\mathbb{1}$, and let $g_1$ denote the solution for $\zeta_1 \equiv 1$ as in Lemma E.29. Set $h = g + \alpha g_1$, where $g = g_{\varepsilon_0}[\zeta]$, and

$$
\alpha = -\frac{\psi(\pi)\mathbb{1}[g]}{\psi(\pi)\mathbb{1}[g_1]+1}.
$$

Using Theorem E.1 to control the norms of $g$ and $g_1$, and using Lemma E.30 to control $|\alpha|$, we have

$$
\begin{aligned}
\|h\|_{L^2} &\leq \|g\|_{L^2} + |a|\|g_1\|_{L^2} \\
&\leq \frac{C\|\zeta\|_{L^2}}{n\log L} + |\alpha|\frac{C\|\zeta_1\|_{L^2}}{n\log L} \\
&\leq \frac{C\|\zeta\|_{L^2}}{n\log L},
\end{aligned}
$$

establishing (E.78).

From our choice of $\alpha$,

$$
\begin{aligned}
\mathbf{\Theta}[h] - \zeta &= \mathbf{\Theta}^\circ[g] - \zeta + \alpha\mathbf{\Theta}^\circ[g_1] + \psi(\pi)\mathbb{1}[g] + \alpha\psi(\pi)\mathbb{1}[g_1] \\
&= \mathbf{\Theta}^\circ[g] - \zeta + \alpha\left(\mathbf{\Theta}^\circ[g_1] - \zeta_1\right). \tag{E.82}
\end{aligned}
$$

Using Lemma E.28 and the fact that $\zeta_1 \in \Phi(\|\zeta_1\|, \varepsilon_1)$ we have

$$
\begin{aligned}
\|\mathbf{\Theta}[h] - \zeta\|_{L^2} &\leq \|\mathbf{\Theta}^\circ[g] - \zeta\|_{L^2} + |\alpha|\|\mathbf{\Theta}^\circ[g_1] - \zeta_1\|_{L^2} \\
&\leq C_{\varepsilon_0}C_\zeta\frac{r_{\varepsilon_0}^3}{\log L}\left(P_3 + \frac{P_1P_2}{\log L} + \frac{P_1^3}{\log^2 L}\right)
\end{aligned}
$$

$$+|\alpha|C_{\varepsilon_1}\|\zeta_1\|_{L^2}\frac{r_{\varepsilon_1}^3}{\log L}\left(P_3+\frac{P_1P_2}{\log L}+\frac{P_1^3}{\log^2 L}\right).$$

Further using Lemma E.30 to bound $\alpha\|\zeta_1\|_{L^2}\le C'\|\zeta\|_{L^2}\le C'C_\zeta$ and $r_{\varepsilon_0}\le r_{\varepsilon_1}$, we have

$$\|\mathbf{\Theta}[h]-\zeta\|_{L^2}\le C'C_\zeta\frac{r_{\varepsilon_1}^3}{\log L}\left(P_3+\frac{P_1P_2}{\log L}+\frac{P_1^3}{\log^2 L}\right) \tag{E.83}$$

for some absolute constant $C'>0$. Notice that from Lemma E.3 $M_2<2\hat\kappa\le 2\Delta_\varepsilon^{-2}$ and thus

$$
\begin{aligned}
P_3+&\frac{P_1P_2}{\log L}+\frac{P_1^3}{\log^2 L}\\
&\le P_3+P_1P_2+P_1^3\\
&=C_3(M_4^3+M_3M_4+M_4M_5+\mathrm{len}(\mathcal{M})\left(\Delta_\varepsilon^{-4}+M_2\Delta_\varepsilon^{-3}+M_3\Delta_\varepsilon^{-2}\right))\\
&\quad+C_2(M_4^2+M_5+\mathrm{len}(\mathcal{M})\left(\Delta_\varepsilon^{-3}+M_2\Delta_\varepsilon^{-2}\right))C_1(M_4+\mathrm{len}(\mathcal{M})\Delta_\varepsilon^{-2})\\
&\quad+C_1^3(M_4+\mathrm{len}(\mathcal{M})\Delta_\varepsilon^{-2})^3\\
&\le C(M_4^3+M_3M_4+M_4M_5\\
&\quad+\mathrm{len}(\mathcal{M})^3\Delta_\varepsilon^{-6}+\mathrm{len}(\mathcal{M})^2\left(\Delta_\varepsilon^{-5}+M_2\Delta_\varepsilon^{-4}\right)\\
&\quad+\mathrm{len}(\mathcal{M})(\Delta_\varepsilon^{-4}+M_2\Delta_\varepsilon^{-3}+M_3\Delta_\varepsilon^{-2}\\
&\quad+M_4^2\Delta_\varepsilon^{-2}+M_5\Delta_\varepsilon^{-2}+M_4\Delta_\varepsilon^{-3}+M_2M_4\Delta_\varepsilon^{-2}))\\
&\le C(M_4^3+M_3M_4+M_4M_5+\mathrm{len}(\mathcal{M})^3\Delta_\varepsilon^{-6}+\Delta_\varepsilon^{-6}\\
&\quad+\mathrm{len}(\mathcal{M})\Delta_\varepsilon^{-2}\left(M_3+M_5\right))\\
&\overset{\mathrm{def}}{=}P\Big(M_3,M_4,M_5,\mathrm{len}(\mathcal{M}),\Delta_{\varepsilon_0}^{-1}\Big) \tag{E.84}
\end{aligned}
$$

where $P\Big(M_3,M_4,M_5,\mathrm{len}(\mathcal{M}),\Delta_{\varepsilon_0}^{-1}\Big)$ is a polynomial of $M_3,M_4,M_5,\mathrm{len}(\mathcal{M}),\Delta_\varepsilon^{-1}$ of degree $\le 9$, with degree $\le 3$ in $M_3,M_4,M_5\,\mathrm{len}(\mathcal{M})$, and degree $\le 6$ in $\Delta_\varepsilon^{-1}$. Here $P_1,P_2$ and $P_3$ are polynomials defined in Lemma E.18, Lemma E.19 and Lemma E.20. This together with (E.83) give us (E.79).

To obtain the tighter bound (E.80) on $\mathbf{P}_{S_{\varepsilon_0}^\perp}(\mathbf{\Theta}[h]-\zeta)$, we begin by applying Lemma E.27 with $\zeta\in\Phi(C_\zeta,\varepsilon_0)$ and $\zeta_1\in\Phi(\|\zeta_1\|_{L^2},\varepsilon_1)$, we have

$$
\begin{aligned}
\mathbf{\Theta}^\circ[g]-\zeta&\in\Phi\Big(C'_{\varepsilon_0}C_\zeta,\varepsilon_0\Big),\\
\mathbf{\Theta}^\circ[g_1]-\zeta_1&\in\Phi\Big(C'_{\varepsilon_1}\|\zeta_1\|_{L^2},\varepsilon_1\Big)\quad\subseteq\quad\Phi\Big(C'_{\varepsilon_1}\|\zeta_1\|_{L^2},\varepsilon_0\Big)
\end{aligned} \tag{E.85}
$$

for certain $C'_{\varepsilon_0},C'_{\varepsilon_1}>0$. Using $\mathbf{\Theta}[h]-\zeta=\mathbf{\Theta}^\circ[g]-\zeta+\alpha(\mathbf{\Theta}^\circ[g_1]-\zeta_1)$, we have

$$C'_{\varepsilon_0}C_\zeta+\frac{C'\|\zeta\|_{L^2}}{\|\zeta_1\|_{L^2}}\times C'_{\varepsilon_1}\|\zeta_1\|_{L^2}\le C''C_\zeta.$$

for some constant $C''>0$ and thus

$$\mathbf{\Theta}[h]-\zeta\in\Phi(C''C_\zeta,\varepsilon_0). \tag{E.86}$$

Applying Lemma E.28 with $w=\mathbf{\Theta}[h]-\zeta$ and simplifying with (E.84) we obtain

$$\|\mathbf{P}_{S_{\varepsilon_0}^\perp}(\mathbf{\Theta}[h]-\zeta)\|_{L^2}\le C''C_\zeta\frac{P\Big(M_3,M_4,M_5,\mathrm{len}(\mathcal{M}),\Delta_{\varepsilon_0}^{-1}\Big)\times r_{\varepsilon_0}^3}{\log L}.$$

(E.80) follows from $r_{\varepsilon_0}<6\pi L^{-4/9}$, which implies that $r_{\varepsilon_0}^3/\log L\le L^{-4/3}$ when $L$ is larger than an appropriate numerical constant.

Finally, since $\mathbf{P}_{S_{\varepsilon_0}}\mathbf{\Theta}^\circ[g]=\mathbf{P}_{S_{\varepsilon_0}}\zeta$,

$$\mathbf{P}_{S_{\varepsilon_0}}[\mathbf{\Theta}[h]-\zeta]\;=\;\alpha\mathbf{P}_{S_{\varepsilon_0}}\left[\mathbf{\Theta}^\circ[g_1]-\zeta_1\right]. \tag{E.87}$$

From Lemma E.22, $\boldsymbol{P}_{S_{\varepsilon_0}}$ commutes with differentiation, and so for any $i$-times differentiable $w$,

$$\|(\boldsymbol{P}_{S_{\varepsilon_0}}w)^{(i)}\|_{L^2} \leq \|w^{(i)}\|_{L^2}.$$

Applying this with $i = 0, 1, 2, 3$, we see that for any $w \in \Phi(C_w, \varepsilon)$, $\boldsymbol{P}_{S_{\varepsilon_0}}w \in \Phi(C_w, \varepsilon)$. Applying this observation to (E.85), we have

$$\boldsymbol{P}_{S_{\varepsilon_0}}\left(\boldsymbol{\Theta}^\circ[g_1] - \zeta_1\right) \in \Phi\left(C'_{\varepsilon_1}\|\zeta_1\|_{L^2}, \varepsilon_0\right).$$

Combining with (E.87), we have

$$\boldsymbol{P}_{S_{\varepsilon_0}}(\boldsymbol{\Theta}[h] - \zeta) \in \Phi\left(|\alpha|C'_{\varepsilon_1}\|\zeta_1\|_{L^2}, \varepsilon_0\right) \subseteq \Phi\left(C'C'_{\varepsilon_1}\|\zeta\|_{L^2}, \varepsilon_0\right),$$

which is (E.81). Here, we have used the bound on $\alpha$ from the previous lemma. This completes the proof. ∎

**Theorem E.32** (Certificates for DC kernel). *Suppose that the conditions of Theorem E.1 are satisfied for both $\varepsilon = \varepsilon_0 = \frac{1}{20}$ and $\varepsilon = \varepsilon_1 = \frac{51}{100}$ and $L \geq \left(\varepsilon^{1/2}\min\{\mathrm{len}(\mathcal{M}_+), \mathrm{len}(\mathcal{M}_-)\}/(12\pi^2)\right)^{-\frac{a_\varepsilon+1}{a_\varepsilon}}$ for both $\varepsilon = \varepsilon_0$ and $\varepsilon = \varepsilon_1$. There exist constants $C, C''$ such that for any number $K > 0$, when*

$$L \geq CK^4 P\left(M_3, M_4, M_5, \mathrm{len}(\mathcal{M}), \Delta_{\varepsilon_0}^{-1}\right)^4, \tag{E.88}$$

*then for any $\zeta \in S_{\varepsilon_0} \cap \Phi(K\|\zeta\|_{L^2}, \varepsilon_0)$ there exists a certificate $h$ satisfying*

$$\|\boldsymbol{\Theta}[h] - \zeta\|_{L^2} \leq \|\zeta\|_{L^\infty}L^{-1}, \tag{E.89}$$

*with*

$$\|h\|_{L^2} \leq \frac{C''\|\zeta\|_{L^2}}{n \log L}.$$

*In (E.88), $P$ is the polynomial defined in Lemma E.31.*

**Proof.** Let $\zeta_{(0)} = \zeta$, and iteratively define

$$\zeta_{(i+1)} = -\boldsymbol{P}_{S_{\varepsilon_0}}\left(\boldsymbol{\Theta}[h_{(i)}] - \zeta_{(i)}\right) \in S_{\varepsilon_0}.$$

where $h_{(i)}$ is the approximate certificate of $\zeta_{(i)} \in S_{\varepsilon_0}$ constructed in Lemma E.31. From (E.81), we have

$$\zeta_{(i+1)} \in \Phi\left(C_1\|\zeta_{(i)}\|_{L^2}, \varepsilon_0\right),$$

where $C_1$ is a numerical constant. Hence, for $i \geq 1$, from Lemma E.31 we have

$$\left\|h_{(i)}\right\|_{L^2} \leq \left(\frac{C_2}{n \log L}\right)\|\zeta_{(i)}\|_{L^2},$$

$$\left\|\boldsymbol{\Theta}[h_{(i)}] - \zeta_{(i)}\right\|_{L^2} \leq C_2 C_1 \|\zeta_{(i-1)}\|_{L^2}\frac{Pr_{\varepsilon_1}^3}{\log L}, \tag{E.90}$$

$$\left\|\boldsymbol{P}_{S_{\varepsilon_0}^\perp}\left[\boldsymbol{\Theta}[h_{(i)}] - \zeta_{(i)}\right]\right\|_{L^2} \leq C_2 C_1 \|\zeta_{(i-1)}\|_{L^2}PL^{-4/3}.$$

For $i = 0$, as $\zeta_{(0)} \in \Phi\left(K\|\zeta\|_{L^2}, \varepsilon_0\right)$, this simplifies to

$$\left\|h_{(0)}\right\|_{L^2} \leq \left(\frac{C_2}{n \log L}\right)\|\zeta\|_{L^2},$$

$$\left\|\boldsymbol{\Theta}[h_{(0)}] - \zeta_{(0)}\right\|_{L^2} \leq C_2 K\|\zeta\|_{L^2}\frac{Pr_{\varepsilon_1}^3}{\log L}, \tag{E.91}$$

$$\left\|\boldsymbol{P}_{S_{\varepsilon_0}^\perp}\left[\boldsymbol{\Theta}[h_{(0)}] - \zeta_{(0)}\right]\right\|_{L^2} \leq C_2 K\|\zeta\|_{L^2}PL^{-4/3}.$$

We use these relationships to control $\|\zeta_{(i)}\|_{L^2}$. As $r_{\varepsilon_1} \leq 6\pi L^{-1/10}$, there exists a constant $C$ such that when $L \geq CK^4 P^4$, $C_2 K\frac{Pr_{\varepsilon_1}^3}{\log L} \leq \tau = \frac{1}{2}$ and $C_2 C_1\frac{Pr_{\varepsilon_1}^3}{\log L} \leq \tau^2$. We argue by induction that

$$\|\zeta_{(i)}\|_{L^2} \leq \tau^i\|\zeta\|_{L^2} \quad \forall\, i \geq 0.$$

This is true by construction for $i = 0$, while for $i = 1$ it follows from (E.91). Finally, for $i \geq 2$, using (E.90) and inductive hypothesis, we have

$$
\begin{aligned}
\|\zeta_{(i)}\|_{L^2} &\leq \|\Theta[h_{(i-1)}] - \zeta_{(i-1)}\|_{L^2} \\
&\leq C_2 C_1 \|\zeta_{(i-2)}\|_{L^2} \frac{P r_{\varepsilon_1}^3}{\log L}, \\
&\leq \tau^2 \|\zeta_{(i-2)}\|_{L^2} \\
&\leq \tau^i \|\zeta\|_{L^2},
\end{aligned}
$$

as claimed.

We set

$$
h = \sum_{i=0}^{k} h_{(i)},
$$

where $k$ will be specified below. By construction,

$$
\begin{aligned}
\|h\|_{L^2} &\leq \sum_{i=0}^{k} \|h_{(i)}\|_{L^2} \\
&\leq \frac{C_1}{n \log L} \sum_{i=0}^{k} \|\zeta_{(i)}\|_{L^2} \\
&\leq \frac{2C_1}{n \log L} \|\zeta\|_{L^2},
\end{aligned}
$$

as claimed.

We next verify that $\Theta[h]$ is an accurate approximation to $\zeta$:

$$
\begin{aligned}
\left\|\Theta[h] - \zeta_{(0)}\right\|_{L^2} &\leq \left\|P_{S_{\varepsilon_0}}[\Theta[h] - \zeta_{(0)}]\right\|_{L^2} + \left\|P_{S_{\varepsilon_0}^\perp} \Theta[h]\right\|_{L^2} \\
&\leq \left\|P_{S_{\varepsilon_0}} \left[\Theta[h_{(k)}] - \zeta_{(k)}\right]\right\|_{L^2} + \sum_{i=0}^{k} \left\|P_{S_{\varepsilon_0}^\perp} \Theta[h_{(i)}]\right\|_{L^2} \\
&\leq \tau^{k+1} \left\|\zeta_{(0)}\right\|_{L^2} + C_2 C_1 P L^{-4/3} \sum_{i=1}^{k} \|\zeta_{(i-1)}\|_{L^2} \\
&\quad + \quad C_2 K P L^{-4/3} \|\zeta\|_{L^2} \\
&\leq \tau^{k+1} \left\|\zeta_{(0)}\right\|_{L^2} + C_2 P L^{-4/3} (2C_1 + K) \|\zeta\|_{L^2}.
\end{aligned}
$$

Choosing $k$ appropriately, and ensuring that $C_2(2C_1 + K) P L^{-4/3} < \frac{1}{2} L^{-1}$, establishes (E.89). The latter condition follows immediately, from $L \geq C K^4 P^4$ for appropriately large $C$. ∎

**Lemma E.33.** *There exist constants $C, C', C'', C'''$ and a polynomial $P = \mathrm{poly}\{M_3, M_4, M_5, \mathrm{len}(\mathcal{M}), \Delta^{-1}\}$ of degree $\leq 36$, with degree $\leq 12$ in $M_3, M_4, M_5 \, \mathrm{len}(\mathcal{M})$, and degree $\leq 24$ in $\Delta^{-1}$ such that for any number $K > 0$, when*

$$
L \geq \max\left\{ \exp(C' \, \mathrm{len}(\mathcal{M}) \hat{\kappa}), \left(\frac{1}{\Delta\sqrt{1+\kappa^2}}\right)^{C'' \circledast(\mathcal{M})}, C''' \hat{\kappa}^{10}, K^4 P, \rho_{\max}^{12} \right\}
$$

*then for any real $\zeta \in \Phi(K\|\zeta\|_{L^2}, \frac{1}{20})$, there exists a real certificate $g : \mathcal{M} \to \mathbb{R}$ with*

$$
\|g\|_{L_\mu^2} \leq \frac{C\|\zeta\|_{L_\mu^2}}{\rho_{\min} n \log L}
$$

*such that*

$$
\|\Theta_\mu[g] - \zeta\|_{L_\mu^2} \leq \|\zeta\|_{L^\infty} L^{-1}
$$

**Proof.** We first show that there exists constants $C, C', C'', C'''$ that under condition of the lemma, conditions of Theorem E.32 are satisfied. From definition, $\Delta = \Delta_{\varepsilon_0} \leq \Delta_{\varepsilon_1}$, as we choose $\delta_0 = 1 - \varepsilon_0, \delta_1 = \delta_0\sqrt{\varepsilon_0}/\sqrt{\varepsilon_1} < 1 - \varepsilon_1$, we have $\delta_0\sqrt{\varepsilon_0} = \delta_1\sqrt{\varepsilon_1}$ and thus $\maltese(\mathcal{M}) = \maltese_{\varepsilon_0,\delta_0}(\mathcal{M}) \geq \maltese_{\varepsilon_1,\delta_1}(\mathcal{M})$ as in (E.76). Thus conditions of Theorem E.1 for $\varepsilon = \varepsilon_1$ can be absorbed into conditions for $\varepsilon = \varepsilon_0$ with a change of constant factors.

Furthermore, as $\Delta \leq \frac{\sqrt{\varepsilon_0}}{\hat{\kappa}} < \frac{1}{2\sqrt{1+\kappa^2}}$, we can alter the condition in Theorem E.1 by $\left(1 + \frac{1}{\Delta\sqrt{1+\kappa^2}}\right)^{C''\maltese(\mathcal{M})} \leq \left(\frac{1}{\Delta\sqrt{1+\kappa^2}}\right)^{2C''\maltese(\mathcal{M})}$ and thus reduce to the form in the statement.

We then notice that from Lemma E.3 $\min\{\mathrm{len}(\mathcal{M}_+), \mathrm{len}(\mathcal{M}_-)\} \geq \frac{1}{\hat{\kappa}}$, and we can choose $C'$ such that $\exp(C'\,\mathrm{len}(\mathcal{M})\hat{\kappa}) \geq \exp(C') \geq C_{\varepsilon_0}$. Similarly, by choosing $C'''$ appropriately, $L \geq C'''\hat{\kappa}^{10}$ implies both $L \geq (\varepsilon^{-1/2}6\pi\hat{\kappa})^{\frac{a_\varepsilon+1}{a_\varepsilon}}$ and $L \geq (\varepsilon^{-1/2}12\pi^2\min\{\mathrm{len}(\mathcal{M}_+), \mathrm{len}(\mathcal{M}_+)\})^{\frac{a_\varepsilon+1}{a_\varepsilon}}$ for both $\varepsilon = \varepsilon_0$ and $\varepsilon = \varepsilon_1$ as $a_{\varepsilon_0} > \frac{4}{5}$ and $a_{\varepsilon_1} > \frac{1}{9}$.

From Theorem E.32, we know there exists $g$ such that

$$\|\boldsymbol{\Theta}[g] - \zeta\|_{L^2} \leq \|\zeta\|_{L^\infty} L^{-1} \tag{E.92}$$

with

$$\|g\|_{L^2} \leq \frac{C\|\zeta\|_{L^2}}{n\log L}. \tag{E.93}$$

We can further require this $g$ to be a real function over the manifold. To see this, notice that for any $\boldsymbol{x}, \boldsymbol{x}' \in \mathcal{M}$, both the kernel $\Theta(\boldsymbol{x}, \boldsymbol{x}')$ and $\zeta(\boldsymbol{x})$ are real, thus if we take the real component of $g$ as $\hat{g} = (g + \bar{g})/2$, then we have $\|\hat{g}\|_{L^2} \leq \|g\|_{L^2}$ and further by the triangle inequality

$$
\begin{aligned}
|\boldsymbol{\Theta}[\hat{g}](\boldsymbol{x}) - \zeta(\boldsymbol{x})| &= \left|\int_{\boldsymbol{x}'\in\mathcal{M}} \Theta(\boldsymbol{x}, \boldsymbol{x}')\hat{g}(\boldsymbol{x}')d\boldsymbol{x}' - \zeta(\boldsymbol{x})\right| \\
&\leq \left(\left|\int_{\boldsymbol{x}'\in\mathcal{M}} \Theta(\boldsymbol{x}, \boldsymbol{x}')\hat{g}(\boldsymbol{x}')d\boldsymbol{x}' - \zeta(\boldsymbol{x})\right|^2 \right.\\
&\qquad \left. + \left|\int_{\boldsymbol{x}'\in\mathcal{M}} \Theta(\boldsymbol{x}, \boldsymbol{x}')(1/2)(g - \bar{g})(\boldsymbol{x}')d\boldsymbol{x}'\right|^2\right)^{1/2} \\
&= \left|\int_{\boldsymbol{x}'\in\mathcal{M}} \Theta(\boldsymbol{x}, \boldsymbol{x}')g(\boldsymbol{x}')d\boldsymbol{x}' - \zeta(\boldsymbol{x})\right|.
\end{aligned}
$$

The last line comes from the fact that $\int_{\boldsymbol{x}'\in\mathcal{M}} \Theta(\boldsymbol{x}, \boldsymbol{x}')\hat{g}(\boldsymbol{x}')d\boldsymbol{x}' - \zeta(\boldsymbol{x})$ and $\int_{\boldsymbol{x}'\in\mathcal{M}} \Theta(\boldsymbol{x}, \boldsymbol{x}')(1/2)(g - \bar{g})(\boldsymbol{x}')d\boldsymbol{x}'$ are the pure real and imaginary part of $\int_{\boldsymbol{x}'\in\mathcal{M}} \Theta(\boldsymbol{x}, \boldsymbol{x}')g(\boldsymbol{x}')d\boldsymbol{x}' - \zeta(\boldsymbol{x})$. Thus $\hat{g}$ is real and also satisfies (E.92) and (E.93).

To include the density, define $g_\mu(\boldsymbol{x}) = g(\boldsymbol{x})/\rho(\boldsymbol{x})$. We get

$$
\begin{aligned}
\|g_\mu\|_{L^2_\mu}^2 &= \int_{\boldsymbol{x}\in\mathcal{M}} |g_\mu(\boldsymbol{x})|^2 \rho(\boldsymbol{x})d\boldsymbol{x} \\
&= \int_{\boldsymbol{x}\in\mathcal{M}} |g(\boldsymbol{x})|^2 \rho^{-1}(\boldsymbol{x})d\boldsymbol{x} \\
&\leq \min_{\boldsymbol{x}'\in\mathcal{M}} \rho^{-1}(\boldsymbol{x}') \int_{\boldsymbol{x}\in\mathcal{M}} |g(\boldsymbol{x})|^2 d\boldsymbol{x} \\
&= \rho_{\min}^{-1}\|g\|_{L^2}^2
\end{aligned}
$$

Then for $\zeta$, we have

$$
\begin{aligned}
\|\zeta\|_{L^2_\mu}^2 &= \int_{\boldsymbol{x}\in\mathcal{M}} |\zeta(\boldsymbol{x})|^2 \rho(\boldsymbol{x})d\boldsymbol{x} \\
&\geq \rho_{\min}\|\zeta\|_{L^2}^2.
\end{aligned}
$$

This gives

$$\|g_\mu\|_{L^2_\mu} \leq \rho_{\min}^{-1/2}\|g\|_{L^2}$$

$$\leq \rho_{\min}^{-1/2} \frac{C\|\zeta\|_{L^2}}{n \log L}$$

$$\leq \frac{C\|\zeta\|_{L_\mu^2}}{\rho_{\min} n \log L}.$$

On the other hand, we have

$$\|\mathbf{\Theta}_\mu[g] - \zeta\|_{L_\mu^2}^2 = \int_{\boldsymbol{x}\in\mathcal{M}} \left( \int_{\boldsymbol{x}'\in\mathcal{M}} \Theta(\boldsymbol{x},\boldsymbol{x}')g_\mu(\boldsymbol{x})\rho(\boldsymbol{x}')d\boldsymbol{x}' - \zeta(\boldsymbol{x}) \right)^2 \rho(\boldsymbol{x})d\boldsymbol{x}$$

$$\leq \max_{\boldsymbol{x}''\in\mathcal{M}} \rho(\boldsymbol{x}'') \int_{\boldsymbol{x}\in\mathcal{M}} \left( \int_{\boldsymbol{x}'\in\mathcal{M}} \Theta(\boldsymbol{x},\boldsymbol{x}')g(\boldsymbol{x}')d\boldsymbol{x}' - \zeta(\boldsymbol{x}) \right)^2 d\boldsymbol{x}$$

$$= \rho_{\max}\|\mathbf{\Theta}[g] - \zeta\|_{L^2}^2.$$

Notice that $P' = CP + \text{len}(\mathcal{M})^3$ is still a polynomial of degree 3 in $\text{len}(\mathcal{M})$. Then when $L \geq \max\{P'^4, \rho_{\max}^{12}\}$, we get

$$\|\mathbf{\Theta}_\mu[g] - \zeta\|_{L_\mu^2}^2 \leq \rho_{\max}^{1/2} CPL^{-\frac{4}{3}}\|\zeta\|_{L^2}$$

$$= L^{-1}\|\zeta\|_{L^\infty} \frac{CP}{L^{1/4}} \frac{\rho_{\max}^{1/2}}{L^{1/24}} \frac{\sqrt{\text{len}(\mathcal{M})}}{L^{1/24}}$$

$$\leq L^{-1}\|\zeta\|_{L^\infty}.$$

As $P$ in Lemma E.31 is a polynomial $\text{poly}\{M_3, M_4, M_5, \text{len}(\mathcal{M}), \Delta^{-1}\}$ of degree $\leq 9$, with degree $\leq 3$ in $M_2, M_4, M_5 \text{len}(\mathcal{M})$, and degree $\leq 6$ in $\Delta^{-1}$, we have $P'^4$ is of the right degree requirement as in the statement of the theorem. ∎

## F  Bounds for the Skeleton Function $\psi$

In this section, we are going to provide sharp bounds on the "skeleton" function $\psi$ and its higher-order derivatives. We recall that the *angle evolution function* is defined as

$$\varphi(t) = \arccos\left( \left(1 - \frac{t}{\pi}\right)\cos t + \frac{1}{\pi}\sin t \right), \quad t \in [0, \pi].$$

Define $\varphi^{[0]} = \text{Id}$, $\varphi^{[\ell]}$ as $\varphi$'s $\ell$-fold composition with itself (which will be referred to as the *iterated angle evolution function*). Then the skeleton is defined as

$$\psi(t) = \frac{n}{2} \sum_{\ell=0}^{L-1} \xi_\ell(t),$$

where

$$\xi_\ell(t) = \prod_{\ell'=\ell}^{L-1} \left(1 - \frac{1}{\pi}\varphi^{[\ell']}(t)\right), \quad \ell = 0, \cdots, L-1.$$

To analyze the function $\psi$, we will establish in this section several "sharp-modulo-constants" estimates that connect $\psi$ to a much simpler function, derived using the local behavior of $\varphi$ at 0 and its consequences for the iterated compositions $\varphi^{[\ell]}$ that appear in the definition of $\psi$. In particular, let us define $\widehat{\varphi} : [0, \pi] \to [0, \pi]$ by $\widehat{\varphi}(t) = t/(1 + t/(3\pi))$, so that

$$\widehat{\varphi}^{[\ell]}(t) = \frac{t}{1 + \ell t/(3\pi)},$$

and moreover define

$$\widehat{\xi}_\ell(t) = \prod_{\ell'=\ell}^{L-1} \left(1 - \frac{\widehat{\varphi}^{[\ell']}(t)}{\pi}\right), \qquad \widehat{\psi}(t) = \frac{n}{2} \sum_{\ell=0}^{L-1} \widehat{\xi}_\ell(t).$$

We will prove that $\widehat{\varphi}^{[\ell]}$ provides a sharp approximation to $\varphi^{[\ell]}$ (Lemmas F.2 and F.3), and then work out a corresponding sharp approximation of $\widehat{\psi}$ to $\psi$ (Lemmas F.7 and F.9). We will then derive

estimates for the low-order derivatives of $\psi$ in Appendix F.4. Unfortunately, it is impossible to obtain $L^1$ estimates for $\psi$ in terms of $\widehat{\psi}$ that are sharp enough to facilitate operator norm bounds for $\Theta_\mu$, which would let us construct certificates for an operator with kernel $\widehat{\psi}$ rather than the NTK $\Theta_\mu$; but the estimates we derive in this section will be nonetheless sufficient to enable our localization and certificate construction arguments in Appendix E.

We note that bounds similar to a subset of the bounds in this section have been developed in an $L$-asymptotic, large-angle setting by [5]. The bounds we develop here are non-asymptotic and hold for all angles, and are established using elementary arguments that we believe are slightly more transparent. We reuse (and restate in Appendix G) some estimates from [6, Section C] here, but the majority of our estimates will be fundamentally improved (a representative example is Lemma F.3).

Throughout this section, we use $\dot{\varphi}, \ddot{\varphi}, \dddot{\varphi}$ to represent first, second and third derivatives of $\varphi$ (see Lemma G.5 for basic regularity assertions for this function and its iterated compositions) and likewise for $\xi$ and $\psi$. In particular, for example, in our notation the function $\dot{\varphi}^{[\ell]}$ refers to the derivative of $\varphi^{[\ell]}$, *not* the $\ell$-fold iterated composition of $\dot{\varphi}$. Although this leads to an abuse of notation, the concision it enables in our proofs will be of use.

### F.1 Sharp Lower Bound for the Iterated Angle Evolution Function

**Lemma F.1.** *One has*

$$\varphi(t) \leq \frac{t}{1 + t/(3\pi)}, \quad t \in [0, \pi].$$

**Proof.** As $\cos$ is monotonically decreasing in $[0, \pi)$, it is the same as proving

$$(1 - \frac{t}{\pi})\cos(t) + \frac{\sin t}{\pi} - \cos\frac{t}{1 + \frac{t}{3\pi}} \geq 0$$

We have the gradient as

$$-(1 - \frac{t}{\pi})\sin t + \sin(\frac{t}{1 + t/(3\pi)})\frac{1}{(1 + t/(3\pi))^2}$$

$$\geq -(1 - \frac{t}{\pi})\sin t + \sin t\frac{1}{(1 + t/(3\pi))^3}$$

$$\geq (-(1 - \frac{t}{\pi}) + \frac{1}{(1 + \frac{t}{3\pi})^3})\sin t$$

$$\geq 0$$

For the first inequality, we use the estimate

$$\sin(ax) \geq x \sin a; \quad 0 \leq x \leq 1, \ 0 \leq a \leq \pi, \tag{F.1}$$

which is easily established using concavity of $\sin$ on $[0, \pi]$ and the secant line characterization, and for the final inequality, we use the estimate $1 - 3a \leq \frac{1}{(1+a)^3}$ for any $a > -1$, which follows from convexity of $a \mapsto (1 + a)^{-3}$ on this domain and the tangent line characterization (at $a = 0$). Since at $t = 0$, we have the inequality holds, we know it holds for the whole interval $[0, \pi]$ by the mean value theorem. ∎

**Lemma F.2** (Corollary of [6, Lemma C.12]). *If $\ell \in \mathbb{N}_0$, one has the "fluid" estimate for the iterated angle evolution function*

$$\varphi^{[\ell]}(t) \leq \frac{t}{1 + \ell t/(3\pi)}.$$

**Proof.** Follow the argument of [6, Lemma C.12], but use Lemma F.1 as the basis for the argument instead of Lemma G.4. ∎

**Lemma F.3.** *There exists an absolute constant $C_0 > 0$ such that for all $\ell \in \mathbb{N}$*

$$\widehat{\varphi}^{[\ell]} - \varphi^{[\ell]} \leq C_0 \frac{\log(1 + \ell)}{\ell^2}. \tag{F.2}$$

*As a consequence, there exist absolute constants $C, C', C'' > 0$ such that for any $0 < \varepsilon \le 1/2$, if $L \ge C\varepsilon^{-2}$ then for every $t \in [0, C'\varepsilon^2]$ one has*

$$\dot{\varphi}^{[L]}(t) \le \frac{1 + \varepsilon}{(1 + Lt/(3\pi))^2},$$

*and for every $t \in [0, \pi]$ one has*

$$\dot{\varphi}^{[L]}(t) \le \frac{C''}{(1 + Lt/(3\pi))^2}.$$

*Finally, we have for $\ell > 0$*

$$\xi_\ell(t) \le (1 + e^{6C_0} \frac{\log(1 + \ell)}{\ell}) \widehat{\xi}_\ell(t), \tag{F.3}$$

*and if $L \ge 3$*

$$\psi(t) \le \widehat{\psi}(t) + 4n e^{6C_0} \log^2 L.$$

**Proof.** Fix $L \in \mathbb{N}$ arbitrary. We prove (F.2) first, then use it to derive the remaining estimates. The main tool is an inductive decomposition: start by writing

$$\widehat{\varphi}^{[L]}(t) - \varphi^{[L]}(t) = \widehat{\varphi} \circ \widehat{\varphi}^{[L-1]}(t) - \varphi \circ \varphi^{[L-1]}(t)$$
$$= \widehat{\varphi} \circ \widehat{\varphi}^{[L-1]}(t) - \widehat{\varphi} \circ \varphi^{[L-1]}(t) + \widehat{\varphi} \circ \varphi^{[L-1]}(t) - \varphi \circ \varphi^{[L-1]}(t),$$

and then use the definition of $\widehat{\varphi}$ to simplify the first term on the RHS of the final equation (via direct algebraic manipulation) to

$$\widehat{\varphi} \circ \widehat{\varphi}^{[L-1]}(t) - \widehat{\varphi} \circ \varphi^{[L-1]}(t) = \frac{\widehat{\varphi}^{[L-1]}(t) - \varphi^{[L-1]}(t)}{\left(1 + \frac{1}{3\pi}\varphi^{[L-1]}(t)\right)\left(1 + \frac{1}{3\pi}\widehat{\varphi}^{[L-1]}(t)\right)}.$$

This gives an expression for the difference $\widehat{\varphi}^{[L]}(t) - \varphi^{[L]}(t)$ as an affine function of the previous difference $\widehat{\varphi}^{[L-1]}(t) - \varphi^{[L-1]}(t)$:

$$\widehat{\varphi}^{[L]}(t) - \varphi^{[L]}(t) = \frac{\widehat{\varphi}^{[L-1]}(t) - \varphi^{[L-1]}(t)}{\left(1 + \frac{1}{3\pi}\varphi^{[L-1]}(t)\right)\left(1 + \frac{1}{3\pi}\widehat{\varphi}^{[L-1]}(t)\right)} + \widehat{\varphi} \circ \varphi^{[L-1]}(t) - \varphi \circ \varphi^{[L-1]}(t),$$

and unraveling inductively, we obtain

$$\widehat{\varphi}^{[L]}(t) - \varphi^{[L]}(t) = \sum_{\ell=0}^{L-1} \left( \prod_{\ell'=\ell+1}^{L-1} \frac{1}{\left(1 + \frac{1}{3\pi}\widehat{\varphi}^{[\ell']}(t)\right)\left(1 + \frac{1}{3\pi}\varphi^{[\ell']}(t)\right)} \right) (\widehat{\varphi} - \varphi) \circ \varphi^{[\ell]}(t),$$

where for concision we write $(\widehat{\varphi} - \varphi)(t) = \widehat{\varphi}(t) - \varphi(t)$. Note that all the product coefficients in this expression are nonnegative numbers. Denoting by $\tilde{C}_1$ the constant attached to $t^3$ in the result Lemma F.13 and defining $C_1 = \max\left\{\tilde{C}_1, 1\right\}$, Lemma F.13 gives

$$\widehat{\varphi}^{[L]}(t) - \varphi^{[L]}(t) \le C_1 \sum_{\ell=0}^{L-1} \left( \prod_{\ell'=\ell+1}^{L-1} \frac{1}{\left(1 + \frac{1}{3\pi}\widehat{\varphi}^{[\ell']}(t)\right)\left(1 + \frac{1}{3\pi}\varphi^{[\ell']}(t)\right)} \right) \left(\varphi^{[\ell]}(t)\right)^3. \tag{F.4}$$

To prove (F.2), we will use a two-stage approach:

1. (**First pass**) First, we will control only the first factor in the product term in (F.4) using Lemma F.13, given that $\varphi \ge 0$ allows us to upper bound by the product term without the second factor. The resulting bound on the LHS of (F.4) will be weaker (in terms of its dependence on $L$) than (F.2).

2. (**Second pass**) After completing this control, we will have obtained a lower bound on $\varphi^{[L]}$; we can then return to (F.4) and use this lower bound to get control of both factors in the product term, which will allow us to sharpen our previous analysis and establish the claimed bound (F.2).

**First pass.** We have

$$\prod_{\ell'=\ell+1}^{L-1} \frac{1}{\left(1 + \frac{1}{3\pi}\widehat{\varphi}^{[\ell']}(t)\right)} = \frac{1 + \frac{(\ell+1)t}{3\pi}}{1 + \frac{Lt}{3\pi}}.$$

Tossing the product term involving $\varphi^{[\ell']}$ and applying Lemma F.2 in (F.4), we thus have a bound

$$\widehat{\varphi}^{[L]}(t) - \varphi^{[L]}(t) \leq \frac{C_1}{1 + \frac{Lt}{3\pi}} \sum_{\ell=0}^{L-1} \frac{t^3}{\left(1 + \frac{\ell t}{3\pi}\right)^2} + \frac{C_1 t/(3\pi)}{1 + \frac{Lt}{3\pi}} \sum_{\ell=0}^{L-1} \frac{t^3}{\left(1 + \frac{\ell t}{3\pi}\right)^3}.$$

For the first term in this expression, we calculate using an estimate from the integral test

$$\sum_{\ell=0}^{L-1} \frac{t^3}{\left(1 + \frac{\ell t}{3\pi}\right)^2} \leq t^3 + \int_0^L \frac{t^3}{\left(1 + \frac{\ell t}{3\pi}\right)^2} \, \mathrm{d}\ell$$

$$= t^3 + 3\pi t^2 \left(1 - \frac{1}{1 + \frac{Lt}{3\pi}}\right)$$

$$= t^3 + \frac{Lt^3}{1 + \frac{Lt}{3\pi}},$$

and for the second term, we calculate similarly

$$\sum_{\ell=0}^{L-1} \frac{t^3}{\left(1 + \frac{\ell t}{3\pi}\right)^3} \leq t^3 + \int_0^L \frac{t^3}{\left(1 + \frac{\ell t}{3\pi}\right)^3} \, \mathrm{d}\ell$$

$$= t^3 + \frac{3\pi t^2}{2} \left(1 - \frac{1}{\left(1 + \frac{Lt}{3\pi}\right)^2}\right)$$

$$= t^3 + Lt^3 \frac{1 + \frac{Lt}{6\pi}}{\left(1 + \frac{Lt}{3\pi}\right)^2}$$

$$\leq t^3 + \frac{Lt^3}{1 + \frac{Lt}{3\pi}}.$$

Combining these results gives

$$\widehat{\varphi}^{[L]}(t) - \varphi^{[L]}(t) \leq \frac{C_1 t^3}{\left(1 + \frac{Lt}{3\pi}\right)} + \frac{C_1 Lt^3}{\left(1 + \frac{Lt}{3\pi}\right)^2} + \frac{C_1 t^4/(3\pi)}{\left(1 + \frac{Lt}{3\pi}\right)} + \frac{C_1 Lt^4/(3\pi)}{\left(1 + \frac{Lt}{3\pi}\right)^2}$$

$$\leq \frac{3\pi C_1 t}{L} \left(3\pi + 2t + \frac{1}{3\pi}t^2\right). \tag{F.5}$$

This bound gives us a nontrivial estimate as far out as $t = \pi$, but the result is weaker there than what we need. We can proceed with a bootstrapping approach to improve our result for large angles. To begin, we have shown via (F.5)

$$\varphi^{[L]}(t) \geq \widehat{\varphi}^{[L]}(t) - \frac{16\pi^2 C_1 t}{L}.$$

Let us write $t_0 = C/\sqrt{L}$, where $C > 0$ is a constant we will optimize below, and define

$$\breve{\varphi}_L(t) = \begin{cases} \widehat{\varphi}^{[L]}(t) - \frac{16 C_1 \pi^2 t}{L} & 0 \leq t \leq t_0 \\ \widehat{\varphi}^{[L]}(t_0) - \frac{16 C_1 \pi^2 t_0}{L} & t_0 \leq t \leq \pi. \end{cases}$$

The notation here is justified by noticing that $\varphi^{[L]}$ is concave and nondecreasing, so that our previous estimates imply $\varphi^{[L]} \geq \breve{\varphi}_L$. It follows

$$\widehat{\varphi}^{[L]} - \varphi^{[L]} \leq \widehat{\varphi}^{[L]} - \breve{\varphi}_L.$$

Our previous bound (F.5) is an increasing function of $t$, and sufficient for $0 \leq t \leq t_0$. For $t \geq t_0$, we have

$$\widehat{\varphi}^{[L]}(t) - \breve{\varphi}_L(t) \leq \frac{16 C_1 \pi^2 t_0}{L} + \widehat{\varphi}^{[L]}(t) - \widehat{\varphi}^{[L]}(t_0),$$

and we can calculate using increasingness of $\widehat{\varphi}^{[L]}$

$$\widehat{\varphi}^{[L]}(t) - \widehat{\varphi}^{[L]}(t_0) \leq \frac{\pi}{1 + L/3} - \frac{C}{\sqrt{L} + CL/(3\pi)}$$

$$= \frac{\pi\sqrt{L} - C}{(1 + L/3)(\sqrt{L} + CL/(3\pi))}$$

$$\leq \frac{9\pi^2}{CL^{3/2}},$$

whence the bound

$$\widehat{\varphi}^{[L]} - \varphi^{[L]} \leq \frac{\pi}{L^{3/2}}\left(16\pi C_1 C + \frac{9\pi}{C}\right)$$

$$\leq \frac{24\pi^2\sqrt{C_1}}{L^{3/2}} \tag{F.6}$$

valid on the entire interval $[0, \pi]$; the final inequality corresponds to the choice $C = \frac{3}{4\sqrt{C_1}}$.

**Second pass.** To start, with an eye toward the unused product term in (F.4), we have from (F.6)

$$1 + \frac{1}{3\pi}\varphi^{[L]}(t) \geq 1 + \frac{1}{3\pi}\widehat{\varphi}^{[L]}(t) - \frac{8\pi\sqrt{C_1}}{L^{3/2}}.$$

Using the numerical inequality $e^{-2x} \leq 1 - x$, valid for $0 \leq x \leq 1/2$ at least, we have if $L \geq \left(256\pi^2 C_1\right)^{1/3}$

$$1 + \frac{1}{3\pi}\varphi^{[L]}(t) \geq \exp\left(-\frac{16\pi\sqrt{C_1}}{L^{3/2}}\right)\left(1 + \frac{1}{3\pi}\widehat{\varphi}^{[L]}(t)\right).$$

Applying this bound to terms in the second product term in (F.4) with index $\ell \geq \left\lceil\left(256\pi^2 C_1\right)^{1/3}\right\rceil \equiv r(C_1)$, we therefore have [8]

$$\widehat{\varphi}^{[L]}(t) - \varphi^{[L]}(t) \leq C_1 \sum_{\ell=0}^{L-1}\left(\varphi^{[\ell]}(t)\right)^3\left(\prod_{\ell'=\max\{r(C_1),\ell+1\}}^{L-1}\frac{1}{\left(1 + \frac{1}{3\pi}\widehat{\varphi}^{[\ell']}(t)\right)\left(1 + \frac{1}{3\pi}\varphi^{[\ell']}(t)\right)}\right)$$

$$\leq C_1 \sum_{\ell=0}^{L-1}\left(\varphi^{[\ell]}(t)\right)^3\exp\left(16\pi\sqrt{C_1}\sum_{\ell'=\max\{r(C_1),\ell+1\}}^{L-1}\frac{1}{(\ell')^{3/2}}\right)$$

$$\times\left(\prod_{\ell'=\max\{r(C_1),\ell+1\}}^{L-1}\frac{1}{\left(1 + \frac{1}{3\pi}\widehat{\varphi}^{[\ell']}(t)\right)^2}\right)$$

$$= C_1 e^{16\pi\sqrt{C_1}\zeta(3/2)}\sum_{\ell=0}^{L-1}\left(\varphi^{[\ell]}(t)\right)^3\left(\frac{1 + \frac{\max\{r(C_1),\ell+1\}t}{3\pi}}{1 + \frac{Lt}{3\pi}}\right)^2$$

$$= \frac{C_1 e^{16\pi\sqrt{C_1}\zeta(3/2)}}{\left(1 + \frac{Lt}{3\pi}\right)^2}\left(\left(1 + \frac{(r(C_1)-1)t}{3\pi}\right)^2\sum_{\ell=0}^{r(C_1)-2}\left(\varphi^{[\ell]}(t)\right)^3\right.$$

$$\left. + \sum_{\ell=r(C_1)-1}^{L-1}\left(\varphi^{[\ell]}(t)\right)^3\left(1 + \frac{(\ell+1)t}{3\pi}\right)^2\right).$$

Now, since $\varphi \leq \widehat{\varphi}$, we have

$$\left(1 + \frac{(r(C_1)-1)t}{3\pi}\right)^2\sum_{\ell=0}^{r(C_1)-2}\left(\varphi^{[\ell]}(t)\right)^3 \leq r(C_1)^2 t^3,$$

---

[8] Although it has a different meaning in our argument at large, here and in some subsequent bounds $\zeta(x) = \sum_{n=1}^{\infty} n^{-x}$ denotes the Riemann zeta function. In this setting, we have $\zeta(3/2) \leq e$.

and

$$\sum_{\ell=r(C_1)}^{L-1} \left(\varphi^{[\ell]}(t)\right)^3 \left(1 + \frac{(\ell+1)t}{3\pi}\right)^2 \leq 2t^3 \sum_{\ell=0}^{L-1} \frac{1}{1 + \frac{\ell t}{3\pi}}$$

$$\leq 2t^3 + 2t^3 \int_0^L \frac{1}{1 + \frac{\ell t}{3\pi}} \, d\ell$$

$$= 2t^3 + 6\pi t^2 \log(1 + Lt/3\pi),$$

whence

$$\widehat{\varphi}^{[L]}(t) - \varphi^{[L]}(t) \leq \frac{C_1 e^{16\pi\sqrt{C_1}\zeta(3/2)}}{\left(1 + \frac{Lt}{3\pi}\right)^2} \left((2 + r(C_1)^2)t^3 + 6\pi t^2 \log(1 + Lt/3\pi)\right)$$

$$\leq \frac{9\pi^2 C_1 e^{16\pi\sqrt{C_1}\zeta(3/2)}}{L^2} \left((2 + r(C_1)^2)t + 6\pi \log(1 + Lt/3\pi)\right)$$

$$\leq 54\pi^3 C_1 (2 + r(C_1)^2) e^{16\pi\sqrt{C_1}\zeta(3/2)} \frac{\log(1+L)}{L^2}.$$

In the final line, we are simply shuffling constants using $t \leq \pi$. This completes the proof of (F.2).

**Derived estimates.** The remaining claims can be derived from the main claim we have just established; we will do so now. Below, we write $C_0 = 54\pi^3 C_1 (2 + r(C_1)^2) e^{16\pi\sqrt{C_1}\zeta(3/2)}$. We will also assume $\ell \geq 1$.

We prove the claim about $\xi_\ell$ first. First, notice that for nonnegative numbers $a, b$, one has $1 - a + b \leq e^{2b}(1 - a)$ provided $a \leq 1/2$. Since $\varphi \leq \pi/2$, we have for each $\ell > 0$

$$\xi_\ell(t) \leq \prod_{\ell'=\ell}^{L-1} \left(1 - \frac{\widehat{\varphi}^{[\ell']}(t)}{\pi} + \frac{C_0 \log(1 + \ell')}{\pi(\ell')^2}\right)$$

$$\leq \exp\left(2C_0 \sum_{\ell'=\ell}^{L-1} \frac{\log(1 + \ell')}{(\ell')^2}\right) \widehat{\xi}_\ell(t).$$

By the integral test estimate, we have for $\ell > 0$

$$\sum_{\ell'=\ell}^{L-1} \frac{\log(1 + \ell')}{(\ell')^2} \leq \frac{\log(1 + \ell)}{\ell^2} + \int_\ell^L \frac{\log(1 + \ell')}{(\ell')^2} \, d\ell'$$

$$\leq \frac{\log(1 + \ell)}{\ell^2} + \log\left(\frac{1 + \frac{1}{\ell}}{1 + \frac{1}{L}}\right) + \log(1 + \ell)/\ell - \log(1 + L)/L$$

$$\leq \frac{3\log(1 + \ell)}{\ell},$$

where we applied $\log(1 + x) \leq x$ for all $x > -1$, whence for $\ell > 0$

$$\xi_\ell(t) \leq e^{6C_0 \log(1+\ell)/\ell}\widehat{\xi}_\ell(t).$$

In particular, using the fact that $\log(1 + \ell)/\ell \leq 1$ and the estimate $e^{cx} \leq 1 + xe^c$ for $x \in [0, 1]$ (by convexity of the exponential function), we obtain

$$\xi_\ell(t) \leq \left(1 + e^{6C_0} \frac{\log(1 + \ell)}{\ell}\right) \widehat{\xi}_\ell(t), \tag{F.7}$$

as claimed. The proof of the second inequality is very similar: first, repeated application of the chain rule gives

$$\dot{\varphi}^{[L]} = \prod_{\ell=0}^{L-1} \dot{\varphi} \circ \varphi^{[\ell]}.$$

Using the expression

$$\dot{\varphi}(t) = \frac{(1 - t/\pi)\sin t}{\sin \varphi(t)},$$

we can exploit a telescopic cancellation in the preceding expression for $\dot\varphi^{[L]}$, obtaining

$$\dot\varphi^{[L]} = \frac{\sin t}{\sin \varphi^{[L]}(t)} \prod_{\ell=0}^{L-1} \left(1 - \frac{\varphi^{[\ell]}(t)}{\pi}\right).$$

As the form of this upper bound is identical to the one we controlled for $\xi_\ell$, only with a different constant factor, we can now apply the first part of that argument to the present setting, obtaining a bound

$$\dot\varphi^{(L)} \le \frac{\sin t}{\sin \varphi^{[L]}(t)} \exp\left(\sum_{\ell=1}^{L-1} \widehat\varphi^{[\ell]} - \varphi^{[\ell]}\right) \exp\left(-\frac{1}{\pi}\sum_{\ell=0}^{L-1}\frac{t}{1+\ell t/(3\pi)}\right)$$

where in simplifying we also used that $\varphi^{[0]} = \widehat\varphi^{[0]}$. To proceed, we split the first sum, obtaining for any index $1 \le \ell_\star \le L-1$

$$\sum_{\ell=1}^{L-1} \widehat\varphi^{[\ell]} - \varphi^{[\ell]} = \sum_{\ell=1}^{\ell_\star-1}(\widehat\varphi^{[\ell]} - \varphi^{[\ell]}) + \sum_{\ell=\ell_\star}^{L-1}(\widehat\varphi^{[\ell]} - \varphi^{[\ell]})$$

$$\le Ct\sum_{\ell=1}^{\ell_\star-1}\frac{1}{\ell} + 3C_0\frac{\log(1+\ell_\star)}{\ell_\star}$$

$$\le Ct\log(\ell_\star) + 3C_0\frac{\log(1+\ell_\star)}{\ell_\star}$$

$$\le C\log(1+\ell_\star)\left(t + \frac{1}{\ell_\star}\right),$$

where in the second line the bound on the first sum used (F.5), and the second used the estimate we proved in the previous section and the integral test estimate above; in the third line we estimated the harmonic series with the integral test; and in the fourth line we worst-cased. Next, for any $t \le 1/\ell_\star$, we have by the above

$$\sum_{\ell=1}^{L-1} \widehat\varphi^{[\ell]} - \varphi^{[\ell]} \le C\log(1+\ell_\star)/\ell_\star,$$

and because the RHS approaches 0 as $\ell_\star \to \infty$, for any $0 < \varepsilon \le 1$ there is an integer $N(\varepsilon) > 0$ such that for all $\ell_\star \ge N$ we have

$$\sum_{\ell=1}^{L-1} \widehat\varphi^{[\ell]} - \varphi^{[\ell]} \le \log(1+\varepsilon).$$

In particular, obtaining a lower bound for the RHS by concavity of $\log$, it is sufficient to take $\ell_\star \ge C\varepsilon^{-2}$ for a suitably large absolute constant $C > 0$. To ensure there exists such a value of $\ell_\star$, it suffices to choose $L \ge C\varepsilon^{-2}$ and therefore $t \le C'\varepsilon^2$. In particular, plugging this estimate into our previous bound, we have shown that for any $\varepsilon > 0$, if $L \ge C'\varepsilon^{-2}$ then for all all $t \le C\varepsilon^2$ we have

$$\dot\varphi^{[L]} \le (1+\varepsilon)\frac{\sin t}{\sin \varphi^{[L]}(t)} \exp\left(-\frac{1}{\pi}\sum_{\ell=0}^{L-1}\frac{t}{1+\ell t/(3\pi)}\right).$$

We then calculate by an estimate from the integral test

$$\sum_{\ell=0}^{L-1}\frac{t}{1+\ell t/(3\pi)} \ge \int_0^L \frac{t}{1+\ell t/(3\pi)}\,d\ell = 3\pi\log(1+Lt/(3\pi)),$$

which establishes under the previous conditions on $L$ and $t$ that

$$\dot\varphi^{[L]} \le \frac{\sin t}{\sin \varphi^{[L]}(t)}\frac{1+\varepsilon}{(1+Lt/(3\pi))^3}.$$

To conclude, we need to simplify the $\sin$ ratio term. Using Lemma F.4, for any $0 < \varepsilon' \le 1/2$, we have for $0 \le t \le C\varepsilon'$ that

$$\frac{\sin t}{\sin \varphi^{[L]}(t)} \le (1+2\varepsilon')(1+Lt/(3\pi)),$$

which suffices to prove the claim for small $t$ after noting $(1 + 2\varepsilon')(1 + \varepsilon) = 1 + 2\varepsilon' + \varepsilon + 2\varepsilon'\varepsilon$, choosing whichever is smaller, and adjusting the preceding conditions on $t$ and $L$ (i.e. the absolute constants in the previous bounds may grow/shrink as necessary). To show the claimed bound on the entire interval $[0, \pi]$, we can follow exactly the argument above, but instead of partitioning the sum of errors $\widehat{\varphi}^{[\ell]} - \varphi^{[\ell]}$ as above we simply use bound the sum of errors as in the bound on $\widehat{\xi}_\ell$ previously to obtain a large constant in the numerator; the $\sin$ ratio is controlled in this case using the first conclusion in Lemma F.4, which is valid on the whole interval $[0, \pi]$.

Finally, we obtain the estimate on $\psi$ by calculating using the estimate involving $\xi_\ell$ and $\widehat{\xi}_\ell$ that we proved earlier. First, we note that although we required $\ell > 0$ above, the fact that $\widehat{\varphi}^{[0]} = \varphi^{[0]}$ implies that we have an estimate $\xi_0 \le (1 + \log(2)e^{6C_0})\widehat{\xi}_0$. We therefore have

$$\psi(t) = \frac{n}{2} \sum_{\ell=0}^{L-1} \xi_\ell(t) \le \frac{n}{2} \sum_{\ell=0}^{L-1} \left(1 + e^{6C_0} \log \frac{(1 + \ell)}{\ell}\right) \widehat{\xi}_\ell(t)$$

$$\le \widehat{\psi}(t) + (n/2)e^{6C_0}\left(\log(2)\widehat{\xi}_0(t) + \sum_{\ell=1}^{L-1} \frac{\log(1 + \ell)}{\ell}\widehat{\xi}_\ell(t)\right).$$

It is easy to see that $\widehat{\xi}_\ell \le 1$. Hence

$$\psi(t) \le \widehat{\psi}(t) + (n/2)e^{6C_0}\left(\log(2) + \sum_{\ell=1}^{L-1} \frac{\log(1 + \ell)}{\ell}\right)$$

$$\le \widehat{\psi}(t) + ne^{6C_0}\left(\log(2) + \sum_{\ell=2}^{L-1} \frac{\log(\ell)}{\ell}\right)$$

$$\le \widehat{\psi}(t) + ne^{6C_0}\left(2\log(2) + \int_{\ell=2}^{L-1} \frac{\log(\ell)}{\ell}\,d\ell\right)$$

$$\le \widehat{\psi}(t) + ne^{6C_0}\left(2\log(2) - (1/2)\log^2 2 + (1/2)\log^2(L-1)\right)$$

$$\le \widehat{\psi}(t) + 4ne^{6C_0}\log^2 L,$$

where the final bound requires $L \ge 3$.

∎

**Lemma F.4.** *For $\ell \in \mathbb{N}_0$, one has for $t \in [0, \pi]$*

$$\frac{\sin(t)}{\sin(\varphi^{[\ell]}(t))} \le 3(1 + \ell t/(3\pi))$$

*and there exists an absolute constant $C > 0$ such that for any $0 < \varepsilon \le 1/2$, if $0 \le t \le C\varepsilon$ one has*

$$\frac{\sin(t)}{\sin(\varphi^{[\ell]}(t))} \le (1 + 2\varepsilon)(1 + \ell t/(3\pi)).$$

**Proof.** We prove the bound on $[0, \pi]$ first. Because $t \mapsto t^{-1}\sin t$ is decreasing on $[0, \pi]$, we apply Lemma G.6 to get

$$\frac{\sin(t)}{\sin(\varphi^{[\ell]}(t))} \le \frac{t}{\varphi^{[\ell]}(t)}$$

$$\le \frac{t}{\frac{t}{1 + \ell t/(\pi)}}$$

$$= 1 + \ell t/(\pi) \le 3(1 + \ell t/(3\pi)).$$

Now fix $0 < \epsilon \le 1/2$. We claim that there is an absolute constant $C > 0$ such that if $t \le C\epsilon$, we have

$$\varphi^{[\ell]}(t) \ge (1 - \epsilon)\widehat{\varphi}^{[\ell]}(t).$$

Assuming this claim, we have for $t \leq C\epsilon$

$$\frac{\sin(t)}{\sin(\varphi^{[\ell]}(t))} \leq \frac{t}{\varphi^{[\ell]}(t)}$$

$$\leq \frac{t}{(1-\epsilon)\widehat{\varphi}^{[\ell]}(t)}$$

$$\leq \frac{1}{1-\epsilon}(1+\ell t/(3\pi))$$

$$\leq (1+2\epsilon)(1+\ell t/(3\pi)),$$

which is enough to conclude after rescaling. Now we want to show the claim. Let $C_0 = \max\{1, C_1\}$ where $C_1$ denotes the constant on $t^3$ in Lemma F.13. We first notice that

$$\varphi(t) \geq \widehat{\varphi}(t) - C_0 t^3$$

$$= \frac{t}{1+t/(3\pi)} - C_0 t^3$$

$$= \frac{t}{1+t/(3\pi)} - \frac{t}{1+\pi/(3\pi)}\frac{4}{3}C_0 t^2$$

$$\geq (1 - \frac{4C_0}{3}t^2)\frac{t}{1+t/(3\pi)}.$$

We are going to proceed with an induction-like approach. Put $\epsilon_1 = 4C_0 t^2/3$, and choose $t \leq \sqrt{3/(4C_0)}$ so that $1 - \epsilon_1 \geq 0$. Supposing that it holds $\varphi^{[\ell-1]} \geq (1 - \epsilon_{\ell-1})\widehat{\varphi}^{[\ell-1]}(t)$ for a positive $\epsilon_{\ell-1}$ such that $1 - \epsilon_{\ell-1} \geq 0$ (we have shown there is such $\varepsilon_1$ and controlled it), we have by some applications of the induction hypothesis, Lemma F.1, and the previous small-$t$ estimate (we use below that $t \leq \sqrt{3/(4C_0)}$)

$$\varphi^{[\ell]}(t) \geq \left(1 - \frac{4C_0}{3}\left(\varphi^{[\ell-1]}(t)\right)^2\right)\widehat{\varphi}(\varphi^{[\ell-1]}(t))$$

$$= \left(1 - \frac{4C_0}{3}\left(\varphi^{[\ell-1]}(t)\right)^2\right)\frac{(1-\epsilon_{\ell-1})\frac{t}{1+(\ell-1)t/(3\pi)}}{1+(3\pi)^{-1}(1-\epsilon_{\ell-1})\frac{t}{1+(\ell-1)t/(3\pi)}}$$

$$\geq \left(1 - \frac{4C_0}{3}\frac{t^2}{(1+(\ell-1)t/(3\pi))^2}\right)\frac{(1-\epsilon_{\ell-1})t}{1+\ell t/(3\pi) - \epsilon_{\ell-1}t/(3\pi)}$$

$$\geq \left(1 - \frac{4C_0}{3}\frac{t^2}{(1+(\ell-1)t/(3\pi))^2}\right)(1-\epsilon_{\ell-1})\widehat{\varphi}^{[\ell]}(t)$$

$$\geq \left(1 - \frac{4C_0}{3}\frac{t^2}{(1+(\ell-1)t/(3\pi))^2} - \epsilon_{\ell-1}\right)\widehat{\varphi}^{[\ell]}(t).$$

This shows that we can take $\epsilon_\ell = \epsilon_{\ell-1} + (4C_0/3)t^2/(1+(\ell-1)t/(3\pi))^2$ as long as this term is not larger than 1. Unraveling inductively to check, we get

$$\epsilon_\ell = \sum_{\ell'=0}^{\ell-1}\frac{4C_0}{3}\frac{t^2}{(1+\ell't/(3\pi))^2}$$

$$\leq \frac{4C_0}{3}t^2\left(1 + \int_{\ell'=0}^{\ell-1}\frac{1}{(1+\ell't/(3\pi))^2}\,\mathrm{d}\ell'\right)$$

$$\leq \frac{4C_0}{3}t^2\left(1 + \frac{\ell-1}{1+(\ell-1)t/(3\pi)}\right)$$

$$\leq \frac{4C_0}{3}\left(\pi + \frac{1}{3\pi}\right)t$$

$$= \frac{16\pi C_0}{3}t.$$

In particular, the induction is consistent as long as $t \leq 3/(16\pi C_0)$. Note as well that since $C_0 \geq 1$ we have $\sqrt{3/(4C_0)} \geq 3/(16\pi C_0)$. Thus by induction, we know that when $0 < \epsilon < 1$ and $t \leq \frac{3C_0\epsilon}{16\pi}$,

we have

$$\varphi^{[\ell]}(t) \geq (1 - \epsilon)\widehat{\varphi}^{[\ell]}(t)$$

as claimed. ∎

### F.2 Sharp Lower Bound for $\psi$

**Lemma F.5.** *There is an absolute constant $C_0 > 0$ such that*

$$\psi(\pi) \leq \frac{n(L-1)}{8} + 6\pi n e^{6C_0} \log^2 L$$

**Proof.** Following Lemma F.3 (worsening constants slightly for convenience), we directly have

$$\psi(\pi) \leq \widehat{\psi}(\pi) + 6\pi n e^{6C_0} \log^2 L.$$

$\widehat{\psi}(t)$ has a closed form expression, by notice that

$$
\begin{aligned}
\widehat{\xi}_\ell(t) &= \prod_{\ell'=\ell}^{L-1} \left(1 - \frac{\widehat{\varphi}^{[\ell']}(t)}{\pi}\right) \\
&= \prod_{\ell'=\ell}^{L-1} \left(1 - \frac{t/\pi}{1 + \ell' t/(3\pi)}\right) \\
&= \prod_{\ell'=\ell}^{L-1} \frac{1 + (\ell' - 3)t/(3\pi)}{1 + \ell' t/(3\pi)} \\
&= \frac{(1 + (\ell - 3)t/(3\pi))\,(1 + (\ell - 2)t/(3\pi))\,(1 + (\ell - 1)t/(3\pi))}{(1 + (L - 3)t/(3\pi))\,(1 + (L - 2)t/(3\pi))\,(1 + (L - 1)t/(3\pi))}
\end{aligned}
\tag{F.8}
$$

and

$$
\begin{aligned}
\widehat{\psi}(t) &= \frac{n}{2} \sum_{\ell=0}^{L-1} \widehat{\xi}_\ell(t) \\
&= \frac{n}{2} \frac{\sum_{\ell=0}^{L-1}(3\pi + (\ell - 3)t)(3\pi + (\ell - 2)t)(3\pi + (\ell - 1)t)}{(3\pi + (L - 3)t)(3\pi + (L - 2)t)(3\pi + (L - 1)t)} \\
&= \frac{n}{2} \frac{1}{4t} \frac{\sum_{\ell=0}^{L-1}(3\pi + (\ell - 3)t)(3\pi + (\ell - 2)t)(3\pi + (\ell - 1)t)(3\pi + \ell t)}{(3\pi + (L - 3)t)(3\pi + (L - 2)t)(3\pi + (L - 1)t)} \\
&\quad - \frac{\sum_{\ell=0}^{L-1}(3\pi + (\ell - 4)t)(3\pi + (\ell - 3)t)(3\pi + (\ell - 2)t)(3\pi + (\ell - 1)t)}{(3\pi + (L - 3)t)(3\pi + (L - 2)t)(3\pi + (L - 1)t)} \\
&= \frac{n}{8t} \frac{(3\pi + (L - 4)t)(3\pi + (L - 3)t)(3\pi + (L - 2)t)(3\pi + (L - 1)t)}{(3\pi + (L - 3)t)(3\pi + (L - 2)t)(3\pi + (L - 1)t)} \\
&\quad - \frac{(3\pi - 4t)(3\pi - 3t)(3\pi - 2t)(3\pi - t)}{(3\pi + (L - 3)t)(3\pi + (L - 2)t)(3\pi + (L - 1)t)} \\
&= \frac{n}{8t}(3\pi + (L - 4)t) - \frac{n}{8t} \frac{(3\pi - 4t)(3\pi - 3t)(3\pi - 2t)(3\pi - t)}{(3\pi + (L - 3)t)(3\pi + (L - 2)t)(3\pi + (L - 1)t)} \\
&= \frac{n(L - 4)}{8} + \frac{n}{8t} \left(3\pi - \frac{(3\pi - 4t)(3\pi - 3t)(3\pi - 2t)(3\pi - t)}{(3\pi + (L - 3)t)(3\pi + (L - 2)t)(3\pi + (L - 1)t)}\right)
\end{aligned}
\tag{F.9}
$$

From the second to the fourth line above, we used a telescopic series cancellation trick to sum. Then we get the claim as

$$
\begin{aligned}
\psi(\pi) &\leq \widehat{\psi}(\pi) + 6\pi n e^{6C_0} \log^2 L \\
&= \frac{n(L - 4)}{8} + \frac{3\pi n}{8t} + 6\pi n e^{6C_0} \log^2 L
\end{aligned}
$$

$$= \frac{n(L-1)}{8} + 6\pi n e^{6C_0} \log^2 L.$$

∎

**Lemma F.6.** *When $L \geq 2$, we have for any $r > 0$*

$$\int_0^r \psi(t)dt \geq \frac{n(L-4)}{8}r + \frac{3\pi n}{8} \log(1 + \frac{L-2}{3\pi}r)$$

**Proof.** From Lemma F.2, we have $\varphi^{[\ell]}(t) \leq \frac{t}{1+\ell t/(3\pi)}$. Thus we get

$$\xi_\ell(t) = \prod_{\ell'=\ell}^{L-1}(1 - \frac{1}{\pi}\varphi^{[\ell']}(t))$$

$$\geq \frac{(3\pi + (\ell-3)t)(3\pi + (\ell-2)t)(3\pi + (\ell-1)t)}{(3\pi + (L-3)t)(3\pi + (L-2)t)(3\pi + (L-1)t)}.$$

As a result, we have

$$\psi(t) = \frac{n}{2}\sum_{\ell=0}^{L-1}\xi_\ell(t)$$

$$\geq \frac{n}{2}\frac{\sum_{\ell=0}^{L-1}(3\pi + (\ell-3)t)(3\pi + (\ell-2)t)(3\pi + (\ell-1)t)}{(3\pi + (L-3)t)(3\pi + (L-2)t)(3\pi + (L-1)t)}$$

$$= \frac{n}{2}\frac{1}{4t}\frac{\sum_{\ell=0}^{L-1}(3\pi + (\ell-3)t)(3\pi + (\ell-2)t)(3\pi + (\ell-1)t)(3\pi + \ell t)}{(3\pi + (L-3)t)(3\pi + (L-2)t)(3\pi + (L-1)t)}$$

$$\quad - \frac{\sum_{\ell=0}^{L-1}(3\pi + (\ell-4)t)(3\pi + (\ell-3)t)(3\pi + (\ell-2)t)(3\pi + (\ell-1)t)}{(3\pi + (L-3)t)(3\pi + (L-2)t)(3\pi + (L-1)t)}$$

$$= \frac{n}{8t}\frac{(3\pi + (L-4)t)(3\pi + (L-3)t)(3\pi + (L-2)t)(3\pi + (L-1)t)}{(3\pi + (L-3)t)(3\pi + (L-2)t)(3\pi + (L-1)t)}$$

$$\quad - \frac{(3\pi - 4t)(3\pi - 3t)(3\pi - 2t)(3\pi - t)}{(3\pi + (L-3)t)(3\pi + (L-2)t)(3\pi + (L-1)t)}$$

$$= \frac{n}{8t}(3\pi + (L-4)t) - \frac{n}{8t}\frac{(3\pi - 4t)(3\pi - 3t)(3\pi - 2t)(3\pi - t)}{(3\pi + (L-3)t)(3\pi + (L-2)t)(3\pi + (L-1)t)}$$

$$= \frac{n(L-4)}{8} + \frac{n}{8t}\left(3\pi - \frac{(3\pi - 4t)(3\pi - 3t)(3\pi - 2t)(3\pi - t)}{(3\pi + (L-3)t)(3\pi + (L-2)t)(3\pi + (L-1)t)}\right)$$

$$\overset{t'=\frac{t}{3\pi}}{=} \frac{n(L-4)}{8} + \frac{n}{8t'}\left(1 - \frac{(1 - 4t')(1 - 3t')(1 - 2t')(1 - t')}{(1 + (L-3)t')(1 + (L-2)t')(1 + (L-1)t')}\right)$$

$$\geq \frac{n(L-4)}{8} + \frac{n}{8t'}\left(1 - \frac{(1 - 3t')(1 - 2t')(1 - t')}{(1 + (L-2)t')^3}\right)$$

$$\quad + \frac{n}{8t'}\frac{4t'(1 - 3t')(1 - 2t')(1 - t')}{(1 + (L-3)t')(1 + (L-2)t')(1 + (L-1)t')}$$

$$\geq \frac{n(L-4)}{8} + \frac{n}{8t'}\left(1 - \frac{1}{(1 + (L-2)t')^3}\right)$$

$$\overset{L'=L-2}{=} \frac{n(L-4)}{8} + \frac{n}{8}\frac{3L' + 3L'^2t' + L'^3t'^2}{(1 + L't')^3}$$

$$= \frac{n(L-4)}{8} + \frac{n}{8}\left(\frac{L'}{1 + L't'} + \frac{L'}{(1 + L't')^2} + \frac{L'}{(1 + L't')^3}\right).$$

In the third and fourth lines above, we used a splitting and cancellation trick to sum similar to what we used in Lemma F.5. In moving from the seventh to the eighth line, we used the inequality $(x-1)(x+1) \leq x^2$ after splitting off a term that can be negative for large $t'$. In moving from

the eighth to the ninth line, we used nonnegativity of the third summand and upper bounded the numerator of the term in the second summand. (In both of the previous simplifications, we are using that $t' \leq 1/3$.) The remaining simplifications obtain a common denominator in the second term and then cancel. Integrating, we thus find

$$\int_0^r \psi(t)dt \geq \frac{n(L-4)}{8}r + \frac{3\pi n}{8}\left(\log(1 + L'\frac{r}{3\pi}) + \left(1 - \frac{1}{1 + \frac{L'r}{3\pi}}\right) + \frac{1}{2}\left(1 - \frac{1}{(1 + \frac{L'r}{3\pi})^2}\right)\right)$$

$$\stackrel{\text{when } L' \geq 0}{\geq} \frac{n(L-4)}{8}r + \frac{3\pi n}{8}\log(1 + \frac{L-2}{3\pi}r).$$

∎

**Lemma F.7.** *There exists an absolute constant $C > 0$ such that when $L \geq 2$, we have for any $r > 0$*

$$\int_0^r (\psi(t) - \psi(\pi))dt \geq \frac{3\pi n}{8}\log(1 + \frac{L-2}{3\pi}r) - Cnr\log^2 L$$

**Proof.** Following Lemma F.6 and Lemma F.5, we directly get

$$\int_0^r (\psi(t) - \psi(\pi))dt \geq \int_0^r \psi(t)dt - \left(\frac{n(L-1)}{8} + 6\pi ne^{6C_0}\log^2 L\right)r$$

$$\geq \frac{3\pi n}{8}\log(1 + \frac{L-2}{3\pi}r) - 6\pi ne^{6C_0}\log^2 Lr - \frac{3n}{8}r$$

$$\geq \frac{3\pi n}{8}\log(1 + \frac{L-2}{3\pi}r) - \left(6\pi e^{6C_0} + \frac{3}{8}\right)nr\log^2 L$$

∎

## F.3 Nearly-Matching Upper Bound

**Lemma F.8.** *There exist absolute constants $C, C' > 0$ and absolute constants $K, K' > 0$ such that for any $0 < \varepsilon \leq 1$, if $L \geq K\varepsilon^{-3}$ then for any $0 \leq t \leq K'\varepsilon^3$ one has*

$$\psi(t) - \psi(\pi) \leq (1 + \varepsilon)\left(1 + \frac{18}{1 + (L-3)t/(3\pi)} + \frac{C\log^2(L)}{L}\right)\frac{n}{8}\frac{L-3}{1 + (L-3)t/(3\pi)}$$

*and for any $0 \leq t \leq \pi$ one has*

$$\psi(t) - \psi(\pi) \leq C'n\frac{L-3}{1 + (L-3)t/(3\pi)}.$$

**Proof.** We try to control the DC subtracted skeleton $\psi(t) - \psi(\pi)$ by its derivative $\dot{\psi}(t)$, which would require us to control the derivatives $\dot{\xi}_\ell(t)$ and further $\dot{\varphi}^{[\ell]}(t)$. Fix $0 < \varepsilon \leq 1/2$. When $L \geq C_0\varepsilon^{-2}$ for some constant $C_0 > 0$, Lemma F.3 provides sharp bound for $\dot{\varphi}^{[\ell]}(t)$ with

$$\dot{\varphi}^{[\ell]}(t) \leq \frac{1 + \varepsilon}{(1 + c\ell t)^2} \qquad t \in [0, C'\epsilon^2]$$

$$\dot{\varphi}^{[\ell]}(t) \leq \frac{C_1}{(1 + c\ell t)^2} \qquad t \in [0, \pi]$$

with absolute constants $C', C_1 > 0$ and $c = 1/(3\pi)$. For notation convenience, define $t_1 = C'\varepsilon^2$ and write

$$M_t = \begin{cases} 1 + \varepsilon & 0 \leq t \leq t_1 \\ C_1 & \text{otherwise.} \end{cases}$$

We can compactly write the previous two bounds together as

$$\dot{\varphi}^{[\ell]}(t) \leq \frac{M_t}{(1 + c\ell t)^2}.$$

This allows us to separate $\psi(t) - \psi(\pi)$ into two components $\psi(t) - \psi(t_1)$ and $\psi(t_1) - \psi(\pi)$, where we get the correct constant $1 + \epsilon$ in the first component and control the second component by the fact that $\psi$ becomes sharp when $L$ is large, making the difference between $\psi(t_1)$ and $\psi(\pi)$ negligible

Now, for $\ell \geq 4$, with $c = \frac{1}{3\pi}$, we have

$$|\dot\xi_\ell(t)| = \frac{\xi_\ell(t)}{\pi} \sum_{\ell'=\ell}^{L-1} \frac{\dot\varphi^{[\ell']}}{1 - \varphi^{[\ell']}/\pi}$$

$$\leq \frac{\xi_\ell(t)}{\pi} \sum_{\ell'=\ell}^{L-1} \frac{\dot\varphi^{[\ell']}}{1 - \frac{t/\pi}{1+c\ell't}}$$

$$= \frac{\xi_\ell(t)}{\pi} \sum_{\ell'=\ell}^{L-1} \frac{1 + c\ell't}{1 + c(\ell'-3)t} \dot\varphi^{[\ell']}$$

$$\leq \frac{\xi_\ell(t)}{\pi} \sum_{\ell'=\ell}^{L-1} \frac{1 + c\ell't}{1 + c(\ell'-3)t} \frac{M_t}{(1+c\ell't)^2}$$

Let $c_{1,\ell} = 1 + e^{6C_0 \frac{\log(1+\ell)}{\ell}}$. (F.3) and (F.8) provide control for $\xi_\ell(t)$ and we have

$$|\dot\xi_\ell(t)| \leq \frac{c_{1,\ell}}{\pi} \frac{(1+c(\ell-3)t)(1+c(\ell-2)t)(1+c(\ell-1)t)}{(1+c(L-3)t)(1+c(L-2)t)(1+c(L-1)t)} \sum_{\ell'=\ell}^{L-1} \frac{1 + c\ell't}{1 + c(\ell'-3)t} \frac{M_t}{(1+c\ell't)^2}$$

$$\leq \frac{c_{1,\ell}M_t}{\pi} \frac{(1+c(\ell-3)t)(1+c(\ell-2)t)(1+c(\ell-1)t)}{(1+c(L-3)t)(1+c(L-2)t)(1+c(L-1)t)} \int_{\ell'=\ell-1}^{L-1} \frac{1}{(1+c(\ell'-2)t)^2} d\ell'$$

$$\leq \frac{c_{1,\ell}M_t}{\pi} \frac{(1+c(\ell-3)t)(1+c(\ell-2)t)(1+c(\ell-1)t)}{(1+c(L-3)t)(1+c(L-2)t)(1+c(L-1)t)} \frac{L - \ell}{(1+c(L-3)t)(1+c(\ell-3)t)}$$

$$\leq \frac{M_t}{\pi} \frac{(1+c(\ell-2)t)(1+c(\ell-1)t)(L-\ell)}{(1+c(L-3)t)^2(1+c(L-2)t)(1+c(L-1)t)}$$

$$+ \frac{C\log(1+\ell)/\ell}{\pi} \frac{L}{(1+c(L-3)t)^2}.$$

In moving from the fifth to the sixth line, we used that $(1+c(\ell-3)t)(1+c(\ell)t) = 1 + c(2\ell - 3)t + c^2(\ell-3)\ell t^2 \geq 1 + c(2\ell-4)t + c^2(\ell^2 - 4\ell + 4)t^2 = (1 + c(\ell-2)t)^2$ provided $\ell \geq 4$ and subsequently the integral test. In the splitting in the last line, we used that $M_t$ is always bounded by a (very large) absolute constant, and worst-cased (as this term will be sub-leading in $L$).

To control derivatives of $\psi$, we need to control sums of the derivatives above. We will derive some further estimates for this purpose. First, we calculate

$$\sum_{\ell=1}^{L-1} (1+c(\ell-2)t)(1+c(\ell-1)t)(L-\ell)$$

$$= \sum_{\ell=1}^{L-1} \left( (L-\ell) + (L-\ell)(2\ell-3)ct + (L-\ell)(\ell-1)(\ell-2)c^2t^2 \right)$$

$$= \frac{L(L-1)}{2} + \frac{L(L-1)(L-\frac{7}{2})}{3}ct + \frac{L(L-1)(L-2)(L-3)}{12}c^2t^2$$

$$\leq \left( \frac{\frac{L(L-1)}{2} + \frac{L(L-1)(L-\frac{7}{2})}{3}ct}{\frac{L(L-3)}{12}(1+c(L-1)t)(1+c(L-2)t)} + 1 \right) \frac{L(L-3)}{12}(1+c(L-1)t)(1+c(L-2)t)$$

$$\leq \left( \frac{6\frac{L-1}{L-3} + 4ct(L-1)}{(1+c(L-1)t)(1+c(L-2)t)} + 1 \right) \frac{L(L-3)}{12}(1+c(L-1)t)(1+c(L-2)t).$$

If $L \geq 4$, we can simplify a term in the last line of the previous expression as

$$\frac{6\frac{L-1}{L-3} + 4ct(L-1)}{(1+c(L-1)t)(1+c(L-2)t)} \leq \frac{18}{1+c(L-1)t},$$

and under $L \geq 4$ we also have

$$\sum_{\ell=2}^{L-1} \frac{\log(1+\ell)}{\ell} \lesssim \log(L) \int_{\ell=1}^{L-1} \frac{1}{\ell} \lesssim \log^2 L.$$

Applying the upper bound from before and adding some terms to the sum (because all terms are nonnegative), we get

$$\pi \sum_{\ell=4}^{L-1} \left| \dot{\xi}_\ell(t) \right| \leq \left( 1 + \frac{18}{1+c(L-1)t} \right) \frac{L(L-3)}{12} \frac{M_t}{(1+c(L-3)t)^2} + \frac{CL\log^2 L}{(1+c(L-3)t)^2}. \quad \text{(F.10)}$$

From Lemma G.9, for $\ell = \{0,1,2,3\}$, we can bound $\xi_\ell(t) \leq \frac{1+\ell t/\pi}{1+Lt/\pi}$. Using that $\xi_\ell$ is decreasing for all $\ell \geq 0$ and nonnegative, for $t, t' \in [0, \pi]$, $t' \geq t$, we are now able to control the DC subtracted skeleton as

$$\psi(t) - \psi(t')$$

$$\leq \frac{n}{2} \frac{4 + (0+1+2+3)t/\pi}{1 + Lt/\pi} - \frac{n}{2} \sum_{\ell=4}^{L-1} \int_{v=t}^{t'} \dot{\xi}_\ell(v) dv$$

$$\leq \frac{2 + 3t/\pi}{1 + Lt/\pi} n$$

$$\quad + \frac{n}{2\pi} \int_{v=t}^{t'} \left( \left( 1 + \frac{18}{1+c(L-1)\nu} \right) \frac{M_\nu L(L-3)/12}{(1+c(L-3)\nu)^2} + \frac{CL\log^2 L}{(1+c(L-3)\nu)^2} \right) dv$$

$$\leq \frac{2 + 3t/\pi}{1 + Lt/\pi} n + \frac{n}{2\pi} \int_{v=t}^{t'} \frac{L \left( (1 + \frac{18}{1+c(L-1)t}) \frac{M_{t'}(L-3)}{12} + C \log^2 L \right)}{(1+c(L-3)\nu)^2} dv$$

$$\leq \frac{2 + 3t/\pi}{1 + Lt/\pi} n$$

$$\quad + \frac{n}{2\pi} L \left( \left( 1 + \frac{18}{1+c(L-1)t} \right) \frac{M_{t'}(L-3)}{12} + C \log^2 L \right) \int_{v=t}^{\pi} \frac{1}{(1+c(L-3)\nu)^2} dv.$$

From the second to the third line, we use the fact that $M_\nu$ is nondecreasing in $\nu$. Thus we have

$$\psi(t) - \psi(t') \leq \frac{2 + 3t/\pi}{1 + Lt/\pi} n + \frac{n}{2\pi} \frac{L \left( (1 + \frac{18}{1+c(L-1)t}) \frac{M_{t'}(L-3)}{12} + C \log^2(L) \right) \nu}{1+c(L-3)\nu} \Big|_{\nu=t}^{\pi}$$

$$\leq \frac{5n}{1 + Lt/\pi} + \frac{n}{2} \frac{L \left( (1 + \frac{18}{1+c(L-1)t}) \frac{M_{t'}(L-3)}{12} + C \log^2(L) \right)}{(1+c(L-3)t)(1+c(L-3)\pi)}$$

$$\leq \frac{5n}{1 + Lt/\pi} + \frac{L-3}{1+c(L-3)\pi} \frac{M_{t'} n}{24} \frac{L \left( 1 + \frac{18}{1+c(L-1)t} + \frac{12C \log^2(L)}{L-3} \right)}{1+c(L-3)t}$$

$$\leq \frac{n}{8} \frac{L M_{t'}}{1+c(L-3)t} \left( 1 + \frac{18}{1+c(L-1)t} + \frac{12C \log^2 L}{L-3} \right) + \frac{5n}{1 + Lt/\pi}$$

$$\leq \frac{n M_{t'}}{8} \frac{L-3}{1+c(L-3)t} \left( 1 + \frac{18}{1+c(L-1)t} + \frac{C \log^2(L)}{L-3} \right)$$

$$\leq \frac{n M_{t'}}{8} \frac{L-3}{1+c(L-3)t} \left( 1 + \frac{18}{1+c(L-3)t} + \frac{C \log^2(L)}{L} \right).$$

In moving from the second to the third line, we simplified/rearranged and used that $C_1 \geq 1$. In moving from the fourth to the fifth line, we replace the numerator of $L$ in the leading term with $L - 3 + 3$, then expand and simplify. In particular we have

$$\psi(t_1) - \psi(\pi) \leq \frac{nC_1}{8} \frac{1}{ct_1} \left( 19 + \frac{C \log^2(L)}{L} \right)$$

$$\leq \frac{C_1 C'(19+C)}{8c\epsilon^2}n$$

and

$$\psi(t) - \psi(t_1) \leq \frac{n(1+\epsilon)}{8}\frac{1}{ct}\left(1 + \frac{18}{1+c(L-3)t} + \frac{C\log^2(L)}{L}\right)$$

$$\leq \frac{n(1+\epsilon)}{8}\frac{L-3}{1+c(L-3)t}\left(1 + \frac{18}{1+c(L-3)t} + \frac{C\log^2(L)}{L}\right).$$

This inequality holds for all $t$ because when $t \geq t_1$, the left hand side is negative. Notice that when $t \leq \epsilon^3/(2C_1C'(19+C))$ and $L - 3 \geq C_1C'(19+C)/(c\epsilon^3)$, we would have

$$\frac{C_1C'(19+C)}{8c\epsilon^2}n = \epsilon\frac{n}{8}\frac{L-3}{2c(L-3)\epsilon^3/(2C_1C'(19+C))}$$

$$\leq \epsilon\frac{n}{8}\frac{L-3}{1+c(L-3)\epsilon^3/(2C_1C'(19+C))}$$

$$\leq \frac{n\epsilon}{8}\frac{L-3}{1+c(L-3)t}.$$

Thus when $L \geq C_1C'(19+C)/(c\epsilon^3) + 3$, for $t \leq \epsilon^3/(2C_1C'(19+C))$,

$$\psi(t) - \psi(\pi) = \psi(t) - \psi(t_1) + \psi(t_1) - \psi(\pi)$$

$$\leq \frac{n(1+2\epsilon)}{8}\frac{L-3}{1+c(L-3)t}\left(1 + \frac{18}{1+c(L-3)t} + \frac{C\log^2(L)}{L}\right).$$

Combining the results and notice that we can absorb the factor of 2 into constants by defining $\varepsilon' = 2\varepsilon$ would give us the claim.

∎

**Lemma F.9.** *There exist absolute constants $C, C' > 0$ and $K, K' > 0$ such that for any $0 < \varepsilon \leq 1$, if $L \geq K\varepsilon^{-3}$, for any $0 \leq a \leq b \leq \pi$, one has*

$$\int_a^b (\psi(t) - \psi(\pi))dt \leq Cn\log\left(\frac{1 + (L-3)b/(3\pi)}{1 + (L-3)a/(3\pi)}\right). \tag{F.11}$$

*And if $r > 0$ satisfies $r \leq K'\varepsilon^3$, one further has*

$$\int_r^b (\psi(t) - \psi(\pi))dt \leq (1+\varepsilon)\frac{3\pi n}{8}\log\left(\frac{1 + (L-3)b/(3\pi)}{1 + (L-3)r/(3\pi)}\right) + C'n\log(\pi/(K'\varepsilon^3)). \tag{F.12}$$

**Proof.** (F.11) follows directly from Lemma F.8 and integration. To achieve an upper bound for integral from $r$ to $b$, we cut the integral at $t_1 = K'\varepsilon^3$ and apply bounds from Lemma F.8 separately. Specifically, set $b' = \min\{b, t_1\}$, from Lemma F.8 and (F.11) we would have

$$\int_r^b (\psi(t) - \psi(\pi))dt = \int_r^{b'} (\psi(t) - \psi(\pi))dt + \int_{b'}^b (\psi(t) - \psi(\pi))dt$$

$$\leq (1+\varepsilon)\left(1 + \frac{C\log^2(L)}{L}\right)\frac{3\pi n}{8}\log\left(\frac{1 + (L-3)b'/(3\pi)}{1 + (L-3)r/(3\pi)}\right)$$

$$+ \int_r^b \frac{18n(1+\varepsilon)}{8}\frac{L-3}{(1+(L-3)t/(3\pi))^2}dt + C'n\log\left(\frac{1 + (L-3)b/(3\pi)}{1 + (L-3)b'/(3\pi)}\right)$$

$$\leq (1+\varepsilon)\left(1 + \frac{C\log^2(L)}{L}\right)\frac{3\pi n}{8}\log(\frac{1 + (L-3)b/(3\pi)}{1 + (L-3)r/(3\pi)}) + \frac{9n}{2(3\pi)^2} + C'n\log(b/b')$$

$$\leq (1+\varepsilon)\frac{3\pi n}{8}\log\left(\frac{1 + (L-3)b/(3\pi)}{1 + (L-3)r/(3\pi)}\right) + 2\frac{C\log^2(L)}{L}\frac{3\pi n}{8}\log\left(1 + (L-3)\pi/(3\pi)\right)$$

$$+ \frac{n}{2\pi^2} + C'n\log(\pi/t_1)$$

$$\leq (1+\varepsilon)\frac{3\pi n}{8}\log\left(\frac{1 + (L-3)b/(3\pi)}{1 + (L-3)r/(3\pi)}\right) + \frac{C\log^3(L)}{L} + C'n\log(\pi/\left(K'\epsilon^{-3}\right)).$$

(F.12) then follows by setting $L \geq K\epsilon^{-3}$ for some $K > 0$.

∎

### F.4 Higher Order Derivatives of $\psi$

**Lemma F.10.** *There exist absolute constants $C, C'$ such that when $L \geq C$, we have for any $r \in [0, \pi]$,*

$$\max_{t \geq r}\left|\dot{\psi}(t)\right| \leq \frac{C'n}{r^2} \tag{F.13}$$

*and we can control the integration*

$$\int_{t=0}^{r} t^3 \left|\dot{\psi}(t)\right| dt \leq C'nr^2 \tag{F.14}$$

**Proof.** From (F.10) (we control $M_t \leq C$ for an absolute constant $C > 0$ in this context, so that we do not need to deal with the conditions on $\varepsilon$ that appear there) and Lemma G.10, we have

$$\left|\dot{\psi}(t)\right| \leq \frac{n}{2} \sum_{\ell=0}^{3}\left|\dot{\xi}_\ell(t)\right|$$

$$+ \frac{Cn}{2}\left(\left(1 + \frac{18}{1+(L-1)t/(3\pi)}\right)\frac{\frac{L(L-3)}{12}}{(1+(L-3)t/(3\pi))^2} + \frac{L\log^2 L}{(1+(L-3)t/(3\pi))^2}\right)$$

$$\leq \frac{n}{2}\frac{12L}{1+Lt/\pi}$$

$$+ \frac{Cn}{2}\left(\left(1 + \frac{18}{1+(L-1)t/(3\pi)}\right)\frac{\frac{L(L-3)}{12}}{(1+(L-3)t/(3\pi))^2} + \frac{L\log^2 L}{(1+(L-3)t/(3\pi))^2}\right)$$

$$\leq \frac{6\pi n}{t} + \frac{Cn}{2}\left(\frac{L(L-3)}{(L-3)^2 t^2} + \frac{L\log^2 L}{(L-3)^2 t^2}\right)$$

$$\leq \frac{Cn}{t^2}.$$

This directly get us (F.13) and (F.14). ∎

**Lemma F.11.** *There exist absolute constants $C, C'$ such that when $L \geq C$, we have for any $r > 0$*

$$\max_{t \geq r}\left|\ddot{\psi}(t)\right| \leq C'\frac{n}{r^3} \tag{F.15}$$

*and*

$$\int_{t=0}^{\pi} t^6 \left|\ddot{\psi}(t)\right| dt \leq C'n.$$

**Proof.** Following Lemmas G.8, G.10 and G.11, we have

$$\dot{\xi}_\ell = -\frac{\xi_1}{\pi}\mathbb{1}_{\ell=0} - \frac{\xi_\ell}{\pi}\sum_{\ell'=\max\{1,\ell\}}^{L-1}\frac{\dot{\varphi}^{[\ell']}}{1-\varphi^{[\ell']}/\pi} \tag{F.16}$$

and

$$\left|\ddot{\xi}_\ell\right| = -\frac{\xi_\ell}{\pi}\sum_{\ell'=\max\{1,\ell\}}^{L-1}\frac{\ddot{\varphi}^{[\ell']}}{1-\varphi^{[\ell']}/\pi} + \frac{\xi_\ell}{\pi^2}\sum_{\substack{\ell',\ell''=\max\{1,\ell\}\\ \ell'\neq\ell''}}^{L-1}\frac{\dot{\varphi}^{[\ell']}\dot{\varphi}^{[\ell'']}}{(1-\varphi^{[\ell']}/\pi)(1-\varphi^{[\ell'']}/\pi)}$$

$$- \frac{2\xi_1}{\pi^2}\mathbb{1}_{\ell=0}\sum_{\ell'=1}^{L-1}\frac{\dot{\varphi}^{[\ell']}}{1-\varphi^{[\ell']}/\pi}$$

$$\leq \left|\frac{\xi_\ell}{\pi}\sum_{\ell'=\max\{1,\ell\}}^{L-1}\frac{\ddot{\varphi}^{[\ell']}}{1-\varphi^{[\ell']}/\pi}\right|$$

$$+ \frac{\xi_\ell}{\pi^2} \left( \sum_{\ell'=\max\{1,\ell\}}^{L-1} \frac{\dot\varphi^{[\ell']}}{1 - \varphi^{[\ell']}/\pi} \right)^2 + \left| \frac{2\xi_1}{\pi^2} \mathbb{1}_{\ell=0} \sum_{\ell'=1}^{L-1} \frac{\dot\varphi^{[\ell']}}{1 - \varphi^{[\ell']}/\pi} \right|,$$

where the diagonal is added to obtain the upper bound on the second term for the inequality. We compute $\ddot\varphi^{[\ell]}(t)$ as

$$\ddot\varphi^{[\ell]} = \left( \dot\varphi^{[\ell-1]} \right)^2 \ddot\varphi \circ \varphi^{[\ell-1]} + \ddot\varphi^{[\ell-1]} \dot\varphi \circ \varphi^{[\ell-1]}$$

and thus

$$\frac{\ddot\varphi^{[\ell]}}{\dot\varphi^{[\ell]}} = \dot\varphi^{[\ell-1]} \frac{\ddot\varphi}{\dot\varphi} \circ \varphi^{[\ell-1]} + \frac{\ddot\varphi^{[\ell-1]}}{\dot\varphi^{[\ell-1]}}$$

$$= \sum_{\ell'=0}^{\ell-1} \dot\varphi^{[\ell']} \frac{\ddot\varphi}{\dot\varphi} \circ \varphi^{[\ell']},$$

which gives

$$\left| \ddot\varphi^{[\ell]} \right| = \left| \dot\varphi^{[\ell]} \sum_{\ell'=0}^{\ell-1} \dot\varphi^{[\ell']} \frac{\ddot\varphi}{\dot\varphi} \circ \varphi^{[\ell']} \right|.$$

From Lemma F.14, we have $|\ddot\varphi| \le c_1 = 4$ on $t \in [0, \pi]$ and $\dot\varphi \ge c_2 = \frac{1}{2}$ on $[0, \frac{\pi}{2}]$. As when $\ell > 0$, we have $\varphi^{[\ell]}(t) \le \frac{\pi}{2}$, we separate the case when $\ell = 0$. From Lemma F.16 we get

$$\sum_{\ell'=0}^{\infty} \dot\varphi^{[\ell']}(t) \le \frac{C}{t}.$$

Using the chain rule to get the expression for $\dot\varphi^{[\ell]}$, and concavity of $\varphi$ to get that $\varphi(t) \ge t/2$, and decreasingness of $\dot\varphi$, we have

$$\left| \ddot\varphi^{[\ell]}(t) \right| \le |\ddot\varphi(t)| \prod_{\ell'=1}^{\ell-1} \dot\varphi \circ \varphi^{[\ell']}(t) + \frac{c_1}{c_2} \dot\varphi^{[\ell]}(t) \sum_{\ell'=1}^{\ell-1} \dot\varphi^{[\ell']}(t)$$

$$\le c_1 \prod_{\ell'=1}^{\ell-1} \dot\varphi \circ \varphi^{[\ell']}(t) + \frac{c_1}{c_2} \dot\varphi^{[\ell]}(t) \sum_{\ell'=1}^{\ell-1} \dot\varphi^{[\ell']}(t)$$

$$\le c_1 \prod_{\ell'=0}^{\ell-2} \dot\varphi \circ \varphi^{[\ell']} \circ \varphi(t) + \frac{C}{t} \dot\varphi^{[\ell]}(t)$$

$$\le c_1 \prod_{\ell'=0}^{\ell-2} \dot\varphi \circ \varphi^{[\ell']}(t/2) + \frac{C}{t} \dot\varphi^{[\ell]}(t)$$

$$\le 2c_1 \prod_{\ell'=0}^{\ell-1} \dot\varphi \circ \varphi^{[\ell']}(t/2) + \frac{C}{t} \dot\varphi^{[\ell]}(t/2)$$

$$\le 8\dot\varphi^{[\ell]}(t/2) + \frac{C}{t} \dot\varphi^{[\ell]}(t/2)$$

$$\le \frac{C}{t} \dot\varphi^{[\ell]}(t/2). \tag{F.17}$$

From Lemma G.5, we know $\xi_\ell$ is monotonically decreasing, so $\xi_\ell(t) \le \xi_\ell(t/2)$. Thus (proceeding from our previous bound)

$$\left| \frac{\xi_\ell(t)}{\pi} \sum_{\ell'=\max\{1,\ell\}}^{L-1} \frac{\ddot\varphi^{[\ell']}(t)}{1 - \varphi^{[\ell']}(t)/\pi} \right| \le \frac{C}{t} \left| \frac{\xi_\ell(t/2)}{\pi} \sum_{\ell'=\max\{1,\ell\}}^{L-1} \frac{\dot\varphi^{[\ell']}(t/2)}{1 - \varphi^{[\ell']}(t)/\pi} \right|$$

$$\le \frac{2C}{t} \left| \frac{\xi_\ell(t/2)}{\pi} \sum_{\ell'=\max\{1,\ell\}}^{L-1} \frac{\dot\varphi^{[\ell']}(t/2)}{1 - \varphi^{[\ell']}(t/2)/\pi} \right|,$$

where in the second line we used that $1/2 \leq 1 - \varphi^{(\ell')}(t)/\pi \leq 1$ and the $t/2$ term is no smaller. Similarly, applying Lemma F.16 again, we get

$$\left| \frac{2\xi_1(t)}{\pi^2} \mathbb{1}_{\ell=0} \sum_{\ell'=1}^{L-1} \frac{\dot\varphi^{[\ell']}(t)}{1 - \varphi^{[\ell']}(t)/\pi} \right| \leq \frac{C}{t} \frac{\xi_1(t)}{\pi} \mathbb{1}_{\ell=0} \leq \frac{C}{t} \frac{\xi_1(t/2)}{\pi} \mathbb{1}_{\ell=0}$$

and

$$\frac{\xi_\ell(t)}{\pi^2} \left( \sum_{\ell'=\max\{1,\ell\}}^{L-1} \frac{\dot\varphi^{[\ell']}(t)}{1 - \varphi^{[\ell']}(t)/\pi} \right)^2 \leq \xi_\ell(t) \left( \frac{C}{t} \frac{1}{1 + \ell t/(3\pi)} \right)^2$$

$$\leq \frac{C\xi_\ell(t)}{t^2} \left( 1 + \ell t/(3\pi) \right)^{-2}$$

Combining all these bounds and applying Lemma G.9, we have obtained

$$\left| \ddot\xi_\ell(t) \right| \leq \frac{C}{t} \left| \dot\xi_\ell(t/2) \right| + \frac{C'}{t^2} \frac{1}{1 + Lt/\pi}$$

$$\lesssim \frac{1}{t} \left| \dot\xi_\ell(t/2) \right| + \frac{1}{Lt^3}.$$

Note this holds for all $\ell = 0, \cdots, L-1$, so we directly get $|\ddot\psi(t)| \leq \frac{C}{t} |\dot\psi(t/2)| + \frac{nL}{2} \frac{C'}{Lt^3}$. Thus from Lemma F.10, there exists constant $C, C_1', C_1''$, when $L \geq C$, we have

$$\max_{t \geq r} \left| \ddot\psi(t) \right| \leq \max_{t \geq r} \left( \frac{C''}{t} \left| \dot\psi(t/2) \right| + \frac{nL}{2} \frac{C'}{Lt^3} \right)$$

$$\leq \frac{1}{r} \frac{C_1'}{(r/2)^2} n + C' \frac{n}{r^3},$$

which provides the bound for $L^\infty$ control. For $L^1$ control, we have

$$\int_{t=r}^{\pi} t^6 \left| \ddot\psi(t) \right| dt \leq \int_{t=0}^{\pi} t^6 \left( \frac{C}{t} \left| \dot\psi(t) \right| + \frac{nL}{2} \frac{C'}{Lt^3} \right) dt$$

$$\leq C \int_{t=r}^{\pi} t^3 \left| \dot\psi(t/2) \right| dt + C'n$$

$$\leq Cn,$$

which finishes the proof. ∎

**Lemma F.12.** *There exist absolute constants $C, C'$ such that when $L \geq C$ we have for any $r > 0$*

$$\max_{t \geq r} \left| \dddot\psi(t) \right| \leq \frac{C'n}{r^4} \tag{F.18}$$

*and*

$$\int_{t=0}^{\pi} t^9 \left| \dddot\psi(t) \right| dt \leq C'n.$$

**Proof.** We calculate with the chain rule starting from the representation in Lemma G.8 (and use the triangle inequality)

$$\left| \dddot\xi_\ell \right| \leq \left| \frac{\xi_\ell}{\pi} \sum_{\ell'=\max\{1,\ell\}}^{L-1} \frac{\dddot\varphi^{[\ell']}}{1 - \varphi^{[\ell']}/\pi} \right| + 3 \left| \frac{\xi_\ell}{\pi^2} \sum_{\substack{\ell',\ell''=\max\{1,\ell\} \\ \ell' \neq \ell''}}^{L-1} \frac{\ddot\varphi^{[\ell']}\dot\varphi^{[\ell'']}}{(1 - \varphi^{[\ell']}/\pi)(1 - \varphi^{[\ell'']}/\pi)} \right|$$

$$+ \left| \frac{\xi_\ell}{\pi^3} \sum_{\substack{\ell',\ell'',\ell'''=\max\{1,\ell\} \\ \ell' \neq \ell'', \ell' \neq \ell''', \ell'' \neq \ell'''}}^{L-1} \frac{\dot\varphi^{[\ell']}\dot\varphi^{[\ell'']}\dot\varphi^{[\ell''']}}{(1 - \varphi^{[\ell']}/\pi)(1 - \varphi^{[\ell'']}/\pi)(1 - \varphi^{[\ell''']}/\pi)} \right|$$

$$+ \left| \frac{\xi_1}{\pi^3} \mathbb{1}_{\ell=0} \sum_{\substack{\ell',\ell''=1 \\ \ell' \neq \ell''}}^{L-1} \frac{\dot{\varphi}^{[\ell']} \dot{\varphi}^{[\ell'']}}{(1 - \varphi^{[\ell']}/\pi)(1 - \varphi^{[\ell'']}/\pi)} \right| + \left| \frac{3\xi_1}{\pi^2} \mathbb{1}_{\ell=0} \sum_{\ell'=1}^{L-1} \frac{\ddot{\varphi}^{[\ell']}}{1 - \varphi^{[\ell']}/\pi} \right|$$

$$\leq \left| \frac{\xi_\ell}{\pi} \sum_{\ell'=\max\{1,\ell\}}^{L-1} \frac{\dddot{\varphi}^{[\ell']}}{1 - \varphi^{[\ell']}/\pi} \right|$$

$$+ \frac{3\xi_\ell}{\pi^2} \sum_{\ell'=\max\{1,\ell\}}^{L-1} \frac{\left| \ddot{\varphi}^{[\ell']} \right|}{1 - \varphi^{[\ell']}/\pi} \sum_{\ell''=\max\{1,\ell\}}^{L-1} \frac{\dot{\varphi}^{[\ell'']}}{1 - \varphi^{[\ell'']}/\pi}$$

$$+ \frac{\xi_\ell}{\pi^3} \left( \sum_{\ell'=\max\{1,\ell\}}^{L-1} \frac{\dot{\varphi}^{[\ell']}}{1 - \varphi^{[\ell']}/\pi} \right)^3$$

$$+ \frac{\xi_1}{\pi^3} \mathbb{1}_{\ell=0} \sum_{\ell'=1}^{L-1} \frac{\left| \dot{\varphi}^{[\ell']} \right|}{1 - \varphi^{[\ell']}/\pi} \sum_{\ell''=1}^{L-1} \frac{\dot{\varphi}^{[\ell'']}}{1 - \varphi^{[\ell'']}/\pi} + \left| \frac{3\xi_1}{\pi^2} \mathbb{1}_{\ell=0} \sum_{\ell'=1}^{L-1} \frac{\ddot{\varphi}^{[\ell']}}{1 - \varphi^{[\ell']}/\pi} \right|. \tag{F.19}$$

Following (F.17) and Lemma F.16, we have

$$\sum_{\ell'=1}^{L-1} \left| \ddot{\varphi}^{[\ell']}(t) \right| \leq \sum_{\ell'=1}^{L-1} \frac{\left| \ddot{\varphi}^{[\ell']}(t) \right|}{1 - \varphi^{[\ell']}(t)/\pi} \leq \frac{2C}{t} \sum_{\ell'=1}^{L-1} |\dot{\varphi}(t/2)| \leq C/t^2, \tag{F.20}$$

which leaves the main unresolved term in (F.19) to be $\dddot{\varphi}$. On the other hand, we have from the chain rule

$$\dddot{\varphi}^{[\ell]} = 3\dot{\varphi}^{[\ell-1]} \ddot{\varphi}^{[\ell-1]} (\ddot{\varphi} \circ \varphi^{[\ell-1]}) + \left( \dot{\varphi}^{[\ell-1]} \right)^3 \left( \dddot{\varphi} \circ \varphi^{[\ell-1]} \right) + \dddot{\varphi}^{[\ell-1]} \left( \dot{\varphi} \circ \varphi^{[\ell-1]} \right).$$

Using the product expression $\dot{\varphi}^{[\ell]} = \dot{\varphi}^{[\ell-1]} \dot{\varphi} \circ \varphi^{[\ell-1]}$ and the triangle inequality, we have

$$\left| \frac{\dddot{\varphi}^{[\ell]}}{\dot{\varphi}^{[\ell]}} \right| \leq 3 \left| \ddot{\varphi}^{[\ell-1]} \frac{\ddot{\varphi}}{\dot{\varphi}} \circ \varphi^{[\ell-1]} \right| + \left( \dot{\varphi}^{[\ell-1]} \right)^2 \left| \frac{\dddot{\varphi}}{\dot{\varphi}} \circ \varphi^{[\ell-1]} \right| + \left| \frac{\dddot{\varphi}^{[\ell-1]}}{\dot{\varphi}^{[\ell-1]}} \right|$$

$$\leq \sum_{\ell'=1}^{\ell-1} \left( 3 \left| \ddot{\varphi}^{[\ell']} \frac{\ddot{\varphi}}{\dot{\varphi}} \circ \varphi^{[\ell']} \right| + \left( \dot{\varphi}^{[\ell']} \right)^2 \left| \frac{\dddot{\varphi}}{\dot{\varphi}} \circ \varphi^{[\ell']} \right| \right) + \left| \frac{\dddot{\varphi}}{\dot{\varphi}} \right|$$

where the second line uses induction. From Lemmas F.14 and F.15, we have $|\ddot{\varphi}| \leq c_1 = 4$, $|\dddot{\varphi}| \leq c_4$ on $t \in [0, \pi]$ and $\dot{\varphi} \geq c_2 = \frac{1}{2}$, $\ddot{\varphi} \leq -c_3$ on $[0, \frac{\pi}{2}]$, Again, for $\ell > 0$, we have $\varphi^{[\ell]}(t) \leq \frac{\pi}{2}$. Applying (F.20) and Lemma F.16, we get

$$\left| \frac{\dddot{\varphi}^{[\ell]}(t)}{\dot{\varphi}^{[\ell]}(t)} \right| \leq 3\frac{c_1}{c_2} \sum_{\ell'=1}^{L} \left| \ddot{\varphi}^{[\ell']} \right| + \frac{c_4}{c_2} \sum_{\ell'=1}^{L} \left( \dot{\varphi}^{[\ell']}(t) \right)^2 + \frac{c_4}{\dot{\varphi}(t)}$$

$$\leq C/t^2 + \frac{c_4}{\dot{\varphi}(t)}.$$

Multiplying both side with $\dot{\varphi}^{[\ell]}$, we get the bound

$$\left| \dddot{\varphi}^{[\ell]}(t) \right| \leq \frac{C}{t^2} \dot{\varphi}^{[\ell]}(t) + c_4 \prod_{\ell'=1}^{\ell-1} \dot{\varphi} \circ \varphi^{[\ell']}(t)$$

$$\leq \frac{C}{t^2} \dot{\varphi}^{[\ell]}(t) + c_4 \prod_{\ell'=0}^{\ell-2} \dot{\varphi} \circ \varphi^{[\ell']} \circ \varphi(t)$$

$$\leq \frac{C}{t^2} \dot{\varphi}^{[\ell]}(t) + 2c_4 \dot{\varphi}^{[\ell]}(t/2)$$

$$\leq \frac{C}{t^2} \dot{\varphi}^{[\ell]}(t/2),$$

where the justifications for this argument are very similar to those used in the proof of Lemma F.14.

Plugging bounds we have here back to (F.19). From Lemma F.16 and monotonicity of $\xi_\ell$ in Lemma G.5, we get

$$
\begin{aligned}
\left|\dddot{\xi}_\ell(t)\right| &\leq \left|\frac{\xi_\ell(t)}{\pi} \sum_{\ell'=\max\{1,\ell\}}^{L-1} \frac{C\dot{\varphi}^{[\ell]}(t/2)/t^2}{1-\varphi^{[\ell']}(t)/\pi}\right| \\
&\quad + \frac{3\xi_\ell(t)}{\pi^2}\frac{C}{t^2} \sum_{\ell'=\max\{1,\ell\}}^{L-1} \frac{\dot{\varphi}^{[\ell']}(t)}{1-\varphi^{[\ell']}(t)/\pi} + \frac{\xi_\ell(t)}{\pi^3}\left(\frac{C}{t}\right)^2 \sum_{\ell'=\max\{1,\ell\}}^{L-1} \frac{\dot{\varphi}^{[\ell']}(t)}{1-\varphi^{[\ell']}(t)/\pi} \\
&\quad + \frac{\xi_1(t)}{\pi^3}\mathbb{1}_{\ell=0}\frac{C}{t} \sum_{\ell'=1}^{L-1} \frac{\dot{\varphi}^{[\ell']}(t)}{1-\varphi^{[\ell']}(t)/\pi} + \left|\frac{3\xi_1(t)}{\pi^2}\mathbb{1}_{\ell=0} \sum_{\ell'=1}^{L-1} \frac{\ddot{\varphi}^{[\ell']}(t)}{1-\varphi^{[\ell']}(t)/\pi}\right| \\
&\leq \frac{C}{t^2}\left|\frac{\xi_\ell(t)}{\pi} \sum_{\ell'=\max\{1,\ell\}}^{L-1} \frac{\ddot{\varphi}^{[\ell']}(t/2)}{1-\varphi^{[\ell']}(t)/\pi}\right| + \frac{C\xi_1(t)}{t^2}\mathbb{1}_{\ell=0} \\
&\leq \frac{C}{t^2}\left|\frac{\xi_\ell(t/2)}{\pi} \sum_{\ell'=\max\{1,\ell\}}^{L-1} \frac{\ddot{\varphi}^{[\ell']}(t/2)}{1-\varphi^{[\ell']}(t/2)/\pi}\right| + \frac{C\xi_1(t/2)}{t^2}\mathbb{1}_{\ell=0} \\
&= \frac{C}{t^2}\left|\dot{\xi}_\ell(t/2)\right|
\end{aligned}
$$

where from the second to third line we also use the fact that $1/2 \leq 1 - \varphi^{[\ell]}(t)/\pi \leq 1$ for all $\ell \geq 1$ and the last line follows from the formula of $\dot{\xi}_\ell$ in (F.16).

From our bounds of $\dot{\psi}(t)$ in (F.13), this leads to

$$
\begin{aligned}
\max_{t\geq r}\left|\dddot{\psi}(t)\right| &\leq \max_{t\geq r} \frac{C}{t^2}\left|\dot{\psi}(t/2)\right| \\
&\leq \frac{C}{r^2}\frac{C'}{(r/2)^2} \\
&= \frac{Cn}{r^4}
\end{aligned}
$$

and

$$
\begin{aligned}
\int_{t=0}^{\pi} t^9\left|\dddot{\psi}(t)\right|dt &\leq \int_{t=0}^{\pi} t^9 \frac{C}{t^2}\left|\dot{\psi}(t/2)\right|dt \\
&\leq Cn,
\end{aligned}
$$

as claimed.

∎

## F.5 Additional Proofs for Some Bounds

**Lemma F.13.** *There exists an absolute constant $C_1 > 0$ such that*

$$
\widehat{\varphi}(t) - \varphi(t) \leq C_1 t^3.
$$

**Proof.** From Lemma G.4, $\varphi$ is 3 times continuously differentiable on $(0, \pi)$, and

$$
\varphi(0) = 0, \quad \dot{\varphi}(0) = 1, \quad \ddot{\varphi}(0) = -\frac{2}{3\pi}.
$$

It is easy to check that

$$
\widehat{\varphi}(0) = 0, \quad \dot{\widehat{\varphi}}(0) = 1, \quad \ddot{\widehat{\varphi}}(0) = -\frac{2}{3\pi}.
$$

Since the Taylor expansions of these two functions around $0$ agree to third order, and both are $3$ times continuously differentiable on $(0, \pi)$, we obtain by Lagrange's remainder theorem that for any $t \in [0, \pi)$,

$$\widehat{\varphi}(t) - \varphi(t) = \int_0^t \left( \ddot{\widehat{\varphi}}(s) - \dddot{\varphi}(s) \right) \frac{s^2}{2} \mathrm{d}s \leq C_1 t^3$$

for some finite constant $C_1 = \sup_{t \in [0,\pi)} \left| \dddot{\widehat{\varphi}}(t) - \dddot{\varphi}(t) \right|$. At $t = \pi$ we have $\widehat{\varphi}(\pi) - \varphi(\pi) = \frac{\pi}{1 + \pi/3} - \frac{\pi}{2} \leq 0$ hence the same bound holds for $t \in [0, \pi]$. ∎

**Lemma F.14.** *One has*

$$\dot{\varphi}(t) \geq \frac{1}{2}, \quad t \in [0, \frac{\pi}{2}]$$
$$|\ddot{\varphi}(t)| \leq 4, \quad t \in [0, \pi]$$

**Proof.** We know $\varphi$ is monotonically increasing and concave on $[0, \pi]$, thus for $t \in [0, \pi/2]$,

$$\dot{\varphi}(t) \geq \dot{\varphi}(\frac{\pi}{2})$$
$$= \frac{1/2}{\sin(\varphi(\pi/2))}$$
$$\geq \frac{1}{2}.$$

Using Lemma G.6 we also have for $t \in [0, \pi]$, $\dot{\varphi}(t) \leq \dot{\varphi}(0) = 1$,

$$\varphi(t) \geq \frac{t}{1 + t/\pi} \geq \frac{t}{2},$$

and the first bound here can be used to obtain

$$t - \varphi(t) \leq \frac{t^2/\pi}{1 + t/\pi} \leq t^2/\pi.$$

Thus since $\varphi \leq \pi/2$

$$\cos t \sin \varphi(t) - \dot{\varphi}(t) \sin t \cos \varphi(t) \geq \cos t \sin \varphi(t) - \sin t \cos \varphi(t)$$
$$\geq -\sin(t - \varphi(t)),$$

and in particular, using the expression for $\ddot{\varphi}$ from Lemma G.4

$$-\ddot{\varphi}(t) = -(1 - \frac{t}{\pi}) \frac{\cos t \sin \varphi(t) - \dot{\varphi}(t) \sin t \cos \varphi(t)}{\sin^2 \varphi(t)} + \frac{\sin t}{\pi \sin(\varphi(t))}$$
$$\leq (1 - \frac{t}{\pi}) \frac{\sin(t - \varphi(t))}{\sin^2 \varphi(t)} + \frac{2}{\pi}$$
$$\leq \frac{t^2/\pi}{\sin^2(t/2)} + \frac{2}{\pi}$$
$$\leq \frac{t^2/\pi}{(t/\pi)^2} + \frac{2}{\pi}$$
$$\leq 4.$$

∎

**Lemma F.15.** *There exist constants $c_3, c_4 > 0$ such that $\ddot{\varphi}(t) < -c_3$ for $t \in [0, \frac{\pi}{2}]$ and $|\dddot{\varphi}| \leq c_4$ for $t \in [0, \pi]$.*

**Proof.** The existence of $c_3$ follows from Lemma G.4 directly. The existence of $c_4$ follows from smoothness of $\varphi$ on $(0, \pi)$ and the fact that $\dddot{\varphi}(0) = -\frac{1}{3\pi^2}$, $\dddot{\varphi}(\pi) = \frac{2}{\pi}$ both exist. ∎

**Lemma F.16.** *There exists an absolute constant $C > 0$ such that for any $0 < t \le \pi$ and $\ell \in \mathbb{N}_0$, one has*

$$\sum_{\ell'=\ell}^{\infty} \dot{\varphi}^{[\ell']}(t) \le \frac{C}{t} \frac{1}{1 + \ell t/(3\pi)}$$

**Proof.** Using Lemma F.3, we have

$$\dot{\varphi}^{[\ell]}(t) \le \frac{C}{(1 + \ell t/(3\pi))^2}.$$

We can then calculate

$$\sum_{\ell'=\ell}^{\infty} \dot{\varphi}^{[\ell']}(t) \le C \sum_{\ell'=\ell}^{\infty} \frac{1}{(1 + \ell't/(3\pi))^2} \le C \left( \frac{1}{1 + \ell t/(3\pi)} + \int_{\ell'=\ell}^{\infty} \frac{1}{(1 + \ell't/(3\pi))^2} \, \mathrm{d}\ell' \right)$$

$$\le C \left( \frac{1}{1 + \ell t/(3\pi)} + \frac{3\pi/t}{1 + \ell t/(3\pi)} \right)$$

$$\le \frac{C}{t} \frac{1}{1 + \ell t/(3\pi)},$$

as claimed.

∎

# G  Auxiliary Results

Results in this section are reproduced from the literature for self-containedness, and for the most part are presented without proofs.

## G.1  Certificates Imply Generalization

**Theorem G.1** ([6, Theorem B.1], specialized slightly). *Let $\mathcal{M}$ be a two curve problem instance. For any $0 < \delta \le 1/e$, choose $L$ so that*

$$L \ge C_1 \max \{ C_\mu \log^9(1/\delta) \log^{24} (C_\mu n_0 \log(1/\delta)), \kappa^2 C_\lambda \},$$

*let $N \ge L^{10}$, set $n = C_2 L^{99} \log^9(1/\delta) \log^{18}(Ln_0)$, and fix $\tau > 0$ such that*

$$\frac{C_3}{nL^2} \le \tau \le \frac{C_4}{nL}.$$

*Then if there exists a function $g \in L^2_{\mu\infty}(\mathcal{M})$ such that*

$$\left\| \Theta_\mu^{\mathrm{NTK}}[g] - \zeta_0 \right\|_{L^2_{\mu\infty}(\mathcal{M})} \le C_5 \frac{\sqrt{\log(1/\delta) \log(nn_0)}}{L \min\{\rho_{\min}^{q_{\mathrm{cert}}}, \rho_{\min}^{-q_{\mathrm{cert}}}\}}; \quad \|g\|_{L^2_{\mu\infty}(\mathcal{M})} \le C_6 \frac{\sqrt{\log(1/\delta) \log(nn_0)}}{n\rho_{\min}^{q_{\mathrm{cert}}}},$$

(G.1)

*with probability at least $1 - \delta$ over the random initialization of the network and the i.i.d. sample from $\mu$, the parameters obtained at iteration $\lfloor L^{39/44}/(n\tau) \rfloor$ of gradient descent on the finite sample loss $\mathcal{L}_{\mu^N}$ yield a classifier that separates the two manifolds.*

*The constants $C_1, \dots, C_4 > 0$ depend only on the constants $q_{\mathrm{cert}}, C_5, C_6 > 0$, the constants $\kappa, C_\lambda$ are respectively the extrinsic curvature constant and the global regularity constant defined in [6, §2.1], and the constant $C_\mu$ is defined as $\max\{\rho_{\min}^q, \rho_{\min}^{-q}\}(1 + \rho_{\max})^6 (\min\{\mu(\mathcal{M}_+), \mu(\mathcal{M}_-)\})^{-11/2}$, where $q = 11 + 8q_{\mathrm{cert}}$.*

## G.2  Concentration of the Initial Random Network and Its Gradients

**Theorem G.2** (Corollary of [6, Theorem B.2, Lemma C.11]). *Let $\mathcal{M}$ be a two curve problem instance. For any $d \ge K \log(nn_0 \mathrm{len}(\mathcal{M}))$, if $n \ge K'd^4 L^5$ then one has on an event of probability at least $1 - e^{-cd}$*

$$\|\Theta - \Theta^{\mathrm{NTK}}\|_{L^\infty(\mathcal{M} \times \mathcal{M})} \le Cn/L,$$

*where $c, C, K, K' > 0$ are absolute constants.*

**Lemma G.3** ([6, Lemma D.11])**.** *There are absolute constants $K, K' > 0$ such that if $d \geq K \log(n n_0 \operatorname{len}(\mathcal{M}))$ and $n \geq K' d^4 L$, then*

$$\mathbb{P}\left[\|f_{\boldsymbol{\theta}_0}\|_{L^\infty} \leq \sqrt{d}\right] \geq 1 - e^{-cd},$$

$$\mathbb{P}\left[\|\zeta_0\|_{L^\infty} \leq \sqrt{d}\right] \geq 1 - e^{-cd}.$$

*Define*

$$\zeta(\boldsymbol{x}) = -f_\star(\boldsymbol{x}) + \int_{\mathcal{M}} f_{\boldsymbol{\theta}_0}(\boldsymbol{x}') \mathrm{d}\mu(\boldsymbol{x}').$$

*Then under the same assumptions*

$$\mathbb{P}\left[\|\zeta_0 - \zeta\|_{L^\infty} \leq \sqrt{\frac{d}{L^2} + d^{5/2}\sqrt{\frac{L}{n}}}\right] \geq 1 - e^{-cd}$$

*for some numerical constant c.*

## G.3 Basic Estimates for the Infinite-Width Neural Tangent Kernel

**Lemma G.4** ([6, Lemma E.5])**.** *One has*

1. *$\varphi \in C^\infty(0, \pi)$, and $\dot\varphi$, $\ddot\varphi$, and $\dddot\varphi$ extend to continuous functions on $[0, \pi]$;*

2. *$\varphi(0) = 0$ and $\varphi(\pi) = \pi/2$; $\dot\varphi(0) = 1$, $\ddot\varphi(0) = -2/(3\pi)$, and $\dddot\varphi(0) = -1/(3\pi^2)$; and $\dot\varphi(\pi) = \ddot\varphi(\pi) = 0$;*

3. *$\varphi$ is concave and strictly increasing on $[0, \pi]$ (strictly concave in the interior);*

4. *$\ddot\varphi < -c < 0$ for an absolute constant $c > 0$ on $[0, \pi/2]$;*

5. *$0 < \dot\varphi < 1$ and $0 > \ddot\varphi \geq -C$ on $(0, \pi)$ for some absolute constant $C > 0$;*

6. *$\nu(1 - C_1\nu) \leq \varphi(\nu) \leq \nu(1 - c_1\nu)$ on $[0, \pi]$ for some absolute constants $C_1, c_1 > 0$.*

**Proof.** Combine the results in [6, Lemma E.5] with Lemma F.15 to obtain the conclusion. ∎

**Lemma G.5** (Corollaries of Lemma G.4, stated in [6, Lemma C.10])**.** *One has:*

1. *The function $\varphi$ is smooth on $(0, \pi)$, and (at least) $C^3$ on $[0, \pi]$.*

2. *For each $\ell = 0, 1, \cdots, L$, the functions $\varphi^{[\ell]}$ are nonnegative, strictly increasing, and concave (positive and strictly concave on $(0, \pi)$).*

3. *If $0 \leq \ell < L$, the functions $\xi_\ell$ are nonnegative, strictly decreasing, and convex (positive and strictly convex on $(0, \pi)$).*

4. *The function $\psi$ is smooth on $(0, \pi)$, $C^3$ on $[0, \pi]$, and is nonnegative, strictly decreasing, and convex.*

**Lemma G.6** ([6, Lemma C.13])**.** *If $\ell \in \mathbb{N}_0$, the iterated angle evolution function satisfies the estimate*

$$\varphi^{[\ell]}(t) \geq \frac{t}{1 + \ell t/\pi},$$

**Lemma G.7** ([6, Lemma C.17])**.** *One has for every $\ell \in \{0, 1, \cdots, L\}$*

$$\varphi^{[\ell]}(0) = 0; \quad \dot\varphi^{[\ell]}(0) = 1; \quad \ddot\varphi^{[\ell]}(0) = -\frac{2\ell}{3\pi},$$

*and for $\ell \in [L]$,*

$$\dot\varphi^{[\ell]}(\pi) = \ddot\varphi^{[\ell]}(\pi) = 0.$$

*Finally, we have $\dot\varphi^{[0]}(\pi) = 1$ and $\ddot\varphi^{[0]}(\pi) = 0$.*

**Lemma G.8** ([6, Lemma C.18]). *For first and second derivatives of $\xi_\ell$, one has*

$$\dot{\xi}_\ell = -\pi^{-1} \sum_{\ell'=\ell}^{L-1} \dot{\varphi}^{[\ell']} \prod_{\substack{\ell''=\ell \\ \ell'' \neq \ell'}}^{L-1} \left(1 - \pi^{-1} \varphi^{[\ell'']}\right),$$

*and*

$$\ddot{\xi}_\ell$$

$$= \frac{-1}{\pi} \sum_{\ell'=\ell}^{L-1} \left[ \ddot{\varphi}^{[\ell']} \prod_{\substack{\ell''=\ell \\ \ell'' \neq \ell'}}^{L-1} \left(1 - \pi^{-1}\varphi^{[\ell'']}\right) - \pi^{-1}\dot{\varphi}^{[\ell']} \sum_{\substack{\ell''=\ell \\ \ell'' \neq \ell'}}^{L-1} \dot{\varphi}^{[\ell'']} \prod_{\substack{\ell'''=\ell \\ \ell''' \neq \ell', \ell''' \neq \ell''}}^{L-1} \left(1 - \pi^{-1}\varphi^{[\ell''']}\right) \right],$$

*where empty sums are interpreted as zero, and empty products as* 1. *In particular, one calculates*

$$\xi_\ell(0) = 1; \qquad \dot{\xi}_\ell(0) = -\frac{L-\ell}{\pi}; \qquad \ddot{\xi}_\ell(0) = \frac{(L-\ell)(L-\ell-1)}{\pi^2} + \frac{L(L-1)-\ell(\ell-1)}{3\pi^2},$$

*and*

$$\xi_0(\pi) = 0; \qquad \dot{\xi}_\ell(\pi) = -\frac{1}{\pi}\xi_1(\pi)\mathbb{1}_{\ell=0}; \qquad \ddot{\xi}_\ell(\pi) = 0.$$

**Lemma G.9** ([6, Lemma C.20]). *For all $\ell \in \{0, 1, \ldots, L-1\}$, one has*

$$\xi_\ell(t) \leq \frac{1 + \ell t/\pi}{1 + Lt/\pi}$$

**Lemma G.10** ([6, Lemma C.21]). *One has*

$$|\dot{\xi}_\ell(t)| \leq 3\frac{L-\ell}{1 + Lt/\pi}.$$

**Lemma G.11** ([6, Lemma C.23]). *There are absolute constants $c, C > 0$ such that for all $\ell \in \{0, \ldots, L-1\}$, one has*

$$\left| \ddot{\xi}_\ell \right| \leq C\frac{L(L-\ell)(1 + \ell\nu/\pi)}{(1 + cL\nu)^2} + C\frac{(L-\ell)^2}{(1 + cL\nu)(1 + c\ell\nu)}.$$