# OpenReview forum: "Deep Networks Provably Classify Data on Curves"
_NeurIPS.cc/2021/Conference — NeurIPS 2021 Poster_

### Official Review · Reviewer_2zgs · 2021-07-08

**Rating:** 7
**Confidence:** 4

**Summary:**

This paper studied a binary classification problem for two disjoint smooth curves using a deep ReLU neural network. It proved that when the network depth is large relative to the geometric properties of the curves and when the network width and number of samples is polynomial in the depth, gradient descent can learn to correctly classify all the points on the two curves. The analysis is in the NTK regime and the main technical contribution is to prove that when the network depth is large enough, the corresponding neural tangent kernel can classify the two curves (certificate problem). Analyzing this neural tangent kernel turns out to be very challenging and requires highly non-trivial techniques.

**Limitations And Societal Impact:**

The authors have adequately addressed the limitations and potential negative societal impact of their work

**Main Review:**

I think this is a solid theory paper studying the training of neural networks given data with low-dimensional structures. Currently, there are very few theoretical results dealing with data with low-dimensional structures. So I think this paper is a good step along this line. Although the result is only for two curves, analyzing this setting is already very challenging.

The analysis is in the NTK regime, and the main contribution of this paper is to solve the certificate problem (showing the NTK can approximate the decision boundary with small L_2 norm). Theorem 3.1 requires novel techniques and intense proofs. It shows that the NTK can approximate the decision boundary when the depth of the network is polynomial in the injectivity radius and exponential in the curvature. In the proof, increasing the depth of the network makes the kernel more localized and help with the certificate problem. This is interesting since it gives one perspective on the advantages of deeper neural networks.

My main concern for this paper is that the result is restricted in the NTK regime. But we know that the neural network in practice is often not in the NTK regime and the NTK performed worse than trained neural networks. This makes me doubt to what extend this theory can explain the neural network training in practice (beyond the NTK regime). For example, what's the role of network depth if we train a network beyond the NTK regime to classify two curves.

That being said, I still think this is a solid theoretical contribution to the analysis of NTK and kernel regression under structured data. I think the techniques used in this paper, in particular the proof for the certificate problem, can be very useful in other NTK research.

**Time Spent Reviewing:**

4

---

> ### Author Response · Authors · 2021-08-10
> **Response to reviewer 2zgs**
>
> We thank the reviewer for their careful review. We respond individually to the issues raised by the reviewer below.
>
> > My main concern for this paper is that the result is restricted in the NTK regime. But we know that the neural network in practice is often not in the NTK regime and the NTK performed worse than trained neural networks. This makes me doubt to what extend this theory can explain the neural network training in practice (beyond the NTK regime). For example, what's the role of network depth if we train a network beyond the NTK regime to classify two curves.
>
> We are in complete agreement with the reviewer here -- as the reviewer is no doubt aware, NTK theory can be useful for describing the very early iterations of training for certain practical architectures, and there is some experimental evidence that certain architectures (especially fully-connected networks) perform close to their NTK limits [1], but it is clear that non-kernel techniques are necessary to characterize the performance of modern architectures operating in sample-efficient or "feature learning" regimes. We hope that the technical approach we bring to bear here may be of some use in this endeavor: for example, as we discuss in the second paragraph in the discussion section, our diagonalization-free approach to certificate construction could find application in the study of convolutional architectures, where the network depth may play a stronger role than in the case of a fully-connected architecture. In a similar vein, one interesting direction for future research could be to understand changes to the NTK during training in terms of a "conservation of localization" from initialization: given our analysis that depth localizes the initial NTK, this could make it possible to guarantee progress of gradient descent decreasing the fitting error while accommodating certain changes to the NTK during training that do not negatively affect its ability to guarantee further progress. In this way, our analysis could be helpful in analyzing a regime beyond the typical one where the NTK stays constant.
>
> ### References above
>
> [1] J. Lee et al., “Finite Versus Infinite Neural Networks: an Empirical Study,” arXiv [cs.LG], 31-Jul-2020 [Online]. Available: http://arxiv.org/abs/2007.15801

---

> > ### Comment · Reviewer_2zgs · 2021-08-28
> > **Thanks for the response**
> >
> > Thanks for the response. The authors addressed my concerns on the NTK regime. I think this is a solid theoretical contribution to the analysis of NTK and kernel regression under structured data. I would like to keep my current positive review.

---

### Official Review · Reviewer_t2Mz · 2021-07-11

**Rating:** 6
**Confidence:** 3

**Summary:**

This paper proves that deep fully-connected neural network with gradient descent is able to successfully classify two separated curves in the NTK regime. In particular, the paper defines several key intrinsic properties of the data (two curves) which characterize the required neural network depth, width and sample complexity to obtain a classifier that separates two curves. The results show that if the nonlinearity of the curves increases, the depth requirement also increases, which shows that neural network depth is important. As far as I know, I agree with the authors' claim that this is the first paper that provides generalization guarantees for training a deep neural network to classify structured data.

**Limitations And Societal Impact:**

This is a theory paper. I don't see any potential negative societal impact. The limitations of the paper are well discussed throughout the paper.

**Main Review:**

The paper tries to answer the following question: how large network and how many samples do we need to learn a classifier on structured data. This is a critical question in deep learning theory especially in NTK theory. Therefore, the theoretical results are novel and important. I also believe the techniques developed, especially about the localization effects of NTK with DNN, might be useful to the community. However, there are a few questions and concerns about the paper.

1. The paper assumes the structured data is a two disjoint smooth curves on the unit sphere. I am wondering if it is possible to extend one-dim curves to multi-dim manifold. What's the technical difficulty?

2. The definition of clover number and injectivity radius includes those between inner-class samples.  I didn't find the intuition why we need to include these pairs in these two properties. For example, what if we define $\Delta = \text{inf}_{x\in \mathcal{M}^+, x'\in \mathcal{M}^-} {\angle(x, x')}$ instead? As the authors mentioned in Line 235-237, a shallow network can learn a linearly-separable binary classification problem, where the geometric properties among the inner-class samples don't matter too much.

3. The required depth in Theorem 3.1 and 3.2 seems quite large that is far away from practical cases.

4. The paper gives a great introduction on low-dimensional structure in real applications.  However, I think there is some room for improving the related work section. For example,  more detailed comparison between the results of this paper and the relevant literature can be discussed. I think this paper is also relevant, Arora, Sanjeev, et al. "Fine-grained analysis of optimization and generalization for overparameterized two-layer neural networks." International Conference on Machine Learning. PMLR, 2019.

5. The left part of Fig. 2 seems to show intersection between two different classes while the right part is showing self-intersection inside one class. Also, t-SNE project is a projection from high-dimension space to 2-dim space. The self-intersection might not happen in the original high-dimension space. So the right part of Fig. 2 seems not a good example to me. Also, as mentioned in Appendix A.1, there is a significant error in Fig. 2, which needs to be fixed.

6. A minor: in Line 140, should $\tau < \lambda_1$ be $\tau  \lambda_1 < 1$?

Overall, I think the paper is towards an important direction but the current version is not ready for publication.

**Time Spent Reviewing:**

4

---

> ### Author Response · Authors · 2021-08-10
> **Response to Reviewer t2Mz**
>
> We thank the reviewer for their careful review. We respond individually to the issues raised by the reviewer below.
>
> > ... I think there is some room for improving the related work section. For example, more detailed comparison between the results of this paper and the relevant literature can be discussed. I think this paper is also relevant, ...
>
> We are happy to include a reference to the mentioned work, as well as other similar relevant works on generalization that we inadvertantly omitted from the related work section. We will include this work in the context of our discussion of 'conditional' guarantees in the literature (line 99), since the referenced work is utilizing Rademacher complexity ideas together with a capacity bound via the RKHS norm of the label function in order to bound the generalization error by the RKHS norm $\boldsymbol{y}^{*} (\boldsymbol{H}^\infty)^{-1} \boldsymbol{y}$. We should point out the approach to establishing generalization in Theorem 3.2 follows the NTK regime gradient descent dynamics analysis of Buchanan et al., which is different from that used by Arora et al.: rather than seeking a lower bound on $\lambda_0$ to control the finite-sample error and controlling the generalization error by making the sample size depend on the target's RKHS norm, this result (in essence) makes the network depth large enough to guarantee that the target's RKHS norm is small, then applies this property to analyze the dynamics while taking the sample size large enough relative to the depth to guarantee generalization. This approach is similar to (a quantitative version of) the approach used by Nitanda and Suzuki in the cited reference.
>
> As we discuss, these kinds of 'conditional' results do not have direct implications for the two curve problem without a separate proof that the RKHS norm $\boldsymbol{y}^{*} (\boldsymbol{H}^\infty)^{-1} \boldsymbol{y}$ is small for a given class of curves (and more broadly, a proof that $\lambda_0$ is not pathologically small either). Our present result contributes a solution to these problems for a broad family of curves, and we hope that the related work section and the above discussion make the scarcity of such guarantees in the literature clear.
>
> > The paper assumes the structured data is a two disjoint smooth curves on the unit sphere. I am wondering if it is possible to extend one-dim curves to multi-dim manifold. What's the technical difficulty?
>
> We believe this extension is possible, and that our present submission takes the critical first step in this direction by resolving the core conceptual issues---in particular, identifying the localization phenomenon of the NTK as a tool for working with structured data. We expect the proof schema discussed in Section 4 to generalize to higher dimensions: critically here, our technical estimates for the kernel $\psi^\circ$ in Appendix F are sharp, which means that a suitably-modified localization argument following the approach outlined in Section 4.1 can be made to work with an additional 'price' reflected in the network depth requirements to localize in high-dimensions. We believe this will enable a generalization of our argument for restricted invertibility of $\boldsymbol{\Theta}$. We then see the remaining challenges in generalizing the approach to unrestricted certificate construction described in Section 4.2 as purely technical. In particular, to construct certificates for one-dimensional manifolds, we find it necessary to control three derivatives of the kernel $\psi^\circ$ and the angle function $\angle(\boldsymbol{x}, \boldsymbol{x'})$; to construct certificates for $d$-dimensional smooth manifolds, we expect to need control of $O(d)$ derivatives.
>
> We cut some discussion of this issue from the paragraph in section 5, line 335, due to space constraints, which we will re-include in the revised version.
>
> > The definition of clover number and injectivity radius includes those between inner-class samples. I didn't find the intuition why we need to include these pairs in these two properties ...
>
> Although we present some empirical evidence that in-class properties play a role in the solution of the certificate problem for the general class of curves we consider here (see Figure 3 left, where regions of high curvature that are distant from the opposite class give relatively-large contributions to the certificate's norm), we ultimately see this as a consequence of our analysis of the NTK regime dynamics with the square loss (and in particular, the certificate problem), which may be removable with a more refined harmonic analysis argument (as we mention in the second paragraph of the discussion section). Intuitively, the structure of $\mathcal M$ as a disjoint union of manifolds lets us express the operator $\boldsymbol \Theta$ that governs the certificate problem in the following block-diagonal form:
>
> $\qquad \qquad\qquad\qquad\qquad\qquad\qquad\qquad\qquad\boldsymbol \Theta = \begin{bmatrix} \boldsymbol\Theta_+ & \boldsymbol\Theta_\times \\\\ \boldsymbol\Theta_\times^\ast & \boldsymbol\Theta_-\end{bmatrix},$
>
>  where $\boldsymbol\Theta_\sigma$ is an integral operator on $L^2$ functions defined on $\mathcal M_\sigma$ for $\sigma \in \{+,-\}$, and $\boldsymbol\Theta_\times$ maps functions defined on $\mathcal M_-$ to functions defined on $\mathcal M_+$. As we discuss in section 4, to solve the certificate problem, we need to understand how the eigenfunctions of $\boldsymbol \Theta$ align with $\zeta$ (essentially the label function $f_\star$). Our approach to accomplishing this (described intuitively in lines 287-296 of section 4.1) is to use localization via the network depth to make the off-diagonal blocks $\boldsymbol \Theta_\times$ negligible in size, measured relative to the on-diagonal blocks in the $L^2 \to L^2$ operator norm; we can then analyze invertibility of $\boldsymbol \Theta$ in terms of invertibility of its diagonal blocks $\boldsymbol \Theta_\sigma$, which naturally leads our results to feature a dependence on the in-class properties of the data manifolds. However, it is clear that this analysis misses some possible cancellation between the on- and off-diagonal blocks, as well as with the label function $f_\star$, that could lead to better rates for the certificate problem or an improved dependence on the in-class properties of the geometry. Evidently, even when training in the NTK regime, certain highly-symmetric datasets (such as the linearly-separable geometry in Buchanan et al., Proposition 1) can be classified perfectly without any in-class dependence; but we think it is unclear that guarantees that apply to general classes of regular curves, as we consider here, can avoid these in-class dependences, and see this as an interesting direction for future work.
> > The required depth in Theorem 3.1 and 3.2 seems quite large that is far away from practical cases.
>
> We agree with the reviewer that our rates for the network depth in these theorems are probably pessimistic compared to practice. For our part, we are unaware of any existing guarantees that could be used as a point of reference for our resource requirements for the two curve problem, whether in the theoretical deep learning literature or in the literature on scattered data interpolation (as we discuss in the related work section). More precisely, we are not aware of any quantitative rates for the RKHS norm of specific functions in kernel regression problems on general smooth manifolds: because the certificate problem is roughly equivalent to a proof of the fitting error having small RKHS norm in the RKHS generated by the NTK operator, our Theorem 3.1 can be appropriately cast as a novel proof that the fitting error's RKHS norm can be made small by making the depth large relative to geometric properties of the data. In this sense, we see our present work as the first step in the pursuit of practically-relevant rates for the classification of structured data with deep networks: future work, whether in the NTK regime or outside, will be able to build off of the tools and techniques we introduce here and sharpen the rates for the network we establish in Theorems 3.1 and 3.2.
>
> > The left part of Fig. 2 seems to show intersection between two different classes while the right part is showing self-intersection inside one class. Also, t-SNE project is a projection from high-dimension space to 2-dim space. The self-intersection might not happen in the original high-dimension space. So the right part of Fig. 2 seems not a good example to me. Also, as mentioned in Appendix A.1, there is a significant error in Fig. 2, which needs to be fixed.
>
> Due to the approximate symmetry of the "1" digit under rotation by $\pi$, the points that appear close in the t-SNE projection are in fact close in geodesic distance in the original high-dimensional space as well (this was verified numerically but not included in the submission). The error that is mentioned in Appendix A has been fixed, by considering a four-dimensional geometry that is similar to the one in the current Figure 2 upon projection onto a three-dimensional space. The resulting curves have low curvature, yet nearly intersect multiple times, leading to a large clover number. We will replace this figure in the revised version.

---

> > ### Comment · Reviewer_t2Mz · 2021-08-20
> > **Thank you for the clarification. Increase my score.**
> >
> > Most of my concerns are addressed. Also, considering this paper shows a promising direction in deep learning theory, I'd like to increase my score.

---

### Official Review · Reviewer_skyx · 2021-07-14

**Rating:** 7
**Confidence:** 4

**Summary:**

This paper studies the binary classification problem of datas from two disjoint curve on the unit ball using deep neural network. The authors present a systematic analysis over the geometric difficulty of the problem and relate this to the depth of the neural network. In the NTK region, they provide the generalization analysis based on certain assumptions on width and number of samples.

**Limitations And Societal Impact:**

Yes.

**Main Review:**

Strength:

The curve classification problem is well-motivated and related works are exhaustively described. They present an insightful theorem about the depth of the neural network to ensure the existence of a certificate problem, which is critical to the convergence analysis of two curves problem in NTK region. They also provide an end-to-end guarantee for the generalization property of two curves problem.

Weakness and comments:

The power 99 of $L$ in the expression of the certificate problem parameter $n$ and the power 10 of $L$ in the expression of the sample number $N$ is a bit large, depending on the lower bound of the neural network depth $L$ is large. I’m wondering if these bounds can be further improved.

It will be helpful to provide numerical experiments even on toy examples to better illustrate the generalization guarantee in Theorem 3.2.

Minors:

In line 122, $f_*$ shall be $\mathbb{R}^{n_0}\to\{-1,1\}$.

The linear operator $\boldsymbol{\Theta}_\mu^\mathrm{NTK}$ is not explicitly defined.

**Time Spent Reviewing:**

6

---

> ### Author Response · Authors · 2021-08-10
> **Response to Reviewer skyx**
>
> We thank the reviewer for their careful review. We respond individually to the issues raised by the reviewer below.
>
> > The power 99 of $L$ in the expression of the certificate problem parameter $n$ and the power 10 of $L$ in the expression of the sample number $N$ is a bit large, depending on the lower bound of the neural network depth $L$ is large. I’m wondering if these bounds can be further improved.
>
> We agree with the reviewer that the dependences of width and sample size over depth are fairly unsavory. We see this as a limitation of the NTK tools we use to prove Theorem 3.2, rather than of the analysis in our present submission: for instance, the resource requirements of the form $n \asymp L^{99}$ and $N \geq L^{10}$ that appear in our Theorem 3.2 are consequences of our application of the NTK regime dynamics analysis of Buchanan et al. (given as Theorem G.1 in the appendices), and although these exponents are rather large, they are at least on par with the 'classical' non-generalization results for deep ReLU networks that appear in the literature (e.g. [1-2] below), and in the case of the width are actually superior in the present regime to other generalization-included results for deep ReLU nets in the literature (we are only aware of reference [3] below, which can have an exponential depth dependence). Although there are some results in the literature that provide sharp rates for quantities like the magnitude of $L^\infty$ changes during training of the NTK in restricted cases (e.g. on the diagonal, studying $\Theta(\boldsymbol{x}, \boldsymbol{x})$), such as [4-5] below, there is still a gap to have sharp rates for all NTK-related quantities necessary to prove a generalization result in the NTK regime for deep ReLU networks, and we see this as a wide-open and interesting direction for future research. We mention changes during training in particular as this is the most stringent part of the analysis in Buchanan et al., which drives the eventual $n \asymp L^{99}$ rate.
>
> The sample complexity requirement $N \geq L^{10}$ does not seem too tightly optimized: a careful study of Lemma B.7 of Buchanan et al. suggests that in the one-dimensional case, residual terms of the form $O(L^{3/2} / \sqrt{N})$ or worse appear, driving the best possible sample complexity in this regime to be $N = O(L^3)$. The actual best possible rates one can obtain in our setting may be necessarily worse than some others in the literature, since we ask our trained neural network to perfectly classify both manifolds: if one is only interested in decreasing the empirical risk or the population risk, or if one works with an exponentially-tailed loss like cross-entropy, one may be able to achieve better rates, as well.
>
> We invite the reviewer to also consider our discussion of a similar issue in the response to reviewer `K4pY`.
>
>
> ### References above
> [1] Zeyuan Allen-Zhu, Yuanzhi Li, and Zhao Song. “A Convergence Theory for Deep Learning
> 415 via Over-Parameterization” (Nov. 2018). arXiv: 1811.03962 [cs.LG].
>
> [2] S. S. Du, J. D. Lee, H. Li, L. Wang, and X. Zhai, “Gradient Descent Finds Global Minima of Deep Neural Networks,” arXiv [cs.LG], 09-Nov-2018 [Online]. Available: http://arxiv.org/abs/1811.03804
>
> [3] Z. Chen, Y. Cao, D. Zou, and Q. Gu, “How Much Over-parameterization Is Sufficient to Learn Deep ReLU Networks?,” arXiv [cs.LG], 27-Nov-2019 [Online]. Available: http://arxiv.org/abs/1911.12360
>
> [4] B. Hanin and M. Nica, “Finite Depth and Width Corrections to the Neural Tangent Kernel,” in International Conference on Learning Representations, 2020 [Online]. Available: https://openreview.net/forum?id=SJgndT4KwB
>
> [5] D. A. Roberts, S. Yaida, and B. Hanin, “The Principles of Deep Learning Theory,” arXiv [cs.LG], 18-Jun-2021 [Online]. Available: http://arxiv.org/abs/2106.10165

---

> > ### Comment · Reviewer_skyx · 2021-08-20
> > **Response to the authors**
> >
> > The authors address my major concerns. I would like to keep my score.

---

### Official Review · Reviewer_K4pY · 2021-07-16

**Rating:** 6
**Confidence:** 4

**Summary:**

This paper considers classification using wide (NTK) neural networks. The data are assumed to concentrate on two separable one-dimensional curves on the unit sphere. As long as the width of the network is sufficiently large, well trained neural network can perfectly classify the two curves under some regularity conditions of the curves.

A synthesis of NTK and low-dimensional data is a reasonably novel direction, despite that NTK theory has been well studied in the past few years. From a technical point of view, since the support of the data is no longer the whole unit sphere, the kernel here exhibits difficulty in analyzing. The paper makes plenty of efforts to explain this idea with both verbal and graphical illustrations, which is helpful. As discovered by the theoretical results, depth plays a role in this curve classification problem, whereas on the whole unit sphere, existing works demonstrate little impact of increased depth to the NTK. I feel like this is an interesting finding and contributes to community.

The presentation of the paper is well-organized, especially the preliminaries of some differential geometry and two curve problem.

**Limitations And Societal Impact:**

The authors discussed limitations of the work in Section 5.

**Main Review:**

I have the following questions for the authors, which cover the clarity and quality of the results.

1) Approximate fixed kernel v.s. Kernel at initialization. In Line 137, the evolution of $\zeta_k$ is written using the kernel at initialization. In fact, this should be considered as an approximation, which is already noted by the authors in Line 129. Does the generalization result (Theorem 3.2) takes into account the shift of the kernel during training, or it simply takes the kernel as a fixed one. If the latter is the case, what is the difficulty to analyze the former case?

2) Injectivity radius and clover number. In Riemannian geometry, injectivity radius refers to the smallest radius so that the exponential map is a local diffeomorphism. The introduction of the injectivity radius in the paper seems to alter the definition, although from a geometric point of view, they are related. A discussion may be needed. Moreover, in injectivity radius and clover number, certain choices of $\tau_1$ and $\tau_2$ is provided without explanation. Does it have a direct geometric meaning? or it is for the sake of theoretical convenience?

3) Network size. In Theorem 3.1 and Theorem 3.2, we see the lower bound of depth $L$ is a polynomial with a relatively high degree. Further, the width $n$ in Theorem 3.2 scales as $L^{99}$, which is extremely large. This is partially due to the NTK analysis, however, I am curious whether the authors attempt to optimize the network size. Maybe more interestingly, can the authors comment on how does the network size depend on the intrinsic dimension of the curves? (This is a relatively minor point, as the authors have already presented a discussion on extensions to image data and CNNs.)



**Time Spent Reviewing:**

24 hours

---

> ### Author Response · Authors · 2021-08-10
> **Response to Reviewer K4pY**
>
> We thank the reviewer for their careful review. We respond individually to the issues raised by the reviewer below.
>
> > Approximate fixed kernel v.s. Kernel at initialization ... does the generalization result (Theorem 3.2) takes into account the shift of the kernel during training, or it simply takes the kernel as a fixed one. If the latter is the case, what is the difficulty to analyze the former case?
>
> Yes, the guarantees in Theorem 3.2 apply to the training of the neural network by randomly-initialized vanilla gradient descent with a constant step-size: the approximate fixed kernel and kernel at initialization are just analytical tools we use to establish the proof. If this point was unclear because of the very concise problem formulation in section 2.1, we note that a more rigorous formulation of the problem appears in section C.1 of the appendix, where all details of the gradient descent algorithm we study (and to which Theorem 3.2 applies) are given.
>
> > Injectivity radius and clover number. In Riemannian geometry, injectivity radius refers to the smallest radius so that the exponential map is a local diffeomorphism. The introduction of the injectivity radius in the paper seems to alter the definition, although from a geometric point of view, they are related. A discussion may be needed.
>
> Thank you for pointing this out -- we are embarassed to have overlooked this collision in terminology. We have taken to calling the parameter $\Delta$ the "injectivity radius" in our analysis because it defines a region in the sphere over which we can guarantee that the parametric angle function $t' \mapsto \angle(\boldsymbol x_\sigma(t), \boldsymbol x_\sigma(t + t'))$ is injective for all $t$, all $\sigma \in \{ +, - \}$, and every $t'\geq0$ such that the second argument of the angle lies in this region. As such, we will change all uses of the term "injectivity radius" to "angle injectivity radius" in the revised version.
>
> >  Moreover, in injectivity radius and clover number, certain choices of $\tau_1$ and $\tau_2$ is provided without explanation. Does it have a direct geometric meaning? or it is for the sake of theoretical convenience?
>
> The size of $\tau_1$ and $\tau_2$ scale as the inverse curvature, which serves as a crucial threshold for when the extrinsic distance (angle) can be well approximated by the intrinsic distance (manifold distance). This is shown with more details in Lemma E.4 in the appendix. The choice for the constants in front of $\tau_1$ and $\tau_2$ is for pure theoretical convenience. We decompose the contribution of the kernel into "local", "near", "far" and "winding" pieces (equation (E.3)-(E.6)). A small $\tau_1$ helps control the contribution of the "near" piece, and any choice of the constant for $\tau_1$ smaller than the setting in the paper would work just as well. $\tau_2$ is chosen to avoid creating a "winding" piece with extra length. The details of this treatment are presented in the paragraph "Parameter choice" at the start of section E.1.
>
> > Network size. In Theorem 3.1 and Theorem 3.2, we see the lower bound of depth $L$ is a polynomial with a relatively high degree. Further, the width $n$ in Theorem 3.2 scales as $L^{99}$, which is extremely large. This is partially due to the NTK analysis, however, I am curious whether the authors attempt to optimize the network size.
>
> There are two drivers of the high-degree polynomials appearing in the hypotheses of Theorem 3.2: the geometric properties in the condition on depth are a consequence of our analysis of the certificate problem in the present submission (encapsulated in Theorem 3.1), and the remaining conditions involving the width, the sample size $N$, and the statistical constants involving the density $\rho$ are a consequence of our application of the NTK-regime dynamics analysis of Buchanan et al., given as Theorem G.1 in the appendices. As such, we believe the optimization of these latter rates is out-of-scope for our present submission, but we do believe there is significant room for improvement here: for example, reference [5] below established that for ReLU networks trained in the NTK regime, the scaling $n = o(L^3)$ is in some sense a threshold at which the concentration of specific elements of the initial random NTK and their $L^\infty$ changes after the first gradient step are either well-behaved or chaotic, marking this as a "best possible" scaling for NTK regime training. Of course, this threshold does not consider the entire training and generalization process, so it may be the case that worse scalings are necessary if one wants to guarantee generalization after the training process. The rates of Buchanan et al. compare favorably to other rates involving generalization guarantees for training deep ReLU networks for classification tasks: among the few applicable works, the most relevant is [1], which involves an exponential dependence of width on depth in certain regimes. We invite the reviewer to also consider our discussion of a similar issue in the response to reviewer `skyx`.
>
> As for the rates arising from our analysis of the certificate problem, as encapsulated in Theorem 3.1, we have only taken significant effort to optimize these dependences in order to obtain a polynomial dependence of the condition on $L$ on $\Delta^{-1}$ among geometries with bounded clover number: we have not attempted to optimize most constants in this result, nor optimize the exponents on various polynomials. Although sharpening these rates seems to be valuable as future research, as the reviewer suggests, we find extensions of these techniques to the settings of CNNs and image data, as well as to higher-dimensional geometries, more pressing directions for future work. In addition, some possible 'qualitative' optimizations of these conditions may be possible and more urgent: for example, as we discuss in our response to a related point raised by reviewer `t2Mz` and allude to briefly in the second paragraph of the discussion section, a more refined harmonic analysis argument beyond what we employ in our present work and describe in section 4.1 may yield e.g. all-polynomial dependences on curvature in these conditions, or even milder dependences on in-class properties of the geometries.
> Nethertheless, we consider our present work as the first step in the pursuit of practically-relevant rates for the classification of structured data with deep networks: future work, whether in the NTK regime or outside, will be able to build off of the tools and techniques we introduce here and sharpen the rates for the network we establish in Theorems 3.1 and 3.2.
>
> > Maybe more interestingly, can the authors comment on how does the network size depend on the intrinsic dimension of the curves?
>
> Our analysis, and the dynamics argument of Buchanan et al. on which Theorem 3.2 relies, only apply to one-dimensional manifolds. We believe that both can be extended to manifolds of dimension $d>1$ following the proof schema that we outline in Section 4: here, the localization argument in Section 4.1 will naturally lead to an increase in the resource requirements, specifically in terms of the network depth. We expect this would incur at best an exponential dependence on dimension $d$, which seems unavoidable and is a consequence of the structure of the problem. Such exponential dependence arises for instance in approximation results for ReLU networks (e.g. [2,3]), and has even been proposed as an explanation for the power law scaling commonly observed in the test loss of models trained beyond the NTK regime [4]. In addition, several fairly conclusive lower bounds for kernel methods have appeared in the recent literature (e.g. [6]), which establish exponential-in-dimension separations via sample complexity -- NTK approaches to dynamics necessarily fall into this category.
>
> ### References above
>
> [1] Z. Chen, Y. Cao, D. Zou, and Q. Gu, “How Much Over-parameterization Is Sufficient to Learn Deep ReLU Networks?,” arXiv [cs.LG], 27-Nov-2019 [Online]. Available: http://arxiv.org/abs/1911.12360
>
> [2] Schmidt-Hieber, Johannes. "Deep ReLU network approximation of functions on a manifold." arXiv preprint arXiv:1908.00695 (2019).
>
> [3] Petersen, Philipp, and Felix Voigtlaender. "Optimal approximation of piecewise smooth functions using deep ReLU neural networks." Neural Networks 108 (2018): 296-330.
>
> [4] Bahri, Yasaman, et al. "Explaining neural scaling laws." arXiv preprint arXiv:2102.06701 (2021).
>
> [5] B. Hanin and M. Nica, “Finite Depth and Width Corrections to the Neural Tangent Kernel,” in International Conference on Learning Representations, 2020 [Online]. Available: https://openreview.net/forum?id=SJgndT4KwB
>
> [6] Z. Allen-Zhu and Y. Li, “Backward Feature Correction: How Deep Learning Performs Deep Learning,” arXiv [cs.LG], 13-Jan-2020 [Online]. Available: http://arxiv.org/abs/2001.04413

---

> > ### Comment · Reviewer_K4pY · 2021-08-21
> > **Thank you for the response. I keep the score.**
> >
> > After reading the detailed response, I would like to keep the score.
> >
> > My main concerns about assuming a fixed kernel and terminology conflicts are carefully addressed in the response. I also appreciate the explanations about the network size and extension to intrinsic dimension larger than 1. Overall, I think after some revision, the paper contributes well to theoretical understandings of overparameterized neural networks learning with structured data.

---

### Author Response · Authors · 2021-08-10
**Response to All Reviewers**

We thank all reviewers for their careful reading of the work and insightful comments. We will update the revised version with all typos and suggested comments: we provide individual responses below.

We make one general comment on our results: in addition to our existing assumptions that the data manifold is formed by two disjoint smooth, regular, simple closed curves, our results additionally require an assumption that the curves lie in a spherical cap of radius $\pi/2$: for example, the intersection of the sphere and the nonnegative orthant. This constraint is only to avoid technical issues that arise when antipodal points are present in $\mathcal{M}$, so defining the cap radius with any constant less than $\pi$ would work just as well. We would like to note that this assumption is naturally satisfied by data corresponding to image manifolds, which have nonnegative pixel intensities: in this sense it is not very restrictive as a modeling assumption.

---

### Decision · Program_Chairs · 2021-09-28

**Decision:**

Accept (Poster)

**Comment:**

This paper studies classification problems where the problem data are two disjoint curves using deep neural networks, and provides a Neural Tangent Kernel (NTK) based analysis of generalization. The authors identify key concepts related to the geometric difficulty of the problem and relate it to the depth of the network. The reviewers all agreed that the paper contains interesting ideas and insightful theoretical results, and recommended acceptance. A few reviewers expressed minor concerns and suggestions. Please take into account the updated reviews when preparing the final version to accommodate the requested changes. Thank you for your submission to NeurIPS.

**Consistency Experiment:**

NeurIPS has a long history of experimentation. In 2014, NeurIPS ran an experiment in which 10% of submissions were reviewed by two independent committees to quantify the randomness in the review process. This year, we repeated a variant of this experiment to see how the quality of the review process has changed over time.  This paper was part of the experiment and was therefore assigned to two committees (consisting of reviewers, an Area Chair, and a Senior Area Chair) that reached independent decisions.  If both committees made the same recommendation, this recommendation was followed. If a single committee recommended acceptance, the paper was accepted (with the exception of a few cases in which the other committee identified what we considered a fatal flaw, e.g., an error in a key result).

Both committees reached the same decision: **Accept (Poster)**

The other committee assigned to the paper recommended **Accept (Poster)**.  You can find the other set of reviews, along with any follow up discussion with the authors here:
https://openreview.net/forum?id=o2qkHsdqw_X